# A cellular basis for mapping behavioural structure

Mohamady El-Gaby[1,2 ✉], Adam Loyd Harris[1,2,3], James C. R. Whittington[2,4], William Dorrell[5], Arya Bhomick[2,6], Mark E. Walton[2,3], Thomas Akam[3] & Timothy E. J. Behrens[1,2,6 ✉]

To flexibly adapt to new situations, our brains must understand the regularities in the world, as well as those in our own patterns of behaviour. A wealth of findings is beginning to reveal the algorithms that we use to map the outside world[1–6]. However, the biological algorithms that map the complex structured behaviours that we compose to reach our goals remain unknown. Here we reveal a neuronal implementation of an algorithm for mapping abstract behavioural structure and transferring it to new scenarios. We trained mice on many tasks that shared a common structure (organizing a sequence of goals) but differed in the specific goal locations. The mice discovered the underlying task structure, enabling zero-shot inferences on the first trial of new tasks. The activity of most neurons in the medial frontal cortex tiled progress to goal, akin to how place cells map physical space. These 'goal-progress cells' generalized, stretching and compressing their tiling to accommodate different goal distances. By contrast, progress along the overall sequence of goals was not encoded explicitly. Instead, a subset of goal-progress cells was further tuned such that individual neurons fired with a fixed task lag from a particular behavioural step. Together, these cells acted as task-structured memory buffers, implementing an algorithm that instantaneously encoded the entire sequence of future behavioural steps, and whose dynamics automatically computed the appropriate action at each step. These dynamics mirrored the abstract task structure both on-task and during offline sleep. Our findings suggest that schemata of complex behavioural structures can be generated by sculpting progress-to-goal tuning into task-structured buffers of individual behavioural steps.

Our behaviours are highly structured. From cooking a meal to solving a maths problem, we compose elaborate sequences of actions to achieve our goals. When elements of this structure are common across tasks, we can build schemata; generalized representations of task states that enable us to infer new behavioural sequences[7,8]. Lesion, imaging and neurophysiological studies broadly implicate the frontal areas of the neocortex in mapping task structure. This involves roles in forming a schema of task structure[9–15], generating and predicting complex behavioural sequences[16–18], encoding goals[19,20], simultaneously tracking a working memory of multiple task variables[21,22] and rapidly switching between tasks[23,24]. A key challenge is to derive biological algorithms that explain how frontal activity generates maps of task structure.

Maps of task structure should comprise neuronal dynamics that evolve as a function of progress in a task, rather than related variables such as elapsed time or the number of actions taken. This would naturally enable representations to generalize across goal-directed behavioural sequences that differ in length and duration. Indeed, frontal neurons track progress relative to individual goals[25–30], regardless of

the location of the goal or the distance covered to reach it[27]. However, behavioural tasks are complex and often composed of multiple, hierarchically organized goals[31,32]. It remains unclear how neurons track progress along such complex, multi-goal tasks. One view holds that neurons in the medial frontal cortex (mFC) encode abstract task states rather than specific stimuli or actions[9,14,33–35]. Such state neurons would then be flexibly bound to neurons representing detailed behavioural sequences that are composed in each new task[36–38]. Alternatively, a separate line of work on recurrent neural networks suggests that schematic inferences can be made in new scenarios without building or binding new representations. Here details of new task examples are stored as patterns of neural activity using network dynamics sculpted, through the learning of previous examples, by the abstract structure of the task[39]. Whether and how such representational logic relates to generating a schema that tracks an animal's progress in task space remains an open question.

Here we sought to elucidate a neuronal algorithm for mapping abstract task structure. We trained mice on a series of tasks, each of which required visiting four goal locations in a repeating, loop-like

[1]Nuffield Department of Clinical Neurosciences, University of Oxford, Oxford, UK. [2]Wellcome Centre for Integrative Neuroimaging, University of Oxford, Oxford, UK. [3]Department of Experimental Psychology, University of Oxford, Oxford, UK. [4]Department of Applied Physics, Stanford University, Palo Alto, CA, USA. [5]Gatsby Computational Neuroscience Unit, University College London, London, UK. [6]Sainsbury Wellcome Centre for Neural Circuits and Behaviour, University College London, London, UK. ✉e-mail: mohamady.el-gaby@ndcn.ox.ac.uk; behrens@fmrib.ox.ac.uk

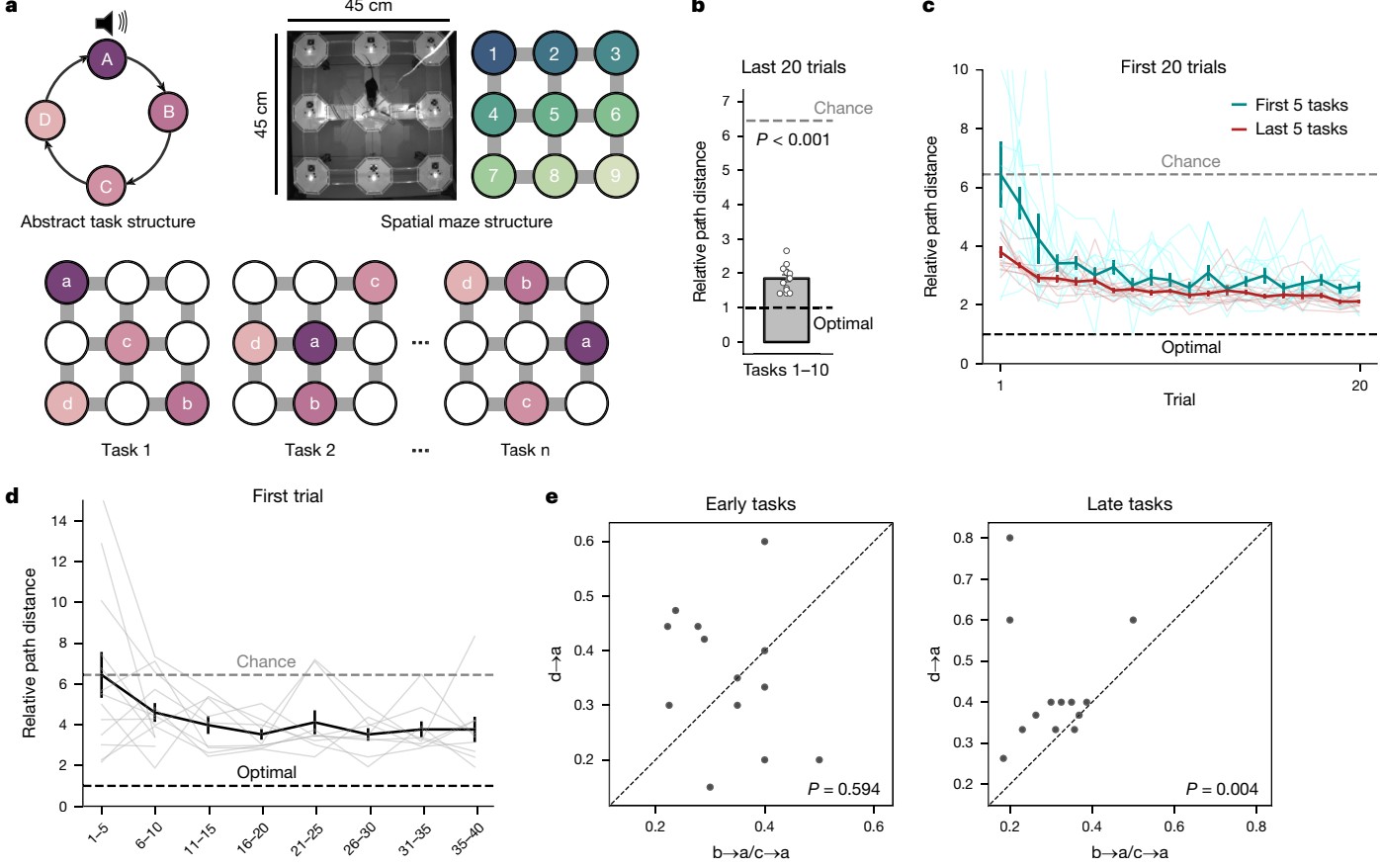

**Fig. 1 | Mice learn an abstract task structure. a**, Task design. Mice learned to navigate between 4 sequential goals on a 3 × 3 spatial grid maze. Reward locations changed across tasks but the abstract structure, four rewards arranged in an ABCD loop, remained the same. A brief tone was played upon reward delivery in location a. **b**, When allowed to reach peak performance (70% shortest path transitions or 200 trial plateau), mice readily reached near-optimal performance in the last 20 trials, as demonstrated by comparing path length between goals to the shortest possible path ('relative path distance'). Two-sided $t$-test against chance (6.44): $n = 13$ mice, $t$-statistic = −43.2, $P = 1.54 \times 10^{-14}$, d.f. = 12. **c**, Performance improved across the initial 20 trials of each new task. This improvement was markedly more rapid for the last five tasks compared to the first five tasks. A two-way repeated-measures ANOVA ($n = 13$ mice) showed a main effect of trial: $F = 11.7$, $P = 1.5 \times 10^{-5}$, d.f.$_1$ = 19,

d.f.$_2$ = 228; task: $F = 35.0$, $P = 7.1 \times 10^{-5}$, d.f.$_1$ = 1, d.f.$_2$ = 12; and trial × task interaction: $F = 2.99$, $P = 0.030$, d.f.$_1$ = 19, d.f.$_2$ = 228. Lines in lighter shades represent performance of individual mice. **d**, Performance on the first trial improved markedly across tasks. One-way repeated measures ANOVA ($n = 9$ mice; only 9 of the 13 mice were presented with all 40 tasks) showed a main effect of task: $F = 2.73$, $P = 0.016$, d.f.$_1$ = 7, d.f.$_2$ = 42. Lines in lighter shades represent performance of individual mice (4 mice only completed 10 tasks). **e**, Mice readily performed zero-shot inference on the first trial of late tasks but not in early tasks. The proportion of tasks in which mice took the most direct path from d to a on the first trial is compared to premature returns from c to a and b to a. Two-sided Wilcoxon test; early tasks: $n = 13$ mice, $W$-statistic = 17.5, $P = 0.168$; late tasks: $n = 13$ mice, $W$-statistic = 3.0, $P = 0.004$. Data are mean ± s.e.m.

sequence. The sequential loop structure relating the goals remained the same across tasks, whereas the goal locations changed. Mice used this abstract structure to perform zero-shot inferences on the first trial of new tasks. Using multi-unit silicon probe recordings, we found that neurons in the medial frontal cortex tracked progress to the next goal, regardless of the behavioural sequences used to reach it. Crucially, these neurons were further sculpted into memory buffers, where neuronal dynamics along each buffer tracked the 'position' of the mouse in task space from a particular behavioural step. Each of these buffers was shaped by the abstract task structure, reflecting the four (or five)-reward loop, and thus enabled prediction of the animals' future actions a long way into the future. These findings point to an algorithm that uses structured memory buffers (SMBs) to encode new behavioural sequences into the dynamics of neural activity without needing associative binding. More broadly, we propose that goal-progress cells in the medial frontal cortex may be elemental building blocks of schemata that can be sculpted into buffers that collectively represent complex behavioural structures.

## The ABCD task

We developed the 'ABCD' task, wherein multiple goals are hierarchically organized by an abstract structure. Mice ($n = 13$) learned to navigate to identical water rewards arranged in a sequence of 4 locations (a–d) on a 3 × 3 grid maze (Fig. 1a and Extended Data Fig. 1a,b). Once reward d was obtained, reward a became available again, allowing the mouse to complete another loop. Each rewarded location (a, b, c or d) defined the beginning of a task 'state' (A, B, C or D, respectively; Fig. 1a) and a brief tone was played upon reward delivery in location a, marking the beginning of every trial. Mice encountered multiple tasks where the reward locations changed but the general ABCD loop structure remained the same (Fig. 1a). Crucially, task structure was made orthogonal to the structure of physical space and the reward sequences were orthogonal across tasks (Fig. 1a and Extended Data Fig. 1b–d). This encouraged the mice to disentangle the abstract task structure from the spatial structure of the maze.

Mice converged on a near-optimal trajectories that routinely took them between goal locations via close-to-shortest routes (Fig. 1b and

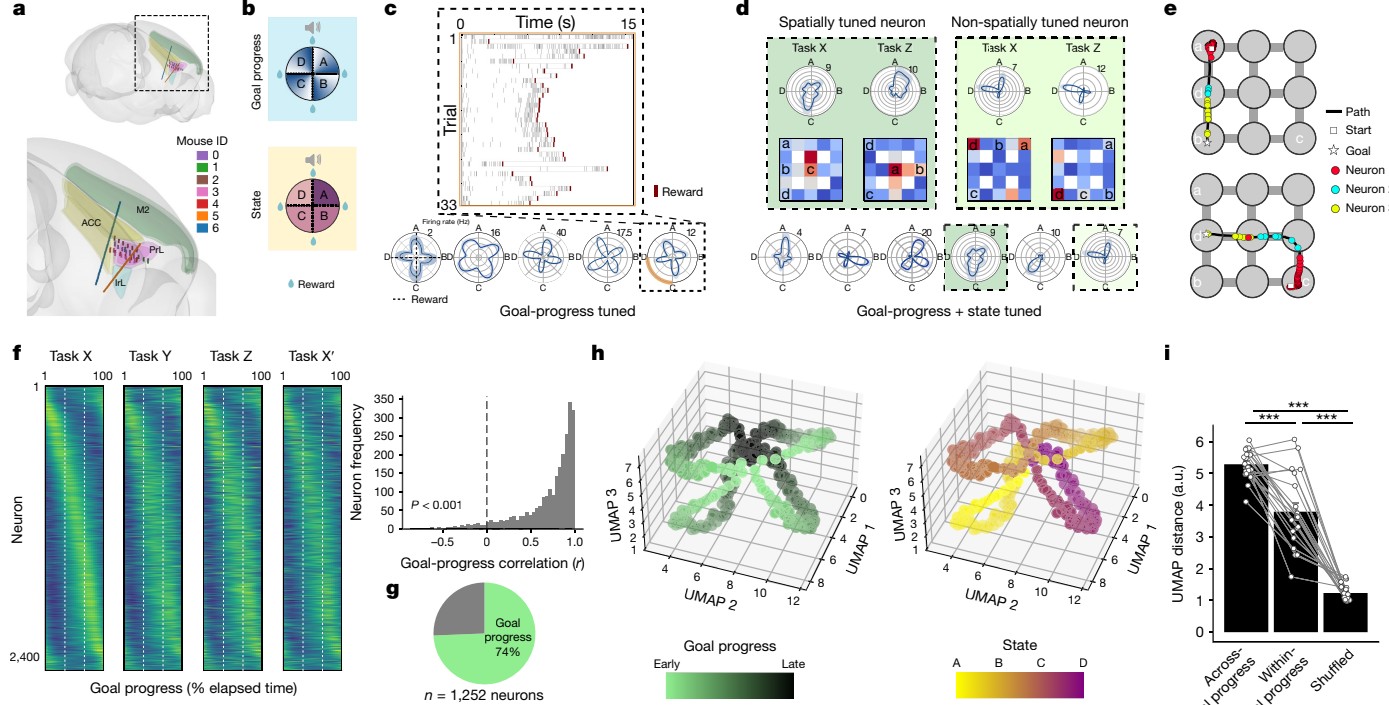

**Fig. 2 | Progress-to-goal is a key feature of task-tuned neurons in the medial frontal cortex. a**, 3D rendering of probe channel positions, with the inset showing mFC regions (Using HERBs[57]). Contacts were mostly in prelimbic cortex (PrL) but also in anterior cingulate cortex (ACC), infralimbic cortex (IrL) and secondary motor cortex (M2). Mouse IDs: 0, me08; 1, ah03; 2, me10; 3, me11; 4, ah04; 5, ab03; 6, ah07. **b**, Schematics of polar plots used to project neuronal activity onto the circular task structure. The radial and angular axes represent firing rate and task position, respectively. Dashed lines along the cardinal directions represent reward (goal) times in each state. **c**, Neurons are tuned to the relative progress to goal of the mouse (goal-progress tuned). Top, a raster plot of firing activity in one state (C: orange segment of polar plot below) of a cell that consistently fires shortly before a goal is reached. Bottom, polar plots of task activity for five separate neurons, with maximum firing rates (Hz) on the top right of each polar plot. **d**, Bottom, some goal-progress-tuned cells are additionally modulated by state in a given task (goal-progress + state-tuned). Top, polar plots and spatial maps for a spatially tuned and state-tuned neuron (left) and a non-spatially tuned and state-tuned neuron (right) across two distinct task configurations. **e**, Two example goal-directed paths and overlaid spiking of three mFC neurons tuned to early, intermediate and late goal progress, regardless of the spatial trajectory of the mouse.

**f**, Goal-progress tuning is consistent across tasks that differ in reward locations. Left, the average firing rate vector of all neurons relative to an individual goal (averaged across all states) arranged by peak goal-progress bin in task X. This alignment is largely maintained in tasks Y and Z as well as a later session of the first task (X'). Right, histogram showing the mean goal-progress vector correlation across tasks for each neuron. One-sample, two-sided *t*-test against 0: $n = 2,461$ neurons; *t*-statistic = 104.3; $P = 0.0$, d.f. = 2,460. **g**, Pie chart showing the proportion of all neurons that are goal-progress tuned: 74%; two proportions test: $n = 1,252$ neurons, $z = 35.5$, $P = 0.0$. **h**, Plot of the mean task manifold derived from UMAP embedding. The same manifold is shown twice to emphasize goal-progress tuning (left) and state tuning (right). The task manifold is composed of goal-progress subloops. **i**, Distances along the three-dimensional manifold across different states and opposite goal-progress bin, across different states but for the same goal-progress bin or across different states and same goal-progress bin for a shuffled control. $n = 20$ double days; two-sided *t*-tests with Bonferroni correction: across-goal progress versus within-goal progress: *t*-statistic = 6.09, $P = 2.25 \times 10^{-5}$, d.f. = 19; across-goal progress versus permuted control: *t*-statistic = 26.0, $P = 7.85 \times 10^{-16}$, d.f. = 19; within-goal progress versus permuted control: *t*-statistic = 8.63, $P = 1.60 \times 10^{-7}$, d.f. = 19. Data are mean ± s.e.m. a.u., arbitrary units.

Extended Data Fig. 1c) using stereotyped trajectories (Extended Data Fig. 1c,e). In the first 10 tasks, we allowed mice to perform as many trials as needed to converge on peak performance (Fig. 1b; trials per task: 335 ± 16). Subsequently, to encourage generalization of task structure, we moved mice to a high-task regime (tasks 11–40). Mice experienced 3 new tasks per day and hence could only complete 35 ± 2 trials per task. Despite this, the mice still performed markedly above chance (Fig. 1c) and were unaffected by the presence of the tone at a (Extended Data Fig. 1f,g). Suboptimal performance was associated with persisting pre-task exposure biases in maze exploration (Extended Data Fig. 1h).

We next tested whether mice learn the abstract task structure. Performance improved across tasks (Fig. 1c and Extended Data Fig. 1i–k) even on the first trial of each new task (Fig. 1d). A subset of mice also subsequently rapidly learned a different abstract structure, performing 'ABCDE' tasks comprising five rewarded goals arranged in a loop (Extended Data Fig. 1l,m). Notably, the ABCD loop allows for a direct test of structural knowledge. If mice understand the loop structure, upon reaching reward location d on the first trial of a new task, they should return directly to a. This zero-shot inference would reflect abstract

knowledge of the ABCD task structure rather than memory retrieval, as the mice are yet to experience the d→a transition. Remarkably, we found that experienced mice took the shortest path between d and a on the first trial more often than chance and more readily than premature returns to a from b or c (Fig. 1e). This was not explained by pre-existing biases in the mice's exploration of the maze, differences in analytical chance levels or differences in the distances of the d-to-a transition compared with those for c-to-a and b-to-a (Extended Data Fig. 1n–p). Moreover, mice not only waited until four rewards were obtained before returning to a (Fig. 1e) but also more readily returned to a than to other reward locations after four rewards (Extended Data Fig. 1q). Mice therefore learned an abstract, task-defined behavioural structure nesting multiple goals.

## Frontal task structure representations

Animals track their 'position' in task space (Fig. 1). To understand the basic neuronal underpinnings of this 'map' of behavioural structure, we used silicon probes to record mFC neurons (Fig. 2a and Extended Data Fig. 2a–c) from mice ($n = 7$) performing late tasks (tasks 21–40),

a stage where we see robust evidence for task structure knowledge (Fig. 1c–e). Each recording day comprised three new tasks (X, Y and Z, presented in separate sessions) and interspersed sleep sessions. Using a generalized linear model, we found that the majority (74%) of mFC neurons were consistently tuned to the relative progress of the mouse towards a rewarded goal (Fig. 2c–g and Extended Data Fig. 2d–f). Goal-progress tuning was highly invariant across tasks and reward locations (Fig. 2f,g) and not explained by simple monotonic responses to locomotion speed, acceleration, elapsed time or physical distance (Fig. 2g and Extended Data Fig. 2d–g).

We then leveraged the hierarchical structure of our ABCD task to test whether mFC neurons are tuned to a given state in an individual ABCD task (for example, state B). We found such state-tuned neurons in abundance (Fig. 2d and Extended Data Fig. 2e). Intriguingly, the majority (81%) of these state-tuned neurons were also goal-progress-tuned (Extended Data Fig. 2e). Such state-tuned neurons comprised both spatial and non-spatial cells (Fig. 2d and Extended Data Fig. 2e,h). Moreover, neuropixels recordings in a subset of mice revealed a similar degree of goal-progress and state tuning along the dorsoventral extent of the medial wall, with a ventrally dominant gradient in place tuning (Extended Data Fig. 2i). Overall, these findings suggest that progress to goal is a key determinant of mFC neuronal firing and that a subset of these goal-progress cells are additionally tuned to a given state in a given task.

To investigate the structure of task representations at the population level, we used uniform manifold approximation and projection (UMAP). Goal-progress sequences were concatenated into a floral structure representing the four states in an individual ABCD task (Fig. 2h,i), even when excluding spatially tuned cells (Extended Data Fig. 2j,k). Goal progress interacted with state such that points on this low-dimensional manifold representing different states were closer if they represented the same goal progress (Fig. 2i). To explore the generality of this hierarchical structuring, we tested whether the same goal-progress sequences could be used to sculpt representations of tasks with a different periodicity. In ABCDE tasks, state-tuned neurons were again predominantly goal-progress-tuned (Extended Data Fig. 3a,b). Intriguingly, mFC neurons maintained their goal-progress tuning across abstract tasks (ABCD and ABCDE: Extended Data Fig. 3c–f). Moreover, the UMAP-derived neuronal manifold was again composed of goal progress sequences concatenated into a floral structure, this time with five petals representing the five states in an ABCDE task (Extended Data Fig. 3g,h). Collectively, these findings point to goal-progress sequences as simple primitives that can be used to construct hierarchical representations of complex, multi-goal task structures.

## Modular organization of mFC task structure mapping

Next, we explored whether the state tuning of mFC neurons generalized across tasks. Intriguingly, rather than invariant state tuning, we found that neurons 'remapped' their state preference across tasks (Fig. 3a,b and Extended Data Fig. 4a–c), but conserved their state tuning across different sessions of the same task (X and X'; Fig. 3b and Extended Data Fig. 4b,c). This was true even for state-tuned neurons with no discernable spatial tuning (Extended Data Fig. 4d). We next tested whether the remapping of mFC neurons across tasks was coherent across the population. We found that, whereas the whole population was not coherent, a significant proportion of neuron pairs remapped coherently across tasks (Fig. 3c and Extended Data Fig. 4e,f). This partial cross-task coherence was seen when only considering non-spatial neurons and when neurons were tuned to distant points in task space (Extended Data Fig. 4g,h). Moreover, whereas coherent pairs of neurons were slightly closer anatomically (Extended Data Fig. 4i), the proportions of generalizing and coherent neurons were similar across the dorsoventral extent of the medial wall of frontal cortex (Extended Data Fig. 4j,k).

The partial pairwise coherence between mFC neurons suggests that such neurons might be organized into task-space modules akin to the modular arrangement of grid cells mapping physical space[40]. To investigate this, we used a clustering approach. We defined a distance metric between pairs of cells which assigned low distances between cells that remapped coherently between tasks. We then applied a low-dimensional embedding on the resulting distance matrix, followed by hierarchical clustering. mFC neurons were significantly clustered (Fig. 3d,e and Extended Data Fig. 4l), indicating that they were organized into modules which remap coherently across tasks. Overall, these findings suggest that mFC neurons do not generalize their state tuning relationships as a coherent whole, but are instead organized into modules that conserve their within-module tuning relationships across tasks.

## The structured memory buffers model

The quasi-coherent remapping of mFC neurons indicates that each module is anchored to a distinct reference point. Since mFC neurons are strongly tuned to both goal progress and spatial location (Fig. 2), we reasoned that these reference points could be 'behavioural steps' defined by conjunctions of goal progress and location. Each module of neurons could be anchored to a particular behavioural step (for example, early goal progress at location 1; Fig. 4a) through a subset of 'anchor' neurons tuned to that specific goal-progress/place combination. The other 'non-anchor' neurons in the module are not tuned to the behavioural step itself, but instead each fire at a specific task-space lag from the behavioural step (Fig. 4a). Under this scheme, the apparent remapping seen when aligning activity to abstract states (ABCD; Fig. 3) occurs because the mice visit the behavioural steps in a different sequence in each task and neurons maintain their lag from their anchoring behavioural step across tasks (Fig. 4b,c).

The central implication of this change in reference frame is that each module is a memory buffer for visits to a particular behavioural step (Fig. 4a,d). These buffers are organized by task structure in two ways. First, the strong goal-progress tuning of mFC neurons means that activity on the buffer evolves as a function of the number of goals obtained. The buffers therefore track true task progress rather than other dimensions such as elapsed time or distance travelled. Second, a buffer is shaped by the structure of the task—in our case, a four or five reward ring (in ABCD and ABCDE tasks). Once a mouse completes a full trial since it last visited a given behavioural step, activity along the buffer will circle back to the anchor point (Fig. 4a,d). This means that the buffers are 'structured' by the task: they are internally organized to reflect the abstract structure of the task—a 4- or 5-reward loop in our case. Because there are buffers for every possible behavioural step, and neurons for every task-space lag from a given behavioural step within each buffer, the instantaneous mFC activity therefore always encodes the entire sequence of past behavioural steps (Fig. 4d). Moreover, because there are anchors representing behavioural steps at intermediate goal progress, not only at rewards, the population encodes the entire sequence of behavioural steps executed by the mouse (the route taken through the maze), not just the sequence of four reward locations (Extended Data Fig. 5a–c). We refer to this over-arching mechanism as the structured memory buffers (SMB) model.

The SMB model posits that activity along mFC buffers could be used to compute task-paced behavioural sequences. The activity of anchor or peri-anchor neurons could be used as a readout that biases the mouse to return to the encoded behavioural step (Fig. 4d and Extended Data Fig. 5b,c). Collectively, activity dynamics along all active SMBs would enable computing a sequence of behavioural steps to solve a given task (Fig. 4d and Extended Data Fig. 5b,c). Consequently, any new behavioural sequence sharing the same abstract structure can be encoded in a programmable way, simply by reconfiguring the order in which SMBs are activated. These network dynamics are sufficient to encode new

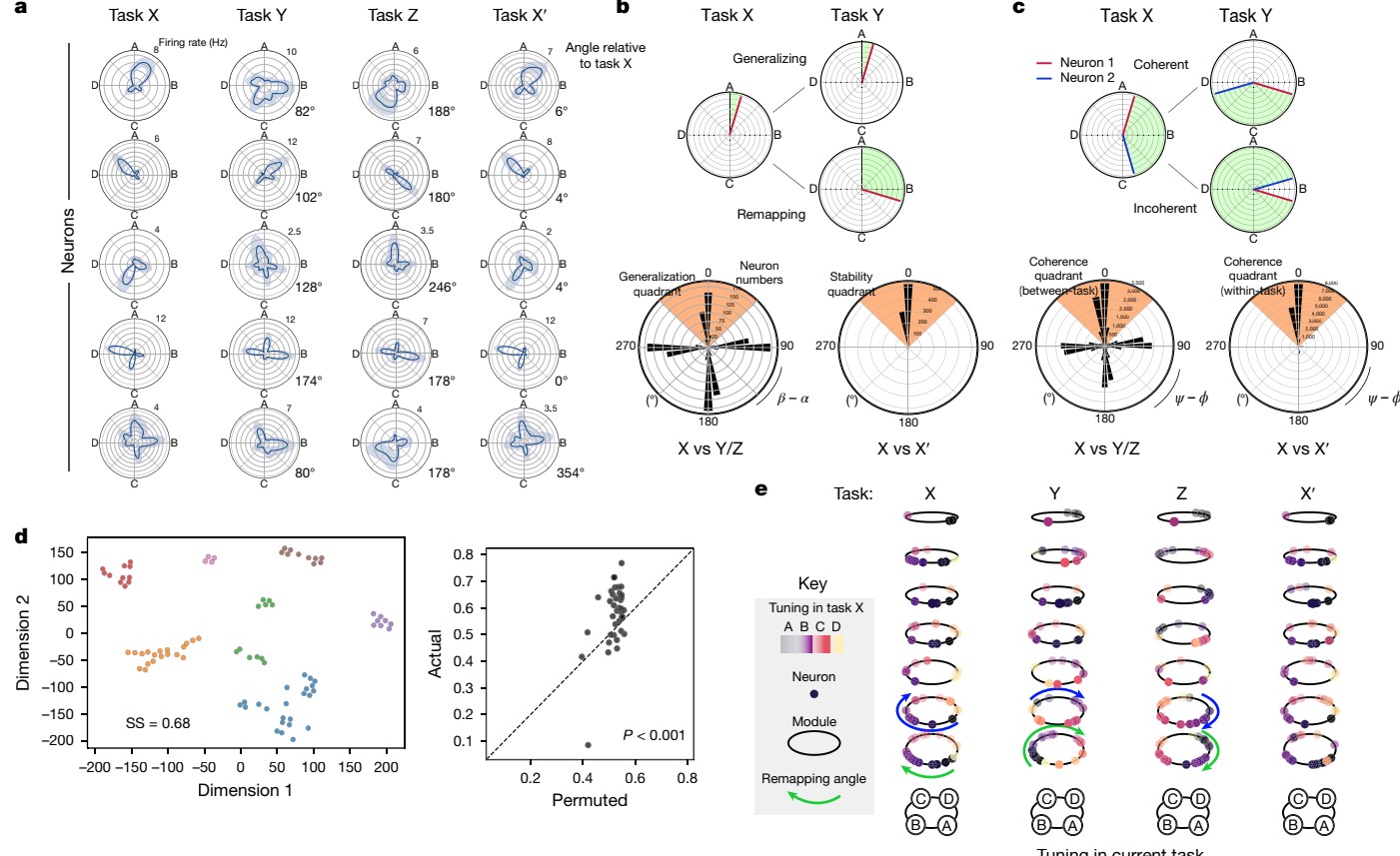

**Fig. 3 | Medial frontal neurons are organized into task-space modules.**
**a**, Simultaneously recorded neurons readily remap their state tuning but maintain their goal-progress preference across tasks. Angles (in degrees) of each cell's rotation relative to session X are shown on the right of each polar plot. **b**, Top, schematic showing quantification of tuning angle across sessions. Bottom left, polar histograms of cross-task angle show no single neuron generalization across tasks, with no clear peak at zero relative to the other cardinal directions (two proportions test against a chance level of 25%; $n = 1,594$ neurons; mean proportion of generalizing neurons across one comparison (mean of X versus Y and X versus Z) = 24%, $z = 0.91$, $P = 0.363$. Bottom right, neurons maintain their state preference across different sessions of the same task (X versus X', two proportions test against a chance level of 25%; $n = 1,160$ neurons; proportion generalizing = 78%, $z = 25.7$, $P = 0.0$). **c**, Top, schematic showing quantification of relative angle difference between pairs of neurons across sessions. Bottom left, polar histograms show that the proportion of coherent pairs of state-tuned neurons (comprising the peak at zero) is higher than chance but less than 100%, indicating that the whole population does not remap coherently. Two-proportions test against a chance level of 25%;

$n = 35,164$ pairs; mean proportion of coherent neurons across one comparison (mean of X versus Y and X versus Z) = 29.3%, $z = 13.0$, $P = 0.0$). Bottom right, as expected from **b**, the large majority of state-tuned neurons keep their relative angles across sessions of the same task (X versus X'; two proportions test against a chance level of 25%; $n = 23,674$ pairs; proportion coherent = 63%, $z = 83.5$, $P = 0.0$). **d**, Left, example from a single recording day showing the result of $t$-distributed stochastic neighbour embedding and hierarchical clustering derived from a distance matrix quantifying cross-task coherence relationships between state-tuned neurons. Each dot represents a neuron. Right, summary silhouette scores for the clustering for real data compared to permuted data that maintains the neuron's goal-progress preference and initial state distribution. Each dot is a recording day. Two-sided Wilcoxon test: $n = 38$ recording days; $W$-statistic = 126.0, $P = 2.13 \times 10^{-4}$. **e**, Modules in a single recording day. The colour code represents the tuning of the neurons in task X. The $x,y$ position defines the tuning in each task. The $z$ position corresponds to (arbitrary) cluster IDs. Neurons remap in task space while maintaining their within-cluster but not between-cluster tuning relationships across tasks.

tasks, without needing to build or bind new representations. Moreover, task-structured buffers can encode future behavioural sequences in a way that is dissociable from their mnemonic activity, for example by top-down signals that simulate alternative routes between goals and hence activate alternative SMBs for intermediate behavioural steps.

The SMB model is computationally attractive because it shows how frontal cortex functions in behavioural schema formation and sequence memory could be unified, while offering a programmable way of computing behavioural sequences. It is also empirically attractive because it explains the spatial-tuning-independent state preference (Fig. 2d and Extended Data Fig. 2e,h), remapping (Fig. 3a,b) and the modular arrangement (Fig. 3c–e) of the neurons across tasks. Crucially, the model makes a number of new empirical predictions about the tuning of single neurons, their relationship to behavioural choices and their organization at the population level. We test these predictions in turn below.

## Lagged fields in task space

The SMB model proposes that neurons consistently fire with a fixed lag in task space relative to a specific behavioural step, regardless of the intervening steps. We tested this prediction using three complementary approaches (Fig. 5 and Extended Data Fig. 6). To ensure our results are due to task-lag preference and not the powerful effect of goal-progress tuning (Fig. 2), these analyses were all done in the preferred goal-progress bin of each neuron.

The first approach involves finding the behavioural step (goal-progress/place conjunction) that aligns neuronal activity across tasks, and thus anchors that neuron. We fitted the anchor by using a sixfold cross-validation across tasks (Fig. 5a and Extended Data Fig. 6a) and found significant alignment between firing lags from the best anchor in the training and test tasks, which was crucially seen even when

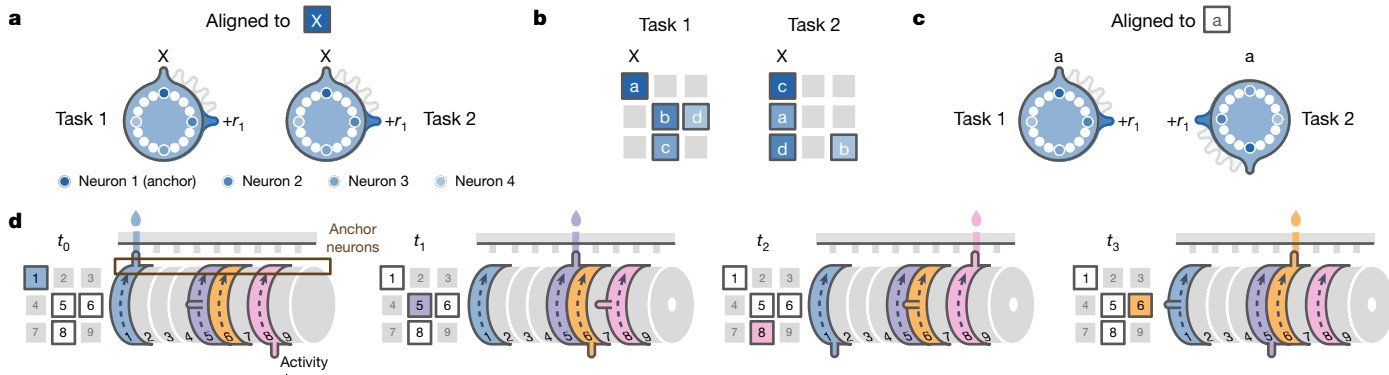

**Fig. 4 | The structured memory buffers model. a**, A hypothetical task-structured memory buffer (SMB) in the ABCD task. This ring-shaped SMB is a buffer for a specific behavioural step (goal in location 1): neurons along the SMB represent task position relative to this behavioural step. This behavioural step is therefore the anchor for this SMB. Aligning neurons by when a rewarded goal is encountered in location 1 reveals the invariant relationships between neurons across any two tasks where location 1 is rewarded. The dark blue anchor neuron (neuron 1) responds directly to goal location 1. Conversely, 3 other neurons fire at lags of 90° (one state away), 180° and 270° from the anchor. Other neurons (white circles) encode lags that are not necessarily multiples of 90°. The SMB is shaped by the task, which in the ABCD loop task means that activity circles back to the anchor point after four rewarded goals. **b**, Two example ABCD tasks that share one goal location (location 1, marked by X). Shaded regions show the spatial firing fields of each of the four neurons shown in **a**. Whereas the anchor neuron (neuron 1; dark blue) fires consistently at goal location 1 across tasks, other neurons (neurons 2–4; lighter shades of blue) fire in different locations in the two tasks. This spatial remapping is not random, but rather preserves encoding of elapsed task progress from goal in location 1. **c**, The same ring as in **a**, when aligned by the abstract task states (for example, state A), appears to rotate by 180° across tasks. This is because location 1 is rewarded in different states across tasks (state A in task 1 and state C in task 2). All neurons on the SMB remap by the same amount, not just the spatially tuned anchor neuron. **d**, A time series showing the flow of activity along 4 SMBs, each anchored to one of the 4 rewarded locations in task 1. A bump of activity is initiated in each SMB when its anchor is visited (top) and moves along the SMB paced by the progress of the mouse in task space. When it circles back close to the start, it biases the mouse to return to the behavioural step encoded by the anchor of the SMB. Multiple SMBs have active bumps at any one time, thereby simultaneously tracking a sequence of behavioural steps for an entire trial. In principle, the same computational logic can also be used even when individual neurons respond to more than one anchor and/or lag. The readout in such a scenario would involve combinatorial activity across anchor neurons from multiple SMBs. Reproduced/adapted with permission from Gil Costa.

only considering non-zero lag neurons (Fig. 5b and Extended Data Fig. 7a). To quantify the degree of alignment further, we measured the correlation between the activity of neurons during the test and training tasks, finding a right-shifted distribution, even for non-zero lag neurons (Fig. 5c and Extended Data Fig. 7b). Neurons were distributed across all lags from the anchors, with an overrepresentation of zero lag neurons, corresponding to the anchors themselves (Fig. 5d and Extended Data Fig. 7c), and anchored to all goal-progress/place combinations (Extended Data Fig. 7d). Moreover, anchoring was robust across mice and did not vary systematically across the dorsoventral extent of the mFC (Extended Data Fig. 7e,f). Intriguingly, when we gave some mice ABCDE tasks, neurons were again anchored at fixed lags from specific behavioural steps, even when only considering lags unique to this periodicity (more than 4 states from anchor; Extended Data Fig. 7g).

The second approach is effectively the reverse of the first. Instead of comparing lag tuning when aligned to particular anchor visits, it compares spatial tuning aligned to particular lags (Fig. 5e). Classically, spatial maps relate neuronal activity to the animal's present location[1]—that is, with 'zero lag' in task space. Our model proposes additional neurons consistently tuned to the animal's location at a precise point in the past or future, regardless of what the animal is currently doing. These neurons will therefore have a peak in their cross-task spatial correlation at a fixed, non-zero lag in task space (Fig. 5e and Extended Data Fig. 6b). To quantify this effect, we used a sixfold cross-validation approach, using training tasks to find the neuron's preferred lag (that is, when cross-task spatial correlation was maximal), and then correlating the spatial maps at the preferred lag between the training tasks and the test task. This train–test correlation was strongly right-shifted (Fig. 5f), even when considering only neurons with non-zero lag relative to the current location of the mouse (Fig. 5f and Extended Data Fig. 7h) and held consistently across mice (Extended Data Fig. 7i).

In the third approach, we implemented a linear regression model to predict the state tuning of neurons across tasks. For each neuron, the model describes state-tuning activity as a function of all possible behavioural steps (goal-progress/place conjunctions) and task lags from each possible behavioural step, again using a sixfold cross-validation (Fig. 5g and Extended Data Fig. 6c). Using this approach, we again observed a strong rightward (positive) shift in the correlation between predicted and actual state tuning in the test task, even when only considering activity at non-zero lags relative to all anchors (Fig. 5h and Extended Data Fig. 8a). These findings held when using a stricter state-tuning threshold, when excluding any neurons with even residual tuning to the current trajectory of the mouse or when running the regression with a nonlinear (log) link function (Extended Data Fig. 8b–d). Moreover, these results were consistent across mice (Extended Data Fig. 8e). Overall, converging lines of evidence indicate that mFC neurons track task progress from specific behavioural steps.

## Distal prediction of behavioural choices

By embedding concrete behavioural steps within task coordinates, SMBs can be used to compute a behavioural sequence for an upcoming trial. At any one point, different cells are active for different future positions at different future lags, meaning that the whole future trajectory is simultaneously available in the instantaneous firing pattern in mFC (Fig. 4d and Extended Data Fig. 5b,c). Although SMBs are activated in a particular order (configured) by a memory of previous trials, their effect on future behaviour can be dissociated from such memories—for example, through top-down modulation or noise. To test this, we need to investigate what happens if neuronal activity in SMBs is temporarily decoupled from previous choices—whether SMB-defined neuronal activity could be used to predict an upcoming choice regardless of the previous choice.

This feature of the SMB model implies spatiotemporally precise predictions—if we know each neuron's anchoring behavioural step and its lag (X) from the anchor in the past we can predict whether the animal will

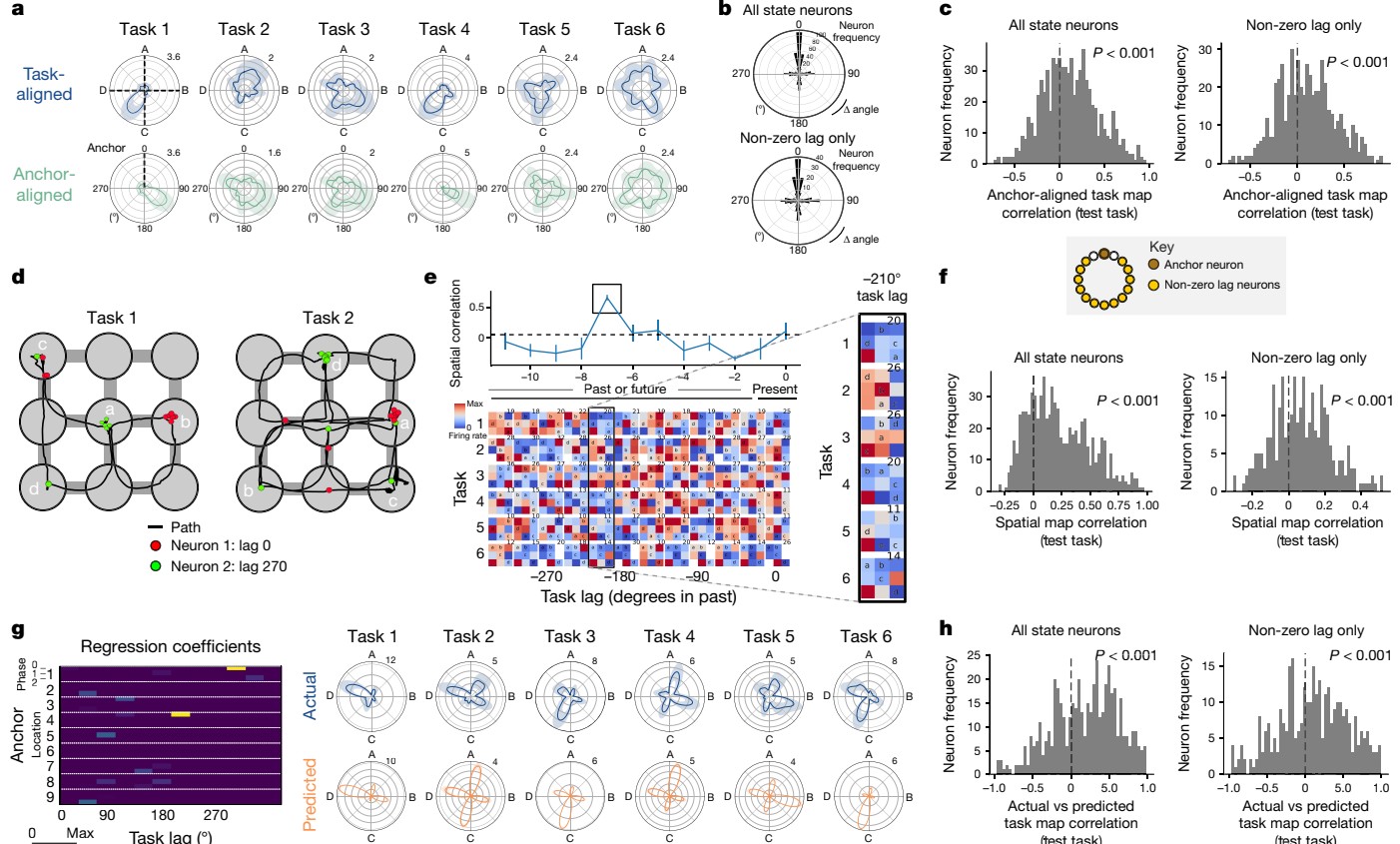

**Fig. 5 | Medial frontal neurons track task progress from specific behavioural steps. a**, Single anchor alignment analysis. Top plot (blue) shows activity aligned by the abstract states; state tuning remaps across tasks. Bottom plot (green) shows the same neuron aligned to the behavioural step (goal-progress/location conjunction) to which it is anchored (dashed vertical line). In this and the other analyses in this figure, we concatenated two recording days, giving a total of up to six new tasks per neuron. **b**, Polar histograms showing the cross-validated alignment of neurons by their preferred anchor (calculated from training tasks) in a left-out test task. The top plot is for all state-tuned neurons, whereas the bottom plot only includes non-zero lag neurons. Two-proportions test against chance (25%), proportion generalizing for all state-tuned neurons: 39%, $n = 738$ neurons, $z = 5.69$, $P = 1.24 \times 10^{-8}$; non-zero lag state-tuned neurons: 36%, $n = 545$ neurons, $z = 3.87$, $P = 1.08 \times 10^{-4}$. **c**, Histograms showing the right-shifted distribution of the mean cross-validated task map correlations between state-tuned neurons aligned to their preferred anchor (from training tasks) and the task map aligned to this same anchor from a left-out test task. This is shown for all state-tuned neurons (top) and only non-zero lag state-tuned neurons (bottom). Two-sided $t$-test against 0 for all state-tuned neurons: $n = 737$ neurons, $t$-statistic = 9.86, $P = 1.32 \times 10^{-21}$, d.f. = 736; non-zero lag state-tuned neurons: $n = 544$ neurons, $t$-statistic = 7.55, $P = 1.86 \times 10^{-13}$, d.f. = 543. **d**, Two example paths of mice during a trial in two distinct tasks with two simultaneously recorded mFC neurons. Neuron 1 is an anchor neuron tuned to reward in location 6. Neuron 2 fires with a lag of roughly 270° in task space from its anchor (reward in location 6). Spikes are jittered to ensure directly overlapping spikes are distinguishable. **e**, Lagged spatial field analysis. Bottom, each row represents a different task and each

column represents a different lag in task space, starting from the current location of the mouse (far right column) and then at successive task space lags in the past or future. Because of the circular nature of the task, past bins at lag $X$ are equivalent to future bins at lag 360 - $X$. Right, zoomed in spatial maps at this neuron's preferred lag. Top, the correlation of spatial maps across tasks at each lag. Colours are normalized per map to emphasize the spatial firing pattern, with maximum firing rates (in Hz) displayed at the top right of each map. **f**, Histograms showing the right-shifted distribution of the mean cross-validated spatial correlations between maps at the preferred lag (from training tasks) and the spatial map at this lag from a left-out test task for all state-tuned neurons (left) and only non-zero lag neurons (right). Two-sided $t$-test against 0 for all state-tuned neurons: $n = 738$ neurons, $t$-statistic = 22.6, $P = 1.45 \times 10^{-86}$, d.f. = 737; non-zero lag state-tuned neurons: $n = 285$ neurons, $t$-statistic = 7.48, $P = 9.07 \times 10^{-13}$, d.f. = 284. **g**, Left, regression analysis reveals neurons with lagged fields in task space from a given anchor (goal-progress/place conjunction). Right, this enables prediction of state tuning and its remapping across tasks for each neuron. **h**, Histograms showing the right-shifted distribution of mean cross-validated correlation values between model-predicted (from training tasks) and actual (from a left-out test task) activity. This correlation is shown for all state-tuned neurons (left) and only state-tuned neurons with non-zero-lag firing from their anchors (right). Two-sided $t$-test against 0 all state-tuned neurons (with non-zero beta coefficients): $n = 489$ neurons, $t$-statistic = 9.3, $P = 5.3 \times 10^{-19}$, d.f. = 488; non-zero lag state-tuned neurons: $n = 329$ neurons, $t$-statistic = 3.9, $P = 1.08 \times 10^{-4}$, d.f. = 328. Data are mean ± s.e.m.

return to this behavioural step at lag (360° − $X$) in the future regardless of the intervening steps. Crucially, this future prediction generalizes across tasks: the same neuron can predict returns to the anchoring behavioural step at the same future lag regardless of the intervening steps (Fig. 6a and Extended Data Fig. 9a). We refer to the neuron's lag ($X$) from its anchor as the neuron's 'bump time', the time where the activity bump would pass through this neuron. To test this prediction, we related the trial-by-trial bump-time activity of non-anchor neurons

to subsequent visits to their anchoring behavioural step, which on average is up to tens of seconds in the future, while controlling for previous choices. Intriguingly, we found that the bump-time activity of neurons was higher before trials where mice visited the neurons' anchoring behavioural step (Extended Data Fig. 9b). To investigate this further, we ran a logistic regression on the trial by trial activity of neurons at non-zero lag from their anchor to predict upcoming behavioural choices by the animal. To control for the autocorrelation in behaviour

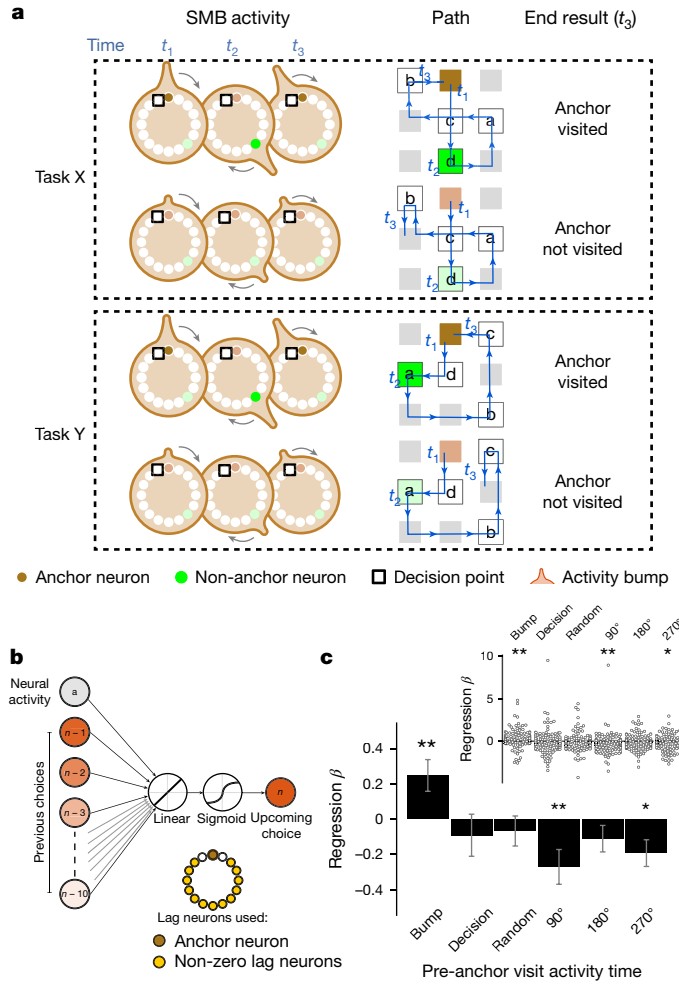

**a**

SMB activity | Path | End result ($t_3$)

Task X
- Anchor visited
- Anchor not visited

Task Y
- Anchor visited
- Anchor not visited

● Anchor neuron  ● Non-anchor neuron  □ Decision point  ⌂ Activity bump

**b**

Neural activity

Previous choices

Linear — Sigmoid — Upcoming choice

Lag neurons used:
● Anchor neuron
● Non-zero lag neurons

**c**

Pre-anchor visit activity time

Regression β

Bump, Decision, Random, 90°, 180°, 270°

**Fig. 6 | Medial frontal activity predicts distal behavioural choices.**
**a**, Schematic showing distal prediction of choices from memory buffers. The SMB model enables us to predict the choices made by the mouse: (1) at a precise lag in the future; and (2) in a way that generalizes across tasks. The size and the timing of the activity bump determines how likely and when (respectively) the animal will visit the SMB's anchor in the next trial. This future prediction should generalize across tasks and thus be independent of where the mouse is at a given time. In this example we show an SMB with an anchor at location 2 (top middle location shaded in brown) at intermediate goal progress (halfway between goals). The brown neuron is the anchor, whereas the green neuron fires at a lag of 1.5 states (135°) from the anchor. Four rows correspond to four possible scenarios across two different tasks, illustrating the key features of the SMB model's predictions. The activity of the green neuron at precisely 1.5 states since the first anchor visit ($t_2$; the bump time of the green neuron) can be used to predict what will happen at $t_3$ (2.5 states forward from $t_2$). A larger activity bump (indicated by the height of the bump along the SMB) increases the likelihood that the animal will return to the anchor at $t_3$. This larger bump is indexed by higher activity of the green neuron at $t_2$ (indicated by a darker green shade). In task Y, the green neuron fires at a different location in order to keep its lag from its anchor. Nevertheless, its activity at $t_2$ can be used in the same way to predict whether the mouse will visit the anchor at $t_3$ (2.5 states in the future). To test this latter point, we only consider non-zero lag cells, which fire in different locations across tasks, in all of the analyses below. Reproduced/adapted with permission from Gil Costa. **b**, Design of logistic regression to assess the effect of each neuron's activity on future visits to its anchor. To control for autocorrelation in behavioural choices, previous choices as far back as ten trials in the past are added as co-regressors. Separate regressions are done for activity at different times: bump time, random times, decision time (the time where the mouse was one spatial step away, and one goal-progress bin away from the anchor of a given neuron) and times shifted by 90° intervals relative to the neuron's bump time. **c**, Bottom, regression coefficients are significantly positive for the bump time but not for any of the other control times. Two-sided t-tests against 0 for bump time: $n = 131$ tasks, t-statistic = 2.75, $P = 0.007$, d.f. = 130; decision time: $n = 131$, t-statistic = −0.77, $P = 0.446$, d.f. = 130; random time: $n = 131$, t-statistic = −0.79, $P = 0.433$, d.f. = 130; 90° shifted time: $n = 131$, t-statistic = −2.74, $P = 0.007$, d.f. = 130; 180° shifted time: $n = 131$, t-statistic = −1.47, $P = 0.143$, d.f. = 130; 270° shifted time: $n = 131$, t-statistic = −2.54, $P = 0.012$, d.f. = 130. Top, swarm plots showing distribution of regression coefficient values across groups. Data are mean ± s.e.m.

of mice, we regressed out previous choices going all the way back to ten trials in the past (Fig. 6b and Extended Data Fig. 9c). We found that mFC neurons significantly predicted future choices only when taking their activity at their bump time (Fig. 6c). This was seen even when only considering neurons with distal lags, more than one state away from the anchor (Extended Data Fig. 9d). This prediction also held when only considering choices to intermediate, non-rewarded locations (Extended Data Fig. 9e–h). Intriguingly, SMBs predicted future choices in the same way across a task with a different periodicity (the ABCDE task; Extended Data Fig. 9i). Overall these results show a generalizable and distal prediction of behavioural choices across tasks with different periodicities, consistent with the use of SMBs to compute task-paced behavioural sequences.

## Internally organized buffers

To compute task-paced behavioural sequences, the SMB model proposes that memory buffers are shaped by the structure of the task. Our findings so far support this task structuring, by showing that mFC modules invariantly track task progress from different behavioural steps, and compute a sequential trajectory that is paced by the periodicity of the task. Here we tested whether neurons maintain an internally organized, task-shaped structure in the absence of any externally structured task input. We therefore investigated whether pairwise coactivity during offline rest or sleep reflects a ring-like neuronal state space (Fig. 7a). We regressed the circular distance between pairs of neurons that share the same anchor, and thus belong to the same module, against their cross-correlation during sleep (Fig. 7b). Regression coefficients were significantly negative for circular distance indicating

that neurons closer on a circular state space are more coactive during sleep (Extended Data Fig. 10a). This held while controlling for forward distance, pairwise spatial map similarity and goal-progress-tuning proximity between neurons. Crucially, we found that pairs of neurons anchored to the same behavioural step showed significantly more negative regression coefficient values than neuron pairs anchored to different behavioural steps (Fig. 7c). Thus, the modular organization of SMBs is maintained offline. We corroborated the regression analyses using a simpler cross-correlation analysis, which showed that pairs of co-anchored neurons close in circular distance but far in forward distance were more coactive during sleep (Fig. 7d,e). Moreover, we observed no differences in the regression coefficients for pairwise forward distance, circular distance or spatial similarity against sleep coactivity when comparing pre-task and post-task sleep or when comparing different sleep epochs (Extended Data Fig. 10b,c). Together, these findings provide offline evidence that SMBs in the mFC are internally organized by the structure of the task.

## Discussion

Our findings identify a cellular algorithm for mapping behavioural structure. We found that mice can learn abstract structures that organize multiple goals and use this knowledge to rapidly compose complex behavioural sequences. Within a task, goal-progress sequences in the mFC were hierarchically elaborated to form representations of a complex, multi-goal task structure. Across tasks, mFC neurons were

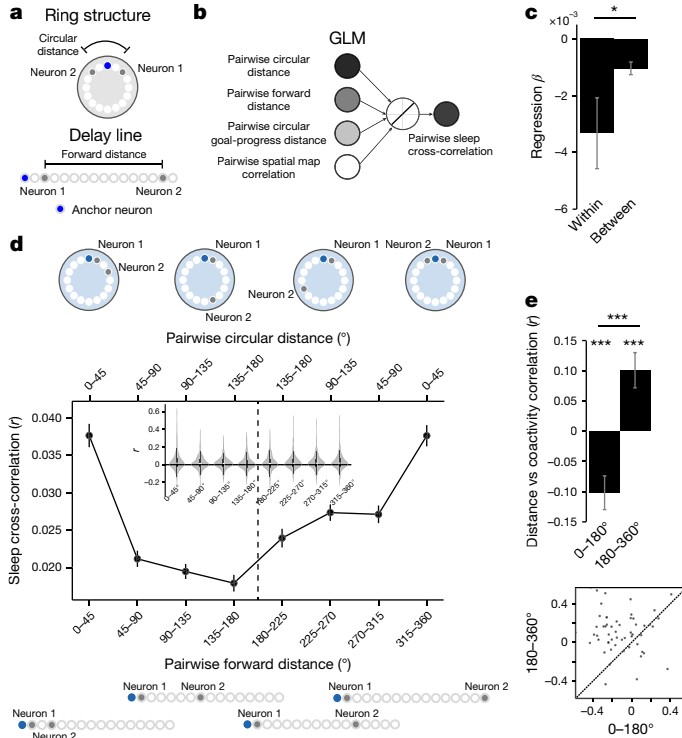

**Fig. 7 | Offline activity of mFC neurons is internally organized by the task structure. a**, Schematic showing potential neuronal state spaces. Neuronal state space relationships are best described by circular distance for a ring and forward distance for a delay line. Reproduced/adapted with permission from Gil Costa. **b**, Schematic showing the linear regression model relating pairwise circular distance and forward distance with coactivity during sleep while regressing out each other, as well as pairwise goal-progress-tuning distance and spatial map similarity. **c**, Regression coefficient values for circular distance against sleep cross-correlation were more negative for pairs sharing the same anchor (within) compared with pairs across anchors (between). One-sided unpaired *t*-test (Welch's *t*-test) for all sleep: *n* = 430 pairs (within), 13,932 pairs (between), *t* = −1.80, *P* = 0.036, d.f. = 14,360. **d**, A plot of sleep cross-correlations (between pairs sharing the same anchor) against forward and circular distance (bottom and top *x* axes, respectively). Schematics at the top and bottom show example pairs that would fall into each category and their circular and forward distances. The v-shaped relationship between forward distance and cross-correlation indicates a circular state space. Inset, kernel density estimate plots showing distribution of cross-correlation values across bins. Reproduced/adapted with permission from Gil Costa. **e**, Bar graph (top) and scatter plot (bottom) of correlation coefficients between pairwise forward distance and pairwise sleep cross-correlations are positive for pairs of neurons with 180–360° forward distance and negative for pairs with a forward distance of 0–180°. Two-sided *t*-tests relative to 0, 0–180°: *n* = 59 sleep sessions, *t*-statistic = −5.16, *P* = 1.07 × 10⁻⁴, d.f. = 58; 180–360°: *n* = 59 sleep sessions, *t*-statistic = 4.16, *P* = 1.08 × 10⁻⁴, d.f. = 58. Two-sided paired *t*-test: *n* = 59 sleep sessions, *t*-statistic = −5.55, *P* = 7.35 × 10⁻⁷, d.f. = 58. Data are mean ± s.e.m. In **c**, this error is analytically derived for each regression; thus there are no individual points to report in this panel.

organized into buffers shaped by the task structure (SMBs). Each SMB comprised neurons that tracked task progress from a specific behavioural step, firing with fixed task-space lags that generalized across tasks. This allowed encoding long sequences of behaviours that were parsed by the task structure into the network dynamics. The resulting algorithm relates mFC roles in schema formation and sequence memory, by internally organizing mnemonic activity according to an abstract structure shared by many tasks.

The use of goal-progress tuning as a primitive ensures that neuronal dynamics evolve as a function of true progress in a multi-goal task, rather than other physical dimensions such as elapsed time or distance. This is in contrast to 'world model' representations, such as grid cells, which use the velocity of the animal in physical space to track its spatial position[41]. Goal-progress-like codes have been found in a range of other mFC studies across distinct tasks[14,25–27,29,30] suggesting they are a robust feature of mFC activity. Here we show that such sequences can be used to build a schema encoding complex (multi-goal) task structures. Such a hierarchical organization suggests that goal-progress sequences could provide an inductive bias for building task coordinates. Indeed we show that, when mice faced a new abstract structure (ABCDE), the same goal-progress sequences found in the ABCD tasks were resculpted to represent the new task periodicity. Notably, whereas our findings reveal goal-progress sequences as primitive building blocks of abstract task structure, they do not address higher-order composition (for example, composing an ABHI task from ABCD and FGHI tasks). Emerging theory and findings are beginning to reveal mechanisms for such higher-order composition[42,43] and will eventually relate them to the lower-order goal-progress primitives we report here.

Our findings reveal that the mFC uses SMBs to track the animal's position in task-space in a sequence comprising multiple goals. SMBs bring together features of both sequence working memory and schema representations under one roof, helping unify two key functions of mFC. The parallel with working memory representations comes from the fact that each SMB tracks a memory of a given behavioural step, and thus at any one point the population simultaneously represents memories of multiple stimuli at different lags in the past. The parallel with schema representations comes from considering that SMBs are shaped by the abstract task structure and generalize to new behavioural sequences. Indeed, we find that the coactivity between neurons in a given SMB is internally organized by the task structure, persisting during offline periods when no externally structured task input is available. This is in line with internal organization being a key organizational feature of cognitive representations in the mFC[26,29] and beyond[44–48]. Moreover, although our data pertain directly to the periodic case, the use of buffers whose dynamics are shaped by the task structure has the potential to generalize to other tasks. In classical sequence working memory tasks, a single SMB would span the neurons for a given observation at different ordinal positions: it is a task-structured pathway that links a single observation across the different subspaces observed empirically in the mPFC of animals holding an instructed sequence in memory (for example, in ref. 49). Indeed, this feature of the SMB model has now been formalized in a normative theory that explains a wide range of neuronal data from the mPFC across distinct tasks that are not necessarily periodic[50].

By embedding concrete behavioural steps into task coordinates, SMBs enable encoding of new behavioural sequences in the dynamics of the mFC network. This offers a programmable solution to mapping new sequences that share the same abstract structure. We found that this prediction of future behavioural steps by SMBs is dissociable from their mnemonic tuning. Thus, whereas SMBs are configured by experience in periodic tasks (for example, the ABCD task), they could also be configured by instruction or simulation to compute a future behavioural sequence that solves a defined problem. This distal prediction is unlike hippocampal splitter cells[51,52] and other latent representations[53] that separate the same location when encountered as part of different stereotypical behavioural sequences. This is also unlike the encoding of future behavioural choices using sequential activity, such as that found in Hippocampal neurons during theta sweeps[54] and awake replay[55], where the same cell encodes the same location now and in the future. Instead, at any one point, different mFC cells are active for different future locations at different future lags: the whole future trajectory is simultaneously available in the instantaneous firing pattern in mFC. The implication for the downstream read out of this activity is that any area using this mFC representation can now compute with the entire plan simultaneously. Moreover, when the plan needs to change to solve a new task sharing the same abstract structure,

SMBs can be rapidly reconfigured without needing new representations. This rapid cross-task generalization may be key to the necessity of mFC for rapid task switching[23,24]. Future studies could test the precise causal role of SMBs in such task switching using optical methods for tagging and manipulating individual, functionally defined cells. Optogenetic manipulations of defined SMBs should enable precisely biasing animals to make specific choices at specific times in the task, tens of seconds after the manipulation has ceased. Moreover, the SMB model suggests that it is possible to encode a complex policy in the dynamics of recurrent neural networks. Future in silico work, especially using deep reinforcement learning models[56], can take inspiration from this to formalize the role of mFC neuronal dynamics in complex planning.

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

## Methods

### Mice

Experiments used adult male C57BL/6 J mice ($n = 13$; Charles River Laboratories). Mice were housed in a dedicated housing facility on a 12 h light/dark cycle (07:00–19:00) at 20–24 °C and humidity of 55 ± 10%. Mouse experiments were run with four cohorts: two cohorts of four, one cohort of three and one cohort of two, and preselected on the basis of criteria outlined in 'Behavioural training' below. No statistical methods were used to pre-determine sample sizes but our sample sizes are similar to those reported in previous publications (for example, in ref. 12). Mice were housed with their littermates up until the start of the experiment with free access to water. Food was available ad libitum throughout the experiments, and water was available ad libitum before the experiments. Mice were 2–5 months old at the time of testing. Experiments were carried out in accordance with Oxford University animal use guidelines and performed under UK Home Office Project Licence P6F11BC25.

### Behavioural training

**The ABCD task.** A set of tasks where subjects must find a sequence of 4 reward locations (termed a, b, c and d) in a $3 \times 3$ grid maze that repeat in a loop. The reward at each location only became available after the previous goal was visited, so the goals had to be visited in sequence to obtain rewards. Once the animal receives reward a the next available reward is in location b (and so on), then once the animal receives reward in location d then reward a becomes available again, creating a repeating loop. An extension of this is the ABCDE task, in which five rewards are arranged in a loop. A brief tone was played upon reward delivery in location a (start of state A), marking the beginning of a loop on every trial. This created an equivalence across tasks for the a location, beyond a single trial memory of the first rewarded location.

**Location.** When the animal is in the physical maze, it could be at a node (one of the circular platforms where reward could be delivered, coded 1–9 as shown in Fig. 1a), or it could be at an edge, which is a bridge between nodes. The nine maze locations were connected as shown in the top right photograph and adjacent schematic in Fig. 1a, with connections only along the cardinal directions. In our particular maze, edges were only arranged along the cardinal direction, so for example, location 1 was only connected to locations 2 and 4 (Fig. 1a, top right).

**Task.** An example of the ABCD loop with a particular sequence of reward locations (for example, 1–9–5–7; reward a is in location 1; reward b in location 9, and so on; Fig. 1a).

**Session.** An uninterrupted block of trials of the same task. We used 20-min sessions. Note that subjects could be exposed to two or more sessions of the same task on a given day. Animals were allowed to complete as many trials as they could in those 20 min. Animals were removed from the maze at the end of a session and either placed back in their home-cage or into a separate enclosed rest/sleep box.

**Trial.** A single run through an entire ABCD loop for a particular task, starting with reward in location a and ending in the next time the animal gets reward in location a again (for example, trial 12 of a task with the following configuration: 1–9–5–7 starts with the 12th time the animal gets reward in location 1 and ends with the 13th time animal gets reward in location 1).

**State.** The time period between an animal receiving reward in a particular location and receiving reward in the next rewarded location. State A starts when animal receives reward a and ends when animal receives reward b; state D starts when animal gets reward d and ends when animal gets reward a, and so on.

**Transition.** A generalized definition of state. For example, progressing from a to b is a transition, and progression from c to d is also a transition.

**Goal progress.** How much progress an animal has made between rewarded locations as a percentage of the time taken between them. Unless otherwise stated, we operationally divide this into early, intermediate and late goal progress, which correspond to one-third increments of the time taken to get from one reward location to the next. For example, if the animal takes 9 s between one reward and another, then early goal progress spans the first 3 s, intermediate goal progress spans the next 3 s and late goal progress spans the last 3 s. This would scale with the length of time it takes for the animal to complete a single state: for example, if it takes 15 s between two rewards, each of the goal-progress bins would be 5 s long. In the ABCD task, goal progress repeats four times because there are four rewards (so there will be an early goal progress for reward a, and early goal progress for reward b, and so on).

**Choice.** We use this to refer to one-step choices in the maze. At every node in the maze the animal has a choice of two or more immediately adjacent nodes to visit next. For example, when in node 1 the animal could choose to move to node 2 or node 4 (Fig. 1a).

**Apparatus.** Maze dimensions: $45 \times 45 \times 30$ cm (length × width × height). Outer dimensions of maze: $66 \times 66 \times 125$ cm. Outer maze walls: instead of solid walls we used electromagnetic field shielding curtains (Electro Smog Shielding (product number: 4260103664431)) that were custom cut to cover the entire outside of the maze. Node dimensions: $11 \times 11$ cm. Bridge length: 7 cm. Water reservoir height (from bottom of syringe to floor): 80–85 cm. Reservoir filled to 30 ml. Details of the design and material of all maze components are available at https://github.com/pyControl/hardware/tree/master/GridMaze.

**Pre-selection.** A total of 13 mice across 4 cohorts were used for experiments. For each cohort, 3–4 mice were pre-selected for the experiment from 10–20 mice based on the following criteria:
(1) Weight above 22 g.
(2) No visible signs of stress upon first exposure to the maze in the absence of rewards. Stress was evidenced by thigmotaxis or defecation in a 20 min exploration session with no rewards delivered.
(3) On a partially connected version of the maze with only 5–7 accessible nodes out of 9, animals learned that poking in wells delivered water reward and that after gaining reward they must go to another node. The number of nodes was fixed for a given session. Animals that obtained 40 or more rewards per 20 min session were taken forward to pretraining. The available nodes were connected pseudo-randomly (Extended Data Fig. 1a), such that animals could access all available nodes but there were always "dead-ends". The identity, number and connectivity of the available nodes was changed for every new 20 min session, to minimize any behavioural biases induced by the exact spatial structure of the maze.
(4) Final selection: for a given cohort, if more than four animals satisfied these criteria, animals that explored the maze with the highest entropy were selected (see 'Behavioural scoring' for entropy calculation).

**Habituation and pre-training.** After at least one week of post-surgery recovery (see 'Surgeries'), animals were placed on water restriction. Animals were maintained at a weight of 88–92% of their baseline weight, which was calculated before water restriction but after implantation and recovery from the surgery (see 'Surgeries'). This is to ensure that they remained motivated to collect water rewards during the task but not overly so, as excessive motivation is known to negatively affect model-based performance[58]. Animals were habituated to being tethered to the electrophysiological recording wire while moving on the

maze, as well as in sleep boxes for at least three days prior to the start of the experiment. During this period, animals were reintroduced to a partially connected maze (only 5–7 out of the 9 nodes available, and not all connected) while tethered to the electrophysiology wire, where water reward was delivered if the animal poked its nose at any node. Reward drops were only available once the animal poked its nose at the node. At this stage, there was no explicit task structure, except that once reward was obtained at one node, animals had to visit a different node to gain further reward (that is, exactly as in step 3 of pre-selection criteria above but while implanted and tethered). Thus, animals (re) learned that poking in wells delivered reward and that after gaining reward they must go to another node. Animals were transitioned to the task when they obtained >40 rewards in a 20-min session. Note that 3 animals (from cohort 1) were not implanted and so for these the rewarded sessions for pre-selection and pre-training were one and the same (see 'Numbers' for more details on mouse cohorts). The volume of water delivered during pretraining or preselection was higher than training, typically 10–15 µl.

Overall, animals received as few as 2 × 20 min sessions in this partially connected maze where reward is presented in all available nodes and a maximum of 8 × 20 min sessions in total across pre-selection and pre-training. All of these sessions had different configurations connecting the available nodes.

**Training.** Animals navigated an automated 3 × 3 grid maze in search of rewards (Fig. 1a), controlled using pyControl[59]. Water rewards (3–4 µl) were presented sequentially at 4 different locations. Animals had to poke in a given reward node, breaking an infrared beam that triggered the release of the reward drop in this well. After reward a was delivered, reward was obtainable from location b, but only after the animal poked in the new location. Once the animal received reward in locations a, then b, then c and then d, reward a becomes available again, thus creating a repeating loop. Animals have 20 min to collect as many rewards as possible and no time outs are given if they make any mistakes. They are then allowed at least a 20-min break away from the maze (in the absence of any water) before starting a new session. For each session, animals were randomly entered from a different side of the square maze, using custom-made electromagnetic field shielding curtains (Electro Smog Shielding (product number: 4260103664431)). This was to ensure that all sides of the maze are equivalent in terms of being entry and exit points from the maze, thereby minimizing any place preference/aversion and minimizing the use of different sides as orienting cues. One cue card was placed high up (at least 50 cm vertically from the maze) on one corner of the maze to serve as an orienting cue. No cues were visible at head level.

While all locations were rewarded identically, a brief pure tone (2 s at 5 kHz) was delivered when the animal consumed reward a. This ensured that task states were comparable across different task sequences. White noise was present throughout the session to avoid distraction from outside noises.

Task configurations (the sequence of reward locations) were selected pseudo-randomly for each mouse, while satisfying the following criteria:

(1) The distance between rewarded locations in physical space (number of steps between rewarded locations) and task space (number of task states between rewarded locations) were orthogonal for each mouse (Extended Data Fig. 1b).
(2) The task cannot be solved (75% performance or more) by moving in a clockwise or anti-clockwise circle around the maze.
(3) The first two tasks have location 5 (the middle location) rewarded— this is to ensure the first tasks the animals are exposed to cannot be completed by circling around the outside $n$ times to collect all rewards. (Note that all late tasks and those used in electrophysiological recordings are not affected by this criterion).
(4) Consecutive tasks do not share a transition (that is, two or more consecutively rewarded locations)

(5) Chance levels are the same for all task transitions (ab, bc, cd and da), and control transitions ca and ba transitions—whether determined analytically by assuming animals diffuse around the maze or empirically by using animal-specific maze-transition statistics from an exploration session before any rewards were delivered on the maze (see 'Behavioural scoring' below for chance level calculations).

For the first 10 tasks, animals were moved to a new task when their performance reached 70% (that is, took one of the shortest spatial paths between rewards for at least 70% of all transitions) on 10 or more consecutive trials or if they plateaued in performance for 200 or more trials. For these first ten tasks, animals were given at most four sessions per day, either all of the same task or, when animals reached criteria, two sessions of the old task and two sessions of the new task. From task 11 onwards, animals learned 3 new tasks a day with the first task being repeated again at the end of each day giving a total of 4 sessions with the pattern X–Y–Z–X'.

To test the behavioural effect of the tone on performance, for cohort 3, 3 mice were additionally exposed to ABCD tasks where the tone was randomly omitted from the start of state A on 50% of trials (the tone was always sounded on the very first A of the session).

To further test the generality of the neuronal representations we uncover here, the fourth cohort (2 mice) experienced ABCDE tasks. This was done after the mice completed all 40 ABCD tasks. On the first day, the mice were exposed to 2 new ABCD tasks, followed by a series of different ABCDE tasks (11 for one mouse and 13 for another) spanning 4 recording days. As before, animals experienced three to four sessions each day.

We tested whether tasks for a given mouse are correlated. For these cross-task correlations, we not only compared sequences with each other, but also with shifted versions of each other to ensure we exhaustively capture any similarities in task sequences. Correlations between tasks with all possible shifts relative to each other were made for a given mouse and then an average correlation value was reported in Extended Data Fig. 1d. Furthermore, the tasks are selected such that no two tasks experienced by the same animal are rotations of each other in physical space.

The sequences of tasks that the animals experienced throughout the experiment were randomized and counterbalanced. Only one experimental group (silicon probe implant in the frontal cortex) was used in this study and so blinding was not necessary.

## Surgeries

Subjects were taken off water restriction 48 h before surgery and then anaesthetized with isoflurane (3% induction, 0.5–1% maintenance), treated with buprenorphine (0.1 mg kg⁻¹) and meloxicam (5 mg kg⁻¹) and placed in a stereotactic frame. A silicon probe mounted on a microdrive (Ronal Tool) and encased in a custom-made recoverable enclosure (ProtoLabs) was implanted into mFC (anterior–posterior (AP): 2.00, medial–lateral (ML): −0.4, dorsal–ventral (DV): −1.0), and a ground screw was implanted above the cerebellum. AP and ML coordinates are relative to bregma, whereas DV coordinates are relative to the brain surface. Mice were given additional doses of meloxicam each day for 3 days after surgery and were monitored carefully for 7 days after surgery and then placed back on water restriction 24 h before pretraining. At the end of the experiment, animals were perfused; and the brains were fixed-sliced and imaged to identify probe locations (Extended Data Fig. 2a). We used the software HERBS[57] (version 0.2.8) to localize the probes to anatomical locations on the mouse brain based on the histology images (Fig. 2a and Extended Data Fig. 2b). For the neuropixels recordings, this was done directly by finding the entire probe track. For Cambridge neurotech probes, we found the top entry point of each of the 6 shanks and the bottom was inferred based on the final DV position reached by the probe post-surgery and nanodrive-based lowering. This was marked for each mouse in Fig. 2a and Extended Data

Fig. 2b and matched to a standardized template to deduce anatomical regions corresponding to the probe positions.

Using this method, we found that 90.7% of all recorded neurons were histologically localized in mFC regions: 68.3% in Prelimbic cortex, 11.3% in anterior cingulate cortex, 6.1% in infralimbic cortex and 5.0% in M2. Of the remaining 9.3%, 4.8% could not be localized to a specific peri-mFC region within the atlas coordinates as they were erroneously localized to peri-mFC white matter areas, likely due to variations between actual region boundaries and those derived from the standard template provided by the HERBs software, 2.2% were found in the dorsal peduncular nucleus, 1.1% in the striatum, 0.6% in the medial orbital cortex, 0.3% in the lateral septal nucleus and 0.3% in olfactory cortex. We used all recorded neurons in the analyses throughout the manuscript but indicate where these pertain to different mFC regions in Extended Data Figs. 2i, 4j,k and 7f.

### Electrophysiology, spike sorting and behavioural tracking

Cambridge NeuroTech F-series 64 silicon channel probes (6 shanks spanning 1 mm arranged front-to-back along the anterior–posterior axis) were used for 3 of the 4 cohorts (5 mice). To record from the mFC, we lowered the probe ~100 μm during the pre-habituation period to reach a final DV position of between −1.3 and −1.5 mm below the brain surface (that is, between −2.05 and −2.25 mm from bregma). This places most channels in the prelimbic cortex (http://labs.gaidi.ca/mouse-brain-atlas/) (Fig. 2a and Extended Data Fig. 2b). For the fourth cohort (2 mice) we used neuropixels 1.0 probes that were fixed at a DV position of 3.8–4 mm from the brain surface, AP 2.0 mm from bregma and ML −0.4 from bregma. This gave us the ability to record from more regions along the entire medial wall of mFC, including regions such as secondary motor cortex (M2), dorsal and ventral anterior cingulate (ACC), and infralimbic cortex, as well as the prelimbic cortex (Fig. 2a and Extended Data Fig. 2b). Neural activity was acquired at 30 kHz with a 32-channel Intan RHD 2132 amplifier board (Intan Technologies) connected to an OpenEphys acquisition board. Behavioural, video and electrophysiological data were synchronized using sync pulses output from the pyControl system. Recordings were spike sorted using Kilosort[60], versions 2.5 and 3, and manually curated using phy (https://github.com/kwikteam/phy). Clusters were classified as single units and retained for further analysis if they had a characteristic waveform shape, showed a clear refractory period in their autocorrelation and were stable over time.

To increase the number of tasks assessed for each neuron, we concatenated pairs of days to obtain six tasks. For concatenated double days, we tracked neurons by concatenating all binary files from sessions recorded across two days. A single concatenated binary file with data from two days was run through the standard kilosort pipeline to automatically extract and sort spikes. We manually curate the output of this file based on standard criteria:

(1) Contamination of a refractory period (2 ms) as determined by a spike autocorrelogram - to be at most 10% of the baseline bin, which is defined as the number of spikes in the 25-ms bin (the maximum value for the autocorrelation plots used for curation in kilosort).

(2) Neurons where the firing rate in 3 or more sessions drops below 20% of the session with the peak firing rate are discarded.

These criteria result in a total percentage of spikes in 1 ms and 2 ms refractory periods of: 0.033% ± 0.003% and 0.076% ± 0.004% of spikes on average respectively. We also note that 99.2% of neurons have a maximum dropoff of spike counts at 2 ms or more.

Dropout rate: by comparing the units designated 'good' by kilosort before curation with the post-curation yield, we find a dropout rate of 51.4% for the concatenated double days. This is in comparison to a dropout rate of 29.7% for single days.

We find that concatenation overwhelmingly succeeds in capturing the same neuron across days. For this we take advantage of the highly conserved goal-progress tuning that is characteristic of mFC neurons (Fig. 2). We assessed 'goal-progress correlation' between different tasks that are taken from the same day and then repeated this for the same neuron for pairs of tasks taken from different days. This allowed us to index the extent to which basic tuning of cells is conserved across days (both in ABCD and ABCDE days). This shows:

(1) An exceptionally tight relationship between each neuron's within and across-day goal-progress correlation—that is, goal-progress correlation between tasks in the same day is itself highly correlated with goal-progress correlation between tasks across days. Pearson correlation $n = 1,540$ neurons, $r = 0.88$, $P = 0.0$.

(2) Another way of looking at this: goal-progress correlation values are indistinguishable within and across days—that is, for a given neuron, goal-progress tuning is equally likely to be conserved across days as it is within the same day. (Within-day correlation: $0.63 ± 0.01$, across-day correlation: $0.62 ± 0.01$ Wilcoxon test: $n = 1,549$, $W$-statistic $= 567,550$, $P = 0.14$, d.f. $= 1,540$)

(3) Almost all neurons that are significantly goal-progress-tuned within a day also maintain their goal-progress tuning across days (95.4%; two proportions test: $n = 1,249$ neurons, $z = 45.2$, $P = 0.0$). Significance was calculated by comparing goal-progress correlation to the 95th percentile of circularly shifted permutations, individually for each neuron.

We performed tracking of the mice in the video data using DeepLabCut[61] (version 2.0), a Python package for marker-less pose estimation based in the TensorFlow machine learning library. Positions of the back of a mouse's head in $xy$ pixel coordinates were converted to region of interest information (which maze node or edge the animal is in for each frame) using a set of binary masks defined in ImageJ that partition the frame into its sub components.

### Data analysis

All data were analysed using Python (3) code. This used custom-made code but made use of libraries such as numpy (1.22.0), scipy (1.10.1), matplotlib (3.7.3), sciKit learn (1.3.2), pandas (2.0.3) and seaborn (0.13.2).

### Behavioural scoring

Performance was assessed by quantifying the percentage of transitions (for example, a to b) where animals took one of the shortest available routes (for example, Extended Data Fig. 1j), or percentage of entire trials where animals took the shortest possible path across all transitions (Extended Data Fig. 1k). We also quantified the path length taken between rewards and divided this by the shortest length to give the relative path distance covered per trial (for example, Extended Data Fig. 1i).

When using percentage of shortest path transitions as a criterion, chance levels were calculated either analytically or empirically. Analytical chance levels were calculated by assuming a randomly diffusing animal and calculating the chances the animal will move from node X to node Y in N steps. This is used in Extended Data Fig. 1o to ensure that the comparison of DA with BA/CA for zero-shot quantification is a fair one. Empirical chance levels were calculated by using the location-to-location transition matrix recorded for each animal in the exploration session before any exposure to reward on the maze. Empirical chance is calculated in two different places:

(1) When finding the probability of a shortest-path transition. Pre-task exploration chance levels are calculated by quantifying each animal's transition probabilities around the maze in an exploration session prior to seeing any ABCD task (or any reward on the maze). This is used in Extended Data Fig. 1n, again to ensure that the DA to BA/CA comparison for zero-shot quantification is a fair one.

(2) When setting a chance level for the relative path distance measure. Chance here is defined as the mean relative path distance for transitions in the first trial averaged across the first five tasks across all animals. This is used in Fig. 1b–d.

Correct transition entropy (the animal's entropy when taking the shortest route between rewards) was calculated for transitions where there was more than one shortest route between rewards. We calculated the probability distribution across all possible shortest paths for a given transition and calculated entropy as follows:

$$\text{Entropy} = \sum \text{pk} \times \log_x(\text{pk})$$

Where $x$ is the logarithmic base, which is set to the number of shortest routes, and pk is the probability of each transition. Thus an entropy of 1 signifies complete absence of a bias for taking any one path and an entropy of 0 means only one of the paths is taken (that is, maximum stereotypy).

To quantify the effect of pre-configured biases in maze exploration on ABCD task performance in Extended Data Fig. 1h we analyse the per mouse correlation between:

1. Relative path distance on a given trial in an ABCD task—that is, ratio of taken path distance versus optimal (shortest) path distance.
2. The mean baseline probability for all steps actually taken by the animal on that same trial (measured from an exploration session before exposure to any ABCD task)—that is, how likely was the animal to take this path on a given trial before task exposure?

This correlation is positive overall, indicating that when animals take more suboptimal (longer) routes they do so through high probability steps—that is, ones that they were predisposed to take prior to any task exposure. This suggests that mistakes are associated with persisting behavioural biases.

## Activity normalization

We aimed to define a task space upon which to project the activity of the neurons. To achieve this, we aligned and normalized vectors representing neuronal activity and maze location to the task states. Activity was aligned such that the consumption of reward a formed the beginning of each row (trial) and consumption of the next reward a started a new row. Normalization was achieved such that all states were represented by the same number of bins (90) regardless of the time taken in each state. Thus, the first 90 bins in each row represented the time between rewards a and b, the second between b and c, the third between c and d and the last between d and a. We then computed the averaged neuronal activity for each bin. Thus the activity of each neuron was represented by an $n \times 360$ matrix, where $n$ is the number of trials and 360 bins represent task space for each trial. This activity was then averaged by taking the mean across trials, and smoothed by fitting a Gaussian kernel (sigma = 10°). To avoid edge effects when smoothing, the mean array was concatenated to itself three times, then smoothed, then the middle third extracted to represent the smoothed array. To reflect the circular structure of the task, the mean and standard error of the mean of this normalized and smoothed activity were projected on polar plots (for example, Fig. 2c,d).

## Generalized linear model

To assess the degree to which mFC neurons are tuned to task space, we used a linear regression to model each neuron's activity and permutation tests to determine significance[18]. Specifically, we aimed to quantify the degree to which goal-progress and location tuning of the neurons is consistent across tasks and states. For this we used a leave-one-out cross-validation design: we divided all tasks into the time periods spanned by each of the four states and used all data except one task–state combination to train the model. The remaining task–state combination (for example, task 3, state B) was used to test the model. This was repeated so that each task–state combination had been left out as a test period once. The training periods were used to calculate mean firing rates for five levels of goal progress relative to reward (five goal-progress bins) and each maze location (nine possible node locations). Edges were excluded from analyses since they are systematically not visited at the earliest goal-progress bin. The mean firing rates for goal progress and place from the training task-state combinations were used as (separate) regressors to test against the binned firing rate of the cell in the test data (held out task-state combination). We note that this procedure gives only one regressor for each variable, where the regressor takes a value equal to the mean firing rate of the cell in a given bin for the variable in question in the training data. For example, if a neuron fired at a mean rate of 0.5 Hz in location 6 in the training data, then whenever the animal is in location 6 in the test data the regressor for 'place' takes a value of 0.5. In effect, this analysis asks whether the tuning of a given neuron to a variable is consistent across different task–state combinations. To assess the validity of any putative task tuning, a number of potentially confounding variables were added to the model. These were: acceleration, speed, time from reward, and distance from reward. This procedure was repeated for all task–state combinations and a separate regression coefficient value was calculated for each.

To assess significance a given neuron was tuned to a given variable, we required it to pass two criteria:

(1) To have a mean regression coefficient higher than a null distribution: the null distribution is derived from repeating the regression but with random circular shifts of each neuron's activity array and computing regression coefficient values for each iteration (100 iterations) and then using the 95th percentile of this distribution as the regression coefficient threshold.

(2) To have a cross-task correlation coefficient significantly higher than 0: activity maps were computed for place (a vector of 9 values corresponding to 9 node locations) or goal progress (a vector of 5 values corresponding to 5 goal-progress bins) and then Pearson correlations calculated between each pairwise combinations of unique tasks. To ensure this analysis is sufficiently powered, we only used data concatenated across two days (giving up to six unique tasks). A two-sided one-sample $t$-test was then conducted to compare these cross-task correlation coefficients against 0.

Only neurons that passed both tests were considered tuned to a particular variable and reported as such in Fig. 2g and Extended Data Fig. 2e. We note that we do not use this tuning status anywhere else in the manuscript. Where we subset away spatial neurons (for example, Extended Data Figs. 2j,k and 4d,g), we deliberately use a less stringent criteria for spatial tuning to ensure that even neurons with weak or residual spatial tuning are excluded from the analysis (this excludes 76% of neurons), and hence ensure that results are robust to any spatial tuning. Where we subset in state neurons we use different criteria outlined in the section immediately below.

To determine whether the population as a whole was tuned to a given variable, we performed two proportions $z$-tests to assess whether the proportion of neurons with significant regression coefficient values for a given variable were statistically higher than a chance level of 5%.

To determine whether goal-progress tuning was also robust in ABCDE tasks, we performed the exact same procedure as above for ABCD tasks (Extended Data Fig. 3b). Moreover, to determine whether goal-progress tuning was conserved across tasks with different abstract structure, we performed an additional GLM on the cohort that experienced ABCD and ABCDE tasks. We used a train-test split where ABCDE tasks served as training tasks to determine mean firing rates for different goal-progress bins and for different place bins and then inputted these values to perform the regression in the ABCD tasks which were left out as test tasks. We again added acceleration, speed, time from reward and distance from reward as co-regressors (Extended Data Fig. 3f).

To exclude any neuron with trajectory tuning from the analysis of lagged tuning below we also ran a separate GLM this time replacing place with conjunctions of current place and the next step. This gave a vector of 24 possible place-next-place combinations which were used as regressors in the same manner as with place tuning above. Namely, the mean firing rates for place–next-place combinations from the training

task-state combinations were used as regressors to test against the binned firing rate of the cell in the test data (held out task-state combination). This was done while regressing out goal progress, acceleration, speed, time from reward, and distance from reward as above. This procedure was repeated for all task–state combinations and a separate regression coefficient value was calculated for each. The mean regression coefficient was then compared to the 95th percentile of coefficients from permuted data (using circular shifts as above). This was the only criteria used, which meant we were deliberately lenient to ensure that any cells with even weak trajectory tuning were excluded from the analysis in Extended Data Fig. 8c.

### State tuning

For state tuning, we first wanted to test whether neurons were tuned to a given state in a given task. We therefore analysed state tuning separately from the GLM above, which explicitly tests for the consistency of tuning across tasks. Instead, we used a $z$-scoring approach. First we took the peak firing rate in each state and trial, giving 4 values per trial: that is, a maximum activity matrix with dimensions $n \times 4$, where $n$ is the number of trials. Then we $z$-scored each row of this maximum activity matrix (that is, giving a mean of 0 and standard deviation of 1 for each trial). We then extracted the $z$-scores for the preferred state across all $n$ trials and subsequently conducted a $t$-test (two-sided) of this array against 0. Neurons with a $P$ value of <0.05 for a given task were taken to be state-tuned in that task. We also used a more stringent $P$ value of <0.01 to assess whether results where we only use a subset of state-tuned neurons are robust even for highly state-tuned neurons. This is used when assessing the degree of generalization and coherence of neurons across tasks (Extended Data Fig. 4b,e: see 'Neuronal generalization') and when assessing invariance of cross-task anchoring (Extended Data Fig. 8b: see 'Lagged task space tuning' section below).

### Manifold analysis

To visualize and further quantify the structure of neuronal activity in individual tasks, we embedded activity into a low dimensional space using UMAP, a non-linear dimensionality reduction technique previously used to visualize mFC population activity[34,49]. For this analysis we only used concatenated double days that: (1) had at least 6 tasks; and (2) had at least 10 simultaneously recorded neurons. As input, we used $z$-scored, averaged, time-normalized activity of each neuron recorded across concatenated double days (see 'Activity Normalization') and repeated this for each task. For ABCD tasks, this gave an $n \times 360$ input matrix where $n$ is the number of neuron tasks (each neuron repeated for 6 tasks) and 360 bins represent $4 \times 90$ bins for each state. This allowed us to assess the manifold structure within (rather than across) tasks. This high-dimensional ($n \times 360$) matrix was then used as an input to the UMAP, with the output being a low-dimensional ($3 \times 360$) embedding. We used parameters (3 output dimensions, cosine similarity as the distance metric, number of neighbours = 50, minimum distance = 0.6) in line with previous studies[34,49].

The output of this was plotted while projecting either a colour map indicating goal-progress (for example, Fig. 2h, left) or task state (for example, Fig. 2h, right) onto the manifold. This showed a hierarchical structure where goal-progress sequences were concatenated into a floral structure that distinguished different states. To quantify this effect we conducted the UMAP analysis separately for each recording (double) day and measured distances in the low dimensional manifold between: (1) bins that have opposite goal-progress bins and different states (across-goal progress); (2) bins that share the same goal-progress bin but represent different states (within-goal progress); and (3) bins that have the same goal-progress bin but where state identity was randomly shuffled across trials to destroy any systematic state-tuning (shuffled). This latter control was used as a floor to test whether distances between states were significantly above chance (which all analyses show that they were). This analysis showed that bins across different states that share the

same goal progress were significantly closer than those across opposite goal progress, providing further support for the hierarchical organization of mFC neurons in a single task into state-tuned manifolds composed of goal-progress sequences. To ensure state-discrimination is not due to any spatial tuning, we repeated this analysis for only non-spatial neurons (excluding even weakly spatially tuned neurons; see 'Generalized linear model'; Extended Data Fig. 2j,k). We also repeated the same procedure for data from the ABCDE task (Extended Data Fig. 3g,h).

Importantly, these manifolds in Fig. 2 and associated extended data figures pertain to individual tasks. They do not show a manifold that generalizes across tasks. Our findings in Figs. 3 and 5 show that the abstract task structures are not encoded by a single manifold, but rather a number of separate SMBs, each anchored to a different location–goal progress combination. It is not possible to visualize the entire multi-SMB manifold using UMAP or any dimensionality reduction method given that the minimum number of possible anchors (9 locations × at least 3 goal-progress bins = a minimum of a 27-dimensional space) makes the full manifold high-dimensional and hence not amenable to being compressed into a lower number of dimensions. In reality we predict many more modules given the high resolution of goal-progress tuning (Fig. 2c–f) and the fact that spatial anchors are typically multi-peaked, giving a large number of possible spatial pattern combinations and hence many more than 9 possible anchors (for example, Fig. 5g). This forces us to use high dimensional methods like the coherence and clustering analysis in Fig. 3 and the anchoring analysis in Fig. 5 to analyse the SMBs. Another way of stating this is to note that dimensionality reduction techniques are useful when visualizing each time point in the task as a single point in a low-dimensional space. However, we find that each time point in the task is actually represented by multiple points on multiple SMBs, each encoding a lag from a different anchor (Fig. 5). This means we cannot meaningfully plot the full SMB structure using the dimensionality reduction techniques used in the field. A useful comparison point are toroidal manifolds of grid cells in the mEC. Here the torus is only visible when a large number of neurons (>100) are isolated from a single module[45]. We can in principle show a manifold for a single SMB. However, given that we are dealing with orders of magnitude more mFC task modules than mEC grid modules (of which there are typically 6) we need orders of magnitude higher neuronal yields than the best cortical yields currently achievable to obtain 100+ neurons in a single mFC module.

### Neuronal generalization

To assess whether individual neurons maintained their state preference across tasks we quantified the angle made between a neuron in one task and the same neuron in another task. Only state-tuned neurons were used in these analyses. To ensure we captured robustly state-tuned neurons, we restricted analyses to neurons state-tuned in more than one-third of the recorded tasks. This subsetting is used throughout the manuscript where state-tuned cells are investigated. Quantifying the angle between neurons was achieved by rotating the neuron in task Y by 10° intervals and then computing the Pearson correlation between this rotated firing rate vector and the mean firing rate vector in task X. Using this approach, we found for each neuron the rotation that gave the highest correlation. For cross-task comparisons we calculated a histogram of the angles across the entire population and averaged this across both comparisons (X versus Y and X versus Z). Within task histograms were computed by comparing task X to task X' (Fig. 3b). To compute the proportion of neurons that generalized their state tuning, we found the maximum rotation across both comparisons (X versus Y and X versus Z). We then set 45° either side of 0 rotation across all tasks as the generalization threshold (orange shaded region in Fig. 3b). Because this represents one-quarter of the possible rotation angles, chance level is equal to $m/4$, where $m$ is the number of comparisons. When calculating generalization across one comparison, chance level is therefore 25%, whereas when two comparisons are taken, the chance level is 1/16 (6.25%). This definition of generalization accounts for the

strong goal-progress tuning, ensuring that the chance level for this 'close-to-zero' proportion is 1/4 for a single task-to-task comparison and 1/16 for 2 task-task comparisons regardless of goal-progress tuning.

Generalization could also be expressed at the level of tuning relationships between neurons. For example, two neurons that are tuned to A and C in one task could then be tuned to B and D in another, thereby maintaining their task-space angle (180°) but remapping in task space across tasks. To test for this, we computed the tuning angle between pairs of neurons and assessed how consistent this was across tasks. This angle was computed by rotating one neuron by 10° intervals and calculating the Pearson correlation between the mean firing vector of neuron $k$ and the rotated firing vector for neuron $j$. The rotation with the highest Pearson correlation gave the between-neuron angle (Fig. 3c). Thus, we compared the angle between a pair of neurons in task X to the same between-neuron angle in tasks Y and Z. Again histograms were averaged across both comparisons (X versus Y and X versus Z) for cross-task histograms while within-task histograms were computed by comparing task X to task X' (Fig. 3c). To compute the proportion of neuron pairs that were coherent across tasks, we found the maximum rotation of the angle between each pair across both comparisons (X versus Y and X versus Z). We then set 45° either side of 0 rotation across all tasks as the coherence threshold (orange shaded region in Fig. 3c). Because this represents one-quarter of the possible rotation angles, chance level is equal to $m/4$, where $m$ is the number of comparisons, and therefore is 25% for one comparison and 1/16 (6.25%) for two comparisons. This definition of coherence accounts for the strong goal-progress tuning, ensuring that the chance level for this 'close-to-zero' proportion is 1/4 for a single task-to-task comparison and 1/16 for 2 task-task comparisons regardless of goal-progress tuning.

This method quantifies remapping by finding the best rotation that matches the same neuron across tasks. While this mostly aligns well with the angles seen by visually inspecting the changes in the firing peak, in some cases (for example, Fig. 3a, neuron 3, session X versus Z) there is a discrepancy between the 'best-rotation' angle and the 'peak-to-peak' angle. This is because the best-rotation measure takes the entire shape of the tuning curve into account. It is therefore robust to small changes in the size of peaks when there is more than one similarly sized peak (for example, neurons 2, 4 and 6 in Extended Data Fig. 4l), which would introduce major inaccuracies in calculating remapping angles when using the peak to measure cross-session changes. Hence our preference for the best-rotation-based approach. However, we note that any ambiguity in calculating angles will introduce unstructured noise that works against us rather than introducing any biases that would induce false coherence. Nevertheless, to make this point robustly, we repeat the single cell generalization and pair-wise coherence analyses while using only state-neurons with concordant remapping angles across both methods (that is, using the best-rotation analysis method and peak-to-peak changes method) for all cross-session comparisons (Extended Data Fig. 4c,f). This would, for example, exclude neuron 3 in Fig. 3a, which on one cross-session comparison rotates differently when using the best rotation versus peak change methods in the X versus Z comparison. We show that the same results hold even under this condition: individual neurons do not generalize but pairs of neurons are partially coherent across tasks Extended Data Fig. 4c,f.

To assess whether the mFC population was organized into modules of coherently rotating neurons, we used a clustering approach. In the trivial case, where all neurons remap randomly, we expect 16 possible clusters. This is because we assess clustering across 3 tasks, which gives two comparisons (X versus Y and X versus Z). For each comparison there are 4 possible ways a given neuron can remap, creating 4 groups (neurons remapping by 0°, 90°, 180° or 270°). In the second comparison, there are another 4 ways the neurons could remap. Thus the number of clusters is 16 (that is, $4^2$). This assumes remapping is always exactly in 90° intervals—that is, perfect goal-progress tuning. In reality, goal-progress tuning is not perfect and so more clusters are

expected in the null condition. To avoid such assumptions, we create a null distribution that preserves the neurons' state tuning in the first task and goal-progress tuning throughout all tasks, but where each neuron otherwise remaps randomly across tasks. The procedure was as follows:

Step 1: take the maximum difference in pairwise, between-neuron angles across all comparisons and convert this into a maximum circular distance $(1 - \cos(\text{angle}))$, thereby generating a distance matrix reflecting coherence relationships between neurons (incoherence matrix).

Step 2: Compute a low dimensional embedding of this incoherence matrix, using $t$-distributed stochastic neighbour embedding (using the TSNE function of scikit learn manifold library, with perplexity = 5).

Step 3: Use hierarchical clustering on this embedded data (using the AgglomerativeClustering function of scikit learn cluster library, with distance threshold = 300). This procedure sorts neurons into clusters reflecting coherence relationships between neurons. We note that this analysis derives the number of clusters (modules) obtained across three tasks rather than the true number of modules across an arbitrarily large number of tasks.

We quantified the degree of clustering by computing the silhouette score for the clusters computed in each recording day:

$$\text{Silhouette score} = \frac{(b - a)}{\max(a, b)}$$

Where $a$ = mean within-cluster distance and $b$ = mean between-cluster distance. We repeated the same procedure but for permuted data, where state tuning in task X and goal-progress tuning in all tasks was identical to the real data but the state preference of each neuron remapped randomly across tasks. This allowed us to compare the Silhouette Scores for the real and permuted data (Fig. 3d). To visualize clusters and the tuning of neurons within them in the same plot, we plotted some example neurons from a single recording day where the $x$ and $y$ axes represented state tuning and the $y$ axis arranged neurons based on their cluster ID (the ordering along the $z$ axis is arbitrary; Fig. 3e).

## Lagged task space tuning

The task-SMB model predicts the existence of neurons that maintain an invariant task space lag from a particular anchor representing a behavioural step, regardless of the task sequence. Concretely, behavioural steps are conjunctions of goal progress (operationally divided into early, intermediate or late) and place (nodes 1–9). To test this prediction, we used three complementary analysis methods. All of these analyses were conducted on data where two recording days were combined and spike-sorted concomitantly, giving a total of six unique tasks per animal (with two exceptions that had four and five tasks each; see exclusions under 'Numbers'). For all of these analyses, only state-tuned neurons were used (see 'State tuning').

The main reason for using concatenated days is to be sufficiently powered for the generalization analyses (in Fig. 5). In essence the aim is to capture multiple instances where animals visit the same anchor points (for example, same reward locations) but in different sequences. The more tasks we can get for this the more we can sample the same reward locations in different task sequences. With 3 tasks, a total of 12 reward locations are presented (4 × 3) meaning each of the 9 reward locations is seen in 1.33 different tasks on average. With 6 tasks, reward locations are experienced in 2.67 different tasks on average. This gives us the ability to assess the same anchor points in different task sequences in a cross-validated manner and hence assess whether lagged tuning to anchor is conserved across tasks. For example, a neuron fires 2 states after reward in location 7 regardless of whether the animal is now in location 1 or 8. For non-rewarded locations, the situation is more complex and dependent on the animal's behavioural trajectories between rewards, but the same qualitative principle applies: more tasks give more visits to a given behavioural step as part of different behavioural sequences.

Method 1: Single anchor alignment. This approach assumes each neuron can only have a single goal-progress/place anchor and quantifies

the degree to which task-space lag for this neuron is conserved across tasks. We fitted the anchor by choosing the goal-progress/place conjunction which maximizes the correlation between lag-tuning-curves in all but one (training) tasks, and again used cross-validation by assessing whether this anchor leads to the same lag tuning in the left-out (test) task. The fitting was conducted by first identifying the times an animal visited a given goal-progress/place and sampling 360 bins (1 trial) of data starting at this visit, then averaging activity aligned to all visits in a given task, and smoothing activity as described above (under 'Activity normalization'). This realigned activity is then compared across tasks to compute the angle ($\theta$) between the neuron's mean aligned/normalized firing rate vector across tasks. This involves essentially doing all the steps for the 'Neuronal generalization' but for the anchor-aligned activity instead of state A-aligned activity. This was done for all possible task combinations and then a distance matrix ($M$) was computed by taking distance $= 1 - \cos(\theta)$. This distance matrix $M$ has dimensions $N_{\text{training tasks}} \times N_{\text{training tasks}} \times N_{\text{anchors}}$ (typically $5 \times 5 \times 27$ as there are usually 6 tasks, meaning 5 training tasks are used, and $3 \times 9$ possible anchors corresponding to 3 possible goal-progress bins (early, intermediate and late) and 9 possible maze locations). The distance can then be averaged across all comparisons to find the mean distance between all comparisons for a given anchor for a given training task, generating a mean-distance matrix ($M_{\text{mean}}$). This $M_{\text{mean}}$ matrix has the dimensions $N_{\text{comparisons}} \times N_{\text{anchors}}$; where $N_{\text{comparisons}} = N_{\text{training tasks}} - 1$; typically this will be $4 \times 27$. The entry with the minimum value in this $N_{\text{mean}}$ matrix gives the combination of training task and goal-progress/place anchor that best aligns the neuron—the training task selected is used as the reference task to do the comparison below. Next, the neuron's mean activity in the test task is aligned to visits to the best anchor calculated from the training tasks. This allows calculating how much this aligned activity array has remapped relative to the aligned activity in the reference training task—if it has remapped by 0° or close to 0° (within a 45° span either side of zero) then the neuron is anchored (that is, maintains a consistent angle with its anchor across tasks). For a given test-train split, we computed a histogram of the angles across all the neurons and then we averaged the histograms across all test-train splits to visualize the overall distribution of angles between training and test tasks (for example, Fig. 5b). To quantify the degree of alignment further, we measured the correlation between the anchor-aligned activity of neurons in the test task versus reference training task. Importantly, to account for the strong goal-progress tuning of cells we only consider activity of neurons in their preferred goal-progress bin when calculating this correlation (Fig. 5c). The neuron's lag from its anchor was identified by finding the lag at which anchor-aligned activity was maximal. This lag is used below for 'Predicting behavioural choices'.

Method 2: Lagged spatial similarity. To detect putative lagged task space neurons, we calculated spatial tuning to where the animal was at different task lags in the past (Fig. 5e). While spatial neurons should consistently fire at the same locations(s) at zero lag, neurons that track a memory of the goal-progress/place anchor will instead show a peak in their cross-task spatial correlation at a non-zero task lag in the past (Fig. 5e). To quantify this effect, we used a cross-validation approach, using all tasks but one to calculate the lag at which cross-task spatial correlation was maximal, and then measuring the Pearson correlation between the spatial maps in the left-out task and the training tasks at this lag (Fig. 5f). To account for the strong goal-progress tuning of the neurons, all maps were computed in each neuron's preferred goal progress. We note that it is the firing rates that are calculated in the preferred goal-progress bins of each neuron. The spatial positions are then derived either in the same bin (that is, the 'present'), or in bins successively further back in the past (making a total of 12 bins spanning the entire 4 states at a resolution of 3 goal-progress bins per state).

Method 3: Model fitting. For each neuron we computed a regression model that described state-tuning activity as a function of all possible combinations of goal-progress/place and all task lags from each

possible goal-progress/place. Thus a neuron could fire at a particular goal-progress/place conjunction but also at a particular lag in task space from this goal-progress/place. We used an elastic net (using the Elastic-Net function from the scikit learn linear_model package) that included a regularization term which was a 1:1 combination of L1 and L2 norms. The alpha for regularization was set to 0.01. A total of $9 \times 3 \times 12$ (312) regressors were used for each neuron, corresponding to 9 locations, 3 goal-progress bins (so 27 possible goal-progress/place anchor points) and 12 lags in task space from the anchor (4 states × 3 goal-progress bins). We trained the model on five (training) tasks and then used the resultant regression coefficients to predict the activity of the neuron in a left-out (test) task. To ensure our prediction results are due to state preference and not the strong effect of goal-progress preference (Fig. 2), both training and cross-validation were only done in the preferred goal progress of each neuron. For non-zero-lag neurons, we only used state-tuned neurons with all of the three highest regression coefficient values at non-zero lag from an anchor (lag from anchor of 30° or more for Fig. 5h, right; 90° (one state) or more for Extended Data Fig. 8a) in the training tasks. Also, for non-zero lag neurons, we only use regression coefficient values either 30° (Fig. 5h, right) or 90° (Extended Data Fig. 8a) either side of the anchor point to predict the state tuning of the cells. This ensured that the prediction was only due to lagged activity and not direct tuning of the neurons to the goal-progress/place conjunction.

The above analysis models neuronal activity as a linear function of anchor lags. To investigate the robustness of our findings to this assumption, we repeated this analysis using a linear–nonlinear–Poisson model which uses a non-linear (logarithmic) link function. We used a regularization alpha = 1. This type of model has been traditionally used to model neuronal tuning while accounting for non-linearities[62].

For per mouse effects reported in Extended Data Figs. 7e,i and 8e, we tested whether the number of mice with a mean cross-validated correlation above 0 is higher than chance, chance level being a uniform distribution (50:50 distribution of per mouse correlation means above and below zero). We used a one-sided binomial test against this chance level. All seven mice need to have mean positive values for this test to yield significance.

**Predicting behavioural choices**
The SMB model proposes that behavioural choices should be predictable from bumps of activity along specific memory buffers long before an animal makes a particular choice. By 'choice' here we mean a decision to move from one node to one of the immediately adjacent nodes on the maze (for example, from location 1 should I go to location 2 or 4?; Fig. 6a). To test whether these choices are predictable from distal neuronal activity we used a Logistic regression model. For this analysis we used only consistently anchored neurons, that is neurons that had the same anchor and same lag to anchor in at least half of the tasks. This relied on the single-anchor analysis (Fig. 5a; see 'Lagged task space tuning', Method 1: Single anchor alignment) to find for each cell its preferred anchor and lag from anchor. Furthermore, to avoid contamination of our results due to simple spatial tuning, we only used neurons with activity lagged far from their anchor (one third of an entire state—that is, in the ABCD task at least 30° in task space either side of the anchor (for example, Fig. 6c); and this was also repeated for lags of at least 90° (one whole state) either side of the anchor in Extended Data Fig. 9d). For the ABCDE task, the equivalent 'close-to-zero-lag' period was 24° (1/3 of one state which is 72°; Extended Data Fig. 9i). We measured the activity of a given neuron during its bump time—that is, the time at which a neuron is lagged relative to its anchor. Precisely, this is the mean firing rate from a period starting with the lag time from the anchor and ending 30° forward in task space from that point (1/3rd of a state). This mean activity was inputted on a trial by trial basis every time the animal was at a goal-progress/place conjunction that was one step before the goal-progress/place anchor in question (for example, if the anchor is at early goal progress in place 2, the possible goal-progress/places before this are: late goal progress in place 1, late goal

progress in place 3 and late goal progress in place 5; see maze structure in Fig. 1a). We used this activity to predict a binary vector that takes a value of 1 when the animal visits the anchor and 0 when the animal could have visited the anchor (that is, was one step away from it) but did not chose to visit the anchor. To remove confounds due to the autocorrelated previous behavioural choices, we added previous choices up to 10 trials in the past into the regression model. For the first regression analysis in Fig. 6b,c and Extended Data Fig. 9c, we add previous choices as individual regressors (each trial being a column in the independent variable matrix). Once we determined the regression coefficients for previous choices in Extended Data Fig. 9c, we fit an exponential decay function to these coefficients and used this kernel in all subsequent regressions in Fig. 6 and Extended Data Fig. 9 to give different weights to the previous choices depending on how many trials back they happened. This creates a single regressor that accounts for all previous choices up to 10 trials in the past. Furthermore, to assess whether any observed prediction was specific to the bump time as predicted by the SMB model, we repeated the logistic regression for other control times: random times and decision time (30° before the potential anchor visit for ABCD and 24° before potential anchor visit in ABCDE). We also do the same for times shifted by one state intervals from the bump time (90°, 180° and 270° for ABCD tasks) and (72°, 144°, 216° and 288° for the ABCDE task). This preserves the cell's goal-progress preference but not lag from anchor, allowing us to test whether the precise lag from anchor in the entire task space is important for predictions of future choices. We further repeated this regression only taking neurons that are more distal from their anchor (that is, at least 90 degree separation either side of the anchor: Extended Data Fig. 9d) and also only for visits to non-zero goal-progress anchors (that is, non-rewarded locations: Extended Data Fig. 9e–h).

### Sleep–rest analysis

To investigate the internal organization of task-related mFC activity we recorded neuronal activity in a separate enclosure containing bedding from the animal's home cage but no reward or task-relevant cues. Animals were pre-habituated to sleep/rest in these 'sleep boxes' before the first task began. We measured neuronal activity across sleep/rest sessions both before any tasks on a given day and after each session. The first (pre-task) sleep/rest session was 1 h long, inter-session sleep/rest sessions were 20 min long and the sleep/rest session after the last task was 30–45 min long. All sessions except the first sleep session were designated as 'post-task' sleep sessions.

In the parts leading up to the sleep analysis we show that neurons are organized sequentially relative to each other (Figs. 2 and 3) and relative to anchor points (Fig. 5), firing consistently at fixed lags from these anchors. We further show task structuring of the SMBs by illustrating that neurons can be used to predict an animal's future choices in a manner paced by the task periodicity (Fig. 6). Having established this task structured, sequential activity, what we aim to do with Fig. 7 is to test: (1) whether this sequential activity is internally organized (that is, present in the absence of any structured task input); and (2) whether internally organized sequential activity is open (creating a delay line) or closed (creating a ring).

Activity was binned in 250-ms bins and cross-correlations between each pair of neurons were calculated using this binned activity. For this analysis, we only used consistently anchored neurons, that is neurons that had the same goal-progress/location conjunction as its best anchor and same lag from this anchor in at least half of the tasks. We then regressed the awake angle difference between pairs of neurons sharing the same anchor against this sleep cross-correlation. This angle was taken from the first task on a given day for pre-task sleep, and from the task immediately before the sleep session for all post-task sleep sessions. The idea is that neurons closer to each other in a given neuronal state-space should be more likely to be coactive within a small time window compared to neurons farther apart. Thus, we assessed the degree to which the regression coefficients were negative (that is, smaller distances correlate

with higher coactivity). If the distribution of lags from the anchor was uniform, forward and circular distances would be orthogonal and so adding forward distance to the regression would be redundant. However, the distribution of lags from the anchor is not uniform (Extended Data Fig. 7c) and so we add forward distance to the regression to remove any possible contribution of delay lines to the results. We measured the forward distance between pairs of co-anchored neurons in reference to their anchor. If neurons are internally organized on a line, then the larger the forward distance between a pair of neurons the further away two neurons are from each other in neuronal state space, and hence the less coactive they will be (Fig. 7a). Circular distance is correlated with forward distance for pairs of neurons with a forward distance of <180 but anti-correlated with forward distance for pairs of neurons with a forward distance of >180, as neurons circle back closer to each other if the state space is a ring. We used a linear regression to compute the regression coefficients for circular and forward distances in the same regression for all consistently anchored neuron pairs sharing the same anchor (Extended Data Fig. 10a), and when comparing those to neuron pairs across anchors (Fig. 7c). To control for place and goal-progress tuning, we added the spatial map correlation and circular goal-progress distance as co-regressors in the regression analyses.

To further analyse whether the state space is circular, we compared the sleep cross-correlation between pairs of neurons (that share the same anchor) at different forward distances. If the state space is circular, this should give a V-shaped curve, with high cross-correlations at the lowest forward distances and highest forward distances which both correspond to low circular distances. In other words, the slope of the cross-correlation versus forward distance curve should be negative for pair-to-pair forward angles <180° and positive for angles >180°. A delay line would instead give a negatively sloping curve at all pairwise angles. In line with a circular state space, we observe a V-shaped curve in Fig. 7d. Further, to investigate the effects of sleep stage or time since sleep on our results, we conducted the same analyses across pre-task and post-task sleep (Extended Data Fig. 10b) and across different times since sleep (Extended Data Fig. 10c).

We further analysed whether neurons that shared the same anchor showed stronger state-space versus sleep coactivity relationships than those that have different anchors. We conducted the regression of circular distance against sleep cross-correlation either for pairs of neurons that share the same anchor, or those that have different anchors (Fig. 7c). As before, we co-regressed spatial correlations and goal-progress distances.

### Numbers

Animals: 13 animals in total were used for behavioural recordings across 4 separate cohorts conducted by 3 different experimenters (A.L.H., M.E.-G. and A.B.)—4 of these animals only completed 10 tasks as part of the first cohort and the remaining 7 completed at least 40 ABCD tasks. Three animals performed additional ABCD tasks with the tone omitted from reward a on 50% of trials. Two animals did additional ABCDE tasks.

Of the 13 animals, 7 animals in total were used for electrophysiological recordings, the remaining 6 animals are accounted for below:
- 3 animals (in cohort 1) were not implanted at any point.
- 1 animal was implanted with silicon probes but was part of the first cohort so did not get to the 3 task days (that is, only completed the first 10 tasks).
- 2 animals were implanted but their signal was lost before the 3 task days.

Exclusions: no animals were excluded from analyses: All animals (13) were included in the behavioural analyses, and all animals for which there was an electrophysiological signal by the 3 task days (7) were included in the electrophysiological analyses.

Neurons: we report 'neuron-days'—that is, by summing up each day's neuron yield throughout the manuscript.

Total number of neuron-days:

(1) ABCD task: 2,929 when splitting all data into single days (that is, while splitting each double day into two and summing the yield across days). (Note: this is used when the analysis pertains specifically to comparisons across 3 tasks: Figs. 2f and 3 and Extended Data Fig. 4). 1677 when considering the yield of double days only once (that is, no splitting of double days).

(2) ABCDE task: 288 neurons on concatenated double days (that is, no splitting of double days).

More detail is provided in Supplementary Table 1, which outlines the numbers of mice, recording days, tasks, sessions, neurons and neuron pairs (as appropriate) for each analysis and the criteria used for inclusion.

### Reporting summary

Further information on research design is available in the Nature Portfolio Reporting Summary linked to this article.

### Data availability

Data used in this study are available via the Open Science Foundation at https://doi.org/10.17605/OSF.IO/3D9R2 (ref. 63). Details of the design and material of all maze components can be found at https://github.com/pyControl/hardware/tree/master/GridMaze. The mouse brain atlas (http://labs.gaidi.ca/mouse-brain-atlas/) was used for checking implantation coordinates.

### Code availability

Code used in this study is available at https://github.com/mohamadyelgaby/mFC_schema.

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

**Acknowledgements** The authors thank G. Costa for designing the model schematics; D. Gupta, M. Bukwich, T. Mrsic-Flogel, R. Dolan, M. Nakano and S. Reinert for valuable input on the manuscript; R. Pinacho, L. Burgeno, B. Godinho, M. B. Pozo, V. Samborska, P. Doohan and G. Daubney for guidance on data collection and data preprocessing; and all members of the Behrens and Walton laboratories for their input and support during the project. A.L.H. is supported by a Wellcome Trust PhD studentship (220047/Z/19/Z), T.E.J.B. is supported by a Wellcome Principal Research Fellowship (219525/Z/19/Z), a Wellcome Collaborator award (214314/Z/18/Z) and the Gatsby Initiative for Brain Development and Psychiatry (GAT3955). The Wellcome Centre for Integrative Neuroimaging and Wellcome Centre for Human Neuroimaging are each supported by core funding from the Wellcome Trust (203139/Z/16/Z and 203147/Z/16/Z). J.C.R.W. is supported by the Sir Henry Wellcome Post-doctoral Fellowship (222817/Z/21/Z). W.D. is supported by the Gatsby Charitable Foundation. T.A. is supported by the Wellcome Trust career development award (225926/Z/22/Z). M.E.W. is supported by a Wellcome Collaborator award (214314/Z/18/Z) and a Wellcome trust SRF (202831/Z/16/Z).

**Author contributions** M.E.-G., T.E.J.B., T.A., J.C.R.W., M.E.W and A.L.H. conceptualized the study and designed the ABCD task. A.L.H., M.E.-G. and A.B. collected the data with input from T.A. and M.E.W. M.E.-G. and A.L.H. analysed and interpreted the data with input from T.E.J.B. and T.A. M.E.-G. conceptualized the SMB model with input from T.E.J.B. and all other authors. M.E.-G. and T.E.J.B. wrote the manuscript with input from A.L.H., T.A., M.E.W. and all other authors.

**Competing interests** The authors declare no competing interests.

**Additional information**
**Correspondence and requests for materials** should be addressed to Mohamady El-Gaby or Timothy E. J. Behrens.

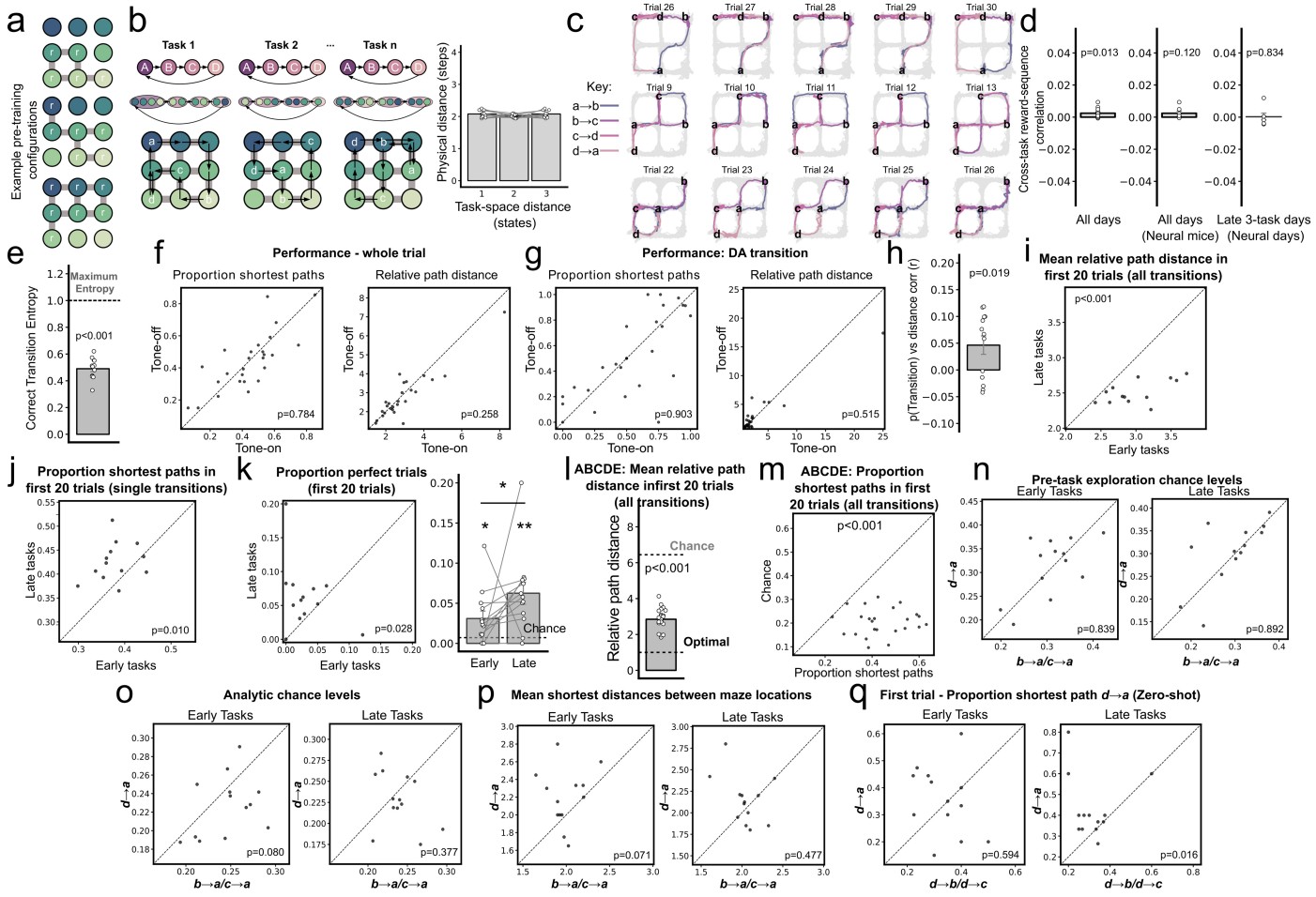

**Extended Data Fig. 1** | See next page for caption.

**Extended Data Fig. 1 | Behavioural measures in the ABCD task.** a) Three example connection configurations for the pre-selection and pre-training sessions done before exposure to the first ABCD task. Here a subset of 5-7 maze locations (nodes) were available to the mouse and each node was rewarded provided the animal did not just receive reward in the same node. b) Tasks were designed such that task space and physical space are orthogonal to each other. Left: schematic showing that optimal path lengths between rewarded goals differed both within and between tasks. Right: a bar plot showing that the task space "distances" between reward locations (how many task states are between the rewards) are not correlated with the physical distances in the maze (the optimal number of steps taken to reach reward). Data points represent individual mice, where physical distances are averaged across all tasks experienced by a given mouse. Pearson correlation: r = 6.04 × 10-18 P = 1.0; one-way ANOVA statistic=2.02, P = 0.147, df = 12 c). Example paths from 3 different mice performing 3 different tasks. Each row is a set of 5 consecutive trials from the same mouse and task. Single trial paths are superimposed upon whole session coverage shown in grey. Mice rapidly converged on near-optimal routes and used only a subset of the available paths. d) Reward sequences showed minimal correlation across all tasks in all mice (left) and no correlations on all tasks for mice where neuronal data was recorded (7/13 mice middle) or on the late 3 task days from neural mice (right). Note that electrophysiological data in this manuscript is all collected from the late 3 task days. T-test (two-sided) against 0 correlation: all tasks: N = 13 mice, r = 0.002, statistic=2.91, P = 0.013, df = 12; all tasks from neural mice (mice where neuronal data was recorded): N = 7 mice, r = 0.002, statistic=1.81, P = 0.120, df = 6; tasks on Late 3-task days from neural mice (tasks where neural data was recorded): N = 7 mice, r = 4.5 × 10$^{-4}$, statistic=0.22, P = 0.634, df = 6. e) Animals used stereotyped routes when taking the shortest route to a goal. The entropy of correct transitions taken is lower than expected if animals took all shortest routes equally. T-test (two-sided) against 1 – N = 13 animals, statistic = −23.6, P = 1.96 × 10$^{-11}$, df = 12 f) Performance is unaffected by the inclusion of a tone at reward *a*. 3 mice were exposed to additional tasks (after completing task 40) where the tone at *a* was randomly omitted in 50% of trials. The tone or no-tone status of a trial refers to whether the tone was omitted at the *a* at the beginning of the trial. Left: mean proportion of transitions where one of the shortest routes was taken N = 26 tasks, Wilcoxon test (two-sided); statistic=164, P = 0.784. Right: mean relative path distance N = 26 tasks, Wilcoxon test (two-sided): statistic=130, P = 0.258 g). Performance on the *d*-to-*a* transition is unaffected by the inclusion of a tone at reward *a* in the previous trial: Left: mean proportion of transitions where one of the shortest routes was taken N = 26 tasks, Wilcoxon test (two-sided); statistic=134, P = 0.903. Right: mean relative path distance N = 26 tasks, Wilcoxon test (two-sided): statistic=149, P = 0.515 h). Suboptimal performance was associated with persisting behavioural biases from before exposure to the task. Y-axis shows the r value calculated from a correlation between the mean relative path distance taken between goals and the probability the steps within this trajectory would have been taken when the animal was naive to any ABCD task (when the animal explored the arena before any rewards or tasks were presented). A net positive correlation indicates that when animals take longer routes (i.e. perform less optimally) they take these routes through steps that they were more likely to take before exposure to any ABCD task. T-test (two-sided) against 0 – N = 13 animals, statistic=2.70, P = 0.019, df = 12 i). Mean relative path distance travelled by the mice between goals in the first 20 trials of early vs late tasks. Wilcoxon test (two-sided) N = 13 animals, Statistic=0.0, P = 2.44 × 10$^{-4}$ j). Mean proportion of transitions where one of the shortest routes was taken in the first 20 trials of early vs late tasks. Wilcoxon test (two-sided) N = 13 animals, Statistic=10.0 P = 0.010 k) Mean proportion of "perfect trials" where <u>all transitions</u> (a → b, b → c, c → d <u>and</u> d → a) in a given trial were taken via the shortest route. Left: scatter plot of mean proportion of perfect trials in the first 20 trials of early vs late tasks. Wilcoxon test (two-sided) N = 13 animals, Statistic=11.0 P = 0.028. Right: bar plot of the same data showing that, for both early and late tasks, the proportion of perfect trials is significantly above chance: T-test (two-tailed) against chance (0.007): Early tasks statistic=2.55, P = 0.025; Late tasks - statistic=4.06, P = 0.002. l) ABCDE task performance (relative path distance): after completing at least 40 ABCD tasks, two animals completed additional ABCDE tasks (11 and 13 tasks each) where tasks comprised a loop of 5 (instead of 4) rewards. Animals readily performed above chance in the first 20 trials, as demonstrated by comparing path length between goals to the shortest possible path (i.e. computing a "relative path distance" measure). T-test (two-sided) against chance (6.44): N = 24 tasks, statistic = −30.0 P = 6.18 × 10$^{-20}$, df = 23. Chance level was calculated empirically using the mean relative path distance across the first trial of the first 5 ABCD tasks. m) ABCDE task performance (proportion correct transitions): animals readily performed above chance in the first 20 trials, as demonstrated by quantifying the proportion of transitions where animals took the shortest possible path. Wilcoxon test (two-sided): N = 24 tasks, statistic=0.0, P = 1.19 × 10$^{-7}$. Chance levels were derived empirically for each mouse using baseline transition probabilities calculated when animals explored the maze before experiencing any ABCD tasks: see Methods under "Behavioural Scoring". n) No difference in the empirical chance levels (baseline transition probabilities calculated when animals explored the maze before experiencing any ABCD tasks: see Methods under "Behavioural Scoring") between *d*-to-*a* and *c*-to-*a*/*b*-to-*a* transitions on the *first* trial in early (left) and late (right) tasks. Wilcoxon test (two-sided); Early tasks: N = 13 animals, statistic=42.0, P = 0.839; Late tasks: N = 13 animals, statistic=43.0, P = 0.893 o). No difference in the analytical chance levels (see Methods under "Behavioural Scoring") between *d* to *a* and *c*-to-*a*/*b*-to-*a* transitions on the *first* trial in early (left) and late (right) tasks. Wilcoxon test (two-sided); Early tasks: N = 13 animals, statistic=20.0, P = 0.080; Late tasks: N = 13 animals, statistic=32.0, P = 0.376 p). No difference in the shortest physical maze distances between *d*-to-*a* and *c*-to-*a*/*b*-to-*a* transitions on the *first* trial in early (left) and late (right) tasks. Wilcoxon test (two-sided); Early tasks: N = 13 animals, statistic=16.0, P = 0.071; Late Tasks: N = 13 animals, statistic=25.0, P = 0.477 q) Zero-shot inference on the first trial of late tasks is associated with animals returning from *d* to *a* more often than *d*-to-*b* or *d*-to-*c*. The proportion of tasks in which animals took the most direct path from *d*-to-*a* on the *first* trial is compared to the same measure but for premature returns from *d*-to-*b* and *d*-to-*c*. Early tasks are shown on the left and late tasks on the right. Wilcoxon test (two-sided); Early tasks: N = 13 animals, statistic=27.0, P = 0.594; N = 13 animals, Late tasks: statistic=6.0, P = 0.016 All error bars represent the standard error of the mean.

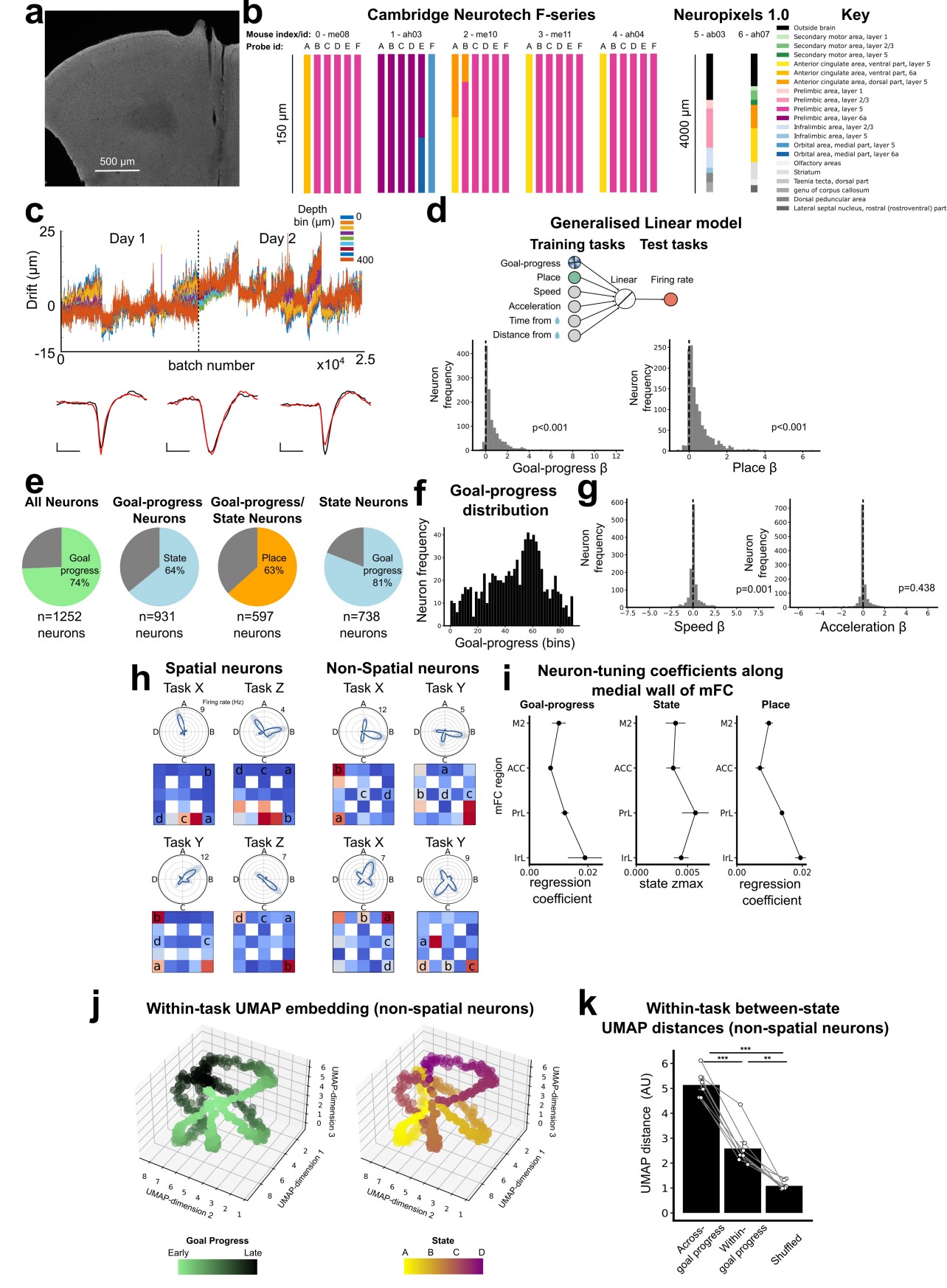

**Extended Data Fig. 2** | See next page for caption.

**Extended Data Fig. 2 | Recording set up and tuning properties of mFC neurons in the ABCD task.** a) Coronal slice from an implanted mouse showing silicon probe track terminating in the prelimbic region of mFC. b) The laminar profile of probe channel positions for each mouse. Shanks A-F in Cambridge neurotech probes are arranged posterior-to-anterior. 90.7% of all recorded neurons were histologically localised in mFC regions based on the inferred channel position: 68.3% in Prelimbic cortex, 11.3% in Anterior Cingulate cortex, 6.1% in Infralimbic cortex and 5.0% in M2. Of the remaining 9.3%, 4.8% could not be localised to a specific peri-mFC region within the atlas coordinates as they were erroneously designated to peri-mFC white matter areas, likely due to variations between actual region boundaries and atlas derived ones, 2.2% were found in the dorsal peduncular nucleus, 1.1% in the striatum, 0.6% in the medial orbital cortex, 0.3% in the lateral septal nucleus and 0.3% in Olfactory cortex. c) Data was spike sorted across concatenated sessions spanning two recording days for the GLM analyses below and later anchoring analysis in Figs. 5–7. Top: Here we show an example "Estimated drift trace" for a concatenated double day, showing a largely stable recording set up. The plot shows the estimated probe drift relative to the brain across the two recording days along the depth of the neuropixels probe. Bottom: Example mean spike waveforms from 3 different neurons across 3 different animals. The plots show the mean of the first 100 spikes on day1 (black) and the mean of the last 100 spikes on day2 (red), illustrating stability of spike detection across days. The spikes are from neuron 1 and neuron 2 in Extended Data Fig. 6b and the neuron in Fig. 5e respectively. Scale bars: Vertical: 200 μV, Horizontal: 0.5 ms. d) Top: a schematic of the variables inputted into a generalised linear model that predicts neuronal activity across tasks and states. The model captured variance as a function of goal-progress, place, speed, acceleration, time from reward and distance from reward. Only data spike sorted across two days (6 unique tasks) was used to ensure this analysis is sufficiently powered. Bottom Left: A histogram showing the mean regression coefficient values for goal-progress as a regressor across task/state combinations for each neuron. One-sample T-test (two-sided) against 0: N = 1252 neurons; statistic=21.7; P = 8.93 × 10-89, df = 1251. Bottom right: A histogram showing the mean regression coefficient values for place as a regressor across task/state combinations for each neuron. One-sample T-test (two-sided) against 0: N = 1252 neurons; statistic=24.9; P = 3.31 × 10-111, df = 1251. e) Pie-charts showing the proportions of cells calculated using the results of the generalised linear model above in addition to cross-task correlations between tuning to goal-progress and place. Only data spike sorted across two days (6 unique tasks) was used to ensure this analysis is sufficiently powered. Plot shows proportions of neurons with i) significant regression coefficient values for goal-progress or place ii) Significantly positive cross-task correlation for goal-progress or place. It also shows proportions of state tuned neurons derived from a separate z-scoring analysis (More details in Methods under "Tuning to basic task variables"). Proportion of all neurons that are goal-progress cells: 74%; Two proportions test: N = 1252 neurons, z = 35.5, P = 0.0.

Proportion of goal-progress neurons that are state tuned: 64% Two proportions test: N = 931 neurons, z = 26.8, P = 0.0. Proportion of neurons tuned to goal-progress and state that are also tuned to place: 63%, Two proportions test: N = 597 neurons, z = 21.2, P = 0.0. Proportion of all state-tuned neurons that are also goal-progress tuned; 81% Two proportions test: N = 738 neurons, z = 29.5, P = 0.0. f) A histogram showing the distribution of significant goal-progress peaks amongst all neurons, all tasks and all states. Only neurons from concatenated double days that are significantly goal progress-tuned and have at least one significant goal-progress peak are shown (N = 873 neurons). The plot shows that such significantly goal-progress tuned cells have peaks throughout the entire range of goal progress values. Note that this plot spans goal-progress space, which is the lag between any two rewarded goals, rather than the full (multi-goal) task space. g) Regression coefficients for animal kinematics (from GLM in Fig. 2g). Two histograms showing the mean regression coefficient values for Speed (Left) and Acceleration (Right) as a regressor across task/state combinations for each neuron. One-sample T-test (two-sided) against 0: Speed: N = 1252 neurons, statistic=3.36, P = 8.01 × 10$^{-4}$ df = 1251; Acceleration: N = 1252 neurons, statistic = −0.78, P = 0.438 df = 1251. h) Polar plots of task tuning and spatial maps for four example neurons that are tuned to both goal-progress and state. Each neuron is plotted across two tasks to illustrate spatial tuning (left two neurons) and lack thereof (right two neurons). i) The subregional distribution of neuron type coefficients along the medial wall of the frontal cortex in neuropixels recordings. One-way ANOVA: Left: Proportion of Goal progress neurons: F = 2.40, P = 0.143, df = 3; Middle: Proportion of state neurons F = 1.04, P = 0.425, dof=3; Right: Proportion of place-tuned neurons F = 18.8, P = 5.54 × 10$^{-4}$, df = 3. Posthoc Tukey HSD tests (Two-tailed): IrL vs PrL P = 0.049; IrL vs ACC P = 0.000; IrL vs M2 statistic=0.003, PrL vs ACC P = 0.021. j) A plot of the mean task manifold derived from a Uniform Manifold Approximation and Projection (UMAP)-embedding along three dimensions restricted to only <u>non-spatial neurons</u>. Note that we use the most permissive threshold for spatial tuning here to ensure that we exclude even neurons with weak/residual spatial tuning. Any neuron that had a spatial regression coefficient above the 95th percentile of the null distribution was excluded from this analysis. The same manifold is shown twice: Left, goal-progress tuning along the manifold; right, state tuning along the same manifold. The entire task manifold is composed of goal-progress subloops. k) Quantifications of distances along the 3-dimensional UMAP-derived manifold - across different states and opposite goal-progress bin (left), across different states but for the same goal-progress bin (middle) or the distances across different states and same goal-progress bin for a shuffled control. N = 8 double-days - T-tests (two-sided) (with bonferroni correction): Across-goal progress vs within goal-progress: statistic =10.3, P = 5.45 × 10$^{-5}$, df = 7; Across-goal progress vs permuted control: statistic =17.5, P = 1.47 × 10$^{-6}$, df = 7; Within goal-progress vs permuted control: statistic =5.2, P = 0.004, df = 7 All error bars represent the standard error of the mean.

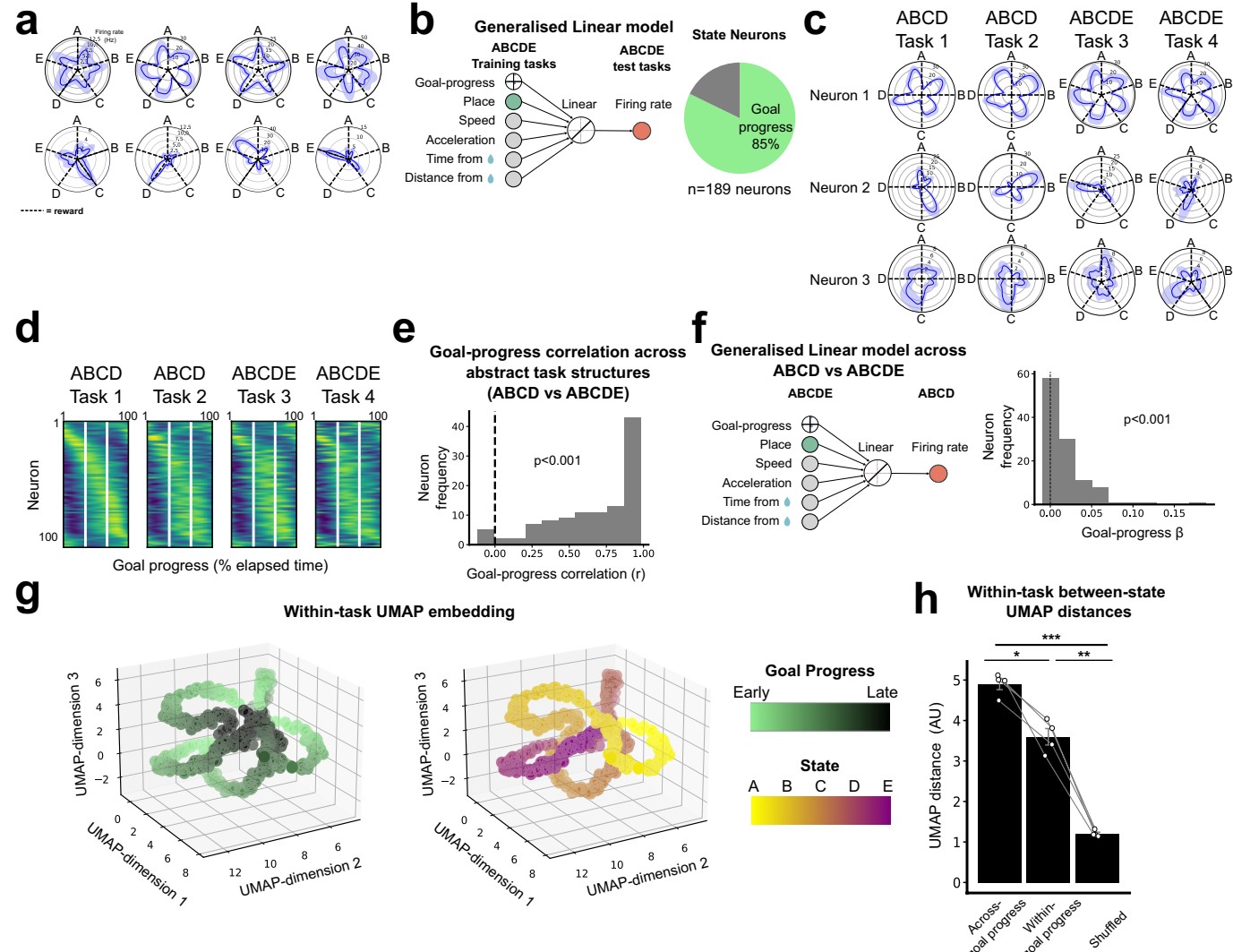

**Extended Data Fig. 3 | Tuning properties of mFC neurons in the ABCDE task.**
a) Polar plots of task-space tuning for 8 example neurons in the ABCDE task -
neurons 1-4 are purely goal-progress tuned while neurons 5-8 are conjunctively
goal-progress and state tuned. b) State neurons in the ABCDE task are
predominantly goal progress tuned. Left: design of GLM to identify
goal-progress tuned neurons in ABCDE tasks. Right pie chart showing the
proportion of state-tuned neurons that are goal-progress tuned: ABCDE
goal-progress/state GLM; Two proportions test: N = 189 state neurons,
proportion goal-progress-tuned= 85%, z = 15.6, P = 0.0) c) Polar plots of
task-space tuning for 3 example neurons recorded across 2 ABCD tasks and
then two ABCDE tasks - neurons 1 is purely goal-progress tuned while neurons 2
and 3 are conjunctively goal-progress and state tuned. d) Goal progress tuning
is maintained across abstract tasks ABCDE vs ABCD: The average firing rate
vector of all neurons relative to an individual goal (from goal "n" to goal "n + 1";
averaged across all states). Animals experienced 2 ABCD tasks followed by 2
ABCDE tasks on these days. Each row represents a single neuron and the
neurons are arranged on the y axis by their peak firing goal-progress in task 1 in
the ABCD condition. This alignment is largely maintained in tasks across both
ABCD and ABCDE structures. White dashes indicate early intermediate and late
goal-progress-cutoffs. e) A histogram showing the mean goal-progress-vector
correlation across tasks for each neuron. One-sample T-test (two-sided)

against 0: N = 111 neurons; statistic=23.8; P = 3.76 × 10$^{-45}$, df = 110. Note that the
neurons used in this panel are those on days where animals experienced both
ABCDE and ABCD tasks. f) Left: design of GLM to identify whether neurons
maintain their goal-progress tuning across ABCDE and ABCD tasks. Right:
A histogram showing the mean regression coefficient values for goal-progress
as a regressor across ABCD and ABCDE tasks for each neuron. One-sample
T-test (two-sided) against 0: N = 111 neurons; statistic=7.43; P = 2.45 × 10$^{-11}$,
df = 110. g) A plot of the mean task manifold derived from a Uniform Manifold
Approximation and Projection (UMAP)-embedding along three dimensions for
mFC activity in the ABCDE. The same manifold is shown twice: Left, goal-progress
tuning along the manifold; right, state tuning along the same manifold. The
entire task manifold is composed of goal-progress subloops. h) Quantifications
of distances along the 3-dimensional UMAP-derived manifold - across different
states and opposite goal-progress bin (left), across different states but for the
same goal-progress bin (middle) or the distances across different states and
same goal-progress bin for a permuted control. N = 4 double-days - T-tests
(two-sided) (with bonferroni correction): Across-goal progress vs within goal-
progress: statistic =6.64, P = 0.021, df = 3; Across-goal progress vs permuted
control: statistic=21.1, P = 7.02 × 10$^{-4}$, df = 3; Within goal-progress vs permuted
control: statistic =10.7, P = 0.005, df = 3 All error bars represent the standard
error of the mean.

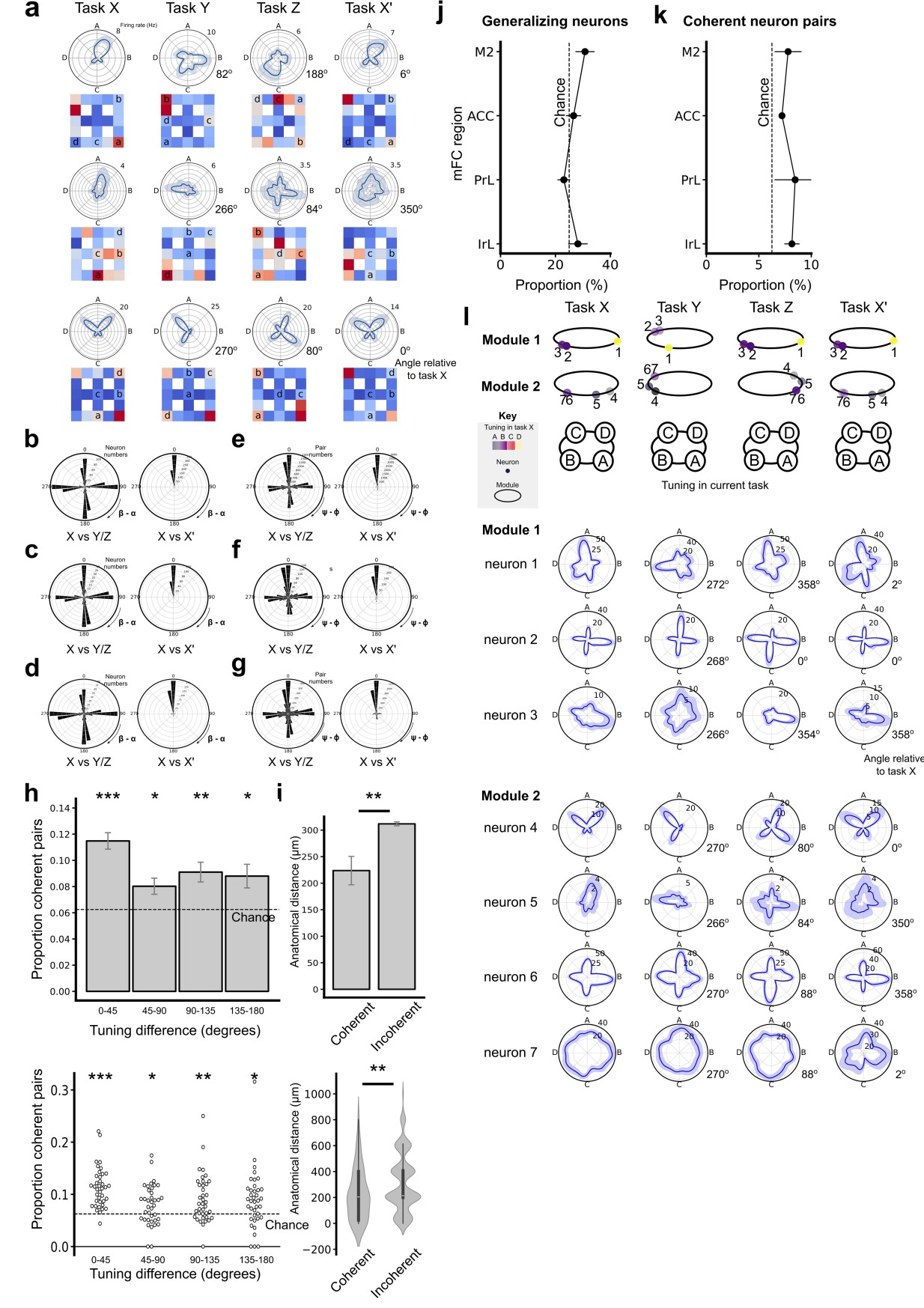

**Extended Data Fig. 4** | See next page for caption.

**Extended Data Fig. 4 | Quasi-coherent task-space remapping of mFC neurons.** a) Three example state-tuned neurons remapping across tasks. The top two neurons remap in a way that is not related to their spatial maps in any given task. The bottom neuron remaps in accordance to its spatial map. Angles (in degrees) of each cell's rotation relative to its tuning in session X are shown to the right of each session's polar plot. Note that these are not all simultaneously recorded neurons. b) Remapping of state neurons that are defined using a stricter threshold (z-score >99th percentile of permuted distribution). Top: A schematic showing how the difference in tuning angles for the same neuron across sessions is quantified. Bottom left: Polar histograms show that state-tuned neurons remap by angles close to multiples of 90 degrees, as a result of conserved goal-progress tuning and the 4 reward structure of the task. No clear peak at zero is seen relative to the other cardinal directions when comparing sessions spanning separate tasks (Two proportions test against a chance level of 25% N = 1061 neurons; mean proportion of generalising neurons across one comparison (mean of X vs Y and X vs Z) = 24%, z = 0.59, P = 0.552. Bottom right: Neurons maintain their state preference across different sessions of the same task (X vs X' Two proportions test against a chance level of 25% N = 770 neurons; proportion generalising=80%, z = 21.5, P = 0.0). c) Remapping when using only state-neurons with concordant remapping angles across two methods (i.e. using the best-rotation analysis method and peak-to-peak changes method). This analysis would for example exclude neuron 3 in Fig. 3a. Left: Polar histograms show that state-tuned neurons remap by angles close to multiples 90 degrees, as a result of conserved goal-progress tuning and the 4 reward structure of the task. No clear peak at zero is seen relative to the other cardinal directions when comparing sessions spanning separate tasks (Two proportions test against a chance level of 25% N = 369 neurons; mean proportion of generalising neurons across one comparison (mean of X vs Y and X vs Z) = 24%, z = 0.41 P = 0.684. Right: state-tuned neurons maintain their state preference across different sessions of the same task (bottom right). Two proportions test against a chance level of 25% N = 240 neurons; proportion generalising=84%, z = 13.0, P = 0.0). d) Remapping of non-spatial neurons. Note that we use the most permissive threshold for spatial tuning here to ensure that we exclude even neurons with weak/residual spatial tuning. Any neuron that had a spatial regression coefficient above the 95th percentile of the null distribution was excluded from this analysis. Left: Polar histograms show that non-spatial state-tuned neurons remap by angles close to multiples 90 degrees, as a result of conserved goal-progress tuning and the 4 reward structure of the task. No clear peak at zero is seen relative to the other cardinal directions when comparing sessions spanning separate tasks (Two proportions test against a chance level of 25% N = 704 neurons; mean proportion of generalising neurons across one comparison (mean of X vs Y and X vs Z) = 22%, z = 1.19 P = 0.233). Right: Non-spatial state-tuned neurons maintain their state preference across different sessions of the same task (bottom right). Two proportions test against a chance level of 25% N = 507 neurons; proportion generalising=68%, z = 13.9, P = 0.0). e) Pairwise coherence of state neurons that are defined using a stricter threshold (z-score >99th percentile of permuted distribution). Top: A schematic showing how the difference in relative angles between pairs of neurons across sessions is quantified. Bottom left: Polar histograms show that the proportion of coherent pairs of state-tuned neurons (comprising the peak at zero) is higher than chance but less than 100%, indicating that the whole population does not rotate coherently. Two proportions test against a chance level of 25% N = 17671 pairs; mean proportion of coherent neurons across one comparison (mean of X vs Y and X vs Z) = 29%, z = 8.9, P = 0.0). Bottom right: As expected from panel b, the large majority of state-tuned neurons keep their relative angles across sessions of the same task (X vs X'; Two proportions test against a chance level of 25% N = 11716 pairs; proportion coherent=64%, z = 59.3, P = 0.0). f) Coherence of state-neuron pairs using only state-neurons with concordant remapping angles across two methods (i.e. using the best-rotation analysis method and peak-to-peak changes method). Left: Polar histograms show that the proportion of coherent pairs of state-tuned neurons (comprising the peak at zero) is higher than chance but far from 1, indicating that the whole population does not rotate coherently (Two proportions test against a chance level of 25% N = 1642 pairs; mean proportion of coherent neurons across one comparison (mean of X vs Y and X vs Z = 30%, z = 3.32, P = 9.04 × 10⁻⁴). Right: As expected from panel b, the large majority of state-tuned neurons keep their relative angles across sessions of the same task (X vs X'; Two proportions test against a chance level of 25% N = 657 pairs; proportion coherent=72%, z = 17.1, P = 0.0). g) Coherence of non-spatial neuron pairs. Note that we use the most permissive threshold for spatial tuning here to ensure that we exclude even neurons with weak/residual spatial tuning. Any neuron that had a spatial regression coefficient above the 95th percentile of the null distribution was excluded from this analysis. Left: Polar histograms show that the proportion of coherent pairs of non-spatial state-tuned neurons (comprising the peak at zero) is higher than chance but far from 1, indicating that the whole population does not rotate coherently (Two proportions test against a chance level of 25% N = 6996 pairs; mean proportion of coherent neurons across one comparison (mean of X vs Y and X vs Z = 30%, z = 3.49, P = 4.74 × 10⁻⁴). Right: As expected from panel b, the large majority of non-spatial state-tuned neurons keep their relative angles across sessions of the same task (X vs X'; Two proportions test against a chance level of 25% N = 4822 pairs; proportion coherent=54%, z = 29.2, P = 0.0). h) Proportion of coherent pairs per recording day (pairs of state-tuned neurons where the relative angle doesn't change by more than 45 degrees across *both* X to Y and X to Z comparisons) relative to all pairs across different pairwise task space angles. T-tests (two-sided) with Bonferroni correction against chance level of 1/16 (probability of neuron pair rotating coherently across two comparisons (i.e. 1/4²)): N = 38 recording days, pairwise circular distance difference: 0-45 degrees statistic=8.17, P = 3.33 × 10⁻⁹, df = 37; 45-90 degrees statistic=2.84, P = 0.013, df = 37; 90-135 degrees statistic=3.88, P = 0.001, df = 37; 135-180 degrees statistic=2.89, P = 0.013, df = 37. Top: bar graph, bottom: individual points (recording days). i) Coherent pairs are slightly closer anatomically than incoherent pairs. Mann-Whitney U-test (Two-sided): N = 3567 pairs (53 coherent; 3514 incoherent), statistic=72872, P = 0.006. Note that, to minimise the effect of noise, this analysis uses only double days and only considers a pair of neurons coherent if they show perfect coherence across all combinations of 6 tasks. Top: bar graph, bottom: kernel density estimate of data distribution. j) The subregional distribution of single neuron generalisation (averaged across X vs Y and X vs Z comparisons) along the medial wall of frontal cortex in neuropixels recordings. One-way ANOVA: F = 1.59, P = 0.323, df = 3. k) The subregional distribution of neuron pair coherence. Coherence is calculated across both X vs Y and X vs Z comparisons along the medial wall of frontal cortex in neuropixels recordings: One-way ANOVA: F = 4.76, P = 0.083, df = 3. l) Top: Visualisation of tuning relationships between two clusters computed in a single recording day. Each dot is a neuron (numbered in correspondence to the polar plots below) and each ring is a cluster derived from the analysis in panel d. The colour code represents the tuning of the neurons in task X. The x,y position defines the tuning in each task. The z position corresponds to cluster ID. Note that the ordering along the z axis is arbitrary. Neurons rotate (remap) in task space while maintaining their within-cluster tuning relationships but not cross-cluster relationships across tasks. Bottom: polar plots for all of the (seven) neurons assigned to each of the two clusters in the above plot. Angles (in degrees) of each cell's rotation relative to its tuning in session X are shown to the right of each session's polar plot. All error bars represent the standard error of the mean.

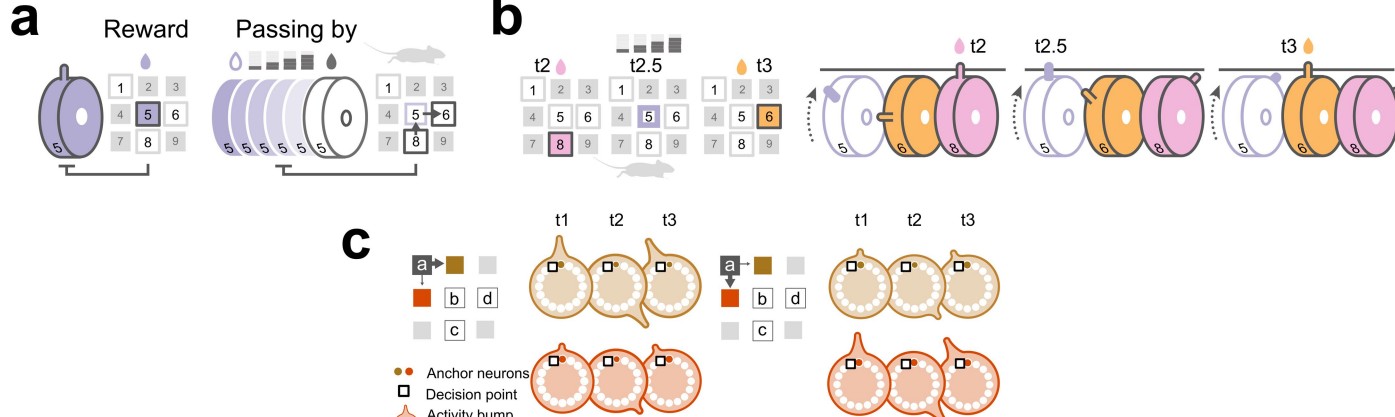

**Extended Data Fig. 5 | The Structured Memory buffers model predicts behavioural sequences.** a) As well as rings tracking task-progress from behavioural steps involving a rewarded place (a conjunction of a place with early goal-progress), there are also rings tracking task-progress from places conjoined with intermediate and late goal-progress. The anchors of these rings are activated when the animal passes through a location, not when it is rewarded, but at a defined, non-zero progress percentage relative to the upcoming goal. b) Non-zero goal-progress anchored rings (e.g. purple outline) allow tracking task-progress from behavioural steps in between two goals. Hence, across all rings, a history of the entire sequence of steps taken by the animal, not just the sequence of reward locations, is encoded at any one point in time. c) Schematic showing distal prediction of an animal's choices from memory buffers. When the animal visits a goal-progress/place (t = 1) in trial N, a bump of activity is initiated in the memory buffer that is anchored to this goal-progress/place. The anchor is location 2 at intermediate goal progress

(brown) in the top memory buffer, and location 4 at intermediate goal progress (red) in the bottom memory buffer. This bump travels around the buffer (e.g. t = 2), paced by progress in the task. When the activity bump circles back to a point close to the anchor (t = 3), it can be read out to bias the animal to return back to the same goal-progress/place in trial N + 1 that was visited in the same task state in trial N. This read-out time defines a "decision point" that is specific for each memory buffer. Left: If, at t = 3 in the example given, the bump on the buffer anchored to intermediate goal-progress in location 2 (brown square) is larger than that for the other option (intermediate goal-progress in location 4; red square) the animal will choose location 2. Right: Location 4 (red square) is chosen if the bump anchored to intermediate goal-progress in location 4 is larger at t = 3. This choice could have been predicted from the bump sizes at an earlier time point (e.g. t = 2) as the bump size will remain highly stable for the duration of a single trial, hence allowing distal prediction of choices from the memory buffers. Reproduced/adapted with permission from Gil Costa.

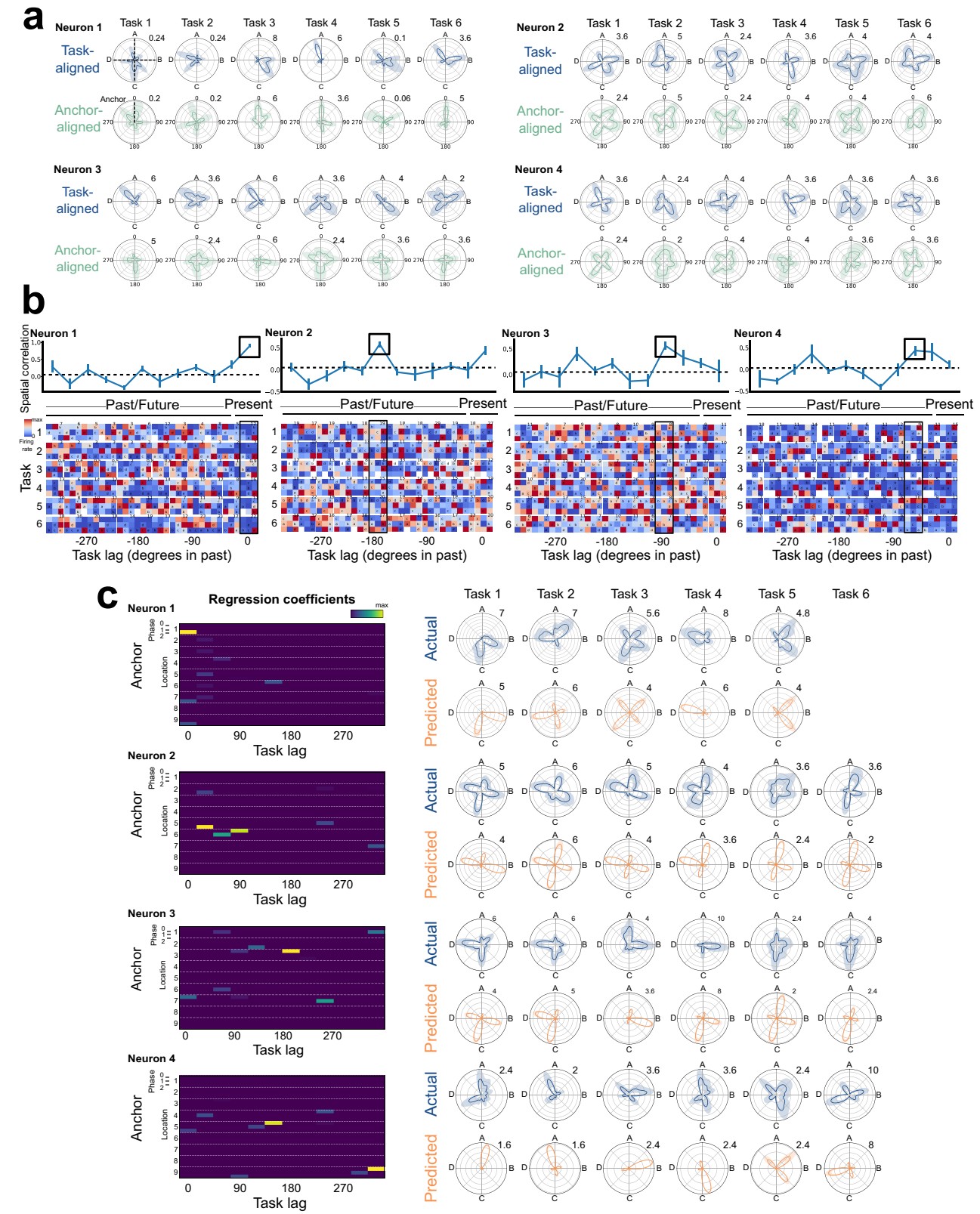

**Extended Data Fig. 6** | See next page for caption.

**Extended Data Fig. 6 | Example anchored mFC neurons.** a) Single anchor alignment analysis. Top (blue) plots for each neuron shows activity aligned by the abstract states (with the dashed vertical line at zero representing reward *a*, i.e. the start of state A; going clockwise, the remaining dashed lines represent reward locations *b*, *c* and *d*, and hence the starts of states B C and D respectively). Neurons appear to remap in task space across tasks. Bottom (green) plots for each cell show that it is possible to find a goal-progress/place conjunction (behavioural step) that consistently aligns neurons across tasks. This behavioural step is therefore said to "anchor" the neuron. Note that the zero line corresponds to visits to the goal-progress/place anchor. b) Lagged spatial field analysis. Example plots showing spatial maps for 4 neurons. Each row represents a different task and each column a different lag in task space. Bottom: Activity of each neuron is plotted as a function of the animal's current location (far right column for each cell) and at successive task space lags in the past for the remaining columns. Because of the circular nature of the task, past bins at lag X are equivalent to future bins at lag 360-X. Top: the correlation of spatial maps across tasks at each lag. To avoid confounds due to goal-progress tuning, all firing rates are calculated only in each neuron's preferred goal-progress bin (i.e. one-third of the entire session). Colours are normalised per map to emphasise the spatial firing pattern, with maximum firing rates (in Hz) displayed at the top right of each map. c) Regression analysis reveals neurons with activity fields lagged in task space from a given goal-progress/place anchor (bottom three neurons), alongside neurons directly tuned to a goal-progress/place (top neuron). The regression coefficients are shown on the left with the actual (blue) and predicted (orange) activity of the neurons shown on the right. All error bars represent the standard error of the mean.

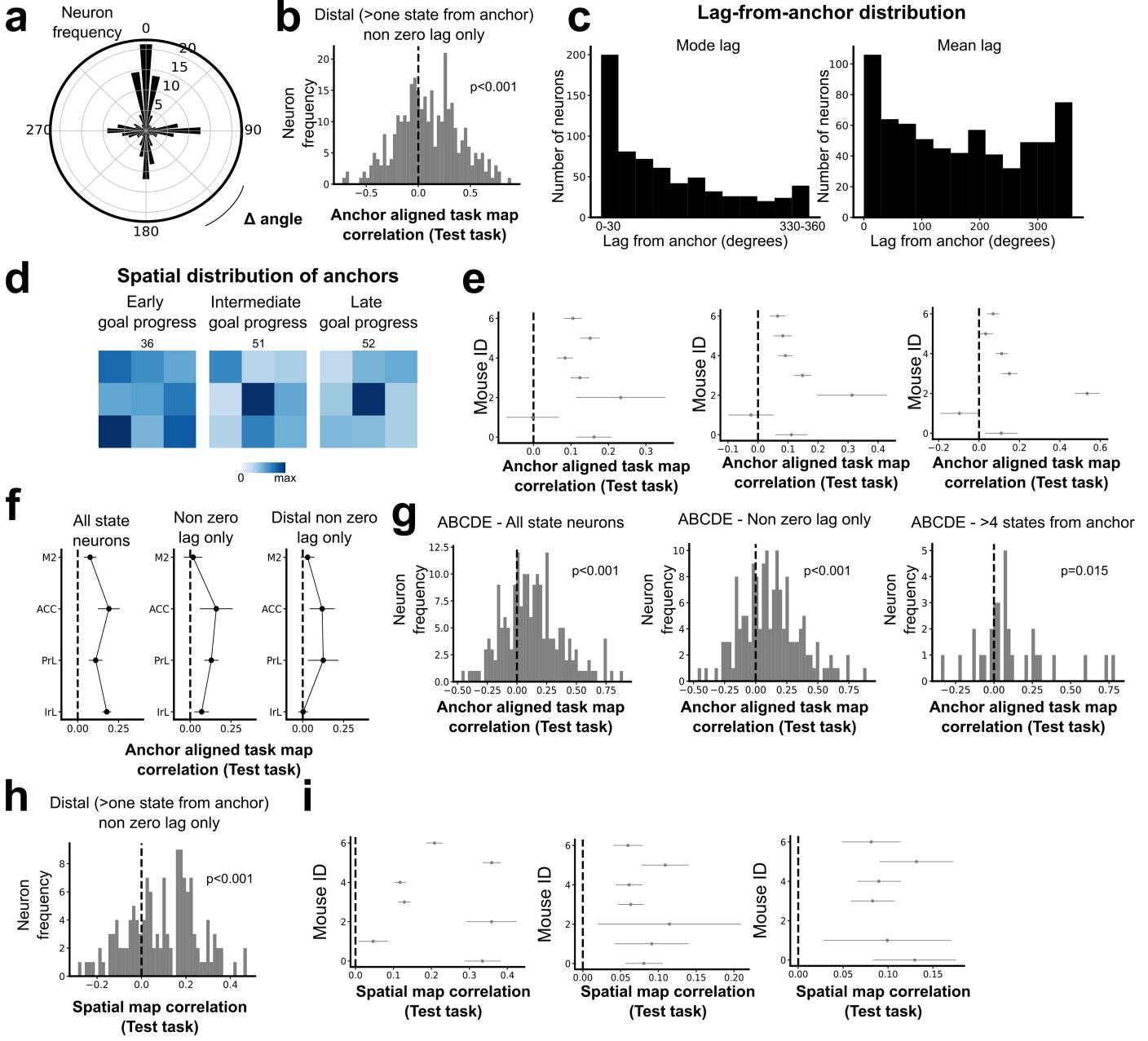

**Extended Data Fig. 7** | See next page for caption.

**Extended Data Fig. 7 | Anchoring analysis using single anchor fitting and lagged spatial map correlations.** a) Polar histogram showing the cross-validated alignment of non-zero-lag neurons by their preferred goal-progress/place (calculated from training tasks) in a left-out test task. Only neurons with a lag of 90 degrees (one state) or more either side of their anchor are shown. Two proportions test against chance (25%): Proportion generalising=35.9%; N = 305 neurons, z = 2.92, P = 0.004 b) Histogram showing the right shifted distribution of the mean cross-validated task map correlations between neurons aligned to their preferred goal-progress/place anchor (from training tasks) and the task map aligned to this goal-progress/place from a left out test task for only non-zero-lag state-tuned neurons with a lag of 90 degrees or more either side of their anchor. T-test (two-sided) against 0: N = 305 neurons, statistic=5.78, P = $1.86 \times 10^{-8}$, df = 296 c) Distribution of task space lags from anchor for all consistently anchored state neurons (neurons with the same anchor in >50% of tasks). Left: Using the most common lag from the anchor across tasks; Right: using the (circular) mean lag from anchor across tasks. Both plots show that consistently anchored neurons have lags from their anchor that span the entire range of possible lags. Note that these plots span the entire (4-goal) task space. d) 2D histograms showing spatial distributions of anchors for all consistently anchored state neurons (neurons with the same anchor in >50% of tasks). The colour bar represents the number of neurons anchored to each maze location at each goal-progress (maze repeated 3 times to display results for early, intermediate and late goal-progress anchors). The maximum number per bin is displayed above each histogram. The plot shows that such consistently anchored cells are anchored to all possible goal-progress/place combinations. e) Mean, per mouse distribution of cross-validated task map correlations between neurons aligned to their preferred goal-progress/place anchor (from training tasks) and the task map aligned to this anchor from a left out test task for: Left: all state-tuned neurons; Middle: non-zero-lag state-tuned neurons (30 degrees or more away from anchor); Right: distal non-zero-lag state-tuned neurons (90 degrees or more away from anchor). One-sided binomial test against chance (chance being mean values equally likely to be above or below 0): All neurons: 6/7 mice with mean positive correlation P = 0.063; Non-zero-lag neurons: 6/7 mice with mean positive correlation P = 0.063 Distal Non-zero-lag neurons: 6/7 mice with mean positive correlation P = 0.063). f) The subregional distribution of cross-validated task map correlations between neurons aligned to their preferred goal-progress/place anchor (from training tasks) and the task map aligned to this anchor from a left out test task along the medial wall of frontal cortex in neuropixels recordings. One-way ANOVA: Left: All state neurons: F = 1.44, P = 0.302, df = 3; Middle: Non-zero lag state neurons: F = 0.92, P = 0.573, df = 3; Right: Distal (>90 degrees from anchor) non-zero lag state neurons F = 0.89, P = 0.485, df = 3. g) Histograms showing the right shifted distribution of mean cross-validated task map correlations between neurons aligned to their preferred goal-progress/place anchor (from training tasks) and the task map aligned to this goal-progress/place from a left out test task in ABCDE tasks. This correlation is shown for all state-tuned neurons (left), non-zero-lag state neurons (middle) and neurons with a lag of more than 4-states from the anchor (right). T-test (two-sided) against 0: All state neurons: N = 188 neurons, statistic=7.21, P = $1.38 \times 10^{-11}$, df = 187; Non-zero-lag state neurons: N = 153 neurons, statistic=6.32, P = $2.47 \times 10^{-9}$, df = 152; >4-state lag from anchor neurons: N = 31 neurons, statistic=2.59, P = 0.015, df = 30 h) Histogram showing the right shifted distribution of the mean cross-validated spatial correlations between spatial maps at the preferred lag in training tasks and the spatial map at this lag from a left out test task for only non-zero-lag state-tuned neurons with spatial correlation peaks a whole state (90 degrees) or further either side of zero-lag. T-test (two-sided) against 0: N = 135 neurons, statistic=7.07, P = $7.93 \times 10^{-11}$, df = 134. i) Mean, per mouse distribution of cross-validated spatial correlations between spatial maps at the preferred lag (from training tasks) and the spatial map at this lag from a left out test task for: Left: All state-tuned neurons; Middle: non-zero-lag state-tuned neurons (30 degrees or more away from anchor); Right: distal non-zero-lag state-tuned neurons (90 degrees or more away from anchor). One-sided binomial test against chance (chance being mean values equally likely to be above or below 0): All state-tuned neurons: 7/7 mice with mean positive correlation P = 0.008; Non-zero lag state-tuned neurons: 7/7 mice with mean positive correlation P = 0.008; Distal Non-zero lag state-tuned neurons: 6/6 mice with mean positive correlation P = 0.016). All error bars represent the standard error of the mean.

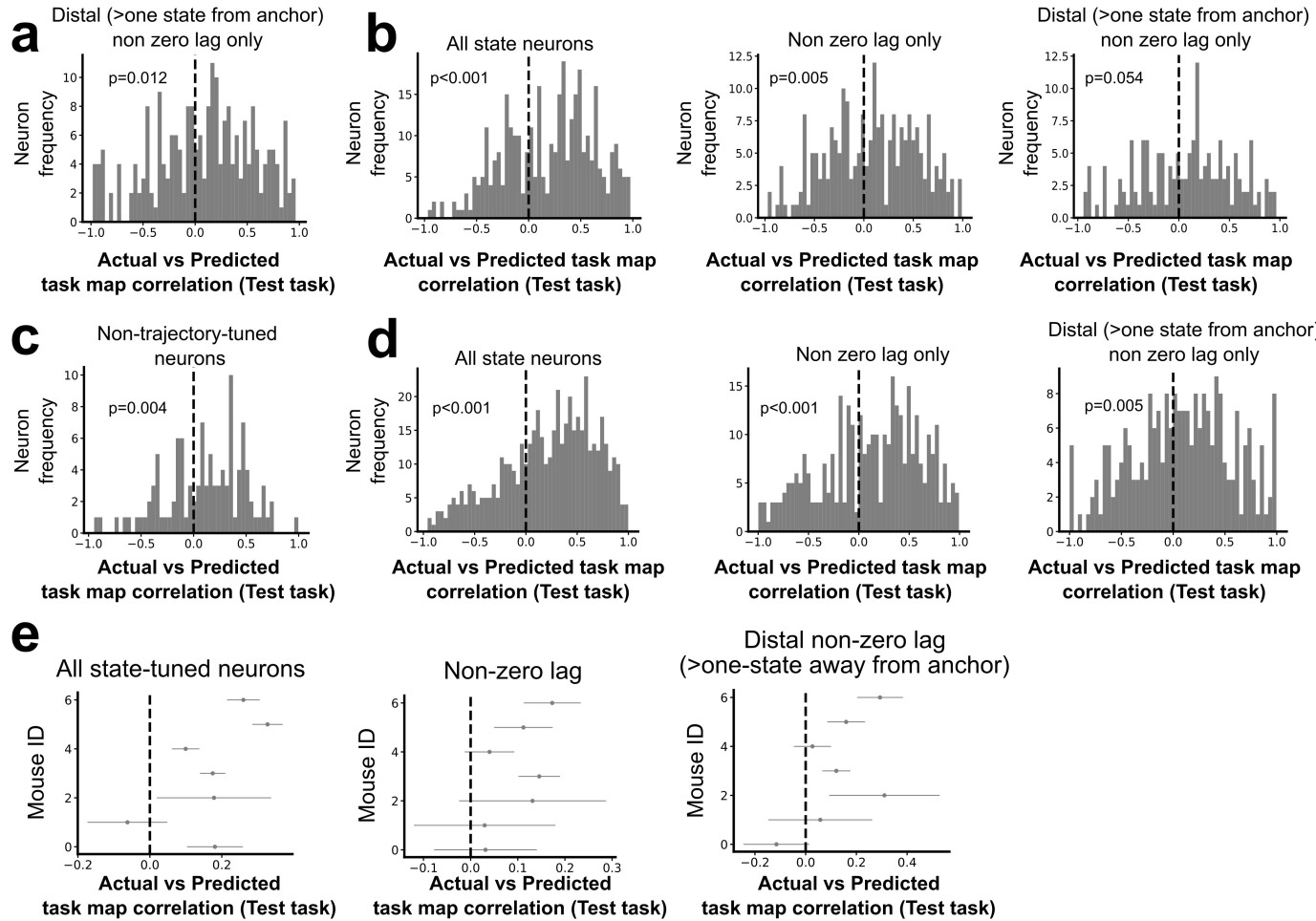

**Extended Data Fig. 8 | Anchoring analysis using linear and non-linear models.** a) Histogram showing the right shifted distribution of mean cross-validated correlation values between model-predicted (from training tasks) and actual activity (from a left out <u>test task</u>) for only non-zero lag state-tuned neurons with the maximum regression coefficient value a whole state (90 degrees) or more either side of the anchor. To avoid contamination due to potential residual spatial-tuning, only regression coefficient values more than 90 degrees in task space either side of the anchor point are used for the prediction. T-test (two-sided) against 0: N = 224 neurons, statistic=2.53, P = 0.012, df = 223. b) Histograms showing the right-shifted distribution of mean cross-validated correlation values between model-predicted (from training tasks) and actual activity (from a left out test task) for <u>state neurons that are defined using a stricter threshold</u> (z-score >99th percentile of permuted distribution). Left: this correlation is shown for all state-tuned neurons; Middle: only state-tuned neurons with non-zero-lag firing from their anchors; and Right: non-zero lag state-tuned neurons with the maximum regression coefficient value a whole state (90 degrees) or more either side of the anchor. T-test (two-sided) against 0: All state-tuned neurons N = 349 neurons, statistic=8.70, P = 1.34 × 10$^{-16}$, df = 348; Non-zero lag state-tuned neurons N = 227 neurons, statistic=2.83, P = 0.005, df = 226; distal (90 degrees) non-zero lag neurons: N = 154 neurons, statistic=1.94, P = 0.054, df = 153. c) Histogram showing the right shifted distribution of mean cross-validated correlation values between model-predicted (from training tasks) and actual activity (from a left out test task) for <u>state neurons that are not tuned to the animal's current trajectory</u>. Note that we use a permissive threshold for trajectory tuning here to ensure we exclude any neurons with even weak/

residual tuning for trajectory. Any neuron that had a trajectory regression coefficient above the 95th percentile of the null distribution was excluded from this analysis. T-test (two-sided) against 0: N = 112 neurons, statistic=4.27, P = 4.13 × 10$^{-5}$, df = 111). d) Histograms showing the right-shifted distribution of mean cross-validated correlation values between model-predicted (from training tasks) and actual activity (from a left out test task) for <u>state neurons using a Poisson regression model</u>. Left: this correlation is shown for all state-tuned neurons; Middle: only state-tuned neurons with non-zero-lag firing from their anchors; and Right: non-zero lag state-tuned neurons with the maximum regression coefficient value a whole state (90 degrees) or more either side of the anchor. T-test (two-sided) against 0: All state-tuned neurons N = 489 neurons, statistic=10.7, P = 2.86 × 10$^{-24}$, df = 488; Non-zero lag state-tuned neurons N = 346 neurons, statistic=4.74, P = 3.09 × 10$^{-6}$, df = 345; distal (90 degrees) non-zero lag neurons: N = 229 neurons, statistic=2.81, P = 0.005, df = 228. e) Mean, per mouse distribution of cross-validated correlation values between model-predicted (from training tasks) and actual activity (from a left out <u>test task</u>) for: Top: All state-tuned neurons, Middle: non-zero lag state-tuned neurons (30 degrees or more away from anchor), Bottom: distal non-zero lag state-tuned neurons (90 degrees or more away from anchor). One-sided binomial test against chance (chance being mean values equally likely to be above or below 0): All state-tuned neurons: 6/7 mice with mean positive correlation P = 0.063; Non-zero lag state-tuned neurons: 7/7 mice with mean positive correlation P = 0.008; Distal Non-zero-lag state-tuned neurons: 6/7 mice with mean positive correlation P = 0.063). All error bars represent the standard error of the mean.

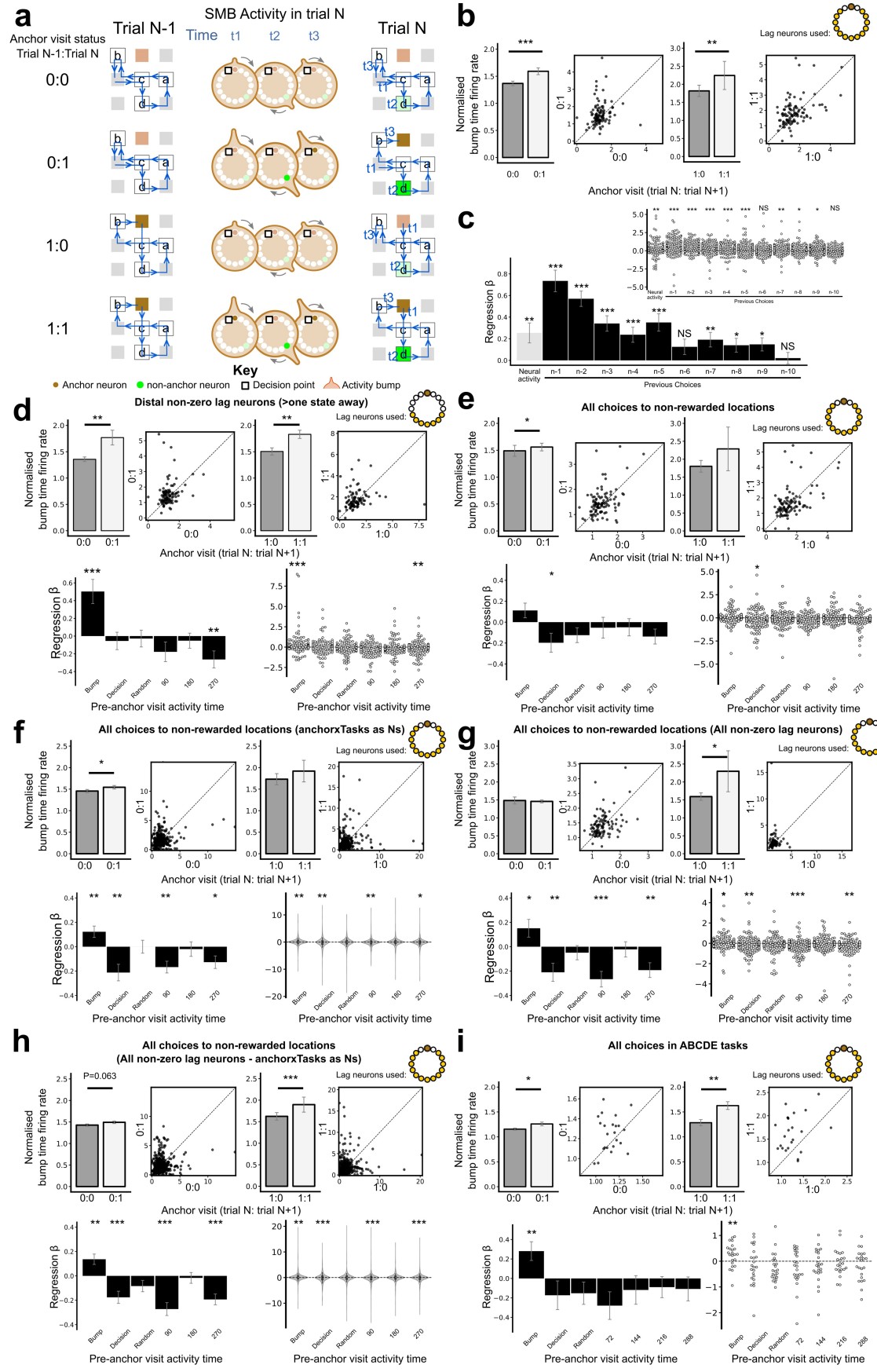

**Extended Data Fig. 9** | See next page for caption.

i.e. excluding choices to reward locations using all non-zero lag neurons (i.e. not only consistently anchored ones as in all other plots) and using Anchor x Task as Ns. Top: Normalised firing rates of neurons during their "bump time": i.e. the lag at which they are active relative to the anchor. Bump time activity is higher before visits to the neuron's anchoring behavioural step in trial $N + 1$ when the anchoring behavioural step was visited in trial N (Top right) and showed a trend towards being higher when the anchor wasn't visited in trial N (Top left). Anchoring behavioural step not visited in trial N: n = 1273 anchor-tasks, statistic=380465, P = 0.063. Anchoring behavioural step visited in trial N: n = 826 anchor-tasks, statistic=141977, P = $8.36 \times 10^{-5}$. In addition, an ANOVA on all data (N = 826 anchor-tasks) showed a main effect of Past: F = 4.66, P = 0.031, df1 = 1, df2 = 821, a main effect of Future: F = 6.65, P = 0.010, df1 = 1, df2 = 821, a Past x Future interaction: F = 5.38, P = 0.021, df1 = 1, df2 = 821. Bottom left: A logistic regression showed coefficients were significantly positive for "bump time" but not other times. T-tests (two-sided) against 0: "bump time": N = 1278 anchor-tasks, statistic=3.23, P = 0.001, df = 1277; "decision time": N = 1278 anchor-tasks, statistic = −3.52, P = $4.39 \times 10^{-4}$, df = 1277; "random time": N = 1278 anchor-tasks, statistic = −1.79, P = 0.073, df = 1277; "90 degree shifted time": N = 1278 anchor-tasks, statistic = −5.16, P = $2.92 \times 10^{-7}$, df = 1277; "180 degree shifted time": N = 1278 anchor-tasks, statistic = −0.37, P = 0.709, df = 1277; "270 degree shifted time": N = 1278 anchor-tasks, statistic = −4.37, P = $1.36 \times 10^{-5}$, df = 1277. Bottom right: Kernel density estimate plots showing distribution of regression coefficient values across groups. i) Prediction of behaviour in the ABCDE tasks. Top: Normalised firing rates of neurons during their "bump time": i.e. the lag at which they are active relative to the anchor. Bump time activity is higher before visits to the neuron's anchor in trial $N + 1$ when the anchoring behavioural step was not visited in trial N (Top left) and also higher before visits to anchoring behavioural step in trial $N + 1$ when the anchoring behavioural step was visited in trial N (Top right). Wilcoxon tests (two-sided): Anchoring behavioural step not visited in trial N: n = 24 tasks, statistic=73, P = 0.027. Anchoring behavioural step visited in trial N: n = 24 tasks, statistic=48, P = 0.003. In addition, an ANOVA on all data (N = 24 tasks) showed a main effect of Past: F = 18.57, P = $2.61 \times 10^{-4}$, df1 = 1, df2 = 23, a main effect of Future: F = 19.9, P = $1.78 \times 10^{-4}$, df1 = 1, df2 = 23, a Past x Future interaction: F = 5.84, P = 0.024, df1 = 1, df2 = 23. Bottom left: A logistic regression showed coefficients were positive for the bump time but not all other control times. T-tests (two-sided) against 0: "bump time": N = 24 tasks, statistic=2.91, P = 0.008, df = 23; "decision time": N = 24 tasks, statistic = −1.17, P = 0.252, df = 23; "random time": N = 24 tasks, statistic = −1.33, P = 0.197, df = 23; "72 degree shifted time": N = 24 tasks, statistic = −1.97, P = 0.061, df = 23; "144 degree shifted time": N = 24 tasks, statistic = −0.8, P = 0.43, df = 23; "216 degree shifted time": N = 24 tasks, statistic = −0.83, P = 0.417, df = 23; "288 degree shifted time": N = 24 tasks, statistic = −0.88, P = 0.390, df = 23. Bottom right: Swarm plots showing distribution of regression coefficient values across groups. All error bars represent the standard error of the mean.

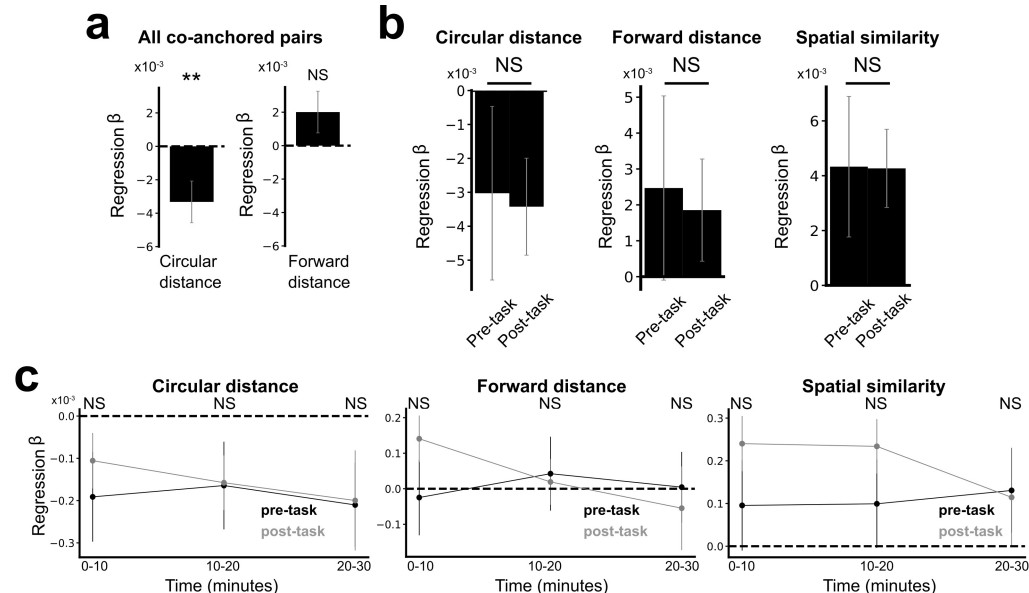

**Extended Data Fig. 10 | Internally organised mFC activity is consistent across sleep epochs.** a) Regression coefficient values for pairwise circular (left) or forward (right) distance regressed against sleep cross-correlation for co-anchored neurons - T-test (two-sided) relative to 0: circular distance: N = 430 pairs, t = −2.66 P = 0.008, df = 429; forward distance: N = 430 pairs, t = 1.61 P = 0.108, df = 429. b) No significant differences are seen when comparing regression coefficients for circular distance, forward distance and spatial similarity between co-anchored pairs of neurons between pre- and post-task sleep across the whole-session. Comparison of regression coefficients (from GLM in Fig. 7a) for left: circular distance: Welch's One-sided T-test: t = 0.14, P = 0.892, df = 919), middle: forward distance: Welch's One-sided T-test: t = 0.21, P = 0.834, df = 919) and right: spatial similarity: Welch's One-sided T-test: t = 0.02, P = 0.983, df = 919). N = 521 pairs (pre-task sleep) 399 pairs (post-task sleep). c) Comparison between pre- and post-task sleep across different time epochs since the beginning of sleep sessions for regression coefficients of circular distance (left), forward distance (middle) and spatial

similarity (right). Unpaired T-test (Welch's One-sided T-test) results (with Bonferroni correction of p values): <u>Circular distance:</u> 0-10 min post-sleep: N = 512 (pre-task) N = 429 (post-task): t = −0.69, P = 0.870, df = 939. 10-20 min post-sleep: N = 512 (pre-task) N = 429 (post-task): t = −0.05, P = 0.997, df = 939. 20-30 min post-sleep: N = 512 (pre-task) N = 429 (post-task): t = −0.07, P = 0.997, df = 939. <u>Forward distance:</u> 0-10 min post-sleep: N = 512 (pre-task) N = 429 (post-task): t = −1.33, P = 0.459, df = 939. 10-20 min post-sleep: N = 512 (pre-task) N = 429 (post-task): t = 0.19, P = 0.91, df = 939. 20-30 min post-sleep: N = 512 (pre-task) N = 429 (post-task): t = 0.38, P = 0.91, df = 939. <u>Spatial similarity:</u> 0-10 min post-sleep: N = 512 (pre-task) N = 429 (post-task): t = −1.17, P = 0.567, df = 939. 10-20 min post-sleep: N = 512 (pre-task) N = 429 (post-task): t = −1.11, P = 0.567, df = 939. 20-30 min post-sleep: N = 512 (pre-task) N = 429 (post-task): t = 0.11, P = 0.915, df = 939. All error bars represent the standard error of the mean, analytically derived for each regression. As such there are no individual points to report.

                             Tim Behrens

# Reporting Summary

## Statistics

For all statistical analyses, confirm that the following items are present in the figure legend, table legend, main text, or Methods section.

| n/a | Confirmed | |
|---|---|---|
| ☐ | ☒ | The exact sample size (*n*) for each experimental group/condition, given as a discrete number and unit of measurement |
| ☐ | ☒ | A statement on whether measurements were taken from distinct samples or whether the same sample was measured repeatedly |
| ☐ | ☒ | The statistical test(s) used AND whether they are one- or two-sided *Only common tests should be described solely by name; describe more complex techniques in the Methods section.* |
| ☐ | ☒ | A description of all covariates tested |
| ☐ | ☒ | A description of any assumptions or corrections, such as tests of normality and adjustment for multiple comparisons |
| ☐ | ☒ | A full description of the statistical parameters including central tendency (e.g. means) or other basic estimates (e.g. regression coefficient) AND variation (e.g. standard deviation) or associated estimates of uncertainty (e.g. confidence intervals) |
| ☐ | ☒ | For null hypothesis testing, the test statistic (e.g. *F*, *t*, *r*) with confidence intervals, effect sizes, degrees of freedom and *P* value noted *Give P values as exact values whenever suitable.* |
| ☒ | ☐ | For Bayesian analysis, information on the choice of priors and Markov chain Monte Carlo settings |
| ☒ | ☐ | For hierarchical and complex designs, identification of the appropriate level for tests and full reporting of outcomes |
| ☐ | ☒ | Estimates of effect sizes (e.g. Cohen's *d*, Pearson's *r*), indicating how they were calculated |

*Our web collection on statistics for biologists contains articles on many of the points above.*

## Software and code

Policy information about availability of computer code

| Data collection | Open source pyControl v1.2 (behaviour) and OpenEphys v0.4.3 (electrophysiology) software was used for data collection |
|---|---|
| Data analysis | Preprocessing was carried out using Kilosort (versions 2.5 and 3) for spike sorting and DeepLabCut (2.0) a Python package for marker-less pose estimation based in the TensorFlow machine learning library for tracking animals in videos. HERBS software (0.2.8) was used to localise the silicon probes to anatomical locations on the mouse brain. All data was analysed using Python (3) code. This used custom-made code but made use of libraries such as numpy (1.22.0), scipy (1.10.1), matplotlib (3.7.3), sciKit learn (1.3.2), pandas (2.0.3) and seaborn (0.13.2). Code used in this study is publicly available here: https://github.com/mohamadyelgaby/mFC_schema |

For manuscripts utilizing custom algorithms or software that are central to the research but not yet described in published literature, software must be made available to editors and reviewers. We strongly encourage code deposition in a community repository (e.g. GitHub). See the Nature Portfolio guidelines for submitting code & software for further information.

## Data

Policy information about availability of data

All manuscripts must include a data availability statement. This statement should provide the following information, where applicable:

- Accession codes, unique identifiers, or web links for publicly available datasets
- A description of any restrictions on data availability
- For clinical datasets or third party data, please ensure that the statement adheres to our policy

Data used in this study is publicly available here (via OSF): https://doi.org/10.17605/OSF.IO/3D9R2
Details of the design and material of all maze components can be found in this open source link:
https://github.com/pyControl/hardware/tree/master/GridMaze
The mouse brain atlas (http://labs.gaidi.ca/mouse-brain-atlas/) was used for checking implantation coordinates.

## Research involving human participants, their data, or biological material

Policy information about studies with human participants or human data. See also policy information about sex, gender (identity/presentation), and sexual orientation and race, ethnicity and racism.

| | |
|---|---|
| Reporting on sex and gender | NA |
| Reporting on race, ethnicity, or other socially relevant groupings | NA |
| Population characteristics | NA |
| Recruitment | NA |
| Ethics oversight | NA |

Note that full information on the approval of the study protocol must also be provided in the manuscript.

# Field-specific reporting

Please select the one below that is the best fit for your research. If you are not sure, read the appropriate sections before making your selection.

☒ Life sciences          ☐ Behavioural & social sciences          ☐ Ecological, evolutionary & environmental sciences

For a reference copy of the document with all sections, see nature.com/documents/nr-reporting-summary-flat.pdf

# Life sciences study design

All studies must disclose on these points even when the disclosure is negative.

| | |
|---|---|
| Sample size | No statistical methods were used to pre-determine sample sizes but our sample sizes are similar to those reported in previous publications. e.g. Zhou et al 2021 (https://doi.org/10.1038/s41586-020-03061-2) |
| Data exclusions | Of the 13 animals, 7 animals in total were used for electrophysiological recordings, the remaining 6 animals are accounted for below:<br>● 3 animals (in cohort 1) were not implanted at any point<br>● 1 animal was implanted with silicon probes but was part of the first cohort so did not get to the 3 task days (i.e. only completed the first 10 tasks)<br>● 2 animals were implanted but their signal was lost before the 3 task days |
| Replication | All Behavioural and neuronal results were replicated across 4 cohorts of 13 mice. |
| Randomization | The sequences of tasks that the animals experienced throughout the experiment were randomised and counterbalanced. |
| Blinding | Only one experimental group (silicon probe implant in the frontal cortex) was used in this study and so blinding was not necessary. |

# Reporting for specific materials, systems and methods

We require information from authors about some types of materials, experimental systems and methods used in many studies. Here, indicate whether each material, system or method listed is relevant to your study. If you are not sure if a list item applies to your research, read the appropriate section before selecting a response.

## Materials & experimental systems

| n/a | Involved in the study |
|---|---|
| ☒ | ☐ Antibodies |
| ☒ | ☐ Eukaryotic cell lines |
| ☒ | ☐ Palaeontology and archaeology |
| ☐ | ☒ Animals and other organisms |
| ☒ | ☐ Clinical data |
| ☒ | ☐ Dual use research of concern |
| ☒ | ☐ Plants |

## Methods

| n/a | Involved in the study |
|---|---|
| ☒ | ☐ ChIP-seq |
| ☒ | ☐ Flow cytometry |
| ☒ | ☐ MRI-based neuroimaging |

## Animals and other research organisms

Policy information about studies involving animals; ARRIVE guidelines recommended for reporting animal research, and Sex and Gender in Research

| | |
|---|---|
| Laboratory animals | 13 male C57BL/6 mice (Charles River Laboratories) aged 2-5 months at the time of testing.  Animals were housed on a 12 hour light/dark cycle (7am-7pm) at 20-24°c and humidity of 55±10%. |
| Wild animals | No wild animals were used in the study. |
| Reporting on sex | Only male animals were used in the study as the weight of the implant used for recording precluded using females. |
| Field-collected samples | No field corrected samples were used in the study. |
| Ethics oversight | Experiments were carried out in accordance with Oxford University animal use guidelines and performed under UK Home Office Project Licence P6F11BC25. |

Note that full information on the approval of the study protocol must also be provided in the manuscript.

## Plants

| | |
|---|---|
| Seed stocks | NA |
| Novel plant genotypes | NA |
| Authentication | NA |

