## [Peer Review File · Nature]

Manuscript Title: A Cellular Basis for Mapping Behavioural Structure

Reviewer Comments & Author Rebuttals

Reviewer Reports on the Initial Version:

Referees' comments:

Referee #1 (Remarks to the Author):

The paper reveals a deeper understanding of how the brain represents and adapts to complex behavioral sequences. Researchers trained mice on tasks with shared structures but different goals, demonstrating that mice could grasp the underlying abstract structure, allowing them to swiftly adapt to new tasks. Neurons in the medial frontal cortex, termed "goal-progress cells," played a crucial role in this process. These neurons effectively mapped the progress towards goals, generalized their activity across different task variations, and encoded entire sequences of future behavioral steps. This discovery suggests that these goal-progress cells serve as fundamental building blocks for representing complex behavioral structures, potentially offering insights into efficient learning and adaptation mechanisms in the brain.

Moreover, the study highlights the potential reconciliation of previous research on the frontal cortex's role in sequencing complex behaviors and draws parallels between the organization of neurons for behavioral structure mapping and cognitive mapping in the brain. It also proposes that learning involves the gradual transformation of naive goal-progress sequences into structured memory buffers, shedding light on the mechanisms underlying memory and learning in the brain. In summary, the paper's findings provide a better understanding of how the brain maps and adapts to abstract behavioral structures, with implications for cognitive neuroscience and our comprehension of neural learning processes.

The task is well-designed and cleverly elicits zero-shot learning in mice. The analysis is sophisticated and rigorous, although quite complex. Nevertheless, I have several concerns. First, it can be difficult to follow what the authors have done, and often the figures are more of a hindrance than a help. Second, although the end results of the analyses show clear effects, it is not always evident that these effects are supported by the raw data. The "working out" needs to be more clearly shown. Finally, there were several places where I simply could not figure out the analysis. My comments are roughly in order of importance.

1. I think the neuronal tuning shown in figure 5C is critical to support the structured memory buffer model, but it is very difficult to see. The figure is truly awful. It's way too small to determine the spatial tuning of the neurons. (It might help to remove the edges from all these plots. There doesn't seem much point including them when they are removed from all the analyses). However, where I can discern the spatial tuning, it doesn't really seem to line up with the correlation coefficients that the authors are reporting. For example, consider neuron 3 shown in Extended Figure 4. This neuron has a relatively high correlation across tasks approaching 0.8, and yet when I look at the spatial

tuning across the tasks, I'd be hard pressed to tell you where this neuron's place field was as it is so variable. I'd really like to understand the discrepancy between the authors' metrics and the single neuron tuning before I would be comfortable believing the structured memory buffer model. Incidentally, these figures also need a color bar so that we know what kind of firing rates we're dealing with.

2. The authors make the claim that goal progress tiles the entire space, but I'm not entirely convinced by this. I'm not a fan of the kind of plots that they show in Figure 2E, because those kinds of plots can arise simply from noise. That concern is somewhat mitigated by the fact that they're only showing significant neurons, but nevertheless, I'd like to see a statistical analysis that supports the claim, and maybe change the figure so that they are showing where significant selectivity (appropriately corrected for multiple comparisons) occurs over the course of the goal progress. Furthermore, the data in the Supplementary Information seems to paint a more nuanced picture. For example, the data shown in Extended Figure 5D seems to suggest that early and late goal progress is represented more strongly than intermediate goal progress. I don't think this necessarily conflicts with the authors' structured memory buffer model. It may be the more salient parts of the task progress (starts and ends) are represented more strongly in neuronal tuning. However, if this is the case then we ought to be made aware of it.

3. I have some concerns about the statistical analysis throughout the paper. The authors are using a relatively liberal alpha level of 0.05 (at least liberal for assessing neuronal tuning) and it is also not clear that they are correcting for multiple comparisons, for example, for the different task phases that they are testing. I think this means that the selective neuron counts are somewhat inflated, and it would probably behoove the authors to clean this up, as most of the neurons with weak tuning are probably just noise.

4. I'm not sure I understand the data being presented in figure 6. I don't understand how there can be a 1:1 condition. According to the authors' explanation, 1 indicates that the animal visited the anchor location, and 1:1 means that they visited the anchor location on trial N and trial N + 1. In the context of the task though, I don't understand how this is possible. Maybe the animal is making an error and returning to the original location? I don't think that's what is happening though, so I think the authors should do a better job of explaining this figure, or at least explaining how the 1:1 condition happens.

5. It is not always clear how the number of neurons for a given analysis was derived, and it can fluctuate from figure to figure. The authors need to clearly explain how the subset of the neurons that they are testing was selected.

6. I wasn't convinced by the clustering analysis shown in figure 3D. If I'm understanding this analysis correctly there are only 12 possible clusters (the pairwise combinations between A, B, C, and D) so I'm not convinced that this clustering analysis shows anything other than the fact that different neurons are responding to different states across the tasks.

7. I thought that the discussion of the results shown in figure 3 needed a little strengthening. The results in figure 3B show not just that the neurons are remapping, but that the zero-lag position is

significantly below chance. In other words, it's not simply a completely new map when it remaps. The new map is orthogonal to the first map, at least if I'm understanding this analysis correctly.

8. In the methods the authors describe using either analytical or empirical methods for calculating the shortest path transitions, but they don't explain when one is used relative to the other in the manuscript.

9. Figure 2F: this is possibly the most horrific pie chart I've ever seen, and I couldn't make head nor tail of it.

10. With respect to the finding that the authors are better able to predict future choices on the low probability sequences compared to the high probability, stereotyped sequences, the authors might also want to cite Chiang and Wallis (2013). They also found that prefrontal spatial tuning on their sequence task was weaker when the animals were following more stereotyped strategies.

11. Figure 5H: this figure would be more impactful if the circles weren't colored in so that we can see neuron 2's firing.

Referee #2 (Remarks to the Author):

In this work, the authors aim to identify a neuronal algorithm for mapping abstract task structure. Broadly speaking, the study consists of two sets of findings: (1) prelimbic neural activity during a novel "ABCD task" in which freely-moving mice must navigate to looping sequences of 4 water reward locations on a 3x3 maze and learn to generalize this goal-directed behavioral sequence across different permutations of the 4 reward locations, and (2) the "Structured Memory Buffer" (SMB) model—a conceptual framework which attempts to explain how the organization of prelimbic neural activity is shaped by task structure in a manner that facilitates task generalization and retrieval of task-relevant behavioral policies.

Electrophysiological recordings revealed the following findings: (1) ~80% of prelimbic neurons in this study track progress towards individual goals. This "goal-progress" tuning is invariant to the spatial location of rewards and elapsed time, as well as invariant to across distinct permutations of reward locations (termed distinct "tasks"). (2) A subset of these goal-progress tuned neurons exhibit preference for distinct "states"—time periods in between two adjacent reward locations. State preference "remaps" across distinct permutations of reward locations, but is maintained across different sessions of the same permutation separated in time, suggesting that prelimbic neural activity is organized into task-space modules. (3) The firing rate of each module of neurons is "anchored" to a particular conjunction of goal-progress and place. Prelimbic neural activity can be explained by a ring-like state space (as a consequence of the task structure), which is preserved during offline states such as sleep. During task performance, activity evolves as a function of the number of goals obtained, eventually circling back to the anchor point.

The authors set out to test three predictions of the SMB model: (1) prelimbic neurons form “mnemonic fields” that track task-progress from specific goal-progress/place conjunctions; (2) “activity bumps” in prelimbic neural activity can be used to predict the mouse’s upcoming behavioral choice; and (3) pairwise coactivity during offline, sleep states preserve a ring-like neuronal state space observed during the ABCD task.

Overall, the authors have made a commendable effort in designing a complex, freely-moving behavioral task in mice and proposing a conceptual framework for understanding how frontal cortical dynamics mediate abstract schema learning. These features of the work are novel, significant and important. However, in its current state, the study falls short of demonstrating that the proposed “unified model for behavioural schema and sequence memory” is implemented in mouse prefrontal cortex, let alone demonstrating how the identified “goal-progress sequences” may facilitate the “formation of a schema encoding any task-structure in the mFC”.

ABCD Task: I have several concerns regarding how the authors have chosen to interpret data and draw conclusions in light of the constraints introduced by their ABCD task.

1. The decision to consider different sequences as distinct “tasks” requires further rationale. While the mice are performing distinct permutations of sequences, I am not fully convinced this actually represents distinct “tasks” or slight variations of the same task. In addition to citing precedence in the literature (if any), authors should consider providing some quantitative details about how “distinct” the sequences were in practice (beyond the qualitative descriptions provided in the Methods). How distinct or similar were the 3 “tasks” chosen for a given day?
2. The presence of the 2-second auditory tone “when the animal consumed reward” at reward location “a” makes it difficult to interpret what mice have learned as the “abstract task structure” and whether this truly constitutes an abstract, generalized task “schema.” Presumably the auditory tone only starts once the mouse pokes for the reward at location “a” and reward is delivered. But this auditory tone is nevertheless present across all “tasks,” thus serving as an explicit reinforcement of the looping structure. Can the authors provide additional insight about why the auditory tone was included at all, and how they consider its role in the “anchoring” of “goal-progress” cells?
3. It is unclear if, at any point, whether animals were required to collect all 4 rewards in the correct sequence without visiting other reward wells. The main performance metric used in this study appears to be based on paths between individual rewards (i.e. shortest spatial paths between rewards for at least 70% of all transitions). If performance is assessed solely based on individual rewards, it is unclear how authors have assessed that mice have learned a “complex behavioral sequence” (which would seem to refer to the entire sequence of 4 locations, as well as the looping nature). Furthermore, it is unclear how the authors are distinguishing between the individual transitions vs. the overall hierarchy of behavioral structure (composed of the complete structure, rather than between two goals).
4. One key conclusion that the authors draw is that task structure—in this case, task periodicity—“paces” a sequential policy. However, in order to make this mechanistic claim (rather than reporting it as an observation), the authors would need to test this hypothesis more directly, e.g. by altering task periodicity to every 3 or 5 rewards. In addition, it is unclear how this would be relevant for tasks which do not have a periodic structure.

In addition, I have significant concerns regarding the presentation and interpretation of data:

1. Key details regarding the maze are missing. Based on the text, it is unclear which components of the maze are “connected,” as well as the size of the 3x3 grid maze. It would be helpful to include a photograph, or at least, a realistic schematic of the actual maze.
2. The use of schematics is appreciated from the perspective of clarity but the lack of accompanying data makes some points difficult to interpret. For example, the authors should include an actual image of the task (as noted above) and some example trajectories to give a better sense of the behavior of the animals. Figure 1d and 1e present the behavior all collapsed into a few line graphs instead of showing individual animal data, which would better illustrate the degree of stereotypy or variability in the behavior. Individual traces for probe locations are not provided (outside a schematic in Figure 2 and a single image in the Extended Data), I would recommend the authors provide actual histology or probe locations for all of their animals.
3. In order to understand how representative the neural activity is, it would be helpful to have a table summarizing the number of units included from each subject for each analysis, considering the authors use different subsets for different analyses (e.g. using only consistently anchored neurons vs. state-tuned neurons vs. neurons with activity lagged at least 30 degrees from their anchor). Although a breakdown of “total number of neurons” is provided in methods, the authors report pooled values (e.g. “neuron-days”), making it difficult to understand if neural activity is representative or if mainly from a subset of subjects.
4. Methodological details are missing. (a) How were neurons tracked across days? The methods section lacks sufficient detail, and it’s unclear what procedure authors used, if any, to rigorously determine which units were the same across days. Please provide quality metrics, exclusion criteria, and dropout rate for assessing unit matching across the “double-days.” (b) What was the laminar distribution of the prelimbic units in this study? Were units in cingulate cortex excluded? (e.g. based on the image provided in Extended Data Fig. 1, it is unclear whether units were considered across the entire shank, or only post-hoc confirmed to be in prelimbic).
5. A minor point: “mFC” (medial frontal cortex) is not a commonly used term to refer to the prelimbic cortex in rodents (e.g. see review by Laubach et al., *eNeuro*, 2018; PMID 30406193). To avoid confusion, and to facilitate broader impact and reproducibility, authors should refer to the actual brain region surveyed (i.e. prelimbic cortex).

I also have several concerns about how the predictions of the SMB model are presented and validated:

1. SMB model predictions are tested using linear and logistic regression models. It is unclear how the authors have considered nonlinearities stemming from neural activity as well as from the behavioral task itself. In the case of predicting behavioral choices, although the authors state that “previous choices up to 5 trials in the past” were included in the regression model to “remove confounds due to the autocorrelated previous behavioural choices”, there is insufficient detail to determine the extent to which the cyclical nature of previous behavioural choices can itself predict behavioural choices. In the case of neural activity, it is unclear whether the authors have considered

nonlinearities emerging from the local recurrent microcircuitry (e.g. reviewed by XJ Wang, 2001; PMID 11476885).

2. There is a somewhat circular logic to the experiments/analyses, and it would be helpful if the authors can provide a more explicit, clear discussion to address this. For example, in Figure 7, rejecting the “delay line” hypothesis is not necessarily the strongest argument for the ring structure hypothesis. The authors are claiming that task structure organizes neural activity, and therefore, they should see this task structure reflected in the neural activity. Why, then, would they expect forward distance to be a good comparison when the “delay line” arrangement is clearly not reflective of the task structure?

3. The assumptions underlying model prediction #2 (“distal prediction of behavioural choices”) are confusing. In particular, the authors state “The SMB model proposes that activity restricted to this ‘bump-time’ can be used to predict whether an animal will return back to this same behavioral step at the same point in the next trial” and furthermore claim this is a “unique prediction” (pg. 17). However, it appears that because of the ABCD task structure (and given the limited options the animal has at any given point), the animal will return to the same spatial position in an allotted time window on the next trial – thus, rendering the prediction of “whether an animal will return...” somewhat obvious.

4. One key conclusion the authors draw is that the SMB model precludes the “need to build or bind new representations.” In general, it is unclear how the SMB model may extend beyond the ABCD task however, and such a blanket statement seems inappropriate.

5. This is a minor point, but it is somewhat confusing that the authors use terminology associated with different machine learning frameworks without providing sufficient rationale and introduction. For example, the authors quite abruptly start to refer to “policy” with no clear indication of how this emerges from the model they propose. Are they referring to “policy” as in a reinforcement learning framework, or a different definition? To a broad audience, the use of such terms with very specific connotations may be misleading.

Referee #3 (Remarks to the Author):

In this manuscript, El-Gaby et al. present evidence that neuronal activity in the medial prefrontal cortex (mPFC) of rodents track the progression of animals towards series of rewards that the animals learn to find in a sequential manner. After learning, animals are trained on a different task in which the location of rewards is switched, and so on. Many neurons tended to be correlated with the specific phase of the task, irrespective of the reward location, while some others were correlated with either the spatial location or one specific reward in the sequence. The study then presents a theoretical framework that accounts for these observations, whereby the mPFC would contain modules of neurons whose population activity shows ring-like topology, reflecting the sequential nature of the task and enabling to flexibly transfer knowledge of the task structure to a new task. Overall, while the study is interesting and offers new insights into task learning correlates in the

mPFC, many findings seem to echo previously published work (which are too often not cited), casting doubt on the novelty of the study.

A first impression while reading the manuscript is the lack of proper citation of the relevant literature. Although manuscripts are constrained by citation limits and may omit some important studies, it is surprising that this manuscript lacks references to several landmark studies on the role of the mPFC in rule-switching (Birrel and Brown, *J Neurosci.*, 2000) and on neuronal correlates of sequence learning in the hippocampus (Bower et al., *J Neurosci.*, 2005; Dupret et al., *Nat Neuro*, 2010) and the mPFC (Euston and McNaughton, *J Neurosci.*, 2006), as well as coding of reward expectancy in the mPFC (Pratt and Mizumori, *Beh Brain Res.*, 2001) and learning-associated sleep activity patterns in the mPFC (Euston et al., *Science*, 2007; Peyrache et al., *Nat Neuro*, 2009). Most critically, a recent study has specifically addressed the internal ring-like representation of task sequence and its generalization across contexts in the mPFC and not the hippocampus (Tang et al., *Cell Reports*, 2023). And this is only to cite a few missing references.

One critical issue with the current study is the task design. By only visiting each goal once in a sequence, this design does not allow to disambiguate state from space and context. Previously published studies have used sequences with repeated locations to address this specific point (e.g. Euston and McNaughton, *J Neurosci.*, 2006). An important observation from this study was that the apparent neuronal correlates of behavioral states were primarily explained by the animal's trajectories. A more detailed analysis of animal trajectories would be necessary to regress what variables are encoded by each neuron (Fig. 2f). In addition, the manuscript states that neurons were not modulated by speed or acceleration, but the associated statistics are not reported in Fig. 2f.

The core tenet of the paper is that the mPFC learns a structure of the task that can be flexibly transferred to a new task, explaining the faster learning rate in early trials after extensive training on this task. However, the manuscript provides limited evidence to establish that the mPFC is essential for this behavior. Indeed, these tasks engage the same sensory modalities while the mPFC is believed to be necessary for switching between tasks entailing extradimensional cues (Birrel and Brown, 2001; see Peyrache et al., *Nat Neuro*, 2009, Benchenane et al., *Neuron*, 2010 and Kaefer, *Neuron*, 2020 for neuronal correlates of rule learning switches). It is still possible that the mPFC plays a role in the transfer between these specific tasks, but a direct demonstration of this view (e.g. manipulation of neuronal activity during behavior) would be informative.

Goal-progress tuning was largely preserved across tasks while, in contrast, the tuning to states remapped for each task. Specifically, the study reports a histogram of single neuron phase difference across tasks (Fig. 3a) which shows that, remarkably, task phase differences are overwhelming represented for every 90-degree shifts. In fact, there seems to be virtually no neurons that show a 45-degree (mod. 90 degree) shift. Yet, it is unclear how the third neuron in Fig. 3a remaps. There seem to be a 20-degree shift between tasks Y and Z and a 135-deg. shift between Z and X. The phase

difference between tasks X and Y for the 1st neuron and X and Z for the 5th neuron are also unclear. Were these neurons recorded simultaneously? More detailed examples and reporting across-task phase differences for each neuron in panel 3a would be informative. In addition, a potential confound of the single cell and pairwise phase difference analysis is that neurons overwhelmingly encode the same phase of the task (Fig. 2e). The manuscript does not clearly explain how this was accounted for.

The idea of separate modules, each independently coding for task phase is appealing. However, Fig. 3e is challenging to interpret. In line with the previous comment, more detailed examples should be provided, showing the tuning of each neuron in a given module and their remapping across tasks. More importantly, the existence of ring-like structure in activity should be directly demonstrated by investigating the topology of population activity (see e.g. Chaudhuri et al., *Nat Neuro*, 2019 for head-direction neurons and Tang et al., *Cell Reports*, 2023 for hippocampal and prefrontal neurons during task execution).

The existence of internally-generated activity patterns is well established, and not only in the medial entorhinal cortex as the citations suggest. Furthermore, it has already been established that neuronal patterns in the mPFC which form during learning are replayed during subsequent sleep (Euston et al., *Science*, 2007; Peyrache et al., *Nat Neuro*, 2009). In Euston et al. (2007), it was even shown that the specific relative timing of spikes was preserved after learning a sequential task similar to what the task design of the present study. In Peyrache et al., (2009), it was shown that these replay events were specifically associated with learning and were likely triggered by hippocampal sharp wave-ripples (also see Kaefer, *Neuron*, 2020 regarding the differences in replay content between the hippocampus and the mPFC). Here, a similar phenomenon was observed (Fig. 7) but a more direct demonstration of the phenomenon is necessary, for example by showing spike train cross-correlograms. Also, the dynamics of these phenomena likely depend on the exact sleep stages, so a more detailed analysis of sleep scoring and time since wake epoch is necessary. Since a difference between pre- and post-sleep was observed, it would be interesting to investigate whether some task-specific information was more likely to be replayed.

Author Rebuttals to Initial Comments:

Response to reviewers: A Cellular Basis for Mapping Behavioural Structure

We would like to thank the reviewers for their thorough and constructive comments. These have led us to add substantial new data, analyses and clarifications that support our findings. The new manuscript now contains a brand new dataset where mice learn two different abstract tasks: the original ABCD task and a new ABCDE task. This has allowed us to better demonstrate the generality of our representational and algorithmic claims. We further show generality anatomically, using new neuropixels data which shows similar representations are prevalent along the medial wall of the frontal cortex. We also add a new analysis of representational geometry using non-linear dimensionality reduction to directly visualise the use of goal-progress sequences as primitives to build task manifolds.

In addition, we include new schematics, examples of neuronal activity and behaviour as well as analyses and clarifications that address all of the reviewers' comments. This includes more explicitly spelling out the novelty of the findings and how they compare to previous findings from other tasks and regions.

Reviewer 1 has kindly stated that our study “reveals a deeper understanding of how the brain represents and adapts to complex behavioural sequences” and described our analyses as “rigorous” and “sophisticated” but noted that “The “working out” needs to be more clearly shown.”. We have now improved the clarity of analyses and figures. We also include new schematics and add more explicit explanations throughout. We thank the reviewer for raising these points which have now made our analyses and figures much clearer.

We are also grateful for Reviewer 2 who described our work as “novel, significant and important” but has concerns about the generality of our findings. We now add a new abstract task (ABCDE) and show empirically that the same representational principles apply across tasks of different periodicities. We further clarify what features of the findings can generalise to non-periodic tasks. We thank the reviewer for encouraging us to discuss these points, which have hopefully made the specific and broad implications of our findings explicit and clear.

Reviewer 3 kindly states that our study is “interesting and offers new insights into task learning correlates in the mPFC” but has concerns about the novelty of some of our findings compared to previous studies. We have now cited all of the studies mentioned by the reviewer (and more) and clarified how our 3 major findings differ from those of previous studies. In doing so we've included extra data, analyses, schematics and text to make this explicit. We thank the reviewer for raising these issues, which have made us more thoroughly and clearly address the novelty of our work.

Below we address each of the reviewers' comments in turn and show the new or modified figures that apply to each comment. We have also highlighted the changed sections in the revised manuscript.

Referee #1 (Remarks to the Author):

The paper reveals a deeper understanding of how the brain represents and adapts to complex behavioral sequences. Researchers trained mice on tasks with shared structures but different goals, demonstrating that mice could grasp the underlying abstract structure, allowing them to swiftly adapt to new tasks. Neurons in the medial frontal cortex, termed "goal-progress cells," played a crucial role in this process. These neurons effectively mapped the progress towards goals, generalized their activity across different task variations, and encoded entire sequences of future behavioral steps. This discovery suggests that these goal-progress cells serve as fundamental building blocks for representing complex behavioral structures, potentially offering insights into efficient learning and adaptation mechanisms in the brain.

Moreover, the study highlights the potential reconciliation of previous research on the frontal cortex's role in sequencing complex behaviors and draws parallels between the organization of neurons for behavioral structure mapping and cognitive mapping in the brain. It also proposes that learning involves the gradual transformation of naive goal-progress sequences into structured memory buffers, shedding light on the mechanisms underlying memory and learning in the brain. In summary, the paper's findings provide a better understanding of how the brain maps and adapts to abstract behavioral structures, with implications for cognitive neuroscience and our comprehension of neural learning processes.

The task is well-designed and cleverly elicits zero-shot learning in mice. The analysis is sophisticated and rigorous, although quite complex. Nevertheless, I have several concerns. First, it can be difficult to follow what the authors have done, and often the figures are more of a hindrance than a help. Second, although the end results of the analyses show clear effects, it is not always evident that these effects are supported by the raw data. The "working out" needs to be more clearly shown. Finally, there were several places where I simply could not figure out the analysis. My comments are roughly in order of importance.

Thanks very much indeed for your generous comments. We very much hope we have clarified the analyses and the data in the new version. We have also added new data from neuropixels recordings, and new analyses to better support our main findings.

1. I think the neuronal tuning shown in figure 5C is critical to support the structured memory buffer model, but it is very difficult to see. The figure is truly awful. It's way too small to determine the spatial tuning of the neurons. (It might help to remove the edges from all these plots. There doesn't seem much point including them when they are removed from all the analyses). However, where I can discern the spatial tuning, it doesn't really seem to line up with the correlation coefficients that the authors are reporting. For example, consider neuron 3 shown in Extended Figure 4. This neuron has a relatively high correlation across tasks approaching 0.8, and yet when I look at the spatial tuning across the tasks, I'd be hard pressed to tell you where this neuron's place field was as it is so variable. I'd really like to understand the discrepancy between the authors' metrics and the single neuron tuning before I would be comfortable believing the structured memory buffer model. Incidentally, these figures also need a color bar so that we know what kind of firing rates we're dealing with.

We thank the reviewer for raising this clarification point. First, we would like to remind the reviewer that the other two analyses in figure 5 (a, b and e,f) test the same point as **figure 5c** just in different ways. In the **Figure 5c** plots we are quantifying the spatial correlation between place maps for where the animal is currently or at a given lag in the past. In visualising individual neurons, we indeed end up with spatial plots that show some spurious peaks even when the cross-task correlation for that particular lag is high. There are two reasons for this. The first is that animals will unevenly visit each maze location depending on the

subset of rewarded locations in each task. The second is that these plots (and associated analyses) are restricted to each neuron's preferred goal-progress bin (as a reminder these neurons are also tuned to goal-progress so predominantly fire in their preferred bin). This means that they represent one-third of the session time (i.e. ~7 minutes). Both of these factors introduce noise as some locations are visited for only a short period of time. This creates the possibility of spurious peaks in a spatial map for a single task at a single lag. Nevertheless, the spatial correlation measure averages away such spurious peaks at the single cell level. We now show examples of neurons that have a single consistent peak and others where there are multiple consistent peaks (**Figure 5c and Extended data figure 5b**). We have also now removed the edges for clarity as requested. We note that, as with other analyses in figure 5, this lagged spatial map analysis is done in a cross-validated manner, which ensures that it is robust to spurious peaks emerging in any given task. We have also added the colour bars. Note that, to visualise each map's spatial firing pattern, each map is individually normalised. This does not affect the correlation measure in any way. We show the figures here for completeness but we recommend viewing the high quality version of these figures provided at the end of the manuscript, as compression on this response document inevitably reduces quality.

Figure 5

- c) Lagged spatial field analysis. Example plots showing spatial maps for 3 neurons. Bottom: Each row represents a different task and each column a different lag in task space. The activity of each neuron is plotted as a function of the animal's current location (far right column for each cell) and at successive task space lags in the past for the remaining columns. For example, the second to last column is the firing of the cell in relation to where the animal was 30 degrees (1/3rd of a state) back from the current point in task space, 7th to last column is the firing of the cell in relation to where the animal was 180 degrees (2 states) back from the current point in task space...etc. Top: the correlation of spatial maps across tasks at each lag. The neuron on the left is an anchor cell (goal-progress/place cell), as seen by the correlation peak at zero lag, while the middle and right-most neurons are neurons lagged by 210 and 180 degrees in the past from their anchors respectively. To avoid confounds due to goal-progress tuning, all firing rates are calculated only in each neuron's preferred goal-progress bin (i.e. one-third of the entire session). Colours are normalised per map to emphasise the spatial firing pattern, with maximum firing rates (in Hz) displayed at the top right of each map.

Below we zoom in on one of the neurons for clarity:

Extended data figure 5b

- b) Lagged spatial field analysis. Example plots showing spatial maps for 3 neurons. Each row represents a different task and each column a different lag in task space. Bottom: Activity of each neuron is plotted as a function of the animal's current location (far right column for each cell) and at successive task space lags in the past for the remaining columns. Top: the correlation of spatial maps across tasks at each lag. The neuron on the left is an anchor cell (goal-progress/place cell), as seen by the correlation peak at zero lag, while the middle and right-most neurons are neurons lagged by 90 and 60 degrees in the past from their anchors respectively. To avoid confounds due to goal-progress tuning, all firing rates are calculated only in each neuron's preferred goal-progress bin (i.e. one-third of the entire session). Colours are normalised per map to emphasise the spatial firing pattern, with maximum firing rates (in Hz) displayed at the top right of each map.

2. The authors make the claim that goal progress tiles the entire space, but I'm not entirely convinced by this. I'm not a fan of the kind of plots that they show in Figure 2E, because those kinds of plots can arise simply from noise. That concern is somewhat mitigated by the fact that they're only showing significant neurons, but nevertheless, I'd like to see a statistical analysis that supports the claim, and maybe change the figure so that they are showing where significant selectivity (appropriately corrected for multiple comparisons) occurs over the course of the goal progress. Furthermore, the data in the Supplementary Information seems to paint a more nuanced picture. For example, the data shown in Extended Figure 5D seems to suggest that early and late goal progress is represented more strongly than intermediate goal progress. I don't think this necessarily conflicts with the authors' structured memory buffer model. It may be the more salient parts of the task progress (starts and ends) are represented more strongly in neuronal tuning. However, if this is the case then we ought to be made aware of it.

We thank the reviewer for raising this important point. We have now added an analysis where we identify the significant peaks and plot a histogram of the goal-progress distribution only for **significantly goal-progress tuned neurons (Figure 2i)**. Importantly, here we use a Bonferroni correction for multiple comparisons across task phase bins. This shows that significant goal-progress peaks tile the entire range of goal-progress values, with some subtle inhomogeneity, over-representing intermediate goal progress values (**Figure 2i**). We note two points:

- 1-The original ordered goal-progress plots in **figure 2f** show all neurons recorded across all 4 sessions on a given day (not just significant goal-progress neurons). Their purpose of this plot is primarily to illustrate how consistent goal-progress tuning is across tasks. We quantify this in the histogram on the right of this panel, where we show the cross-task goal-progress correlations across all neurons is highly positive (**Figure 2f right**). A t-test against zero shows that this correlation is highly positive and is reported in the panel and in the figure legend (**Figure 2f right**). We also separately illustrate this goal-progress invariance using the GLM in **figure 2g,h**
- 2-The plot that was in **Extended data figure 5d** (now in **Extended data figure 7e**) shows the distribution of lags from anchors along the entire ABCD loop (i.e. 4 goal progress cycles) rather than the distribution of goal-progress lags which concern progress relative to a single goal. We now clarify this distinction in the figure legends.

Figure 2

- f) Goal-progress tuning is consistent across tasks that differ in reward locations. Left: The average firing rate vector of all neurons relative to an individual goal (from goal "n" to goal "n+1"; averaged across all states). Animals experience 3 tasks per day (tasks X, Y and Z) and then a further 4th session which task X is repeated (task X'). Each row represents a single neuron and the neurons are arranged on the y axis by their peak firing goal-progress in task X. This alignment is largely maintained in tasks Y and Z as well as a later session of the first task (X'). White dashes indicate early intermediate and late goal-progress-cut-offs. Right: A histogram showing the mean goal-progress-vector correlation across tasks for each neuron. One-sample T test against 0: N=2461 neurons; statistic=104.3; P=0.0, df=2460. Note that the neurons used in this panel are those that were tracked for all 4 sessions on a given day.
- g) Left: a schematic of the variables inputted into a generalised linear model that predicts neuronal activity across tasks and states. The model captured variance as a function of goal-progress, place, speed, acceleration, time from reward and distance from reward. Only data spike sorted across two days (6 unique tasks) was used to ensure this analysis is sufficiently powered. Middle: A histogram showing the mean regression coefficient values for *goal-progress* as a regressor across task/state combinations for each neuron. One-sample T test against 0: N=1252 neurons; statistic=21.7; P=8.93x10⁻⁸⁹, df=1251. Right: A histogram showing the mean regression coefficient values for *place* as a regressor across task/state combinations for each neuron. One-sample T test against 0: N=1252 neurons; statistic=24.9; P=3.31x10⁻¹¹¹, df=1251.
- h) Pie-charts showing the proportions of cells calculated using the results of the generalised linear model above in addition to cross-task correlations between tuning to goal-progress and place. Only data spike sorted across two days (6 unique tasks) was used to ensure this analysis is sufficiently powered. Plot shows proportions of neurons with i) significant regression coefficient values for goal-progress or place ii) Significantly positive cross-task correlation for goal-progress or place. It also shows proportions of state tuned neurons derived from a separate z-scoring analysis (More details in Methods under "Tuning to basic task variables"). Proportion of all neurons that are goal-progress cells: 74%; Two proportions test: N=1252 neurons, z=35.5, P=0.0. Proportion of goal-progress neurons that are state tuned: 64% Two proportions test: N=931 neurons, z=26.8, P=0.0. Proportion of neurons tuned to goal-progress and state that are also tuned to place: 63%, Two proportions test: N=597 neurons, z=21.2, P=0.0. Proportion of all state-tuned neurons that are also goal-progress tuned; 81% Two proportions test: N=738 neurons, z=29.5, P=0.0.

- i) A histogram showing the distribution of significant goal-progress peaks amongst all neurons, all tasks and all states. Only neurons from concatenated double days that are significantly goal progress-tuned and have at least one significant goal-progress peak are shown (N=873 neurons). The plot shows that such significantly goal-progress tuned cells have peaks throughout the entire range of goal progress values. Note that this plot spans goal-progress space, which is the lag between any two rewarded goals, rather than the full (multi-goal) task space.

Extended data figure 7

- e) Distribution of task space lags from anchor for all consistently anchored state neurons (neurons with the same anchor in >50% of tasks). Left: Using the most common lag from the anchor across tasks; Right: using the (circular) mean lag from anchor across tasks. Both plots show that consistently anchored neurons have lags from their anchor that span the entire range of possible lags. Note that these plots span the entire (4-goal) task space.

3. I have some concerns about the statistical analysis throughout the paper. The authors are using a relatively liberal alpha level of 0.05 (at least liberal for assessing neuronal tuning) and it is also not clear that they are correcting for multiple comparisons, for example, for the different task phases that they are testing. I think this means that the selective neuron counts are somewhat inflated, and it would probably behoove the authors to clean this up, as most of the neurons with weak tuning are probably just noise.

We thank the reviewer for raising these important statistical points. We now show that the key findings where such tuning thresholds apply hold with a more stringent alpha level ($p < 0.01$) and clarify that multiple comparisons are not used when determining neuronal tuning.

First with regards to the use of an alpha of 0.05 as a threshold for the percentages of neuron-types reported in **figure 2h**. While we have used alpha level of 0.05 mostly in concordance with previous studies, we agree with the reviewer that, for subsetting individual neurons tuned to a given variable, the value chosen for significance matters and this in turn may affect results downstream of this. We note however that, depending on the question, our use of a slightly liberal p value actually makes our results more stringent. This is the case when we **subset away** spatial neurons in some iterations of the same analyses (e.g. **Extended data figures 4d,g**) to illustrate the findings are robust to any spatial tuning. In this case, using an even lower p value to determine significance may actually introduce a self-serving bias as we would be keeping some neurons with weak spatial tuning in an analysis where we want to exclude as many of those neurons as possible. Nevertheless, we agree with the reviewer that in other cases we might want to use more stringent criteria, and that using more permissive ones actually works against us, potentially introducing noise. This is relevant when we **subset in** state-tuned neurons - to explain the activity only of neurons (or pairs of neurons) that exhibit some state-preference in one or more tasks. We now add an extra criteria of requiring all neurons to not only pass the 95th percentile of its null distribution but to also show cross-task correlation coefficients that are significantly above zero when conducting a correlation between the tuning vectors across tasks. We also use an alpha of 0.01 for state tuning and observe similar partially coherent remapping of pairs of neurons but no single-cell generalisation (**Extended data figure 4b,e**). Moreover, we use an alpha of 0.01 and show similarly robust anchoring of more stringently defined state-tuned neurons (**Extended data Figure 6b**). We note that the choice of alpha primarily affects our subsetting of neurons for analyses downstream of figure 2, rather than the statistical tests on the overall proportion of neurons tuned to different variables. This is because in order to determine whether the overall proportion of neurons significantly tuned to a particular variable is statistically significant, we perform a two

proportions Z-test against the null proportion used (i.e. 0.05 in our original analysis). Thus, the alpha value chosen is reflected in the statistical test itself.

Extended data figure 4

- b) Remapping of state neurons that are defined using a stricter threshold (z-score >99th percentile of permuted distribution). Top: A schematic showing how the difference in tuning angles for the same neuron across sessions is quantified. Bottom left: Polar histograms show that state-tuned neurons remap by angles close to multiples of 90 degrees, as a result of conserved goal-progress tuning and the 4 reward structure of the task. No clear peak at zero is seen relative to the other cardinal directions when comparing sessions spanning separate tasks (Two proportions test against a chance level of 25% N=1061 neurons; mean proportion of generalising neurons across one comparison (mean of X vs Y and X vs Z)=24%, $z=0.59$, $P=0.552$). Bottom right: Neurons maintain their state preference across different sessions of the same task (X vs X'; Two proportions test against a chance level of 25% N=770 neurons; proportion generalising=80%, $z=21.5$, $P=0.0$).
- c) Remapping when using only state-neurons with concordant remapping angles across two methods (i.e. using the best-rotation analysis method and peak-to-peak changes method). This analysis would for example exclude neuron 3 in figure 3a. Left: Polar histograms show that state-tuned neurons remap by angles close to multiples 90 degrees, as a result of conserved goal-progress tuning and the 4 reward structure of the task. No clear peak at zero is seen relative to the other cardinal directions when comparing sessions spanning separate tasks (Two proportions test against a chance level of 25% N=369 neurons; mean proportion of generalising neurons across one comparison (mean of X vs Y and X vs Z)=24%, $z=0.41$ $P=0.684$). Right: state-tuned neurons maintain their state preference across different sessions of the same task (bottom right). Two proportions test against a chance level of 25% N=240 neurons; proportion generalising=84%, $z=13.0$, $P=0.0$).
- d) Remapping of non-spatial neurons. Note that we use the most permissive threshold for spatial tuning here to ensure that we exclude even neurons with weak/residual spatial tuning. Any neuron that had a spatial regression coefficient above the 95th percentile of the null distribution was excluded from this analysis. Left: Polar histograms show that non-spatial state-tuned neurons remap by angles close to multiples 90 degrees, as a result of conserved goal-progress tuning and the 4 reward structure of the task. No clear peak at zero is seen relative to the other cardinal directions when comparing sessions spanning separate tasks (Two proportions test against a chance level of 25% N=704 neurons; mean proportion of generalising neurons across one comparison (mean of X vs Y and X vs Z)=22%, $z=1.19$ $P=0.233$). Right: Non-spatial state-tuned neurons maintain their state preference across different sessions of the same task (bottom right). Two proportions test against a chance level of 25% N=507 neurons; proportion generalising=68%, $z=13.9$, $P=0.0$).
- e) Pairwise coherence of state neurons that are defined using a stricter threshold (z-score >99th percentile of permuted distribution). Top: A schematic showing how the difference in relative angles between pairs of neurons across sessions is quantified. Bottom left: Polar histograms show that the proportion of coherent pairs of state-tuned neurons (comprising the peak at zero) is higher than chance but less than 100%, indicating that the whole population does not rotate coherently. Two proportions test against a chance level of 25% N=17671 pairs; mean proportion of coherent neurons across one comparison (mean of X vs Y and X vs Z)=29%, $z=8.9$, $P=0.0$). Bottom right: As expected from panel b, the large majority of state-tuned neurons keep their relative angles across sessions of the same task (X vs X'; Two proportions test against a chance level of 25% N=11716 pairs; proportion coherent=64%, $z=59.3$, $P=0.0$).
- f) Coherence of state-neuron pairs using only state-neurons with concordant remapping angles across two methods (i.e. using the best-rotation analysis method and peak-to-peak changes method). Left: Polar histograms show that the proportion of coherent pairs of state-tuned neurons (comprising the peak at zero) is higher than chance but far from 1, indicating that the whole population does not rotate coherently (Two proportions test against a chance level of 25% N=1642 pairs; mean proportion of coherent neurons across one comparison (mean of X vs Y and X vs Z)=30%, $z=3.32$, $P=9.04 \times 10^{-4}$). Right: As expected from panel b, the large majority of state-tuned neurons keep their relative angles across

sessions of the same task (X vs X'; Two proportions test against a chance level of 25% N=657 pairs; proportion coherent=72%, $z=17.1$, $P=0.0$).

- g) Coherence of non-spatial neuron pairs. Note that we use the most permissive threshold for spatial tuning here to ensure that we exclude even neurons with weak/residual spatial tuning. Any neuron that had a spatial regression coefficient above the 95th percentile of the null distribution was excluded from this analysis. Left: Polar histograms show that the proportion of coherent pairs of non-spatial state-tuned neurons (comprising the peak at zero) is higher than chance but far from 1, indicating that the whole population does not rotate coherently (Two proportions test against a chance level of 25% N=6996 pairs; mean proportion of coherent neurons across one comparison (mean of X vs Y and X vs Z =30%, $z=3.49$, $P=4.74 \times 10^{-4}$). Right: As expected from panel b, the large majority of non-spatial state-tuned neurons keep their relative angles across sessions of the same task (X vs X'; Two proportions test against a chance level of 25% N=4822 pairs; proportion coherent=54%, $z=29.2$, $P=0.0$).

Extended data figure 6

- b) Histograms showing the right-shifted distribution of mean cross-validated correlation values between model-predicted (from training tasks) and actual activity (from a left out test task) for state neurons that are defined using a stricter threshold (z-score >99th percentile of permuted distribution). Left: this correlation is shown for all state-tuned neurons; Middle: only state-tuned neurons with non-zero-lag firing from their anchors; and Right: non-zero lag state-tuned neurons with the maximum regression coefficient value a whole state (90 degrees) or more either side of the anchor. T test against 0: All state-tuned neurons N=349 neurons, statistic=8.70, $P=1.34 \times 10^{-16}$, $df=348$; Non-zero lag state-tuned neurons N=227 neurons, statistic=2.83, $P=0.005$, $df=226$; distal (90 degrees) non-zero lag neurons: N=154 neurons, statistic=1.94, $P=0.054$, $df=153$.

Second, on the issue of correcting for multiple comparisons, we note that we are not using multiple comparisons across the various task phases/places to determine a given neuron's tuning for the glm in **figure 2g,h**. Instead we are using the entire vector for a given variable to do the regression. For example, to determine if a neuron is spatially tuned, we compute a vector of 9 bins corresponding to the firing rate of a neuron in all 9 locations from data restricted to a defined training set (all but one combination of tasks and task state). We then use this vector as a regressor in a test set (a single task state in a single task), regressing it against the bin-by-bin firing rates of the same neuron in the test set. This allows us to determine a single regression coefficient for a given variable (e.g. place), effectively describing how consistent the firing rate vectors are for this variable between the training and test data. As such, multiple comparisons are not used here. We have now clarified this further in the methods section under "Tuning to basic task variables" and "Generalised linear model" subsection. We do now however use multiple comparisons when determining the significant peaks of each neuron and for this we have used a Bonferroni correction for multiple comparisons across task phase bins, which is used to generate the histogram of significant peaks now seen in the new **figure 2i**.

4. I'm not sure I understand the data being presented in figure 6. I don't understand how there can be a 1:1 condition. According to the authors' explanation, 1 indicates that the animal visited the anchor location, and 1:1 means that they visited the anchor location on trial N and trial N + 1. In the context of the task though, I don't understand how this is possible. Maybe the animal is making an error and returning to the original location? I don't think that's what is happening though, so I think the authors should do a better job of explaining this figure, or at least explaining how the 1:1 condition happens.

We thank the reviewer for the opportunity to clarify this point. This analysis asks: given that a neuron is anchored to a particular behavioural step in the past, does the same neuron predict whether the animal will return to this same behavioural step at the same point on the next trial? This future prediction is possible

because there is variability in the specific trial-by-trial specific routes animals take between goals. A trial here refers to an entirely new iteration of the **full ABCD sequence** (see new **Figure 1b**). We can tap into this trial-by-trial variability to ask whether neurons can predict distal choices of the animal. Animals can potentially take multiple routes (including multiple optimal routes) and so can be at one location at one point in trial N and at a different location at the same point in trial N+1. This is true for non-rewarded locations, as animals can take different routes between rewards. It is also true for rewarded locations, as animals could be one-step away from the rewarded location in trial N+1 and yet erroneously choose a different alternative location despite making the correct choice at the same point in trial N. In addition to now showing examples of the animal's own trial-by-trial behaviour (**Figure 1b**), we now add 2 new schematics (**Figure 6a, Extended data figure 8a**) to illustrate how the predictions made by the SMB model relate to specific scenarios and how our analysis gets at those predictions. Inspired by the reviewer's question, we have made the schematic in **Extended data figure 8a** to illustrate examples of the 0:0, 0:1, 1:0 and 1:1 conditions. This is one way we control for previous choices (e.g. in **Figure 6b**) the other being regressing out choices up to 10 trials in the past when performing the logistic regression to predict future choices (e.g. **Figure 6d**). The broad implication of this analysis is that we can use the activity of neurons on SMBs to precisely predict the animal's future choices while controlling for the animal's previous choices, in line with the SMB algorithm (**Figure 4f,g**). In addition to the schematics, we have also added additional clarification in the results section (under "2-Distal prediction of behavioural choices") and discussion to explain the set up and broader implications of this analysis.

Figure 1

- a) Task design: animals learned to navigate between 4 sequential goals on a 3x3 spatial grid-maze (measuring 45cm x 45 cm). The 9 maze locations were connected as shown in the top right photograph and adjacent schematic, with connections only along the cardinal directions. Reward locations changed across tasks but the abstract structure, 4 rewards arranged in an ABCD loop, remained the same.
- b) Example paths from 3 different mice performing 3 different tasks. Each row is a set of 5 consecutive trials from the same mouse and task. Single trial paths are superimposed upon whole session coverage shown in grey. Mice rapidly converged on near-optimal routes and used only a subset of the available paths.

Figure 6

- a) Schematic showing distal prediction of animal's choices from memory buffers. The SMB model allows us to predict the animal's choices 1) at a precise lag in the future and 2) in a way that generalises across tasks. The larger the size of an activity bump on a given SMB the more likely the animal will subsequently visit the SMB's anchor. The timing of this bump determines when the anchor will be visited. In this example we show an SMB with an anchor at location 2 (top middle location shaded in brown) at intermediate goal progress (i.e. half way between goals). We highlight two neurons on this SMB - one in brown which is at zero lag from the anchor (i.e. fires immediately when the anchor is visited) and another in green which fires at a lag of 135 degrees (1.5 states) from the anchor. We show 4 rows corresponding to 4 possible scenarios across two different tasks to illustrate the key features of the SMB model's predictions. In all trials, the animal starts off at the anchor point (brown location) at t1. In the first scenario (top row) of Task X the anchor visit triggers a large bump of activity at t1, which travels around the ring (e.g. at t2) and when it reaches the SMB's decision point at the end (t3) it biases the animal to return to the anchor. In the second row, the bump initiated by the first anchor visit (t1) is smaller and so the animal is less likely to visit the anchor at t3. The activity of the green neuron at precisely 1.5 states since the first anchor visit (t2; i.e. the green neuron's "bump time") can be used to predict what will happen at t3 (i.e. 2.5 states forward from t2). If the green neuron has a high firing rate at t2 (darker green shading), the animal should return to the anchor at t3. Conversely, low firing rate of this neuron at t2 (light green shading) predicts no return to the anchor at t3. Thus the SMB model allows predicting behavioural steps at precise lags in the future. The SMB model also posits that this future prediction should generalise across tasks and hence be independent of where the animal is at a given time. This can be seen in Task Y. Now the green neuron fires at a different location in order to keep its lag from its anchor. And yet, its activity at t2 can be used in the same way to predict whether the animal will visit the anchor at t3 (i.e. 2.5 states in the future). Again the higher the activity at t2 the more likely the anchor will be visited at t3. To test this latter point, we only consider non-zero lag cells in all of the analyses below, which means we only use neurons that fire in different locations across different tasks. We indicate the lag threshold separately for each panel.

Extended data figure 8

- a) Schematic showing distal prediction of animal's choices from memory buffers as a function of previous trial choices. By harnessing variability in the coupling between anchor visits and bump initiation, we can test whether SMB activity can predict future choices when controlling for previous choices. In this example we show an SMB with an anchor at location 2 (top middle location shaded in brown) at intermediate goal progress (i.e. half way between goals). Each row shows a different scenario across two consecutive trials (trial N-1 and trial N) in the same task and the expected activity of neurons anchored to this location/goal-progress conjunction. Scenario 1 (0:0): the animal doesn't visit the anchor in trial N-1 and hence the activity of neurons on the SMB for this anchor is low. This results in the animal not visiting the anchor in trial N. Scenario 2 (0:1): the animal again doesn't visit the anchor in trial N-1 but this time the activity of neurons on the SMB is high (e.g. due to noise or top down modulation). This results in the animal visiting the anchor in trial N. Scenario 3 (1:0): the animal visits the anchor in trial N-1 but the activity of neurons on the SMB for this anchor is low (e.g. due to noise or top down modulation). This results in the animal not visiting the anchor in trial N. Scenario 4 (1:1): the animal visits the anchor in trial N-1 and the activity of neurons on the SMB for this anchor is high. This results in the animal visiting the anchor again in trial N. In effect the variability in the size of the bump in relation to the anchor visit allows us to decouple the SMBs' activity from previous choices. This makes it possible to test whether SMB activity predicts future choices of the animal. Note that the time stamps shown on the SMB (t1, t2 and t3) are all in trial N.

5. It is not always clear how the number of neurons for a given analysis was derived, and it can fluctuate from figure to figure. The authors need to clearly explain how the subset of the neurons that they are testing was selected.

We apologise for the lack of clarity regarding cell numbers. The numbers change as each analysis has different requirements, (e.g. excluding spatial cells, including only anchored cells...etc). We have now added a table indicating the criteria for inclusion and the Ns for each figure panel (**Table 1**).

6. I wasn't convinced by the clustering analysis shown in figure 3D. If I'm understanding this analysis correctly there are only 12 possible clusters (the pairwise combinations between A, B, C, and D) so I'm not convinced that this clustering analysis shows anything other than the fact that different neurons are responding to different states across the tasks.

We thank the reviewer for the opportunity to clarify this. What we show here is that there are **fewer** clusters (i.e. a higher degree of clustering) than expected if neurons were remapping by chance.

In the trivial case, where all neurons remap randomly, we expect 16 possible clusters. This is because we assess clustering across 3 tasks, which gives two comparisons (X vs Y and X vs Z). For each comparison there are 4 possible ways a given neuron can remap, creating 4 groups (neurons remapping by 0, 90, 180 or 270 degrees). In the second comparison, there are another 4 ways the neurons could remap. Thus the number of clusters is 16 (4^2). This assumes remapping is always exactly in 90 degree intervals, i.e. perfect

goal-progress tuning. In reality, goal-progress tuning is not perfect and so more clusters are expected in the null condition. To avoid such assumptions, we create a null distribution that preserves the neurons' state tuning in the first task and goal-progress tuning throughout all tasks, but where each neuron otherwise remaps randomly across tasks. This results in neurons remapping independently of each other. We then compute clusters and quantify the silhouette scores (which report the normalised difference between the between-cluster and the within-cluster distances) for this permuted data and compare such scores to those of the real data. We find that the real data shows a significantly higher degree of clustering than the permuted data (**Figure 3d**).

Figure 3

- d) Left: Example from a single recording day showing the result of tSNE embedding and hierarchical clustering derived from a distance matrix quantifying cross-task coherence relationships between state-tuned neurons. Each dot represents a neuron. Right: Summary silhouette scores for the clustering for real data compared to permuted data that maintains the neuron's goal-progress preference and initial state distribution. Each dot is a recording day. Wilcoxon test: $N=38$ recording days; statistic=126.0, $P=2.13 \times 10^{-4}$

We note that we separately demonstrate this clustering using a simpler analysis, where we assess the proportion of coherently remapping neurons and compare it to chance across one comparison (i.e. by averaging proportions for X vs Y and X vs Z; **Figure 3c**) or across both comparisons (i.e. by finding proportion of neurons that remap coherently across **both** X vs Y and X vs Z comparisons; **Extended data figure 4h**). In these analyses we can compare the proportion of coherent pairs of neurons to a predefined chance level because we quantify the proportion of neuron pairs within a window 45 degrees either side of zero (i.e. a total area spanning 90 degrees) and compare this proportion to the rest of the population. This means that, regardless of the degree of goal-progress tuning, the chance level for this “close-to-zero” proportion is 25% (90/360). We now show a shaded region to illustrate which parts of the histogram are used to signify generalisation (**Figure 3b**) or coherence (**Figure 3c**). We use a similar approach when we assess combined remapping across both cross-task comparisons (X vs Y and X vs Z), quantifying the proportion of pairs of neurons that remap by a maximum of less than 45 degrees either side of zero across both comparisons (**Extended data figure 4h**). Here, chance level is now $1/16$ ($1/4^2$). We now clarify these points in the revised methods section, under “Neuronal generalisation”.

7. I thought that the discussion of the results shown in figure 3 needed a little strengthening. The results in figure 3B show not just that the neurons are remapping, but that the zero-lag position is significantly below chance. In other words, it's not simply a completely new map when it remaps. The new map is orthogonal to the first map, at least if I'm understanding this analysis correctly.

We thank the reviewer for highlighting this. It was indeed the case that there was a slight underrepresentation of generalising neurons in figure 3 in the original manuscript (generalising neurons were 20% compared to a chance level of 25%). Now that we have added new data, including neuropixels data along the medial wall of mFC, this generalisation is no longer significantly below chance. The proportion of generalising neurons is now 24% and not significantly different from chance. In contrast, the pairwise coherence and clustering analyses in **Figure 3** remain highly above chance as well as all the other major results pertaining to the SMB model (**Figures 5-7**). Further, these results did not reflect the addition

of neurons along a larger dorsoventral extent of the mFC as we have directly quantified the degree of generalisation and coherence along the DV extent of the mFC, spanning prelimbic and infralimbic cortices as well as anterior cingulate and M2 (New **Figure 2a** and **Extended data figure 2b**), and find no significant differences between regions (**Extended data figure 4j,k**). On a related note, we find no differences along the DV extent for both the basic tuning properties of the mFC cells (**Extended data figure 2e**) and for the SMB-predicted mnemonic activity relative to anchors (**Extended data figure 7h**).

- b) Top: A schematic showing how the difference in tuning angles for the same neuron across sessions is quantified. Bottom left: Polar histograms show that state-tuned neurons remap by angles close to multiples of 90 degrees, as a result of conserved goal-progress tuning and the 4 reward structure of the task. The orange shaded area represents neurons that maintain a consistent angle across sessions (with a tolerance of 45 degrees either side of 0 change). No clear peak at zero is seen relative to the other cardinal directions when comparing sessions spanning separate tasks (Two proportions test against a chance level of 25% $N=1594$ neurons; mean proportion of generalising neurons across one comparison (mean of X vs Y and X vs Z)=24%, $z=0.91$, $P=0.363$). Bottom right: Neurons maintain their state preference across different sessions of the same task (X vs X'. Two proportions test against a chance level of 25% $N=1160$ neurons; proportion generalising=78%, $z=25.7$, $P=0.0$).
- c) Top: A schematic showing how the difference in relative angles between pairs of neurons across sessions is quantified. Bottom left: Polar histograms show that the proportion of coherent pairs of state-tuned neurons (comprising the peak at zero) is higher than chance but less than 100%, indicating that the whole population does not rotate coherently. The orange shaded area represents pairs of neurons that maintain a consistent angle across sessions (with a tolerance of 45 degrees either side of 0 change). Two proportions test against a chance level of 25% $N=35164$ pairs; mean proportion of coherent neurons across one comparison (mean of X vs Y and X vs Z)=29.3%, $z=13.0$, $P=0.0$). Bottom right: As expected from panel b, the large majority of state-tuned neurons keep their relative angles across sessions of the same task (X vs X'. Two proportions test against a chance level of 25% $N=23674$ pairs; proportion coherent=63%, $z=83.5$, $P=0.0$).

Extended data figure 4

- j) The subregional distribution of single neuron generalisation (averaged across X vs Y and X vs Z comparisons) along the medial wall of frontal cortex in neuropixels recordings. One-way ANOVA: $F=1.59$, $P=0.323$, $df=3$.
- k) The subregional distribution of neuron pair coherence. Coherence is calculated across both X vs Y and X vs Z comparisons along the medial wall of frontal cortex in neuropixels recordings: One-way ANOVA: $F=4.76$, $P=0.083$, $df=3$.

8. In the methods the authors describe using either analytical or empirical methods for calculating the shortest path transitions, but they don't explain when one is used relative to the other in the manuscript.

We thank the reviewer for raising this which made us realise that our wording was unclear. To clarify, we use analytical or empirical methods only to calculate chance levels for animal transitions. The analytical method is to simply calculate the probability that a given shortest path transition could happen by chance if the animal is moving randomly around the maze. This is used in **Extended data figure 1k**, to ensure that the DA compared to BA/CA comparison for zero-shot quantification is a fair one. Empirical chance is calculated in two different places:

1-When finding the probability of a shortest-path transition - pre-task exploration chance levels are calculated by quantifying each animal's transition probabilities around the maze in an exploration session prior to seeing any ABCD task (or any reward on the maze). This is used in **Extended data figure 1j**, again to ensure that the DA compared to BA/CA comparison for zero-shot quantification is a fair one.

2-When setting a chance level for the "relative path distance" measure. Chance here is defined as the mean relative path distance for transitions in the first trial averaged across the first 5 tasks across all animals. This is used in **figures 1c, d and e**.

We now clarify these points in the methods section of the manuscript under "Behavioural Scoring".

9. Figure 2F: this is possibly the most horrific pie chart I've ever seen, and I couldn't make head nor tail of it.

We apologise for the lack of clarity of this figure. What we're aiming to do here is to show the proportions of neurons tuned to each variable as well as the subpopulations tuned to conjunctions of variables. We have now simplified the figure (now **Figure 2h**) to show separate pie charts for each point we want to make. Note that these proportions are different from those shown in the first submission as we now use an additional criteria. We now require all neurons to not only pass the 95th percentile of its null distribution with respect to a given measure but to also cross-task correlation coefficients that are significantly above zero. We also only assess this for concatenated double days, which give us 6 tasks and hence more power to assess tuning.

Figure 2

h) Pie-charts showing the proportions of cells calculated using the results of the generalised linear model above in addition to cross-task correlations between tuning to goal-progress and place. Only data spike sorted across two days (6 unique tasks) was used to ensure this analysis is sufficiently powered. Plot shows proportions of neurons with i) significant regression coefficient values for goal-progress or place ii) Significantly positive cross-task correlation for goal-progress or place. It also shows proportions of state tuned neurons derived from a separate z-scoring analysis (More details in Methods under "Tuning to basic task variables"). Proportion of all neurons that are goal-progress cells: 74%; Two proportions test: N=1252 neurons, $z=35.5$, $P=0.0$. Proportion of goal-progress neurons that are state tuned: 64% Two proportions test: N=931 neurons, $z=26.8$, $P=0.0$. Proportion of neurons tuned to goal-progress and state that are also tuned to place: 63%, Two proportions test: N=597 neurons, $z=21.2$, $P=0.0$. Proportion of all state-tuned neurons that are also goal-progress tuned; 81% Two proportions test: N=738 neurons, $z=29.5$, $P=0.0$.

10. With respect to the finding that the authors are better able to predict future choices on the low probability sequences compared to the high probability, stereotyped sequences, the authors might also

want to cite Chiang and Wallis (2013). They also found that prefrontal spatial tuning on their sequence task was weaker when the animals were following more stereotyped strategies.

We thank the reviewer for highlighting this important parallel. Now that we have included additional data, this prediction is now significant for both low and high probability transitions. Given that this result is less informative, we have removed it from the manuscript. However, we present it below and are happy to re-add it if the reviewer advises us to. We note that in either case we already cite Chiang and Wallis 2013 in our introduction when discussing mFC roles in working memory.

- a) Prediction of behaviour for only the high probability choices (choices that animals made with a probability in top half of maze transition probabilities during pre-task exploration). Normalised firing rates of neurons during their “bump time”: i.e. the lag at which they are active relative to the anchor. Bump time activity was indistinguishable before visits to the neuron’s anchor compared to non visits in trial N+1 whether the anchor was not visited in trial N (left) or when it was visited in trial N (right). Wilcoxon tests: Anchor not visited in trial N: n=119 tasks, statistic=2255, $P=4.88 \times 10^{-4}$, Anchor visited in trial N: n=104 tasks, statistic=1930, $P=0.009$. In addition, an ANOVA on all data (N=104 tasks) showed a main effect of Past: $F=4.66$, $P=0.033$, $df_1=1$, $df_2=103$, a trend towards a main effect of Future: $F=3.57$, $P=0.062$, $df_1=1$, $df_2=103$. Right: Regression coefficients were significantly positive for the bump time but no other control times. T tests against 0: “bump time”: N=119 tasks, statistic=2.19, $P=0.030$, $df=118$; “decision time”: N=119 tasks, statistic=-1.73, $P=0.086$, $df=118$; “random time”: N=119 tasks, statistic=-0.86, $P=0.392$, $df=118$; “90 degree shifted time”: N=119 tasks, statistic=-1.51, $P=0.133$, $df=118$; “180 degree shifted time”: N=119 tasks, statistic=-1.92, $P=0.057$, $df=118$; “270 degree shifted time”: N=119 tasks, statistic=0.28, $P=0.777$, $df=118$;
- b) Prediction of behaviour for only the low probability choices (choices that animals made with a probability in bottom half of maze transition probabilities during pre-task exploration). Normalised firing rates of neurons during their “bump time”: i.e. the lag at which they are active relative to the anchor. Bump time activity is higher before visits to the neuron’s anchor in trial N+1 whether the anchor was not visited in trial N (left) or when it was visited in trial N (right). Wilcoxon tests: Anchor not visited in trial N: n=124 tasks, statistic=2839, $P=0.010$. Anchor visited in trial N: n=105 tasks, statistic=2321, $P=0.140$. In addition, an ANOVA on all data (N=105 tasks) showed no main effect of Past: $F=2.12$, $P=0.149$, $df_1=1$, $df_2=104$, a trend towards a main effect of Future: $F=3.07$, $P=0.083$, $df_1=1$, $df_2=104$, no Past x Future interaction: $F=2.34$, $P=0.129$, $df_1=1$, $df_2=104$. Right: Regression coefficients are insignificant, but positive, for the bump time but not any of the other control times. T tests against 0: “bump time”: N=124 tasks, statistic=1.4, $P=0.163$, $df=123$; “decision time”: N=124 tasks, statistic=0.28, $P=0.777$, $df=123$; “random time”: N=124 tasks, statistic=-0.77, $P=0.441$, $df=123$; “90 degree shifted time”: N=124 tasks, statistic=-3.37, $P=0.001$, $df=123$; “180 degree shifted time”: N=124 tasks, statistic=-2.15, $P=0.033$, $df=123$; “270 degree shifted time”: N=124 tasks, statistic=-2.99, $P=0.003$, $df=123$.

11. Figure 5H: this figure would be more impactful if the circles weren't colored in so that we can see neuron 2's firing.

We have now changed this figure as requested, making the maze circles grey to allow more easily visualising the coloured spiking.

Figure 5

- h) Two example paths each spanning an entire trial across two distinct tasks and overlaid spiking of 2 mFC state-tuned neurons anchored to early goal-progress in location 6 (middle right location on the maze). Neuron 1 is an anchor neuron and is hence tuned to this goal-progress/place. Neuron 2 fires with a lag of roughly 270 degrees in task space from the anchor and so fires 3 states after the animal visits early goal-progress in location 6. Thus neuron 2 fires when the animal

gets reward in location 5 in task 1 (reward **a**) as this is 270 degrees (3 states) in task space from reward in location 6 (reward **b**). In task 2, this neuron fires most in location 2 (reward **d**) which is also 270 degrees (3 states) from reward in location 6 (reward **a**). For visualisation purposes, spikes were jittered to ensure directly overlapping spikes are distinguishable.

Referee #2 (Remarks to the Author):

In this work, the authors aim to identify a neuronal algorithm for mapping abstract task structure. Broadly speaking, the study consists of two sets of findings: (1) prelimbic neural activity during a novel “ABCD task” in which freely-moving mice must navigate to looping sequences of 4 water reward locations on a 3x3 maze and learn to generalize this goal-directed behavioral sequence across different permutations of the 4 reward locations, and (2) the “Structured Memory Buffer” (SMB) model—a conceptual framework which attempts to explain how the organization of prelimbic neural activity is shaped by task structure in a manner that facilitates task generalization and retrieval of task-relevant behavioral policies.

Electrophysiological recordings revealed the following findings: (1) ~80% of prelimbic neurons in this study track progress towards individual goals. This “goal-progress” tuning is invariant to the spatial location of rewards and elapsed time, as well as invariant to across distinct permutations of reward locations (termed distinct “tasks”). (2) A subset of these goal-progress tuned neurons exhibit preference for distinct “states”—time periods in between two adjacent reward locations. State preference “remaps” across distinct permutations of reward locations, but is maintained across different sessions of the same permutation separated in time, suggesting that prelimbic neural activity is organized into task-space modules. (3) The firing rate of each module of neurons is “anchored” to a particular conjunction of goal-progress and place. Prelimbic neural activity can be explained by a ring-like state space (as a consequence of the task structure), which is preserved during offline states such as sleep. During task performance, activity evolves as a function of the number of goals obtained, eventually circling back to the anchor point.

The authors set out to test three predictions of the SMB model: (1) prelimbic neurons form “mnemonic fields” that track task-progress from specific goal-progress/place conjunctions; (2) “activity bumps” in prelimbic neural activity can be used to predict the mouse’s upcoming behavioral choice; and (3) pairwise coactivity during offline, sleep states preserve a ring-like neuronal state space observed during the ABCD task.

Overall, the authors have made a commendable effort in designing a complex, freely-moving behavioral task in mice and proposing a conceptual framework for understanding how frontal cortical dynamics mediate abstract schema learning. These features of the work are novel, significant and important.

However, in its current state, the study falls short of demonstrating that the proposed “unified model for behavioural schema and sequence memory” is implemented in mouse prefrontal cortex, let alone demonstrating how the identified “goal-progress sequences” may facilitate the “formation of a schema encoding any task-structure in the mFC”.

ABCD Task: I have several concerns regarding how the authors have chosen to interpret data and draw conclusions in light of the constraints introduced by their ABCD task.

Thanks very much indeed for your kind comments, and for your constructive criticism. We are certain it has improved the paper. Below we try to address your comments with clearer figures, new analyses, and new data. In particular, we have acquired a new dataset with neuropixels which includes some days with a sequence of 5, ABCDE, and some days with no tone at A, so we can provide new data directly answering some of the reviewer’s concerns.

1. The decision to consider different sequences as distinct “tasks” requires further rationale. While the mice are performing distinct permutations of sequences, I am not fully convinced this actually

represents distinct “tasks” or slight variations of the same task. In addition to citing precedence in the literature (if any), authors should consider providing some quantitative details about how “distinct” the sequences were in practice (beyond the qualitative descriptions provided in the Methods). How distinct or similar were the 3 “tasks” chosen for a given day?

We thank the reviewer for raising this point. We should clarify that the central claims of the paper are about how this abstract structure common amongst many sequences is encoded in the rodent brain. Indeed, the neuronal representation/algorithm we reveal concerns how to encode a new sequence that shares an abstract structure with old ones. As such, what we call “tasks” here indeed refers to different sequences that share the same abstract structure. We now clarify this in the second paragraph of the results section and in the methods under “Definitions”.

An important point is that, in the ABCD paradigm, task space is orthogonal to physical space - e.g. reward **a** and **b** are closer in task space than **a** and **c** but this has no bearing on how far they are in physical space (e.g. the physical distance between **a-to-b** is equally likely to be higher or lower than **a-to-c**; **Figure 1b**, **Extended data figure 1a**). Thus, to generalise task structure between sequences, animals need to form representations that track the animal’s position in task coordinates in a manner that is orthogonal to their position in spatial coordinates. This is key for appreciating the significance of the later finding that mFC schema representations are built on top of goal-progress sequences that track task position relative to goals rather than the distance/time taken between rewarded goals (**Figure 2 and Extended data figures 2 and 3**). In addition to showing orthogonality of task and physical space distances across tasks (**Extended data figure 1a**) we now provide additional quantification of the correlation between reward sequences experienced by each animal, showing that these are **uncorrelated** in all of the tasks from mice where the neuronal data was recorded (**Extended data figure 1b middle**) and when isolating the late 3-task days (tasks 21-40) where all neuronal data in this manuscript is collected (**Extended data figure 1b right**) and show a correlation coefficient (r)=0.002 when taking all recording days across all mice (**Extended data figure 1b left**). To compute these correlations, we not only compared sequences with each other, but also with shifted versions of each other to ensure we exhaustively capture any similarities in task sequences. We note that the small residual positive correlation (r =0.002) seen when taking all mice is largely due to the constraint on the first two tasks always having location 5 rewarded (to avoid mice circling around the outside on their first task exposures). We also note that, by design, the tasks are selected such that no two tasks experienced by the same animal are rotations of each other in physical space. These points are now clarified in the Methods section under “Training”.

Extended data figure 1

- Tasks were designed such that task space and physical space are orthogonal to each other. Left: schematic showing that optimal path lengths between rewarded goals differed both within and between tasks. Right: A bar plot showing that the task space “distances” between reward locations (how many task states are between the rewards) are not correlated with the physical distances in the maze (the optimal number of steps taken to reach reward). Data points are individual tasks. Pearson correlation: $r=6.04 \times 10^{-18}$ $P=1.0$; One-way ANOVA statistic=2.02, $P=0.147$, $df=12$
- Reward sequences showed minimal correlation across all tasks in all mice (left) and no correlations on all tasks for mice where neuronal data was recorded (7/13 mice middle) or on the late 3 task days from neural mice (right). Note that electrophysiological data in this manuscript is all collected from the late 3 task days. T-test against 0 correlation: All tasks: $N=13$ mice, $r=0.002$, statistic=2.91, $P=0.013$, $df=12$; All tasks from neural mice (mice where neuronal data was recorded): $N=7$ mice, $r=0.002$, statistic=1.81, $P=0.120$, $df=6$; tasks on Late 3-task days from neural mice (tasks where neuronal data was recorded): $N=7$ mice, $r=4.5 \times 10^{-4}$, statistic=0.22, $P=0.634$, $df=6$.

Crucially, we have now added a new data set where animals learn a new abstract task structure (ABCDE) and show that the same type of neuronal sequences represent the animal's position in this task space with different periodicity (**Extended data figures 3,7**). We expand on this under point 4 below.

2. The presence of the 2-second auditory tone “when the animal consumed reward” at reward location “a” makes it difficult to interpret what mice have learned as the “abstract task structure” and whether this truly constitutes an abstract, generalized task “schema.” Presumably the auditory tone only starts once the mouse pokes for the reward at location “a” and reward is delivered. But this auditory tone is nevertheless present across all “tasks,” thus serving as an explicit reinforcement of the looping structure. Can the authors provide additional insight about why the auditory tone was included at all, and how they consider its rule in the “anchoring” of “goal-progress” cells?

Thanks for prompting us to clarify this. We indeed present the tone only when the mouse pokes for reward at location **a**. We include this tone to create an equivalence across tasks for the **a** location, beyond a single trial memory of the first rewarded location. This would've been needed if the mFC neurons were anchored to the start of a trial (A), in order to measure neurons that generalise across equivalent states across tasks. We instead found that mFC neurons are not anchored to A (and hence not anchored to the tone: i.e. they do not fire with a fixed lag from the tone across tasks: **Figure 3a,b**).

We have now added additional tasks where a subset of animals saw new examples of the ABCD task with the tone occasionally omitted. We find that performance is unaffected by the presence or absence of the tone on a trial-by-trial basis (i.e. on trials beginning with a state A where the tone is omitted compared to trials beginning with A where the tone is presented **Extended data figure 1d,e**). This is true whether taking the proportion of transitions via shortest route or the number of steps taken relative to the optimal path length (**Extended data figure 1d**). It is also true whether considering the entire trial or when only considering the **d-to-a** transition (at the end of a trial where tone was omitted from state A), where the memory of the tone after reaching reward **a** might be relevant (**Extended data figure 1e**).

In regards to the reviewer's point about the role of the tone in determining the animals' generalisation behaviour, we note that the zero-shot inference (**Figure 1f, Extended data figure 1m**), which is our main measure for generalisation, cannot be explained by the tone as animals only hear the tone once they receive reward at location **a**. The tone itself cannot possibly bias animals to more readily take the shortest route selectively from **d** to **a** compared to **c-to-a** or **b-to-a** on the first trial because all of these involve returning to the same location (**a**) where the tone was sounded. As such, this zero-shot inference demonstrates that mice learn the abstract periodicity of the task. We now clarify these behavioural and neuronal points in the manuscript results section under “The ABCD paradigm: an abstract task structure guides rapid sequence learning and inference”.

Figure 3

- a) Polar plots showing state tuning of example neurons, simultaneously recorded from the same animal across multiple tasks on a given day. Each row is a neuron and each column is a session. Neurons readily remap their state tuning but maintain their goal-progress preference across tasks. State preference is maintained across different sessions of the same task (X vs X'). Angles (in degrees) of each cell's rotation relative to its tuning in session X are shown to the right of each session's polar plot.
- b) Top: A schematic showing how the difference in tuning angles for the same neuron across sessions is quantified. Bottom left: Polar histograms show that state-tuned neurons remap by angles close to multiples of 90 degrees, as a result of conserved goal-progress tuning and the 4 reward structure of the task. The orange shaded area represents neurons that maintain a consistent angle across sessions (with a tolerance of 45 degrees either side of 0 change). No clear peak at zero is seen relative to the other cardinal directions when comparing sessions spanning separate tasks (Two proportions test against a chance level of 25% N=1594 neurons; mean proportion of generalising neurons across one comparison (mean of X vs Y and X vs Z)=24%, $z=0.91$, $P=0.363$). Bottom right: Neurons maintain their state preference across different sessions of the same task (X vs X'. Two proportions test against a chance level of 25% N=1160 neurons; proportion generalising=78%, $z=25.7$, $P=0.0$).
- c) Top: A schematic showing how the difference in relative angles between pairs of neurons across sessions is quantified. Bottom left: Polar histograms show that the proportion of coherent pairs of state-tuned neurons (comprising the peak at zero) is higher than chance but less than 100%, indicating that the whole population does not rotate coherently. The orange shaded area represents pairs of neurons that maintain a consistent angle across sessions (with a tolerance of 45 degrees either side of 0 change). Two proportions test against a chance level of 25% N=35164 pairs; mean proportion of coherent neurons across one comparison (mean of X vs Y and X vs Z)=29.3%, $z=13.0$, $P=0.0$). Bottom right: As expected from panel b, the large majority of state-tuned neurons keep their relative angles across sessions of the same task (X vs X'. Two proportions test against a chance level of 25% N=23674 pairs; proportion coherent=63%, $z=83.5$, $P=0.0$).

Extended data figure 1

- d) Performance is unaffected by the inclusion of a tone at reward **a**. 3 mice were exposed to additional tasks (after completing task 40) where the tone at **a** was omitted on random trials. The tone or no-tone status of a trial refers to whether the tone was omitted at the **a** at the beginning of the trial. Left: Mean proportion of transitions where one of the shortest routes was taken N=26 tasks, Wilcoxon test; statistic=164, $P=0.784$. Right: Mean relative path distance N=26 tasks, Wilcoxon test: statistic=130, $P=0.258$
- e) Performance on the **d**-to-**a** transition is unaffected by the inclusion of a tone at reward **a** in the previous trial: Left: Mean proportion of transitions where one of the shortest routes was taken N=26 tasks, Wilcoxon test; statistic=134, $P=0.903$. Right: Mean relative path distance N=26 tasks Wilcoxon test: statistic=149, $P=0.515$

3. It is unclear if, at any point, whether animals were required to collect all 4 rewards in the correct sequence without visiting other reward wells. The main performance metric used in this study appears to be based on paths between individual rewards (i.e. shortest spatial paths between rewards for at least 70% of all transitions). If performance is assessed solely based on individual rewards, it is unclear how authors have assessed that mice have learned a “complex behavioral sequence” (which would seem to refer to the entire sequence of 4 locations, as well as the looping nature). Furthermore, it is unclear how the authors are distinguishing between the individual transitions vs. the overall hierarchy of behavioral structure (composed of the complete structure, rather than between two goals).

We thank the reviewer for the opportunity to clarify this. Animals were required to collect all 4 rewards in sequential order, e.g. reward **d** would only be available after reward **c** was delivered. However, we did not punish animals for taking longer routes or visiting other reward wells *en route* to the correct well. Their main motivation was to collect as many rewards as possible within the limited time allotted for a single session (20 mins), thus creating the pressure to minimise path length. Indeed we now show, in addition to quantifying individual transitions, that animals take shorter/shortest paths along the **entire sequence** significantly above chance. We show this by quantifying the proportion of trials in which animals completed an entire sequence of 4 reward locations on a given trial via the shortest distance (i.e. completed a “perfect” trial) and showing that this is significantly higher than chance and improves from early to late tasks (**Extended data figure 1i**). In a similar vein, in **figure 1d** we quantify the distance taken for the entire sequence of 4 locations (entire trial) as a proportion of shortest distance - we show that animals rapidly learn to take shorter distances across the entire trial more readily than chance within the first 20 trials, and that this improves in late compared to early tasks.

Importantly, we also note that the main measure we use to determine whether animals have learned the abstract task structure is the zero-shot inference (**Figure 1f**, **Extended data figure 1m**) which can only be assessed at a single transition per task (i.e. the d-to-a transition on trial 1). Crucially, despite this being a single transition, performing this zero shot inference on the first trial requires animals to know the entire task structure (4 rewards arranged in a loop) and track their position in order to return to **a** from **d** rather than from **b** or **c**.

Extended data figure 1

- i) Mean proportion of “perfect trials” where all transitions were taken via the shortest route. Left: Scatter plot of mean proportion of perfect trials in the first 20 trials of early vs late tasks. Wilcoxon test N=13 animals, Statistic=11.0 P=0.028. Right: bar plot of the same data showing that, for both early and late tasks, the proportion of perfect trials is significantly above chance: Ttest against chance (0.007): Early tasks statistic=2.55, P=0.025; Late tasks - statistic=4.06, P=0.002.

Figure 1

- d) Performance improved across the initial 20 trials of each new task. This improvement was markedly more rapid for the last 5 tasks compared to the first 5 tasks. A two-way repeated-measures ANOVA (N=13 mice) showed a main effect of Trial $F=11.7$ $P=1.5 \times 10^{-5}$, $df_1=19$, $df_2=228$, Task $F=35.0$ $P=7.1 \times 10^{-5}$, $df_1=1$, $df_2=12$ and a Trial x Task interaction $F=2.99$ $P=0.030$, $df_1=19$, $df_2=228$. Individual mouse performance lines are shown in a lighter shade for each task group.
- e) Performance on the very first trial improved markedly across tasks. A one-way repeated-measures ANOVA (N=9 mice – N.B. only 9 of the 13 mice were presented with all 40 tasks) showed a main effect of Task $F=2.73$ $P=0.016$, $df_1=7$, $df_2=42$. Individual mouse performance lines are shown in a lighter shade of grey (Note: 4 mice only completed 10 tasks each).

4. One key conclusion that the authors draw is that task structure—in this case, task periodicity—“paces” a sequential policy. However, in order to make this mechanistic claim (rather than reporting it as an observation), the authors would need to test this hypothesis more directly, e.g. by altering task periodicity to every 3 or 5 rewards. In addition, it is unclear how this would be relevant for tasks which do not have a periodic structure.

We thank the reviewer for this comment which has prompted us to add additional experiments, analyses and explanations which support and clarify our central claims. We have now added data from mice performing a new abstract task structure with a different periodicity (**ABCDE**: 5 rewards arranged in a loop) that speaks directly to this comment.

Before we address this specific point about task periodicity, we want to start with addressing the more general question of how our model is relevant to tasks without a periodic structure. We completely agree that the data, and particular instantiation of the model in the current study, do not trivially generalise to all tasks. We are sorry if we gave the impression that this was our claim. That was not our goal. Here, we wanted to demonstrate (i) how a schema is represented for a particular class of tasks, and (ii) how this representation allows generalisation over tasks with common structure. We further wanted to uncover principles that have the potential to generalise to many different forms of schemata. We think we have achieved this, and we hope that this is enough for a single paper!

The principles that we believe have the potential to generalise are:

- (a) that sequential structure is built on top of “goal progress” primitives -> this is a powerful principle because it means that if tasks are built on sub-elements, then generalisation can work even if sub-elements take different amounts of time or state transitions in different instantiations.
- (b) That mFC neurons are organised into structured memory buffers with intrinsic dynamics that mirror the structure of the task. These SMBs have properties of both sequence working-memory-like representations (in tracking a memory of a specific behavioural step) and schema representations (as they generalise across distinct tasks).

This shows that, in principle, these two functions of mFC can be unified under one representation/algorithm. We expand on this below, and have clarified these claims in the revised Discussion section, spelling out clearly what can be directly gleaned from our study and what principles have potential to generalise.

a-Our study reveals that SMBs are superimposed on more primitive goal-progress sequences. We additionally show that the same goal progress sequences can be sculpted into representations of different

abstract task structure periodicities (ABCD or ABCDE tasks). Our data therefore reveals, at the cellular level, highly primitive elements used to build different task-structure representations. Because goal-progress dynamics evolve in relation to goals rather than time or distance, they can act as primitives that construct more complex task progress representations, even if the task is not periodic.

While our data reveals primitive building blocks of abstract task structure, it does not deal with higher order composition. We do not address the question of how to re-use parts of different abstract task structure representations (e.g. ABCD and FGHI) to compose representations of a new abstract task structure (e.g. ABHI). Emerging theory and findings from other studies are beginning to reveal such higher order composition (e.g. Tafazoli et al bioRxiv 2024; Riveland and Pouget, Nature Neuroscience 2024) but this is beyond the scope of our study.

b-A key feature of the SMB representation is that it is not purely encoding abstract task coordinates but also embeds anchors (concrete behavioural steps). This brings together features of both working memory and schema representations under one roof. The parallel with working memory representations (e.g. Chiang and Wallis 2018) comes from the fact that each SMB tracks a memory of a given behavioural step (**Figure 5**), and hence at any one point the population simultaneously represents memories of multiple stimuli at different lags in the past. The parallel with schema representations comes from considering that SMBs are shaped by the abstract task structure (**Figure 6,7**) and immediately generalise to new behavioural sequences without the need for new representational binding (**Figure 5,6**). The same SMBs encode future steps across distinct tasks regardless of the specific reward sequence or trajectory taken (**Figure 6**).

Can this apply to non-periodic tasks?

The periodic nature of the ABCD task means that the SMBs are configured (i.e. activated in a particular order) based on a memory of the immediate last trial (**Figure 5**). Once configured, the SMBs allow retrieving the upcoming behavioural sequence (**Figure 6**). In principle, this mechanism can also work when the task schema encodes a sequence that is not explicitly periodic. In this case, the SMBs can be configured via instruction or simulation rather than direct experience. This is consistent with existing data. For example, neuronal data from the PFC of monkeys performing a non-periodic sequence working memory task shows that, during the post-instruction delay phase, the same observations are duplicated across different neuronal subspaces, with each subspace encoding a different ordinal position in the sequence (Xie et al, Science 2022). The SMB model provides a mechanism for how this organisation may come about. A single SMB under our scheme would span the neurons for a given observation at different ordinal positions: it's a pathway that links a single observation across the different subspaces. Each subspace would therefore comprise neurons that have the same lag from their anchor (observation) across different anchors. The SMB model postulates that neurons end up in these subspaces in the delay periods because the relevant SMBs are configured by instruction during the sample period to be active in a given order. The SMB model also explains how these representations can be "read out" to execute the sequence, postulating that in data like Xie et al's, we should be able to predict errors to retrieve a given observation at a given ordinal position from the timing of activity of a given SMB, analogous to our findings in **figure 6**. We note that this organisation into subspaces is a special case and SMBs can also exist without neurons of the same lag from their anchor being organised into a coherent subspace. More generally, the dual properties of SMBs embedding a rich representation of multiple behavioural steps while generalising across tasks that share the same structure, can also allow the flexible retrieval of never before seen sequences. For example, if the task is to perform the reverse of an instructed sequence, the same SMBs that would've encoded the forward sequence could be reconfigured by a top-down signal to fire in reverse order - no new representation would need to be built to encode the reverse sequence.

Thus, our data only speaks directly to the periodic case but the general principle of memory buffers whose dynamics are shaped by the task structure has the potential to explain the zero-shot emergence of representations in other types of tasks. Indeed, this feature of our SMB model has already inspired a normative theory that explains why such SMB-like representations emerge in the mFC, and in doing so

explains a wide range of neuronal data from the mFC across distinct tasks that are not necessarily periodic (Whittington et al 2023). We now clarify these points in the Discussion section of the revised manuscript.

We now turn to the question regarding task periodicity pacing a sequential policy.

Previous support

The neuronal data from the ABCD task show that activity of mFC neurons encodes the animal's future behavioural choices in a manner consistent with the claim that a sequential policy is "paced by the task periodicity" - this is because neurons that fire from a lag X (degrees in task space) from a previous visit to an anchor point predict behavioural choices at lag $360-X$ in the future even when accounting for autocorrelation in behaviour (**Figure 6, Extended data figure 8**). This supports the SMB model which proposes:

- 1) memory buffers are structured by the task periodicity
- 2) activity along these buffers is used for reading out the animal's next choice

However, we acknowledge that in the initial submission we had only shown this for one task periodicity (4 rewards: ABCD task) and that a more general demonstration of this principle would necessitate tasks with different periodicity. We provide such support below.

Additional support

1-We have now added a new data set where we tested animals on a series of ABCDE tasks (i.e. tasks with a periodicity of 5 rewards). These were presented after the animals initially learned ABCD tasks. Animals rapidly learned the ABCDE task (**Extended data figure 1 n,o**).

Extended data figure 1

- n) ABCDE task performance (relative path distance): After completing at least 40 ABCD tasks, two animals completed additional ABCDE tasks (11 and 13 tasks each) where tasks comprised a loop of 5 (instead of 4) rewards. Animals readily performed above chance in the first 20 trials, as demonstrated by comparing path length between goals to the shortest possible path (i.e. computing a "relative path distance" measure). T-test against chance (6.44): $N=24$ tasks, statistic=-30.0 $P=6.18 \times 10^{-20}$, $df=23$. Chance level was calculated empirically using the mean relative path distance across the first trial of the first 5 ABCD tasks.
- o) ABCDE task performance (proportion correct transitions): Animals readily performed above chance in the first 20 trials, as demonstrated by quantifying the proportion of transitions where animals took the shortest possible path. Wilcoxon test: $N=24$ tasks, statistic=0.0, $P=1.19 \times 10^{-7}$. Chance levels were derived empirically for each mouse using baseline transition probabilities calculated when animals explored the maze before experiencing any ABCD tasks: see Methods under "Behavioural Scoring".

2-We now show that, as in the ABCD task, state-tuned neurons are also strongly goal-progress-tuned in the ABCDE tasks (**Extended data figure 3**). In fact, neurons maintain their goal progress tuning across abstract task structures - i.e. each neuron's goal progress tuning pattern in ABCD tasks generalises to ABCDE tasks (**Extended data figure 3c-f**). In addition, state-tuned neurons in ABCDE tasks are predominantly conjunctive with goal-progress tuning (**Extended data figure 3a,b**). Moreover, we now use a nonlinear dimensionality reduction method which shows that task manifolds in both the ABCD (**Figure 2j,k**) and ABCDE tasks (**Extended data figure 3g,h**) are built from a backbone of goal-progress sequences. In point 3 immediately below we show that, just as in the ABCD tasks, conjunctive state-goal progress cells

can be organised into SMBs reflecting the ABCDE task structure. These results now lend more direct support to a key claim of the paper “that goal-progress cells in the medial frontal cortex may be elemental building blocks of schemata that can be sculpted to represent complex behavioural structures.”

Figure 2

- j) A plot of the mean task manifold derived from a Uniform Manifold Approximation and Projection (UMAP)-embedding along three dimensions. The same manifold is shown twice: Left, goal-progress tuning along the manifold; right, state tuning along the same manifold. The entire task manifold is composed of goal-progress subloops.
- k) Quantifications of distances along the 3-dimensional UMAP-derived manifold - across different states and opposite goal-progress bin (left), across different states but for the same goal-progress bin (middle) or the distances across different states and same goal-progress bin for a shuffled control. N= 20 double-days, T-tests (with bonferroni correction): Across-goal progress vs within goal-progress: statistic =6.09, $P=2.25 \times 10^{-5}$, $df=19$; Across-goal progress vs permuted control: statistic =26.0, $P=7.85 \times 10^{-16}$, $df=19$; Within goal-progress vs permuted control: statistic =8.63, $P=1.60 \times 10^{-7}$, $df=19$.

Extended Data Figure 3

- a) Polar plots of task-space tuning for 8 example neurons in the ABCDE task - neurons 1-4 are purely goal-progress tuned while neurons 5-8 are conjunctively goal-progress and state tuned.
- b) State neurons in the ABCDE task are predominantly goal progress tuned. Left: design of GLM to identify goal-progress tuned neurons in ABCDE tasks. Right pie chart showing the proportion of state-tuned neurons that are goal-progress tuned: ABCDE goal-progress/state GLM; Two proportions test: N=189 state neurons, proportion goal-progress-tuned= 85%, $z=15.6$, $P=0.0$
- c) Polar plots of task-space tuning for 3 example neurons recorded across 2 ABCD tasks and then two ABCDE tasks - neurons 1 is purely goal-progress tuned while neurons 2 and 3 are conjunctively goal-progress and state tuned.

- d) Goal progress tuning is maintained across abstract tasks ABCDE vs ABCD: The average firing rate vector of all neurons relative to an individual goal (from goal “n” to goal “n+1”; averaged across all states). Animals experienced 2 ABCD tasks followed by 2 ABCDE tasks on these days. Each row represents a single neuron and the neurons are arranged on the y axis by their peak firing goal-progress in task 1 in the ABCD condition. This alignment is largely maintained in tasks across both ABCD and ABCDE structures. White dashes indicate early intermediate and late goal-progress-cutoffs.
- e) A histogram showing the mean goal-progress-vector correlation across tasks for each neuron. One-sample T test against 0: N=111 neurons; statistic=23.8; $P=3.76 \times 10^{-45}$, $df=110$. Note that the neurons used in this panel are those on days where animals experienced both ABCDE and ABCD tasks.
- f) Left: design of GLM to identify whether neurons maintain their goal-progress tuning across ABCDE and ABCD tasks. Right: A histogram showing the mean regression coefficient values for goal-progress as a regressor across ABCD and ABCDE tasks for each neuron. One-sample T test against 0: N=111 neurons; statistic=7.43; $P=2.45 \times 10^{-11}$, $df=110$.
- g) A plot of the mean task manifold derived from a Uniform Manifold Approximation and Projection (UMAP)-embedding along three dimensions for mFC activity in the ABCDE. The same manifold is shown twice: Left, goal-progress tuning along the manifold; right, state tuning along the same manifold. The entire task manifold is composed of goal-progress subloops.
- h) Quantifications of distances along the 3-dimensional UMAP-derived manifold - across different states and opposite goal-progress bin (left), across different states but for the same goal-progress bin (middle) or the distances across different states and same goal-progress bin for a permuted control. N= 4 double-days - T-tests (with bonferroni correction): Across-goal progress vs within goal-progress: statistic =6.64, $P=0.021$, $df=3$; Across-goal progress vs permuted control: statistic =21.1, $P=7.02 \times 10^{-4}$, $df=3$; Within goal-progress vs permuted control: statistic =10.7, $P=0.005$, $df=3$

3-We now show evidence that state-tuned mFC neurons are organised into SMBs in the ABCDE task - neurons are anchored with a fixed lag from given location/goal-progress conjunctions that is fixed across tasks that share the same abstract (ABCDE) structure (**Extended data figure 7i**) - this is true even when considering lags that are unique to ABCDE tasks and not found in the ABCD tasks (i.e. neurons with a mnemonic field more than 4 states from the anchor).

Extended data figure 7i

- i) Histograms showing the right shifted distribution of mean cross-validated task map correlations between neurons aligned to their preferred goal-progress/place anchor (from training tasks) and the task map aligned to this goal-progress/place from a left out test task in ABCDE tasks. This correlation is shown for all state-tuned neurons (left), non-zero-lag state neurons (middle) and neurons with a lag of more than 4-states from the anchor (right). T test against 0: All state neurons: N=188 neurons, statistic=7.21, $P=1.38 \times 10^{-11}$, $df=187$; Non-zero-lag state neurons: N=153 neurons, statistic=6.32, $P=2.47 \times 10^{-9}$, $df=152$; >4-state lag from anchor neurons: N=31 neurons, statistic=2.59, $P=0.015$, $df=30$.

4-We also tested whether the future choice prediction in figure 6 is possible in the ABCDE tasks. These data show that, just as in the ABCD task, activity of mFC neurons predict the animal’s future behavioural choices in the ABCDE tasks. Neurons that fire at a lag X (degrees in task space) from an anchor point in ABCDE tasks predict behavioural choices at lag 360-X in the future even when accounting for autocorrelation in behaviour (**Extended data figure 8g**). This speaks directly to the point about task pacing as now the task has an extra reward, meaning that animals must use a new set of SMBs to predict where they will be in an extra state in the future (360 degrees now represents 5 rewarded states, i.e. 5 goal-progress cycles). Thus, SMBs can be built to represent behavioural sequences for different task periodicities.

Extended data figure 8

g

- g) Prediction of behaviour in the ABCDE tasks. Normalised firing rates of neurons during their “bump time”: i.e. the lag at which they are active relative to the anchor. Bump time activity is higher before visits to the neuron’s anchor in trial N+1 when the anchor was not visited in trial N (left) and also higher before visits to anchor in trial N+1 when the anchor was visited in trial N (right). Wilcoxon tests: Anchor not visited in trial N: n=24 tasks, statistic=73, P=0.027. Anchor visited in trial N: n=24 tasks, statistic=48, P=0.003. In addition, an ANOVA on all data (N=24 tasks) showed a main effect of Past: F=18.57, P=2.61x10⁻⁴, df1=1, df2=23, a main effect of Future: F=19.9, P=1.78x10⁻⁴, df1=1, df2=23, a Past x Future interaction: F=5.84, P=0.024, df1=1, df2=23. Right: Regression coefficients were positive for the bump time but not all other control times. T tests against 0: “bump time”: N=24 tasks, statistic=2.91, P=0.008, df=23; “decision time”: N=24 tasks, statistic=-1.17, P=0.252, df=23; “random time”: N=24 tasks, statistic=-1.33, P=0.197, df=23; “72 degree shifted time”: N=24 tasks, statistic=-1.97, P=0.061, df=23; “144 degree shifted time”: N=24 tasks, statistic=-0.8, P=0.43, df=23; “216 degree shifted time”: N=24 tasks, statistic=-0.83, P=0.417, df=23; “288 degree shifted time”: N=24 tasks, statistic=-0.88, P=0.390, df=23.

In addition, I have significant concerns regarding the presentation and interpretation of data:

1. Key details regarding the maze are missing. Based on the text, it is unclear which components of the maze are “connected,” as well as the size of the 3x3 grid maze. It would be helpful to include a photograph, or at least, a realistic schematic of the actual maze.

We now add a photograph of the maze in **figure 1a** (see figure under point 2 below) and annotate this with the maze dimensions (45x45 cm). We note that the connectivity shown via grey bridges in the schematic in **figure 1a** is true to the connectivity of the maze. We now clarify this in the figure legend.

2. The use of schematics is appreciated from the perspective of clarity but the lack of accompanying data makes some points difficult to interpret. For example, the authors should include an actual image of the task (as noted above) and some example trajectories to give a better sense of the behavior of the animals. Figure 1d and 1e present the behavior all collapsed into a few line graphs instead of showing individual animal data, which would better illustrate the degree of stereotypy or variability in the behavior. Individual traces for probe locations are not provided (outside a schematic in Figure 2 and a single image in the Extended Data), I would recommend the authors provide actual histology or probe locations for all of their animals.

We have now included an image of the task in **figure 1a**. We also include example trajectories across trials in the new **figure 1b**. We have selected trials that make the point that behaviour is often, but not always, stereotyped. Animals sometimes choose different routes to execute the same task. This is what gives us power to predict their upcoming actions over and above their history of previous choices. These changes in behaviour are what is predicted in **figure 6**.

Figure 1

- Task design: animals learned to navigate between 4 sequential goals on a 3x3 spatial grid-maze (measuring 45cm x 45 cm). The 9 maze locations were connected as shown in the top right photograph and adjacent schematic, with connections only along the cardinal directions. Reward locations changed across tasks but the abstract structure, 4 rewards arranged in an ABCD loop, remained the same.
- Example paths from 3 different mice performing 3 different tasks. Each row is a set of 5 consecutive trials from the same mouse and task. Single trial paths are superimposed upon whole session coverage shown in grey. Mice rapidly converged on near-optimal routes and used only a subset of the available paths.

Furthermore, as requested, we include individual animal data in Figures 1d and 1e.

Figure 1

- Performance improved across the initial 20 trials of each new task. This improvement was markedly more rapid for the last 5 tasks compared to the first 5 tasks. A two-way repeated-measures ANOVA (N=13 mice) showed a main effect of Trial $F=11.7$ $P=1.5 \times 10^{-5}$, $df_1=19$, $df_2=228$, Task $F=35.0$ $P=7.1 \times 10^{-5}$, $df_1=1$, $df_2=12$ and a Trial x Task interaction $F=2.99$ $P=0.030$, $df_1=19$, $df_2=228$. Individual mouse performance lines are shown in a lighter shade for each task group.
- Performance on the very first trial improved markedly across tasks. A one-way repeated-measures ANOVA (N=9 mice – N.B. only 9 of the 13 mice were presented with all 40 tasks) showed a main effect of Task $F=2.73$ $P=0.016$, $df_1=7$, $df_2=42$. Individual mouse performance lines are shown in a lighter shade of grey (Note: 4 mice only completed 10 tasks each).

We have now provided histologically-derived probe locations for all of the animals (Figure 2a, Extended data figure 2b). We expand on this under our response to comment 4 below.

3. In order to understand how representative the neural activity is, it would be helpful to have a table summarizing the number of units included from each subject for each analysis, considering the authors use different subsets for different analyses (e.g. using only consistently anchored neurons vs. state-tuned neurons vs. neurons with activity lagged at least 30 degrees from their anchor). Although a breakdown of “total number of neurons” is provided in methods, the authors report pooled values (e.g. “neuron-days”), making it difficult to understand if neural activity is representative or if mainly from a subset of subjects.

We have now made such a table, indicating the criteria for inclusion and the Ns for each figure panel (Table 1). We should also emphasise that electrophysiological data is collected across 7 mice for most conditions. We also show that the main results supporting SMBs are robust across mice (Extended data figures 6e and Extended data figures 7b,g).

4. Methodological details are missing. (a) How were neurons tracked across days? The methods section lacks sufficient detail, and it's unclear what procedure authors used, if any, to rigorously determine which units were the same across days. Please provide quality metrics, exclusion criteria, and dropout rate for assessing unit matching across the "double-days." (b) What was the laminar distribution of the prelimbic units in this study? Were units in cingulate cortex excluded? (e.g. based on the image provided in Extended Data Fig. 1, it is unclear whether units were considered across the entire shank, or only post-hoc confirmed to be in prelimbic).

Below we clarify both of these points in turn

a) We track neurons across days by concatenating all binary files from sessions recorded across two days, rather than relying on unit matching across days. A single concatenated binary file with data from 2 days is run through the standard kilosort pipeline to automatically extract and sort spikes. We manually curate the output of this file based on standard criteria:

- i) Less than 10% contamination of a refractory period (2ms) as determined by a spike autocorrelogram
- ii) Neurons where the firing rate in 3 or more sessions drops below 20% of the session with the peak firing rate are discarded.

Drop-out rate: By comparing the units designated "good" by kilosort before curation with the post-curation yield, we find a drop out rate of 51.4% for the concatenated double days. This is in comparison to a dropout rate of 29.7% for single days. These criteria and rates are now reported in the manuscript methods section under "Electrophysiology, spike sorting and behavioural tracking".

b) We have now included a detailed histology-based map of where all probe channels are for each mouse (**Figure 2a, Extended data figure 2b**). For the data set in the first submission (mouse id 0-4), we used Cambridge neurotech F-series (6-shank) probes implanted along a 1mm anteroposterior extent parallel to the midline and targeted to between -1.3 and -1.5mm from the brain surface. The contacts have a dorsoventral extent of 150 μm with a large majority sited in the prelimbic cortex (PrL; **Extended data figure 2b**). The revised manuscript now has an additional dataset using neuropixels probes (mouse id 5 and 6). This has allowed us to substantially extend the dorsoventral extent of our recordings to record from 3-3.5 mm along the DV axis, including regions such as secondary motor cortex (M2), dorsal and ventral anterior cingulate (ACC), and infralimbic cortex, as well as the prelimbic cortex. Crucially, we report that similar tuning properties are seen along these regions. Place tuning showed a slight dorsoventral gradient, with the proportion of place-tuned neurons being highest in infralimbic and lowest in anterior cingulate cortex. Other than this, we found no significant differences in basic tuning parameters (goal-progress and state: **Extended data figure 2e**), cross-task generalisation/coherence properties (**Extended data figure 4 j,k**), and the anchoring predicted by the SMB model (**Extended data figure 7h**).

We also used this histological data to infer the locations of each recorded neuron, by localising its channel in a given subregion based on a standardised mouse atlas in the HERBs software. Using this method, we found that 90.7% of all recorded neurons were histologically localised in mFC regions: 68.3% in Prelimbic cortex, 11.3% in Anterior Cingulate cortex, 6.1% in Infralimbic cortex and 5.0% in M2. Of the remaining 9.3%, 4.8% could not be localised to a specific peri-mFC region within the atlas coordinates as they were erroneously localised to peri-mFC white matter areas, likely due to variations between actual region boundaries and atlas derived ones, 2.2% were found in the dorsal peduncular nucleus, 1.1% in the striatum, 0.6% in the medial orbital cortex, 0.3% in the lateral septal nucleus and 0.3% in olfactory cortex. We use all recorded neurons in the analyses throughout the manuscript but indicate where these pertain to different mFC regions in **Extended data figure 2e, Extended data figure 4j,k and Extended data figure 7h**. We have detailed these proportions in the figure legend of extended data figure 2c below.

Figure 2

- a) Multi-unit recording set-up: animals were implanted with either Cambridge neurotech silicon probes with 6 shanks targeting 1mm of the anterior-posterior extent of the prelimbic region of the medial Frontal cortex, or single shank neuropixels probes. The diagram shows a 3D rendering of probe channel positions, with the inset showing mFC regions. This was made by matching probe tracks in perfusion-fixed coronal slices from recorded mice to mouse atlas positions via the HERBs software (Fuglstad et al 2023). M2: Secondary Motor cortex, ACC: Anterior Cingulate cortex, PrL: Prelimbic cortex, IRL: Infralimbic cortex.

Extended data figure 2

- a) Coronal slice from an implanted mouse showing silicon probe track terminating in the prelimbic region of mFC.
 b) The laminar profile of probe channel positions for each mouse. Shanks A-F in Cambridge neurotech probes are arranged posterior-to-anterior. 90.7% of all recorded neurons were histologically localised in mFC regions based on the inferred channel position: 68.3% in Prelimbic cortex, 11.3% in Anterior Cingulate cortex, 6.1% in Infralimbic cortex and 5.0% in M2. Of the remaining 9.3%, 4.8% could not be localised to a specific peri-mFC region within the atlas coordinates as they were erroneously designated to peri-mFC white matter areas, likely due to variations between actual region boundaries and atlas derived ones, 2.2% were found in the dorsal peduncular nucleus, 1.1% in the striatum, 0.6% in the medial orbital cortex, 0.3% in the lateral septal nucleus and 0.3% in Olfactory cortex.

Extended data figure 2

- e) The subregional distribution of neuron type coefficients along the medial wall of the frontal cortex in neuropixels recordings. One-way ANOVA: Left: Proportion of Goal progress neurons: $F=2.40$, $P=0.143$, $df=3$; Middle: Proportion of

state neurons $F=1.04$, $P=0.425$, $df=3$; Right: Proportion of place-tuned neurons $F=18.8$, $P=5.54 \times 10^{-4}$, $df=3$. Posthoc Tukey HSD tests: IrL vs PrL $P=0.049$; IrL vs ACC $P=0.000$; IrL vs M2 $statistic=0.003$, PrL vs ACC $P=0.021$.

Extended data figure 4

- j) The subregional distribution of single neuron generalisation (averaged across X vs Y and X vs Z comparisons) along the medial wall of frontal cortex in neuropixels recordings. One-way ANOVA: $F=1.59$, $P=0.323$, $df=3$.
- k) The subregional distribution of neuron pair coherence. Coherence is calculated across both X vs Y and X vs Z comparisons along the medial wall of frontal cortex in neuropixels recordings: One-way ANOVA: $F=4.76$, $P=0.083$, $df=3$.

Extended data figure 7

- h) The subregional distribution of cross-validated task map correlations between neurons aligned to their preferred goal-progress/place anchor (from training tasks) and the task map aligned to this anchor from a left out test task along the medial wall of frontal cortex in neuropixels recordings. One-way ANOVA: Left: All state neurons: $F=1.44$, $P=0.302$, $df=3$; Middle: Non-zero lag state neurons: $F=0.92$, $P=0.573$, $df=3$; Right: Distal (>90 degrees from anchor) non-zero lag state neurons $F=0.89$, $P=0.485$, $df=3$.

5. A minor point: “mFC” (medial frontal cortex) is not a commonly used term to refer to the prefrontal cortex in rodents (e.g. see review by Laubach et al., eNeuro, 2018; PMID 30406193). To avoid confusion, and to facilitate broader impact and reproducibility, authors should refer to the actual brain region surveyed (i.e. prefrontal cortex).

We appreciate this point regarding the term mFC not being commonly used. As mentioned above, we have now detailed both in the text and Extended data figure 2, the subregions of mFC from which we record in each mouse. Furthermore, we have assessed the major findings along areas of mFC including area prefrontal, infralimbic, anterior cingulate and M2 areas. Our analysis shows tuning properties are largely conserved across these medial frontal regions, with a slight gradient in place tuning. Other than this, no differences were seen for basic tuning parameters (goal-progress and state: **Extended data figure 2e**), cross-task generalisation/coherence properties (**Extended data figure 4 j,k**), and the anchoring predicted by the SMB model (**Extended data figure 7h**). In all other neuronal figures we report tuning in the “mFC” to refer to regions along the medial wall of the mouse frontal cortex.

I also have several concerns about how the predictions of the SMB model are presented and validated:

1. SMB model predictions are tested using linear and logistic regression models. It is unclear how the authors have considered nonlinearities stemming from neural activity as well as from the behavioral task itself. In the case of predicting behavioral choices, although the authors state that “previous choices up to 5 trials in the past” were included in the regression model to “remove confounds due to the autocorrelated previous behavioural choices”, there is insufficient detail to determine the extent to which the cyclical nature of previous behavioural choices can itself predict behavioural choices. In the case of neural activity, it is unclear whether the authors have considered nonlinearities emerging from the local recurrent microcircuitry (e.g. reviewed by XJ Wang, 2001; PMID 11476885).

We thank the reviewer for raising this question in regards to the effect of nonlinearities on the SMB model predictions.

1-With regards to the effect of previous behavioural choices. We now show the extent to which behavioural choices themselves predict behaviour, by reporting the regression coefficients from the logistic regression for choices up to 10 trials in the past (**Extended data figure 8b**). Note that we’ve extended this from the submitted manuscript (where we considered 5 trials in the past). For the first regression analysis in **figure 6c,d** and **Extended data figure 8b**, we add previous choices up to 10 trials in the past as individual regressors (each trial being a column in the independent variable matrix). Once we determine the regression coefficients for previous choices in **Extended data figure 8b**, we fit an exponential decay function to these coefficients and use this kernel in all subsequent regressions in **figure 6** and **Extended data figure 8** to weight the previous choices differently depending on how many trials back they happened. This creates a single regressor that accounts for **all previous choices up to 10 trials in the past**. These analyses show that, while previous choices impact the upcoming choice, mFC neuronal activity predicts behavioural choices above and beyond the effect of previous choices. Note that we only partially sample the neurons (i.e. record the activity of only a fraction of the neurons in the mFC) but we fully sample the previous behavioural choices so we cannot make comparisons between regression coefficients for neurons and those for previous behavioural choices. Moreover, this analysis does not assume that the effect of previous choices is linear or otherwise, but rather asks the question: if neuronal activity in SMBs is temporarily decoupled from previous choices - can we use the neurons to predict an upcoming choice regardless of the previous choice?. We discuss this further under point 3 below and illustrate precise predictions using data and schematics.

Extended data Figure 8

- b) Regression coefficients were significantly positive for the neuronal activity at the “bump time” and also for previous behavioural choices, gradually decreasing with trials in the past. T tests against 0: “bump time”: N=131 tasks, statistic=2.74, P=0.007, df=130; “n-1”: N=131, statistic=7.39, P=1.60x10⁻¹³, df=130; “n-2” N=131, statistic=8.03 P=5.04x10⁻¹³, df=130; “n-3” N=131, statistic=4.77 P=4.80x10⁻⁶, df=130; “n-4” N=131, statistic=3.38, P=9.45x10⁻⁴, df=130; “n-5” N=131, statistic=4.36, P=2.58x10⁻⁵, df=130; “n-6”: N=131, statistic=1.76, P=0.080, df=130; “n-7” N=131, statistic=2.83 P=0.005, df=130; “n-8” N=131, statistic=2.19 P=0.030, df=130; “n-9” N=131, statistic=2.42, P=0.017, df=130; “n-10” N=131, statistic=0.40, P=0.691, df=130.

2- We are a little confused by the questions of nonlinearities with respect to the neural data (**Figure 5**) given the analyses we have carried out. We believe that, given the nature of the analyses, any nonlinear

effects could not change the conclusions in the manuscript. We outline the reasoning below, and we also repeat the regression analysis with a linear-nonlinear-poisson model.

These analyses are designed to **extract tuning curves** as a function of anchor and lag. The first (**Figure 5a**) finds anchor-lag tuning by a linear regression. This is equivalent to finding spatially-tuned cells by binning behaviour into a 2D grid and regressing the visits to each grid location onto the activity of the neurons. In this analysis, we are computing average firing rates across all the examples where the animal is at a particular lag from a particular anchor, but we have to be more careful because the animal can be 2 rewards from anchor A and 3 rewards from anchor B at the same time. The regressors are not orthogonal. To account for this covariance, we fit the different regressors together in the same linear model. All we are doing therefore is figuring out the best tuning curve for each neuron. Critically, we then test this in cross-validated data.

Nonlinearities in such a model:

The analysis described above uses linear regression to find the tuning curve that best explains the data after accounting for covariances. What would nonlinearities mean in such a model?

- (1) It could mean that the different regressors could **interact** to further explain the neural data. For example, there might be a neuron that fires on average x spikes when you are 2 rewards from anchor A, y spikes when you are 3 rewards from anchor B, but $x + y + z$ spikes when you are **both** 2 rewards from anchor A **and** 3 rewards from anchor B. In such a cell z is the (2-way) nonlinear portion of the tuning curve. Critically though, if we were misattributing variance from x or y to z , then this would not work on cross-validated data because the interactions are different in the test dataset as the behavioural sequence is orthogonal between test and training data.
- (2) It could mean that there was a nonlinear link function between behaviour and neural response, such as a linear-nonlinear-Poisson (LNP) model traditionally used to model neuronal tuning (e.g. Hardcastle et al Nature Neuroscience 2017). The effect of this is again to perform an average of all the spikes according to different conditions, but to weight the different data points differently - for example: high firing bins given less weight (as variance is proportional to mean in the poisson model). Again, if this different weighting of different data points were the cause of our effects, they would not generalise to a held out (orthogonal) task. Nevertheless, we now run an LNP model in **Extended data figure 6d** and show that the same result holds even when the link function is nonlinear (logarithmic).
- (3) We also find strong support for the same predictions using two additional methods that do not rely on regression. The lagged spatial map analysis in **Figure 5c,d** shows that neurons consistently fire in relation to where the animal was a set lag in task space in the past regardless of where it is currently (across many tasks). This analysis does not assume any linear relationship between location visits and neuronal firing. The same is true for the anchor-aligned task map analysis in **figure 5e-g** which also makes no assumptions about linearity. Crucially, in both cases, the critical test is the firing in a left out test session not used to fit the optimal anchors/lags of the cell.

Extended data figure 6

- d) Histograms showing the right-shifted distribution of mean cross-validated correlation values between model-predicted (from training tasks) and actual activity (from a left out test task) for state neurons using a Poisson regression model. Left: this correlation is shown for all state-tuned neurons; Middle: only state-tuned neurons with non-zero-lag firing from their anchors; and Right: non-zero lag state-tuned neurons with the maximum regression coefficient value a whole state (90 degrees) or more either side of the anchor. T test against 0: All state-tuned neurons N=489 neurons, statistic=10.7, $P=2.86 \times 10^{-24}$, df=488; Non-zero lag state-tuned neurons N=346 neurons, statistic=4.74, $P=3.09 \times 10^{-6}$, df=345; distal (90 degrees) non-zero lag neurons: N=229 neurons, statistic=2.81, $P=0.005$, df=228.

2. There is a somewhat circular logic to the experiments/analyses, and it would be helpful if the authors can provide a more explicit, clear discussion to address this. For example, in Figure 7, rejecting the “delay line” hypothesis is not necessarily the strongest argument for the ring structure hypothesis. The authors are claiming that task structure organizes neural activity, and therefore, they should see this task structure reflected in the neural activity. Why, then, would they expect forward distance to be a good comparison when the “delay line” arrangement is clearly not reflective of the task structure?

We thank the reviewer for raising this important point. We want to clarify that the analysis in **figure 7** shows that neurons sharing the same anchor are **internally organised into a task-shaped ring, even in the absence of the task itself**. The analysis does not simply reject the delay line hypothesis, but also directly affirms the ring hypothesis. We show that circular distance is negatively correlated with sleep coactivity using a regression that takes into account other variables, including forward distance (as well as spatial similarity and goal-progress distance). If the distribution of lags from the anchor was uniform, forward and circular distances would be orthogonal and so adding forward distance to the regression would be redundant. However, the distribution of lags from the anchor is not uniform (**Extended data figure 7e**) and so we add forward distance to the regression to remove any possible contribution of delay lines to the results. We elaborate more on this below.

In the parts leading up to the sleep analysis we show that neurons are organised sequentially relative to each other (**Figure 3**) and relative to anchor points (**Figure 5**), firing consistently at fixed lags from these anchors. We further show task structuring of the SMBs by illustrating that neurons can be used to predict animal’s future choices in a manner paced by the task periodicity (**Figure 6**; see response to point 3 below for more detail on this point). Having established this task structured, sequential activity, what we’re aiming to do with **figure 7** is to test i) whether this sequential activity is internally organised (i.e. present in the absence of any structured task input) and ii) whether the internally organised sequential activity is open (creating a delay line) or closed (creating a ring).

Visits to the anchor could, in principle, elicit neuronal sequences that **end** at neurons with firing fields at the farthest point from the anchor; creating a delay line. If these sequences are instead closed to create a loop, neurons farthest from the anchor in the forward direction are actually close to the anchor in neuronal state space (**Figure 7a**). In such a circular neuronal state-space, neurons far from the anchor in the forward direction are functionally close to the anchor itself. This has implications for how information could be “read out” to trigger the next step in the behavioural sequence. If the neuronal state space is a loop, activity could flow back to the anchor to trigger animals to return to the anchor location. In this scenario, the same neurons can, in principle, be used as both the anchors that trigger the sequence and “read out neurons”

that are used to guide the animal to return back to the anchor in the following trial. A scenario where SMBs are instead structured into a delay-line would implicate another read-out mechanism, potentially relying on activity of separate output neurons at the end of the delay line to guide behaviour back to the anchor point. We note that, in both cases, the SMBs would be structured by the abstract task periodicity (i.e. there should be 4 rewarded states between equivalent visits to an anchor).

We have clarified this point in the text and also added another analysis that quantifies/visualises the relationship between circular distance, forward distance and pairwise sleep cross-correlations of co-anchored neurons (**Figure 7c**). Moreover, this analysis allows us to assess the nature of internal organisation within **individual** memory buffers - we find that neurons that share the same anchor are more co-organised than those with different anchors (**Figure 7d**). This supports the specific predictions of the SMB model, which relies on i) each memory buffer being structured by the task and ii) memory buffers being independent, in order to flexibly encode new tasks. We have now clarified these points in the results section of the manuscript (under “3-Internally organised memory buffers”) and in the methods section under “Sleep/Rest analysis”.

Figure 7

- a) Left: Schematic showing potential neuronal state spaces - if neurons are arranged on a ring, then circular distance is a better description of how close two neurons are in state space than forward distance relative to the anchor. Conversely, if neurons lie on a delay line, forward distance is a better description of neuron-neuron co-firing relationships. Right: Schematic showing the inputs and outputs of linear regression model relating pairwise circular distance and forward distance with coactivity during sleep while regressing out each other as well as pairwise goal-progress tuning distance and spatial map similarity.

- b) Regression coefficient values for circular (left) or forward (right) distance regressed against sleep cross-correlation for co-anchored neurons - T test relative to 0: circular distance: N=430 pairs, $t=-2.66$ $P=0.008$, $df=429$; forward distance: N=430 pairs, $t=1.61$ $P=0.108$, $df=429$.
- c) Left: A plot of cross-correlations during sleep between pairs of neurons sharing the same anchor against forward distance (bottom x axis) or circular distance (top x axis). Schematics at the top and bottom show example pairs that would fall into each category and their circular and forward distances. A delay-line state space would result in a uniformly negative relationship between forward distance and cross-correlation. A circular state space would result in a v-shaped relationship between forward distance and cross-correlation, with a negative slope when forward distance is between 0-180 degrees (where forward and circular distances are identical) and a positive slope when forward distance is between 180-360 degrees (where circular distance is negatively correlated with forward distance). Right: Correlation coefficients between pairwise forward distance and pairwise sleep cross-correlations are positive for pairs of neurons with 180-360 degrees forward distance and negative for pairs with a forward distance of 0-180 degrees. T tests relative to 0: 0-180 degrees: N=59 sleep sessions, $statistic=-5.16$, $P=1.07 \times 10^{-4}$, $df=58$; 180-360 degrees: N=59 sleep sessions, $statistic=4.16$, $P=1.08 \times 10^{-4}$, $df=58$, Paired T test: N=59 sleep sessions, $statistic=-5.55$, $P=7.35 \times 10^{-7}$, $df=58$.
- d) Regression coefficient values for circular distance against sleep cross-correlation using pairs of neurons consistently anchored to the same anchor (within) vs pairs of consistently anchored neurons that have different anchors (between). Regression coefficient values were more negative for pairs sharing the same anchor (within) compared to pairs across anchors (between) across all sleep. One-tailed unpaired T test (Welch's T-test): All sleep: N=430 pairs (within), 13932 pairs (between), $t = -1.80$, $P=0.036$, $df=14360$.
All error bars represent the standard error of the mean.

3. The assumptions underlying model prediction #2 (“distal prediction of behavioural choices”) are confusing. In particular, the authors state “The SMB model proposes that activity restricted to this ‘bump-time’ can be used to predict whether an animal will return back to this same behavioral step at the same point in the next trial” and furthermore claim this is a “unique prediction” (pg. 17). However, it appears that because of the ABCD task structure (and given the limited options the animal has at any given point), the animal will return to the same spatial position in an allotted time window on the next trial – thus, rendering the prediction of “whether an animal will return...” somewhat obvious.

We thank the reviewer for asking us to clarify this crucial point, Prediction 2 “distal prediction of behavioural choices” is possible because there is variability in the specific trial-by-trial routes animals take between goals. We can tap into this variability to ask whether neurons can predict distal choices of the animal. Animals can potentially take multiple routes (including multiple optimal routes) and so can be at one location at one point in trial N and at a different location at the same point in trial N+1. This is true for non-rewarded locations, as animals can take different routes between rewards. It is also true for rewarded locations, as animals could be one-step away from the rewarded location in trial N+1 and yet erroneously choose a different alternative location despite making the correct choice at the same point in trial N. In addition to now showing examples of the animal’s own trial-by-trial behaviour (**Figure 1b**), we now add 2 new schematics (**Figure 6a, Extended data figure 8a**) to illustrate how the predictions made by the SMB model relate to specific scenarios and how our analysis gets at those predictions.

The SMB model allows us to make spatio-temporally precise predictions - if we know each neuron’s anchor and its lag from the anchor (its position on an SMB) we can determine which neurons can predict which choices at which lags in the future. Crucially, this prediction **generalises across tasks with different reward locations and trajectories**. By knowing only the neuron’s anchor and its lag (X) from the anchor we can predict whether the animal will return to this anchor at lag (360-X) in the future regardless of the intervening steps (**Figure 6, Extended data figure 8**). Importantly, this is true while controlling for previous choices, which we do throughout **figures 6 and Extended data figure 8** (see response to point 1 above for more detail on this). If the activity of SMBs was completely deterministic, noise-free and perfectly coupled to behaviour we would expect perfect stereotypy. Once an SMB is set off at a particular point in the task it will continue to bias the animal to return to this anchor point in all trials to come. This would indeed make it impossible to test whether SMBs predict choices on a trial by trial basis, beyond simply showing that SMBs are anchored to these choices (**Figure 5**). However, if there is variability due to e.g. noise, or a deliberate top-down input that signals a change in choices, then we should be able to relate the activity of SMBs to changes in behavioural choices (**Figure 6a, Extended data figure 8a**). In effect we’re asking: what happens if neuronal activity in SMBs is temporarily decoupled from previous choices - can we use the

neurons to predict an upcoming choice regardless of the previous choice? The data in **Figure 6** and **Extended data figure 8** show that we can indeed make this prediction.

We also want to comment on what precisely is novel about this result, and how this follows from the SMB model. Because the SMBs are not purely encoding abstract task coordinates but also embed anchors (concrete behavioural steps) this allows encoding new behavioural sequences in the activity of mFC neurons. This in turn results in neurons that predict the future **in a way that generalises across tasks and trajectories**. This is unlike Hippocampal splitter cells (Wood et al 2000; Frank et al 2000), and other latent representations (e.g. Bower et al 2005; Dupret et al 2010; Liu et al 2023) which predict specific transitions given a specific starting position - the mFC neurons predict future steps regardless of current position or subsequent trajectory taken to reach this step. In fact we exclude cells that are at close-to-zero lag from the anchor in this analysis. Crucially, we can tell precisely which neurons predict which choice at which lag in the future - given each neuron's mnemonic lag from its anchor on a given SMB (**Figure 6; Extended data figure 8**). At any one point in time, different cells are active for different future positions at different future lags. This means the whole future trajectory is simultaneously available in the instantaneous firing pattern in mFC. This is therefore fundamentally unlike the encoding of future behavioural choices that is itself based on sequential activity, such as that found in Hippocampal neurons during theta sweeps (e.g. **Johnson and Redish 2007**) and awake replay (e.g. **Pfeiffer and Foster 2013**). This has concrete computational implications. For example, any area using this mFC representation for computation can now compute with the entire plan simultaneously.

Figure 6

- a) Schematic showing distal prediction of animal's choices from memory buffers. The SMB model allows us to predict the animal's choices 1) at a precise lag in the future and 2) in a way that generalises across tasks. The larger the size of an activity bump on a given SMB the more likely the animal will subsequently visit the SMB's anchor. The timing of this bump determines when the anchor will be visited. In this example we show an SMB with an anchor at location 2 (top middle location shaded in brown) at intermediate goal progress (i.e. half way between goals). We highlight two neurons on this SMB - one in brown which is at zero lag from the anchor (i.e. fires immediately when the anchor is visited) and another in green which fires at a lag of 135 degrees (1.5 states) from the anchor. We show 4 rows corresponding to 4 possible scenarios across two different tasks to illustrate the key features of the SMB model's predictions. In all trials, the animal starts off at the anchor point (brown location) at t1. In the first scenario (top row) of Task X the anchor visit triggers a large bump of activity at t1, which travels around the ring (e.g. at t2) and when it reaches the SMB's decision point at the end (t3) it biases the animal to return to the anchor. In the second row, the bump initiated by the first anchor visit (t1) is smaller and so the animal is less likely to visit the anchor at t3. The activity of the green neuron at precisely 1.5 states since the first anchor visit (t2; i.e. the green neuron's "bump time") can be used to predict what will happen at t3 (i.e. 2.5 states

forward from t_2). If the green neuron has a high firing rate at t_2 (darker green shading), the animal should return to the anchor at t_3 . Conversely, low firing rate of this neuron at t_2 (light green shading) predicts no return to the anchor at t_3 . Thus the SMB model allows predicting behavioural steps at precise lags in the future. The SMB model also posits that this future prediction should generalise across tasks and hence be independent of where the animal is at a given time. This can be seen in Task Y. Now the green neuron fires at a different location in order to keep its lag from its anchor. And yet, its activity at t_2 can be used in the same way to predict whether the animal will visit the anchor at t_3 (i.e. 2.5 states in the future). Again the higher the activity at t_2 the more likely the anchor will be visited at t_3 . To test this latter point, we only consider non-zero lag cells in all of the analyses below, which means we only use neurons that fire in different locations across different tasks. We indicate the lag threshold separately for each panel.

Extended data figure 8

- a) Schematic showing distal prediction of animal's choices from memory buffers as a function of previous trial choices. By harnessing variability in the coupling between anchor visits and bump initiation, we can test whether SMB activity can predict future choices when controlling for previous choices. In this example we show an SMB with an anchor at location 2 (top middle location shaded in brown) at intermediate goal progress (i.e. half way between goals). Each row shows a different scenario across two consecutive trials (trial N-1 and trial N) in the same task and the expected activity of neurons anchored to this location/goal-progress conjunction. Scenario 1 (0:0): the animal doesn't visit the anchor in trial N-1 and hence the activity of neurons on the SMB for this anchor is low. This results in the animal not visiting the anchor in trial N. Scenario 2 (0:1): the animal again doesn't visit the anchor in trial N-1 but this time the activity of neurons on the SMB is high (e.g. due to noise or top down modulation). This results in the animal visiting the anchor in trial N. Scenario 3 (1:0): the animal visits the anchor in trial N-1 but the activity of neurons on the SMB for this anchor is low (e.g. due to noise or top down modulation). This results in the animal not visiting the anchor in trial N. Scenario 4 (1:1): the animal visits the anchor in trial N-1 and the activity of neurons on the SMB for this anchor is high. This results in the animal visiting the anchor again in trial N. In effect the variability in the size of the bump in relation to the anchor visit allows us to decouple the SMBs' activity from previous choices. This makes it possible to test whether SMB activity predicts future choices of the animal. Note that the time stamps shown on the SMB (t_1, t_2 and t_3) are all in trial N.

4. One key conclusion the authors draw is that the SMB model precludes the “need to build or bind new representations.” In general, it is unclear how the SMB model may extend beyond the ABCD task however, and such a blanket statement seems inappropriate.

We agree with the reviewer that clarification is needed here. This statement that “*the SMB model ...precludes the need to build or bind new representations*” is true for new sequences sharing the **same** abstract task structure to old ones. Having built the SMBs for a given abstract task structure during learning, a new task (a new sequence) that shares this same abstract task structure can be represented by reconfiguring pre-existing, pre-anchored SMBs, without the need for building/binding representations. We demonstrate this empirically in figures 5 and 6 where we show that SMBs maintain their sequential structure across task sequences (**Figure 5**) and predict the animal's behavioural choices across task sequences (**Figure 6**). We nevertheless agree that some type of plasticity could be needed to build

representations of **new** abstract task structures (e.g. the ABCDE task), which require building new SMBs with a different structure.

We also want to highlight our responses to point 4 in the section above “4. One key conclusion that the authors draw is that task structure—” which illustrate what aspects of the SMB model generalise beyond periodic tasks and what aspects are specific to our ABCD loop (and other sized loops).

5. This is a minor point, but it is somewhat confusing that the authors use terminology associated with different machine learning frameworks without providing sufficient rationale and introduction. For example, the authors quite abruptly start to refer to “policy” with no clear indication of how this emerges from the model they propose. Are they referring to “policy” as in a reinforcement learning framework, or a different definition? To a broad audience, the use of such terms with very specific connotations may be misleading.

We apologise for the liberal use of machine learning terminology. To make the manuscript clearer and more accessible to a broad audience, we now replace the word “policy” with behavioural trajectory or sequence in the main text but introduce it in the discussion when referring to implications for reinforcement learning models.

Referee #3 (Remarks to the Author):

1-In this manuscript, El-Gaby et al. present evidence that neuronal activity in the medial prefrontal cortex (mPFC) of rodents track the progression of animals towards series of rewards that the animals learn to find in a sequential manner. After learning, animals are trained on a different task in which the location of rewards is switched, and so on. Many neurons tended to be correlated with the specific phase of the task, irrespective of the reward location, while some others were correlated with either the spatial location or one specific reward in the sequence. The study then presents a theoretical framework that accounts for these observations, whereby the mPFC would contain modules of neurons whose population activity shows ring-like topology, reflecting the sequential nature of the task and enabling to flexibly transfer knowledge of the task structure to a new task. Overall, while the study is interesting and offers new insights into task learning correlates in the mPFC, many findings seem to echo previously published work (which are too often not cited), casting doubt on the novelty of the study.

We thank the reviewer for acknowledging that our study is “*interesting and offers new insights into task learning correlates in the mPFC*”. We also thank them for raising important points that need to be clarified regarding the relationship between our findings and previous work on the mFC and hippocampus. We have now added new data sets, analyses and clarifications that address all of the reviewer’s concerns. We believe these additions make our findings better supported and more clearly articulated.

2-A first impression while reading the manuscript is the lack of proper citation of the relevant literature. Although manuscripts are constrained by citation limits and may omit some important studies, it is surprising that this manuscript lacks references to several landmark studies on the role of the mPFC in rule-switching (Birrel and Brown, J Neurosci., 2000) and on neuronal correlates of sequence learning in the hippocampus (Bower et al., J Neurosci., 2005; Dupret et al., Nat Neuro, 2010) and the mPFC (Euston and McNaughton, J Neurosci., 2006), as well as coding of reward expectancy in the mPFC (Pratt and Mizumori, Beh Brain Res., 2001) and learning-associated sleep activity patterns in the mPFC (Euston et al., Science, 2007; Peyrache et al., Nat Neuro, 2009). Most critically, a recent study has specifically addressed the internal ring-like representation of task sequence and its generalization across contexts in the mPFC and not the hippocampus (Tang et al., Cell Reports, 2023). And this is only to cite a few missing references.

Thanks very much indeed for highlighting these important studies. We are happy to cite them. We would just like to be clear though that the current paper is showing a very different type of representation to any of these previous studies. We summarise here, and provide an expanded explanation, with additional data, below.

In short, our study reveals a generalizable neuronal schema in the mFC that represents behavioural tasks comprising multiple, hierarchically organised goals. The representation we find is:

1-Hierarchical: Our findings (**Figure 2; Extended data figures 2,3**) reveal mFC represents the animal’s position in a **multi-goal task** using a hierarchical structure composed of **concatenated goal progress sequences**. This is different from other studies showing schema representations in the mFC such as those we cite in the original manuscript (e.g. Rubin et al 2019; Kaefer et al 2020, Basu et al 2021; Samborska et al 2021) and those we now add (e.g. Pratt and Mizumori 2001, Peyrache et al 2009, Tang et al 2023). These previous representations concern either tasks that can be solved one goal at a time (e.g. Peyrache et al 2001, Kaefer et al 2020) or where the analysis concerns neural correlates of progress in relation to a single goal (e.g. Basu et al 2020; Samborska et al 2021) or subgoal (e.g. Tang et al 2023). They don’t therefore reveal the hierarchical organisation we see across a sequence of goals.

2-Modular: We find that mFC neurons are divided into multiple modules each of which tracks progress in the overall task from a distinct behavioural step - we call them structured memory buffers (SMBs). The key point here is that **this is not one ring (or even one manifold) that generalises across tasks**. Rather the sequential activity within (but not across) SMBs is invariant across tasks (**Figure 3-5 and Extended data figures 4-7**). This is different from previous mFC schema papers (e.g. Pratt and Mizumori 2001, Peyrache et al 2009, Kaefer et al 2020, Basu et al 2021; Samborska et al 2021) that all concern a single generalizable manifold. Moreover, SMBs bring together features of both working memory and schema representations under one roof, helping unify two key functions of mFC. The parallel with working memory representations comes from the fact that each SMB tracks a memory of a given behavioural step (**Figure 5**), and hence at any one point the population simultaneously represents memories of multiple stimuli at different lags in the past. The parallel with schema representations comes from considering that SMBs are shaped by the abstract task structure (**Figure 6,7**) and generalise to new behavioural sequences (**Figure 5,6**). Again we now better clarify in the revised Discussion section how this relates to the multi-goal nature of our task and analyses.

3-Predictive: Because the SMBs are not purely encoding abstract task coordinates but also embed anchors (concrete behavioural steps), this allows encoding new behavioural sequences in the activity of mFC neurons. This in turn implies neurons that predict the future in a way **that generalises across tasks and trajectories**, which we demonstrate empirically (**Figure 6; Extended data figure 8**). This is unlike Hippocampal splitter cells (Wood et al 2000; Frank et al 2000), and other latent representations (e.g. Bower et al 2005; Dupret et al 2010; Liu et al 2023) which predict specific transitions given a specific starting position - the mFC neurons predict future steps **regardless of current position or subsequent trajectory** taken to reach this step. In fact we exclude cells that are at close-to-zero lag from the anchor in this analysis. Crucially, we can tell precisely which neurons predict which choice at which lag in the future - given its position on an SMB (**Figure 6; Extended data figure 8**). At any one point different cells are active for different future positions at different future lags. This means the whole future trajectory is simultaneously available in the instantaneous firing pattern in mFC. This is therefore fundamentally unlike the encoding of future behavioural choices that is itself based on sequential activity, such as that found in Hippocampal neurons during theta sweeps (e.g. Johnson and Redish 2007) and awake replay (Pfeiffer and Foster 2013). This has concrete computational implications. For example, any area using this mFC representation for computation can now compute with the entire plan simultaneously.

This same feature of SMBs, that they embed concrete behavioural steps (they have a fixed anchoring to a specific conjunction of place and goal-progress), means that new task sequences can be rapidly encoded in the activity of mFC SMBs without needing new binding between abstract task structure and location-specific representations. We speculate that this rapid cross-task generalisation property of SMBs is key to the necessity of mFC for rapid task switching (Birrel and Brown 2000; Dias et al 1996; Owen et al 1991). However, we caution that a robust causal demonstration of this specific mechanism (rather than a replication of mFC's role in task switching) requires complex causal interventions that are beyond the scope of this study (we discuss this in more detail under our response to point 4 below).

We discuss the relationship between our findings and Euston and McNaughton 2006 under our response to point 3 below. We also discuss how our findings relate to learning-associated sleep activity patterns in the mPFC (e.g. Euston et al 2007; Peyrache et al 2009; Kaefer et al 2020) under our response to point 7 below. But before doing this we expand on the points above by highlighting additional support in the revised manuscript.

Additional support

We have now added a new data set where animals complete a series of tasks from a new abstract task structure (ABCDE) where the periodicity of the abstract structure is now a loop of 5 instead of 4 rewarded goals. This has allowed us to further validate the generality of the 3 major findings mentioned above:

1-Hierarchical representation: Goal-progress sequences as a primitive for mFC schema representations.

We now show that goal progress tuning is conserved between ABCD and ABCDE tasks and state-tuned neurons are strongly goal-progress-tuned in both ABCD and ABCDE tasks (**Extended data figure 3**). The same goal-progress sequences are reused to represent this new abstract structure with different periodicity. This points to goal-progress sequences as a general primitive in the mFC that can in be used to construct representations of different periodicities. Moreover, inspired by the reviewer’s suggestion below, we used a non-linear dimensionality reduction approach (Uniform Manifold Approximation and Projection: UMAP) to demonstrate that the mFC manifold encoding task states in a single task is hierarchically composed of goal-progress manifolds both for ABCD and ABCDE tasks (**Figure 2g,h; Extended data figure 2f,g and Extended data figure 3g,h**). We discuss this population analysis in more detail under our response to point 6 below.

Extended Data Figure 3

- Polar plots of task-space tuning for 8 example neurons in the ABCDE task - neurons 1-4 are purely goal-progress tuned while neurons 5-8 are conjunctively goal-progress and state tuned.
- State neurons in the ABCDE task are predominantly goal progress tuned. Left: design of GLM to identify goal-progress tuned neurons in ABCDE tasks. Right pie chart showing the proportion of state-tuned neurons that are goal-progress tuned: ABCDE goal-progress/state GLM; Two proportions test: $N=189$ state neurons, proportion goal-progress-tuned=85%, $z=15.6$, $P=0.0$
- Polar plots of task-space tuning for 3 example neurons recorded across 2 ABCD tasks and then two ABCDE tasks - neurons 1 is purely goal-progress tuned while neurons 2 and 3 are conjunctively goal-progress and state tuned.
- Goal progress tuning is maintained across abstract tasks ABCDE vs ABCD: The average firing rate vector of all neurons relative to an individual goal (from goal “n” to goal “n+1”; averaged across all states). Animals experienced 2 ABCD tasks followed by 2 ABCDE tasks on these days. Each row represents a single neuron and the neurons are arranged on the y axis by their peak firing goal-progress in task 1 in the ABCD condition. This alignment is largely maintained in tasks across both ABCD and ABCDE structures. White dashes indicate early intermediate and late goal-progress-cutoffs.
- A histogram showing the mean goal-progress-vector correlation across tasks for each neuron. One-sample T test against 0: $N=111$ neurons; statistic=23.8; $P=3.76 \times 10^{-45}$, $df=110$. Note that the neurons used in this panel are those on days where animals experienced both ABCDE and ABCD tasks.

- f) Left: design of GLM to identify whether neurons maintain their goal-progress tuning across ABCDE and ABCD tasks. Right: A histogram showing the mean regression coefficient values for goal-progress as a regressor across ABCD and ABCDE tasks for each neuron. One-sample T test against 0: $N=111$ neurons; statistic=7.43; $P=2.45 \times 10^{-11}$, $df=110$.
- g) A plot of the mean task manifold derived from a Uniform Manifold Approximation and Projection (UMAP)-embedding along three dimensions for mFC activity in the ABCDE. The same manifold is shown twice: Left, goal-progress tuning along the manifold; right, state tuning along the same manifold. The entire task manifold is composed of goal-progress subloops.
- h) Quantifications of distances along the 3-dimensional UMAP-derived manifold - across different states and opposite goal-progress bin (left), across different states but for the same goal-progress bin (middle) or the distances across different states and same goal-progress bin for a permuted control. $N=4$ double-days - T-tests (with bonferroni correction): Across-goal progress vs within goal-progress: statistic =6.64, $P=0.021$, $df=3$; Across-goal progress vs permuted control: statistic =21.1, $P=7.02 \times 10^{-4}$, $df=3$; Within goal-progress vs permuted control: statistic =10.7, $P=0.005$, $df=3$

2-Modularity: mFC neurons are organised into modules that act as structured memory buffers (SMBs)

We find that mFC neurons form SMBs in the ABCDE task. Neurons are anchored with a fixed lag from given location/goal-progress conjunctions that is fixed across tasks (**Extended data figure 7i**). This is true even when considering lags that are unique to ABCDE tasks and not found in ABCD tasks (i.e. neurons with a mnemonic field more than 4 rewards from the anchor). Thus SMBs can be built to map different task periodicities.

Extended data figure 7

- i) Histograms showing the right shifted distribution of mean cross-validated task map correlations between neurons aligned to their preferred goal-progress/place anchor (from training tasks) and the task map aligned to this goal-progress/place from a left out test task in ABCDE tasks. This correlation is shown for all state-tuned neurons (left), non-zero-lag state neurons (middle) and neurons with a lag of more than 4-states from the anchor (right). T test against 0: All state neurons: $N=188$ neurons, statistic=7.21, $P=1.38 \times 10^{-11}$, $df=187$; Non-zero-lag state neurons: $N=153$ neurons, statistic=6.32, $P=2.47 \times 10^{-9}$, $df=152$; >4-state lag from anchor neurons: $N=31$ neurons, statistic=2.59, $P=0.015$, $df=30$

3-Predictive coding: The SMB model precisely predicts how the firing of mFC neurons relates to the animal's future choices in new tasks.

We extend the analyses of future choice prediction in **figure 6** to data from mice performing the ABCDE abstract task structure. These data show that, just as in the ABCD task, activity of mFC neurons predict the animal's future sequential behavioural choices in the ABCDE tasks. Neurons that fire at a lag X (degrees in task space) from an anchor point in ABCDE tasks predict behavioural choices at lag $360-X$ in the future even when accounting for autocorrelation in behaviour (**Extended data figure 8g**). This is significant as now the task has an extra reward, meaning that animals must use a new set of SMBs to predict where they will be in an extra state in the future (360 degrees now represents 5 rewarded states, i.e. 5 goal-progress cycles). Thus, SMBs can represent behavioural sequences for different task periodicities.

We note that this analysis, just as for the ABCD task (**Figure 6**), uses only neurons at non-zero lag from their anchor and hence by design only deals with non-spatial cells. Moreover, this prediction is done across tasks with distinct spatial trajectories (see more on this point under comment 3 below). This makes this type of predictive coding fundamentally different from other predictive codes that are based on the current location as well as its future trajectory such as splitter cells (**Wood et al 2000; Frank et al 2000**) and theta sequences/awake replay (**Johnson and Redish 2007, Pfeiffer and Foster 2013**).

Extended data figure 8

g

- g) Prediction of behaviour in the ABCDE tasks. Normalised firing rates of neurons during their “bump time”: i.e. the lag at which they are active relative to the anchor. Bump time activity is higher before visits to the neuron’s anchor in trial N+1 when the anchor was not visited in trial N (left) and also higher before visits to anchor in trial N+1 when the anchor was visited in trial N (right). Wilcoxon tests: Anchor not visited in trial N: n=24 tasks, statistic=73, P=0.027. Anchor visited in trial N: n=24 tasks, statistic=48, P=0.003. In addition, an ANOVA on all data (N=24 tasks) showed a main effect of Past: F=18.57, P=2.61x10⁻⁴, df1=1, df2=23, a main effect of Future: F=19.9, P=1.78x10⁻⁴, df1=1, df2=23, a Past x Future interaction: F=5.84, P=0.024, df1=1, df2=23. Right: Regression coefficients were positive for the bump time but not all other control times. T tests against 0: “bump time”: N=24 tasks, statistic=2.91, P=0.008, df=23; “decision time”: N=24 tasks, statistic=-1.17, P=0.252, df=23; “random time”: N=24 tasks, statistic=-1.33, P=0.197, df=23; “72 degree shifted time”: N=24 tasks, statistic=-1.97, P=0.061, df=23; “144 degree shifted time”: N=24 tasks, statistic=-0.8, P=0.43, df=23; “216 degree shifted time”: N=24 tasks, statistic=-0.83, P=0.417, df=23; “288 degree shifted time”: N=24 tasks, statistic=-0.88, P=0.390, df=23.

Thus, our existing and new results show a representation for multi-goal schema. The hierarchical and modular nature of these representations is new and motivates an algorithm for reading out complex behavioural sequences from the activity of mFC neurons that generalises across tasks.

3-One critical issue with the current study is the task design. By only visiting each goal once in a sequence, this design does not allow to disambiguate state from space and context. Previously published studies have used sequences with repeated locations to address this specific point (e.g. Euston and McNaughton, J Neurosci., 2006). An important observation from this study was that the apparent neuronal correlates of behavioral states were primarily explained by the animal's trajectories. A more detailed analysis of animal trajectories would be necessary to regress what variables are encoded by each neuron (Fig. 2f). In addition, the manuscript states that neurons were not modulated by speed or acceleration, but the associated statistics are not reported in Fig. 2f.

Thanks - this is a misunderstanding that we should clarify. We are sorry we were not clear in the analyses. In fact, in **figure 5**, we carried out this control in multiple different ways with detailed trajectory information in the analysis already. Perhaps most strikingly, we were able to predict the firing of cells in locations where they had never previously fired in any of the training data, which certainly cannot be trajectory coding because the trajectories are different from the training data.

To back up a bit, the reviewer is right that each individual task sequence does not dissociate trajectory from behavioural state, but the critical point is that we recorded cells across **many task sequences**. Each task sequence had a different trajectory, but we were able to show **generalisation across sequences**. The factors that were invariant **across** sequences were the **anchor and the lag**. Knowing just these two parameters, we could predict where a cell would fire in a new sequence with a new trajectory. In fact, the more trajectory specific a cell is the more likely it will fail this analysis, as it will not generalise across tasks.

To be completely sure that these were not trajectory-coding cells, we excluded all cells that had lags near zero (and therefore could potentially be coding something spatial). We did this in all three analyses in **figure 5**. Furthermore in **figure 5a,b** we put in the animal's current spatial position as a regressor, along with all previous positions as well as their conjunction with goal-progress (making 9x3x12 possible regressors corresponding to location x goal-progress x lag in task). Because these regressors include the whole history of choices, any variance that is explained by a consistent anchor and lag is by definition orthogonal to variance explained by a particular trajectory in a particular task.

We agree that trajectory-history encoding can predict the future in tasks with a single repeated sequence. There are many examples like this in the literature, including famous ones such as splitter cells (Wood et al. 2000; Frank et al 2000). These are often explained by latent behavioural states. The key difference in our data (**Figure 6**) is that cells **do not predict the future simply because they correlate with the trajectory or the latent state**. Instead they predict future locations across any trajectory, even new ones. They are place cells for the past (**Figure 5**) and the future (**Figure 6**). We have now clarified this further in the Results section (under "2-Distal prediction of behavioural choices") and Discussion section (Paragraph 4).

As a side point, we think our results provide a potential reconciliation between findings like that from Euston and McNaughton 2006 and those showing mnemonic coding in mPFC that is robust to trajectory coding of which ours is an example. This is because, while even neurons with maximal mnemonic lags from the anchor maintain reliable mnemonic coding across tasks (**Figure 5, Extended data figures 6a,b,d; Extended data figure 7a,d**), we nevertheless observe a gradient where more cells fire at lags close to the anchor (**Extended data figure 7e**). Thus, we expect more cells to disambiguate the two Bs in ABCBA in the Euston and McNaughton task because disambiguating these two locations relies on cells with a mnemonic lag one step away from their anchor (either the first A or the C). In the ABCDBCE sequence, the disambiguation of the two Cs would rely on cells with a lag of two steps away from an anchor (with the anchor being either A or D), which are fewer in number. The different computational demands of our task and the Euston and McNaughton tasks (context discrimination for theirs versus cross-task generalisation for ours) preclude a direct, quantitative comparison on the precise proportions of cells at different lags. Nevertheless, the points made above suggest that the two findings are, in principle, reconcilable.

Additional support

To further address trajectory tuning confounds, we repeated the analysis in **figure 5** while removing neurons whose activity was significantly explained by any combination of the animal's current location and next choice. We note that in this analysis we are working against ourselves as this trajectory-tuning measure will also capture neurons that are mnemonically tuned to fire just before or just after the anchor, where the trajectory options are limited. Hence it would be difficult to disentangle mnemonic from trajectory tuning for these neurons. This throws away a lot of potentially mnemonic neurons. The analysis is in essence equivalent to removing all but the most distal neurons (as we do in **Extended data figures 6a and Extended data figure 7a,d**). We find that we can still robustly predict the activity of non-trajectory encoding neurons as firing at fixed mnemonic lags from a given anchor (**Extended data figure 6c**). Thus the mnemonic fields predicted by the SMB model are robust to tuning to the animal's current trajectory.

Extended data figure 6

- c) Histogram showing the right shifted distribution of mean cross-validated correlation values between model-predicted (from training tasks) and actual activity (from a left out test task) for state neurons that are not tuned to the animal's current trajectory. Note that we use a permissive threshold for trajectory tuning here to ensure we exclude any neurons with even weak/residual tuning for trajectory. Any neuron that had a trajectory regression coefficient above the 95th percentile of the null distribution was excluded from this analysis. T test against 0: N=112 neurons, statistic=4.27, $P=4.13 \times 10^{-5}$, $df=111$).

Extended data figure 6

- e) Distribution of task space lags from anchor for all consistently anchored state neurons (neurons with the same anchor in >50% of tasks). Left: Using the most common lag from the anchor across tasks; Right: using the (circular) mean lag from anchor across tasks. Both plots show that consistently anchored neurons have lags from their anchor that span the entire range of possible lags. Note that these plots span the entire (4-goal) task space.

We would also like to clarify our claims with respect to speed and acceleration. With the GLM in **figure 2g,h**, we are not claiming that neurons are not modulated by speed/acceleration - rather that they are tuned to goal progress above and beyond any tuning to speed/acceleration. When adding speed and acceleration into the regression, variance is still significantly explained by goal progress for most neurons and by location for some neurons. Nevertheless, we now report the regression coefficients for speed and acceleration in **Extended data figure 2d**. Neurons show a small but significant modulation by speed.

Extended data figure 2

- d) Regression coefficients for animal kinematics (from GLM in figure 2g). Two histograms showing the mean regression coefficient values for Speed (Top) and Acceleration (Bottom) as a regressor across task/state combinations for each neuron. One-sample T test against 0: Speed: N=1252 neurons, statistic=3.36, $P=8.01 \times 10^{-4}$, $df=1251$; Acceleration: N=1252 neurons, statistic=-0.78, $P=0.438$, $df=1251$.

4-The core tenet of the paper is that the mPFC learns a structure of the task that can be flexibly transferred to a new task, explaining the faster learning rate in early trials after extensive training on this task. However, the manuscript provides limited evidence to establish that the mPFC is essential for this behavior. Indeed, these tasks engage the same sensory modalities while the mPFC is believed to be necessary for switching between tasks entailing extradimensional cues (Birrel and Brown, 2001; see Peyrache et al., Nat Neuro, 2009, Benchenane et al., Neuron, 2010 and Kaefer, Neuron, 2020 for

neuronal correlates of rule learning switches). It is still possible that the mPFC plays a role in the transfer between these specific tasks, but a direct demonstration of this view (e.g. manipulation of neuronal activity during behavior) would be informative.

We agree with the reviewer that a central point of our manuscript is that mFC learns a representation of an abstract task schema that can be transferred to new tasks. However, we do not agree that the reviewer's suggested experiment would strengthen the claims in the manuscript. The primary aim of the manuscript is not to describe the functional specificity of mPFC. It is instead to describe the detailed form of the representation and how it supports a detailed algorithm for encoding task sequences. We show this by demonstrating how the SMB model explains both the activity of the neurons and the behavioural choices of an animal. While we agree that a causal manipulation of neuronal activity may be interesting, below we explain why we believe it would be neither necessary nor sufficient to support our claims.

Would a causal manipulation be sufficient to support the claims?

Manipulations to the mFC have already been shown to induce deficits in schema tasks (e.g. Tse et al 2007). However, we agree with the reviewer that there are potentially many reasons for this: such mFC manipulations can cause effects on attention, sensorimotor transformations, switches in strategy, action sequencing and so on. Simply manipulating mFC and showing task errors will therefore not test the claims in the manuscript, which are about the nature of schema/goal representations in mFC.

There are, in principle, manipulation studies that could test the causal nature of the representation we have uncovered, but these require precise stimulation of particular cells belonging to individual SMBs at specific times in the trial, to induce particular changes in behaviour. This could potentially be achieved using an all-optical approach, including a head-fixed version of the task and the ability to manipulate individual, functionally-defined cells, but this is a multi-year endeavour that involves forming new collaborations, and optimising new techniques. No existing study has achieved manipulations at this level of precision during cognitive behaviours.

Is a lesion experiment necessary to support our claims?

Most importantly, the manuscript is not making claims about functional specificity in mPFC. It makes claims about the nature of the representation. If we were to silence the prelimbic area of mFC and observe no effect, this would not negate the representational or algorithmic claims. There are many other frontal areas that receive similar inputs and are similarly organised to the prelimbic area. Indeed, in the revised manuscript we provide additional neuropixels data showing that regions like the infralimbic, anterior cingulate and M2 have similar representations to what we originally describe in the prelimbic cortex (**Extended data figure 2e; Extended data figure 4j,k; Extended data figure 7h**). Preliminary human fMRI data from our lab suggests we may also find a similar representation and algorithm in the orbitofrontal cortex. There are also theoretical reasons to believe alternative strategies could be used to solve this task by regions like the medial Entorhinal cortex (Whittington et al 2023). Teasing apart these compensatory and complementary effects will require multiple, careful representational/causal studies which we and others intend to conduct over the coming decade. We hope these studies will, in time, reveal the detailed functional anatomy of algorithms along the medial wall of the frontal cortex and beyond.

Overall, we hope that the reviewer agrees with us that a causal manipulation is not necessary for the current manuscript.

Extended data figure 2

- e) The subregional distribution of neuron type coefficients along the medial wall of the frontal cortex in neuropixels recordings. One-way ANOVA: Left: Proportion of Goal progress neurons: $F=2.40$, $P=0.143$, $df=3$; Middle: Proportion of state neurons $F=1.04$, $P=0.425$, $df=3$; Right: Proportion of place-tuned neurons $F=18.8$, $P=5.54 \times 10^{-4}$, $df=3$. Posthoc Tukey HSD tests: IrL vs PrL $P=0.049$; IrL vs ACC $P=0.000$; IrL vs M2 statistic=0.003, PrL vs ACC $P=0.021$.

Extended data figure 4

- j) The subregional distribution of single neuron generalisation (averaged across X vs Y and X vs Z comparisons) along the medial wall of frontal cortex in neuropixels recordings. One-way ANOVA: $F=1.59$, $P=0.323$, $df=3$.
- k) The subregional distribution of neuron pair coherence. Coherence is calculated across both X vs Y and X vs Z comparisons along the medial wall of frontal cortex in neuropixels recordings: One-way ANOVA: $F=4.76$, $P=0.083$, $df=3$.

Extended data figure 7

- h) The subregional distribution of cross-validated task map correlations between neurons aligned to their preferred goal-progress/place anchor (from training tasks) and the task map aligned to this anchor from a left out test task along the medial wall of frontal cortex in neuropixels recordings. One-way ANOVA: Left: All state neurons: $F=1.44$, $P=0.302$, $df=3$; Middle: Non-zero lag state neurons: $F=0.92$, $P=0.573$, $df=3$; Right: Distal (>90 degrees from anchor) non-zero lag state neurons $F=0.89$, $P=0.485$, $df=3$.

5-Goal-progress tuning was largely preserved across tasks while, in contrast, the tuning to states remapped for each task. Specifically, the study reports a histogram of single neuron phase difference across tasks (Fig. 3a) which shows that, remarkably, task phase differences are overwhelmingly represented for every 90-degree shifts. In fact, there seems to be virtually no neurons that show a 45-degree (mod. 90 degree) shift. Yet, it is unclear how the third neuron in Fig. 3a remaps. There seem to be a 20-degree shift between tasks Y and Z and a 135-deg. shift between Z and X. The phase difference between tasks X and Y for the 1st neuron and X and Z for the 5th neuron are also unclear. Were these neurons recorded simultaneously? More detailed examples and reporting across-task phase differences for each neuron in panel 3a would be informative. In addition, a potential confound of the single cell and pairwise phase difference analysis is that neurons overwhelmingly encode the same phase of the task (Fig. 2e). The manuscript does not clearly explain how this was accounted for.

We thank the reviewer for scrutinising this key figure. It is true that the strong goal-progress tuning means the remapping overwhelmingly happens in 90 degree intervals. However, there are exceptions to this, which are visible when zooming into the left circular histogram in **figure 3b**. The neurons displayed in figure 3a are all indeed simultaneously recorded from the same mouse and recording day. As requested, we now add more details to the example remapping plots showing the rotation angles relative to session X (**Figure 3a**).

We note that, as described in the original manuscript, the angles are calculated by finding the best rotation of the tuning curve of each neuron in one session relative to its tuning curve in session X. This aligns well with the angles seen by visually inspecting the changes in the firing peak, but in some cases (e.g. **Figure 3a** neuron 3, session X vs Z) there is a discrepancy between the “best rotation” angle and the “peak-to-peak” angle. This is because the best-rotation measure takes the entire shape of the tuning curve into account. It is therefore robust to small changes in the size of peaks when there is more than one similarly sized peak (e.g. neurons 2,4 and 6 in the new **Extended data figure 4I**), which would introduce major inaccuracies in calculating remapping angles when using the peak to measure cross-session changes. Hence our preference for the best-rotation-based approach. However, we note that any ambiguity in calculating angles will introduce unstructured noise that works against us rather than introducing any biases that would induce false coherence. Nevertheless, to make this point robustly, we now repeat the single cell generalisation and pair-wise coherence analyses while using only state-neurons with concordant remapping angles across both methods (i.e. using the best-rotation analysis method and peak-to-peak changes method) for **all cross-session comparisons (Extended data figure 4c,f)**. This would, for example, exclude neuron 3 in **figure 3a**, which on one cross-session comparison rotates differently when using the best rotation vs peak change methods in the X vs Z comparison. We show that the same results hold even under this condition: individual neurons do not generalise but pairs of neurons are partially coherent across tasks **Extended data figure 4c,f**. We now clarify these methodological considerations in the methods section under (“Neuronal generalisation”).

Importantly, the decisions made here, about which method to use to calculate remapping, have no bearing on the analyses in **figures 5a-d** where we use two different analysis methods to illustrate that subsets of neurons are anchored to different location/goal-progress conjunctions. This result explains the modularity observed in **figure 3** (via the SMB model) but is also a separate replication of such modularity. The data in **figure 5** shows that individual neurons maintain invariant lags to an anchor across tasks, meaning that groups of neurons that share the same anchor remap in a coherent manner across tasks, giving rise to modules. Further, the data in **figure 7** show that such anchor-defined modules maintain their internally organised structure in sleep.

Figure 3

- a) Polar plots showing state tuning of example neurons, simultaneously recorded from the same animal across multiple tasks on a given day. Each row is a neuron and each column is a session. Neurons readily remap their state tuning but maintain their goal-progress preference across tasks. State preference is maintained across different sessions of the same task (X vs X'). Angles (in degrees) of each cell's rotation relative to its tuning in session X are shown to the right of each session's polar plot.
- b) Top: A schematic showing how the difference in tuning angles for the same neuron across sessions is quantified. Bottom left: Polar histograms show that state-tuned neurons remap by angles close to multiples of 90 degrees, as a result of conserved goal-progress tuning and the 4 reward structure of the task. The orange shaded area represents neurons that maintain a consistent angle across sessions (with a tolerance of 45 degrees either side of 0 change). No clear peak at zero is seen relative to the other cardinal directions when comparing sessions spanning separate tasks (Two proportions test against a chance level of 25% N=1594 neurons; mean proportion of generalising neurons across one comparison (mean of X vs Y and X vs Z)=24%, $z=0.91$, $P=0.363$. Bottom right: Neurons maintain their state preference across different sessions of the same task (X vs X' Two proportions test against a chance level of 25% N=1160 neurons; proportion generalising=78%, $z=25.7$, $P=0.0$).
- c) Top: A schematic showing how the difference in relative angles between pairs of neurons across sessions is quantified. Bottom left: Polar histograms show that the proportion of coherent pairs of state-tuned neurons (comprising the peak at zero) is higher than chance but less than 100%, indicating that the whole population does not rotate coherently. The orange shaded area represents pairs of neurons that maintain a consistent angle across sessions (with a tolerance of 45 degrees either side of 0 change). Two proportions test against a chance level of 25% N=35164 pairs; mean proportion of coherent neurons across one comparison (mean of X vs Y and X vs Z)=29.3%, $z=13.0$, $P=0.0$. Bottom right: As expected from panel b, the large majority of state-tuned neurons keep their relative angles across sessions of the same task (X vs X'; Two proportions test against a chance level of 25% N=23674 pairs; proportion coherent=63%, $z=83.5$, $P=0.0$).

In regards to the potential confound introduced by neurons largely maintaining the “same phase”, we assume the reviewer means the neurons’ tuning to goal progress, which is indeed highly conserved across tasks and hence may introduce a confound to the coherence analyses. We account for this confound in **figures 3b** and **c** by quantifying the proportion of neurons (**Figure 3b**) or neuron pairs (**Figure 3c**) within a window 45 degrees either side of zero (i.e a total area spanning 90 degrees) compared to the rest of the population. This means that, despite the strong goal-progress tuning, the chance level for this “close-to-zero” proportion is still 25%. We now show a shaded region to illustrate which parts of the histogram are used to signify generalisation (**Figure 3b**) or coherence (**Figure 3c**). We use a similar approach when we assess combined remapping across **both** cross-task comparisons (X vs Y and X vs Z), quantifying the proportion of pairs of neurons that remap by a maximum of less than 45 degrees either side of zero across both comparisons (**Extended data figure 4h**). Here, chance level is now $1/16$ ($1/4^2$). In the clustering analysis in **figure 3d** we account for goal progress tuning by creating a null distribution that preserves not only the baseline goal-progress tuning but also the stability of goal-progress tuning across tasks, effectively remapping in a way that preserves goal progress tuning but with otherwise random pairwise relationships across tasks. In **figure 5** we account for the potential goal progress confound by only predicting the neuron’s activity in its preferred goal-progress bins across all three methods. This removes any bias introduced by the fact that goal-progress tuning is conserved across tasks. It makes the task of the prediction harder, where we are now purely predicting the neurons’ state tuning (or lagged spatial map in **figure 5c,d**) within its preferred goal-progress bin in a new task that was left out of the training set. In

figure 6 goal-progress tuning has no direct effect on our analyses, as these are about predicting the animals' future choices. Nevertheless, we only quantify the effect of a neuron within its preferred goal-progress bin and show that such future prediction is lost in controls where the activity is shifted by 90 degree intervals to preserve goal-progress tuning but destroy the precise lag to the anchor (e.g. **figure 6d**). In **figure 7** we add the pairwise goal-progress tuning distance as a co-regressor in all of the regression analyses in order to separate the effect of goal-progress distance from distance to anchors. We now clarify these controls throughout the manuscript Methods section.

6-The idea of separate modules, each independently coding for task phase is appealing. However, Fig. 3e is challenging to interpret. In line with the previous comment, more detailed examples should be provided, showing the tuning of each neuron in a given module and their remapping across tasks. More importantly, the existence of ring-like structure in activity should be directly demonstrated by investigating the topology of population activity (see e.g. Chaudhuri et al., Nat Neuro, 2019 for head-direction neurons and Tang et al., Cell Reports, 2023 for hippocampal and prefrontal neurons during task execution).

We thank the reviewer for recognising that "*The idea of separate modules, each independently coding for task phase is appealing.*". Indeed this was the initial inspiration for formulating the SMB model.

We apologise for the lack of clarity in **figure 3e**. It is indeed dense as we have chosen to show all neurons recorded on a given day on this plot (101 on that particular day) to illustrate the broad picture. To improve clarity, we have made an additional figure from a different animal and recording day where we show two simultaneously recorded clusters (modules) of neurons that have been detected using our clustering analysis (in **figure 3d**). We show more detail in this plot as requested, displaying polar plots showing the tuning of **all cells** in the two modules across all sessions as well as the angles with which they rotate across sessions (**Extended data figure 4!**).

Extended data figure 4I

- 1) Top: Visualisation of tuning relationships between two clusters computed in a single recording day. Each dot is a neuron (numbered in correspondence to the polar plots below) and each ring is a cluster derived from the analysis in panel d. The colour code represents the tuning of the neurons in each task. The x,y position defines the tuning in each task. The z position corresponds to cluster ID. Note that the ordering along the z axis is arbitrary. Neurons rotate (remap) in task space while maintaining their within-cluster tuning relationships but not cross-cluster relationships across tasks. Bottom: polar plots for all of the (seven) neurons assigned to each of the two clusters in the above plot. Angles (in degrees) of each cell's rotation relative to its tuning in session X are shown to the right of each session's polar plot.

In regards to investigating the topology of population activity. This is an important point that highlights the relationship between our analyses and findings with those of previous studies. We show below that such manifold analysis is informative for showing, at the population level, the hierarchical structure of mFC state neurons in **a single task**. This **within-task** manifold analysis provides a clearer visualisation that such state tuning is built on top of a more primitive goal progress sequence (Main finding 1: **Figure 2**). However, given the high dimensionality of the SMB-based representation, there is no single **cross-task manifold** that can be visualised using standard dimensionality reduction methods. We unpack this dimensionality issue more below but first we discuss where a dimensionality reduction approach has helped bolster another aspect of our study.

We now show that applying a non-linear dimensionality reduction (UMAP) to our data reveals a hierarchical **within-task** manifold representing task structure. This manifold is composed of goal progress sequences

that are concatenated into a floral structure representing the number of rewarded states in a given task (**Figure 2j,k; Extended data figure 2f,g and Extended data figure 3g,h**). When this is done for data from ABCD tasks, 4 petals of goal progress sequences are apparent and differentiate the 4 states in an individual ABCD task (**Figure 2j,k**). This is seen even when excluding cells with even residual spatial tuning (**Extended data figure 2g,h**). The same principle applies to the newly added ABCDE tasks where the manifold is now composed of 5 goal-progress petals instead of 4 (**Extended data figure 3g,h**). This visualisation and associated analyses across two abstract task structures with distinct periodicity (ABCD and ABCDE) provides direct, population-level support for a key finding in our study: that task-structure manifolds in the mFC are composed of goal progress sequences. We thank the reviewer for this suggestion as we have previously relied on single cell analysis to make this claim (**Figure 2 c-i**) whereas the UMAP-derived manifolds we have uncovered allow us to more clearly visualise this result.

Figure 2

- j) A plot of the mean task manifold derived from a Uniform Manifold Approximation and Projection (UMAP)-embedding along three dimensions. The same manifold is shown twice: Left, goal-progress tuning along the manifold; right, state tuning along the same manifold. The entire task manifold is composed of goal-progress subloops.
- k) Quantifications of distances along the 3-dimensional UMAP-derived manifold - across different states and opposite goal-progress bin (left), across different states but for the same goal-progress bin (middle) or the distances across different states and same goal-progress bin for a shuffled control. $N=20$ double-days, T-tests (with bonferroni correction): Across-goal progress vs within goal-progress: statistic =6.09, $P=2.25 \times 10^{-5}$, $df=19$; Across-goal progress vs permuted control: statistic =26.0, $P=7.85 \times 10^{-16}$, $df=19$; Within goal-progress vs permuted control: statistic =8.63, $P=1.60 \times 10^{-7}$, $df=19$.

Extended data figure 2

- f) A plot of the mean task manifold derived from a Uniform Manifold Approximation and Projection (UMAP)-embedding along three dimensions restricted to only non-spatial neurons. Note that we use the most permissive threshold for spatial tuning here to ensure that we exclude even neurons with weak/residual spatial tuning. Any neuron that had a spatial regression coefficient above the 95th percentile of the null distribution was excluded from this analysis. The same manifold is shown twice: Left, goal-progress tuning along the manifold; right, state tuning along the same manifold. The entire task manifold is composed of goal-progress subloops.
- g) Quantifications of distances along the 3-dimensional UMAP-derived manifold - across different states and opposite goal-progress bin (left), across different states but for the same goal-progress bin (middle) or the distances across different states and same goal-progress bin for a shuffled control. $N=8$ double-days - T-tests (with bonferroni correction): Across-

goal progress vs within goal-progress: statistic =10.3, $P=5.45 \times 10^{-5}$, $df=7$; Across-goal progress vs permuted control: statistic =17.5, $P=1.47 \times 10^{-6}$, $df=7$; Within goal-progress vs permuted control: statistic =5.2, $P=0.004$, $df=7$

Extended data figure 3

g

h

- g) A plot of the mean task manifold derived from a Uniform Manifold Approximation and Projection (UMAP)-embedding along three dimensions for mFC activity in the ABCDE. The same manifold is shown twice: Left, goal-progress tuning along the manifold; right, state tuning along the same manifold. The entire task manifold is composed of goal-progress subloops.
- h) Quantifications of distances along the 3-dimensional UMAP-derived manifold - across different states and opposite goal-progress bin (left), across different states but for the same goal-progress bin (middle) or the distances across different states and same goal-progress bin for a permuted control. $N=4$ double-days - T-tests (with bonferroni correction): Across-goal progress vs within goal-progress: statistic =6.64, $P=0.021$, $df=3$; Across-goal progress vs permuted control: statistic =21.1, $P=7.02 \times 10^{-4}$, $df=3$; Within goal-progress vs permuted control: statistic =10.7, $P=0.005$, $df=3$

Importantly, these manifolds in **figure 2** and associated extended data figures pertain to **individual tasks**. They do not show a manifold that generalises **across tasks**. Our findings in **figures 3** and **5** show that the abstract task structures are not encoded by a single manifold, but rather a number of separate SMBs, each anchored to a different location/goal-progress combination. This contrasts with previous single manifold studies (e.g. Pratt and Mizumori 2001, Peyrache et al 2009, Tang et al 2023, Kaefer et al 2020, Basu et al 2021; Samborska et al 2021). It is not possible to visualise the entire multi-SMB manifold using UMAP or any dimensionality reduction method given that the minimum number of possible anchors (9 locations x at least 3 goal-progress bins = a minimum of a 27-dimensional space) makes the full manifold high-dimensional and hence not amenable to being compressed into a lower number of dimensions. In reality we predict many more modules given the high resolution of goal-progress tuning (**Figures 2c-i**) and the fact that spatial anchors are typically multi-peaked, giving a large number of possible spatial pattern combinations and hence many more than 9 possible anchors (**Figure 5a-d**). This forces us to use high dimensional methods like the coherence and clustering analysis in **figure 3** and the anchoring analysis in **figure 5** to analyse the SMBs.

Another way of stating this is to note that dimensionality reduction techniques are useful when visualising each time point in the task as a single point in a low dimensional space. However, we find that each time point in the task is actually represented by multiple points on multiple SMBs, each encoding a lag from a different anchor (**Figure 5**). This means we cannot meaningfully plot the full SMB structure using the dimensionality reduction techniques used in the field.

A useful comparison point are toroidal manifolds of grid cells in the mEC. Here the torus is only visible when a large number of neurons (>100) are isolated from a **single** module (Gardner et al 2022). We can in principle show a manifold for a **single** SMB across tasks. However, given that we are dealing with orders of magnitude more mFC task modules than mEC grid modules (of which there are typically 6) we need orders of magnitude higher neuronal yields than the best cortical yields currently achievable to obtain 100+ neurons in a single mFC module.

These points are now clarified in the Results section (under “Progress to goal is a primary feature of Frontal task structure representations”) and Methods section (under “Manifold analysis”) sections of the manuscript.

7-The existence of internally-generated activity patterns is well established, and not only in the medial entorhinal cortex as the citations suggest. Furthermore, it has already been established that neuronal patterns in the mPFC which form during learning are replayed during subsequent sleep (Euston et al., Science, 2007; Peyrache et al., Nat Neuro, 2009). In Euston et al. (2007), it was even shown that the specific relative timing of spikes was preserved after learning a sequential task similar to what the task design of the present study. In Peyrache et al., (2009), it was shown that these replay events were specifically associated with learning and were likely triggered by hippocampal sharp wave-ripples (also see Kaefer, Neuron, 2020 regarding the differences in replay content between the hippocampus and the mPFC). Here, a similar phenomenon was observed (Fig. 7) but a more direct demonstration of the phenomenon is necessary, for example by showing spike train cross-correlograms. Also, the dynamics of these phenomena likely depend on the exact sleep stages, so a more detailed analysis of sleep scoring and time since wake epoch is necessary. Since a difference between pre- and post-sleep was observed, it would be interesting to investigate whether some task-specific information was more likely to be replayed.

We agree with the reviewer that internally-generated activity patterns are well-established and ubiquitous in the brain. We focused on the mEC in the submitted manuscript as internal organisation here has been implicated in similar types of generalisation, but we appreciate that the ubiquity of internal organisation needs to be emphasised and we have now done so in the Discussion section of the manuscript. We are also not suggesting that we are the first to demonstrate this in the mFC or elsewhere. Rather the purpose of **figure 7** is to demonstrate that the structured memory buffers (SMBs) we observe in the previous figures are themselves internally organised. We have now cited the suggested studies (and others) to ensure this point is made clear to our readers (Euston et al 2007; Peyrache et al 2009; Dragoi and Tonegawa 2011; Peyrache et al 2015; Grosmark and Buzsaki 2016; Trettel et al 2019; Kaefer et al 2020; Gardner et al 2022). Nevertheless, while the cited studies are relevant and important, none of them make any of the 3 main neuronal findings we mention above (under the reviewer’s second comment).

The main similarity between us and the mFC replay reported in other studies (e.g. Euston et al 2007; Peyrache et al 2009; Kaefer et al 2020) is that they reveal internally-organised structure which we agree is a common feature of representations throughout the brain and across many learned tasks. The internal organisation we observe in **figure 7** supports the main claim of the paper by showing that memory buffers maintain their internal structure in the absence of task-structured sensory/behavioural input. Crucially we are not merely demonstrating internal organisation in a broad sense. Instead we are showing internal organisation *within* individual memory buffers - the sleep coactivity of neurons anchored to the same anchor is highest when the circular distance (but not necessarily forward distance) is low (**Figure 7a,b**). This implies that co-anchored neurons are organised into a task-shaped ring. Moreover, this analysis allows us to assess the nature of internal organisation within **individual** memory buffers - we find that neurons that share the same anchor are more co-organised than those with different anchors (**Figure 7d**). This supports the specific predictions of the SMB model, which relies on i) each memory buffer being structured by the task and ii) memory buffers being independent, in order to flexibly encode new tasks.

Further to this point, we have now provided a more direct demonstration of the sleep effect as requested by the reviewer. Namely, we show that cross-correlations between neurons within the same SMB are highest at low circular distances, even when those correspond to high forward distances (**Figure 7c**). This supports the ring-like organisation of neurons with an SMB, in line with the results from the GLM in **figure 7c**.

We also now assess differences between pre- and post-task sleep across measures. We find that, when taking the entire sleep period, there were no significant differences in the regression coefficients between

pre- and post-task sleep for circular distance or forward distance versus sleep cross correlation (**Extended data figure 9a**). Moreover, no significant difference was found between pre- and post-task sleep when assessing the regression coefficients for spatial correlation between pairs of neurons versus their cross correlation during sleep (**Extended data figure 9a**). Furthermore, when we broke down sleep into epochs of time since wake period, as requested by the reviewer, we still did not observe any differences between pre-task and post-task sleep (**Extended data figure 9b**).

The lack of change in beta coefficients for circular and forward angles against sleep coactivity between pre-task and post-task sleep, even when controlling for time since sleep, suggests that the coactivity structure of the SMBs in relation to the abstract task structure is largely conserved before and after a task on a given day. We note that all of this data is from well-trained animals that have had a chance (in tasks 1-20) to form such internally organised SMBs before we record their activity (in tasks 21-40). Our results are consistent with the SMBs being already learned at this stage and not altered by subsequent learning of new tasks.

Figure 7

- a) Left: Schematic showing potential neuronal state spaces - if neurons are arranged on a ring, then circular distance is a better description of how close two neurons are in state space than forward distance relative to the anchor. Conversely, if neurons lie on a delay line, forward distance is a better description of neuron-neuron co-firing relationships. Right: Schematic showing the inputs and outputs of linear regression model relating pairwise circular distance and forward distance with coactivity during sleep while regressing out each other as well as pairwise goal-progress tuning distance and spatial map similarity.
- b) Regression coefficient values for circular (left) or forward (right) distance regressed against sleep cross-correlation for co-anchored neurons - T test relative to 0: circular distance: N=430 pairs, $t=-2.66$ $P=0.008$, $df=429$; forward distance: N=430 pairs, $t=1.61$ $P=0.108$, $df=429$.

- c) Left: A plot of cross-correlations during sleep between pairs of neurons sharing the same anchor against forward distance (bottom x axis) or circular distance (top x axis). Schematics at the top and bottom show example pairs that would fall into each category and their circular and forward distances. A delay-line state space would result in a uniformly negative relationship between forward distance and cross-correlation. A circular state space would result in a v-shaped relationship between forward distance and cross-correlation, with a negative slope when forward distance is between 0-180 degrees (where forward and circular distances are identical) and a positive slope when forward distance is between 180-360 degrees (where circular distance is negatively correlated with forward distance). Right: Correlation coefficients between pairwise forward distance and pairwise sleep cross-correlations are positive for pairs of neurons with 180-360 degrees forward distance and negative for pairs with a forward distance of 0-180 degrees. T tests relative to 0: 0-180 degrees: N=59 sleep sessions, statistic=-5.16, $P=1.07 \times 10^{-4}$, $df=58$; 180-360 degrees: N=59 sleep sessions, statistic=4.16, $P=1.08 \times 10^{-4}$, $df=58$, Paired T test: N=59 sleep sessions, statistic=-5.55, $P=7.35 \times 10^{-7}$, $df=58$.
- d) Regression coefficient values for circular distance against sleep cross-correlation using pairs of neurons consistently anchored to the same anchor (within) vs pairs of consistently anchored neurons that have different anchors (between). Regression coefficient values were more negative for pairs sharing the same anchor (within) compared to pairs across anchors (between) across all sleep. One-tailed unpaired T test (Welch's T-test): All sleep: N=430 pairs (within), 13932 pairs (between), $t = -1.80$, $P=0.036$, $df=14360$. All error bars represent the standard error of the mean

Extended data figure 9

- a) No significant differences are seen when comparing regression coefficients for circular distance, forward distance and spatial similarity between co-anchored pairs of neurons between pre- and post-task sleep across the whole-session. Comparison of regression coefficients (from GLM in figure 7a) for left: circular distance: Welch's T-test: $t = 0.14$, $P=0.892$, $df=919$, middle: forward distance: Welch's T-test: $t=0.21$, $P=0.834$, $df = 919$ and right: spatial similarity: Welch's T-test: $t = 0.02$, $P=0.983$, $df = 919$. N=521 pairs (pre-task sleep) 399 pairs (post-task sleep).
- b) Comparison between pre- and post-task sleep across different time epochs since the beginning of sleep sessions for regression coefficients of circular distance (left), forward distance (middle) and spatial similarity (right). Unpaired T test (Welch's T-test) results (with Bonferroni correction of p values): Circular distance: 0-10 minutes post-sleep: N=512 (pre-task) N=429 (post-task): $t=-0.69$, $P=0.870$, $df=939$. 10-20 minutes post-sleep: N=512 (pre-task) N=429 (post-task): $t=-0.05$, $P=0.997$, $df=939$. 20-30 minutes post-sleep: N=512 (pre-task) N=429 (post-task): $t=-0.07$, $P=0.997$, $df=939$. Forward distance: 0-10 minutes post-sleep: N=512 (pre-task) N=429 (post-task): $t=-1.33$, $P=0.459$, $df=939$. 10-20 minutes post-sleep: N=512 (pre-task) N=429 (post-task): $t=0.19$, $P=0.91$, $df=939$. 20-30 minutes post-sleep: N=512 (pre-task) N=429 (post-task): $t=0.38$, $P=0.91$, $df=939$. Spatial similarity: 0-10 minutes post-sleep: N=512 (pre-task) N=429 (post-task): $t=-1.17$, $P=0.567$, $df=939$. 10-20 minutes post-sleep: N=512 (pre-task) N=429 (post-task): $t=-1.11$, $P=0.567$, $df=939$. 20-30 minutes post-sleep: N=512 (pre-task) N=429 (post-task): $t=0.11$, $P=0.915$, $df=939$.

Reviewer Reports on the First Revision:

Referees' comments:

Referee #1 (Remarks to the Author):

The authors have been very responsive to my prior comments and the manuscript is much improved. I'm happy to recommend publication. I do have a couple of remaining minor points.

1. The figure legend in Fig. 5c and EDFig. 5B states that the max firing rates are in the top right of the figure, but that doesn't seem to be the case.

2. The authors kilosort exclusion parameters don't make sense to me. I'm guessing their firing rates are usually <10 Hz (that's where knowing those max firing rates would come in handy!) and so if you had two neurons mistakenly grouped together the expected probability of spikes within 2 ms of one another is only 0.02. So a 10% threshold makes no sense at all.

Also, 2 ms is very long for a refractory period cutoff. They can figure out what a sensible cutoff in their data is by examining spikes from well-isolated neurons and gradually adjusting the cutoff. They will notice a step function where the true refractory period is. I'd be very surprised if it is 2 ms.

For reference, the ISI violation threshold we use is 0.1% of spikes < 1.2 ms.

Referee #2 (Remarks to the Author):

The authors have made an impressive effort to address my concerns. New experiments and extensive, additional analyses were key to addressing the generality and robustness of the findings. The results of the ABCDE task—a new abstract task structure with a different periodicity as the original ABCD task—provide strong evidence that goal-progress cells in mFC can flexibly be sculpted by task structure, thereby positioning them as compelling candidates for 'elemental building blocks of schemata'. Furthermore, these findings highlight the potential of this study's findings and framework to generalize to many different forms of schemas. Additional analyses of task performance, as well as new experiments omitting the tone associated with reward A, have helped to rigorously demonstrate that it is indeed the abstract structure of the task (the schema) which is being learned and used by subjects to achieve optimal performance and zero-shot inference. The authors have also extended the rigor of the study in several ways. They have provided key methodological details as requested (e.g. Table 1 specifies the inclusion criteria and sample size for each figure panel). New analyses have further validated their methodology used to test SMB model predictions (e.g. ruling out the potential effect of nonlinearities in their regression model by using a linear-nonlinear Poisson model; including a more granular presentation of metrics used to characterize co-anchored neurons during sleep in Figure 7).

One concern, which I found not fully addressed in the revision, is regarding the authors' approach to spike sorting data concatenated across consecutive days. I expand on this point below and offer

several suggestions for the authors to address this.

Another concern, which should be easily addressed, is the overall clarity of the manuscript. I have made several suggestions below regarding aspects of the terminology, data presentation, and the title, which I found to be confusing.

1. Spike sorting data concatenated across consecutive days (“double days”)

The authors concatenate pairs of consecutive recording days to increase the number of tasks assessed for each neuron. The degree to which concatenated data had been used was not very obvious in the first version of the manuscript, which is why I requested additional information in the revision. The extensive documentation of which analyses were performed using concatenated data (Table 1) is much appreciated. The authors have also made an effort to annotate the manuscript/legends to indicate that concatenated data had been used. That being said, the extensive use of the concatenated data stands in contrast with the limited extent of quality metrics and validation.

Although the authors now include their criteria used to curate single units (based on refractory period contamination and firing rate fluctuation over sessions) and report the drop-out rate, I remain concerned that they have not adequately addressed the potential impact of probe drift across 2 consecutive days. I understand that the Kilosort versions used in the study (2.5 and 3.0) have built-in drift correction algorithms well accepted for standard recordings; however, given that (1) pooling across consecutive days is not standard practice in the field, and (2) the implications for how the data are interpreted (explained below), I think it’s important for the authors to more rigorously assess how units have been matched across consecutive days.

One key feature of the mFC single unit tuning reported in the study is that they can generalize or remap across distinct task sequences. Given that the task sequences were pooled across two days, it is critical to ensure that the activity of distinct units across days are not inadvertently collapsed into one. We can consider a simple example where the case of goal-progress tuned unit #1 firing selectively for state A on task X (day 1), and another goal-progress tuned unit #2 firing selectively for state B on task Z (day 2). If these units were indeed identical, the activity would be interpreted as remapping; if the units were in fact distinct and therefore erroneously matched, the activity would be interpreted as the unit being modulated by 2 states (or perhaps lacking state tuning).

I would suggest the authors show they have indeed rigorously matched units. For example, they could show actual waveforms and quality metrics of unit stability. They could do this for several units, including those whose tuning properties are showcased in the figures. The authors should also examine the extent of drift detected and corrected by Kilosort, and consider showing a drift plot (e.g. one of the KS output metrics/figures) for concatenated sessions. For example, the authors could use approaches in a one recent work which provided a characterization of unit stability over sleep periods recorded over >12 hour long recordings (see Methods and Extended Data Fig. 8 from Maboudi...Diba, Nature, 2024 (<https://www.nature.com/articles/s41586-024-07397-x>)).

In addition, I think it would help if the authors better communicated the necessity of concatenating data. Although I understand their goal to increase the number of tasks assessed for each neuron, the

authors have stated that using a single day (3 tasks) is “insufficiently powered”. It is unclear if and how they have quantified this. At the very least, authors should consider adding an explicit discussion of potential caveats for interpretation (or lack thereof).

Clarity of the manuscript

1. Data presentation: on more than several occasions, the style of data presentation appears to preclude important details or makes it very difficult for the reader to extract the information. For example:

- Fig. 2d: it is not obvious that the polar plots on the top row are organized into sets of polar plots from the same cell, while the polar plots on the bottom are from 6 different cells. Please consider using more informative labels in figures (e.g. “spatially tuned cell” rather than “spatially tuned”).

- There are many details which are impossible to see due to their small font size and are often not explained in the legend. e.g. Fig. 2d, numbers showing firing rate in polar plots and the max firing rates in Fig. 5c heatmaps.

2. Terminology: The manuscript uses a variety of terminology (e.g. ‘behavioral step’, ‘anchor’, ‘goal-progress/place conjunction’) interchangeably and often in a confusing manner (e.g. Figure 5 includes the notion of ‘behavioral steps’ in the title but then doesn’t refer to this again in the legend and instead use ‘anchor’ and ‘goal-progress/place’. I’d recommend clarifying the manuscript by trying to use consistent terms and less jargon throughout.

Minor Comments

- I am still a bit confused about how animals achieve ‘near perfect’ performance. On one hand, authors show that more than half of the subjects exhibit suboptimal performance (i.e. taking longer routes) associated with persisting behavioral biases from before task exposure (Extended Data Fig. 1f). It is difficult to understand how this is compatible with the striking proportion of ‘perfect trials’ which require all transitions in a trial to take the shortest route (and presumably pooled from the same 13 subjects) (Extended Data Fig. 1i). Is this simply because the ‘perfect trials’ considers only the first trial’s $d \rightarrow a$ transition?

- Authors should provide more details on the partially connected maze (6 out of 9 ports available) used during pre-training. It appears that subjects were introduced to this partially configured maze twice before the full ABCD paradigm, so it’s important to understand which parts of the ABCD task structure (if any) were present in this pre-training stage. The authors note that there was ‘no explicit task structure’ (pg. 44). How were the 6 out of 9 available ports connected? Did any of these configurations allow for a similar ‘loop’ structure that would be present in the ABCD paradigm? Furthermore, additional details of the task design/construction should be provided to facilitate reproducibility efforts. E.g. dimensions, materials of the maze walls; materials of the corridors, lighting conditions, etc.

- The new histology-based mapping of probe channels, in addition to new analyses of subregional

distribution of neuron properties, provides a better understanding of why the authors have chosen to use the blanket term “medial frontal cortex”. One detail which remains unclear is if the authors actually included the 9.3% neurons that were histologically localized outside of mFC.

- The section “Tuning to basic task variables” could benefit from one or two citations to indicate if and how the methods of model fitting and performance assessment are standard practice.

- Fig. 1a: the ‘d→a’ trajectory paths are hard to see. Extended Data Fig. 1b: these plots are difficult to see.

- Extended Data Fig. 1a, right: it is unclear how the authors determined physical distance. How is the range of values on the y-axis so narrow? From the schematic on the left, the optimal number of steps taken to each reward can range between 1 and 4 steps, yet, all values on the plot are close to 2.

Referee #3 (Remarks to the Author):

I would like to thank the authors for their careful response to my (and other reviewer's) comments. They have clarified several aspects of their study, included new data and many new analyses. The central claim of the study will be of interest to the community. I have no further comment and the manuscript is suitable for publication in my opinion.

Author Rebuttals to First Revision:

Referees' comments:

Referee #1 (Remarks to the Author):

The authors have been very responsive to my prior comments and the manuscript is much improved. I'm happy to recommend publication. I do have a couple of remaining minor points.

We thank the reviewer for their constructive and helpful comments and for recommending publication. Their comments have helped us significantly improve the clarity of the manuscript and strengthen our analyses. Our study is much clearer because of their input.

1. The figure legend in Fig. 5c and EDFig. 5B states that the max firing rates are in the top right of the figure, but that doesn't seem to be the case.

The maximum firing rates are displayed in the top right of each individual firing rate matrix. This is because each firing rate matrix is normalised individually to emphasise firing rate "pattern" similarities across maps, rather than mean firing rates. Although size constraints mean that they are indeed very small. We apologise for this. We have now doubled the size of these so they are now more visible. Although we note that they are much more visible in the high quality version of the figure. Please also note the change in figure panel numbers.

Figure 5e

e) Lagged spatial field analysis. Bottom: Each row represents a different task and each column a different lag in task space, starting from the animal's current location (far right column) and then at successive task space lags in the past. Because of the circular nature of the task, past bins at lag X are equivalent to future bins at lag 360-X. Inset: zoomed in spatial maps at preferred lag. Top: the correlation of spatial maps across tasks at each lag. Colours are normalised per map to emphasise the spatial firing pattern, with maximum firing rates (in Hz) displayed at the top right of each map.

Extended data Figure 6b

b) Lagged spatial field analysis. Example plots showing spatial maps for 4 neurons. Each row represents a different task and each column a different lag in task space. Bottom: Activity of each neuron is plotted as a function of the animal's current location (far right column for each cell) and at successive task space lags in the past for the remaining columns. Because of the circular nature of the task, past bins at lag X are equivalent to future bins at lag $360-X$. Top: the correlation of spatial maps across tasks at each lag. To avoid confounds due to goal-progress tuning, all firing rates are calculated only in each neuron's preferred goal-progress bin (i.e. one-third of the entire session). Colours are normalised per map to emphasise the spatial firing pattern, with maximum firing rates (in Hz) displayed at the top right of each map.

2. The authors kilosort exclusion parameters don't make sense to me. I'm guessing their firing rates are usually <10 Hz (that's where knowing those max firing rates would come in handy!) and so if you had two neurons mistakenly grouped together the expected probability of spikes within 2 ms of one another is only 0.02. So a 10% threshold makes no sense at all.

Also, 2 ms is very long for a refractory period cutoff. They can figure out what a sensible cutoff in their data is by examining spikes from well-isolated neurons and gradually adjusting the cutoff. They will notice a step function where the true refractory period is. I'd be very surprised if it is 2 ms.

For reference, the ISI violation threshold we use is 0.1% of spikes < 1.2 ms.

We apologise for this misunderstanding. By "10% threshold" we don't mean that we allow 10% of spikes to be in the refractory period, that would indeed be far too permissive.

Instead, we mean that we allow the number of spikes in the refractory period to be at most 10% of the baseline bin, which is defined as the number of spikes in the 25ms bin (the maximum value for the autocorrelation plots we use for curation in kilosort).

We now quantify the total percentage of spikes in 1 ms and 2 ms refractory periods to be: $0.033\% \pm 0.003\%$ and $0.076\% \pm 0.004\%$ of spikes on average respectively. This is comparable with the reviewer's threshold of $<0.1\%$ of spikes in a refractory period of 1.2 ms. Indeed 93% of our neurons would pass this threshold ($<0.1\%$ of spikes in a refractory period of 1.2 ms).

A 2 ms refractory period is used as a first pass to account for neuron-to-neuron variability in refractory periods but we note that this allows neurons who's refractory period is shorter albeit with an effectively stricter threshold. Indeed we have conducted the cutoff adjustment analysis suggested by the reviewer and report that 99.2% of neurons have a maximum dropoff of spike counts at 2ms or more. These details are now reported in the revised Methods section under "**Electrophysiology, spike sorting and behavioural tracking**".

Referee #2 (Remarks to the Author):

The authors have made an impressive effort to address my concerns. New experiments and extensive, additional analyses were key to addressing the generality and robustness of the findings. The results of the ABCDE task –a new abstract task structure with a different periodicity as the original ABCD task– provide strong evidence that goal-progress cells in mFC can flexibly be sculpted by task structure, thereby positioning them as compelling candidates for ‘elemental building blocks of schemata’. Furthermore, these findings highlight the potential of this study’s findings and framework to generalize to many different forms of schemas. Additional analyses of task performance, as well as new experiments omitting the tone associated with reward A, have helped to rigorously demonstrate that it is indeed the abstract structure of the task (the schema) which is being learned and used by subjects to achieve optimal performance and zero-shot inference. The authors have also extended the rigor of the study in several ways. They have provided key methodological details as requested (e.g. Table 1 specifies the inclusion criteria and sample size for each figure panel). New analyses have further validated their methodology used to test SMB model predictions (e.g. ruling out the potential effect of nonlinearities in their regression model by using a linear-nonlinear Poisson model; including a more granular presentation of metrics used to characterize co-anchored neurons during sleep in Figure 7).

We thank the reviewer for their kind words. Their thorough comments have helped us significantly strengthen our main results and clarify methodological details. Our manuscript is markedly improved because of their comments and suggestions.

One concern, which I found not fully addressed in the revision, is regarding the authors’ approach to spike sorting data concatenated across consecutive days. I expand on this point below and offer several suggestions for the authors to address this.

Another concern, which should be easily addressed, is the overall clarity of the manuscript. I have made several suggestions below regarding aspects of the terminology, data presentation, and the title, which I found to be confusing.

1. Spike sorting data concatenated across consecutive days (“double days”)

The authors concatenate pairs of consecutive recording days to increase the number of tasks assessed for each neuron. The degree to which concatenated data had been used was not very obvious in the first version of the manuscript, which is why I requested additional information in the revision. The extensive documentation of which analyses were performed using concatenated data (Table 1) is much appreciated. The authors have also made an effort to annotate the manuscript/legends to indicate that concatenated data had been used. That being said, the extensive use of the concatenated data stands in contrast with the limited extent of quality metrics and validation.

Although the authors now include their criteria used to curate single units (based on refractory period contamination and firing rate fluctuation over sessions) and report the drop-out rate, I remain concerned that they have not adequately addressed the potential impact of probe drift across 2 consecutive days. I understand that the Kilosort versions used in the

study (2.5 and 3.0) have built-in drift correction algorithms well accepted for standard recordings; however, given that (1) pooling across consecutive days is not standard practice in the field, and (2) the implications for how the data are interpreted (explained below), I think it's important for the authors to more rigorously assess how units have been matched across consecutive days.

One key feature of the mFC single unit tuning reported in the study is that they can generalize or remap across distinct task sequences. Given that the task sequences were pooled across two days, it is critical to ensure that the activity of distinct units across days are not inadvertently collapsed into one. We can consider a simple example where the case of goal-progress tuned unit #1 firing selectively for state A on task X (day 1), and another goal-progress tuned unit #2 firing selectively for state B on task Z (day 2). If these units were indeed identical, the activity would be interpreted as remapping; if the units were in fact distinct and therefore erroneously matched, the activity would be interpreted as the unit being modulated by 2 states (or perhaps lacking state tuning).

I would suggest the authors show they have indeed rigorously matched units. For example, they could show actual waveforms and quality metrics of unit stability. They could do this for several units, including those whose tuning properties are showcased in the figures. The authors should also examine the extent of drift detected and corrected by Kilosort, and consider showing a drift plot (e.g. one of the KS output metrics/figures) for concatenated sessions. For example, the authors could use approaches in a one recent work which provided a characterization of unit stability over sleep periods recorded over >12 hour long recordings (see Methods and Extended Data Fig. 8 from Maboudi...Diba, Nature, 2024 (<https://www.nature.com/articles/s41586-024-07397-x>)).

In addition, I think it would help if the authors better communicated the necessity of concatenating data. Although I understand their goal to increase the number of tasks assessed for each neuron, the authors have stated that using a single day (3 tasks) is "insufficiently powered". It is unclear if and how they have quantified this. At the very least, authors should consider adding an explicit discussion of potential caveats for interpretation (or lack thereof).

We thank the reviewer for raising this important point. We agree that it is important to rule out if remapping across days could be due to spike sorting artefact. However, if this was the case, it would work against us for any tuning properties that we show are conserved across days. For example, goal progress tuning (**Figure 2**) and stability of mnemonic tuning to anchor for each cell (**Figure 5**). We also note that the remapping effects shown in **Figure 3** are all within the same day and so are unaffected by two-day concatenation.

Nevertheless, we now provide more explicit quantification showing that the **concatenation overwhelmingly succeeds in capturing the same neuron across days**. For this we take advantage of the highly conserved goal-progress tuning that is characteristic of mFC neurons (**Figure 2**). We assessed "goal-progress correlation" between different tasks that are taken from the same day and then repeated this for the same neuron for pairs of tasks taken from different days. This allowed us to index the extent to which basic tuning of cells is conserved across days (both in ABCD and ABCDE days). This shows:

1-An exceptionally tight relationship between each neuron's within and across-day goal-progress correlation - i.e. goal-progress correlation between tasks in the same day is itself highly correlated with goal-progress correlation between tasks across days. Pearson correlation N=1540 neurons, $r=0.88$, $P=0.0$).

2-Another way of looking at this - Goal-progress correlation values are indistinguishable within and across days - i.e. for a given neuron, goal-progress tuning is equally likely to be conserved across days as it is within the same day. (Within-day correlation: 0.63 ± 0.01 , Across-day correlation: 0.62 ± 0.01 Wilcoxon test: N=1549, statistic=567550, $P=0.14$, df=1540)

3-Almost all neurons that are significantly goal-progress tuned within a day also maintain their goal progress tuning across days (95.4% - Two proportions test: N=1249 neurons, $z=45.2$, $P=0.0$). Significance was calculated by comparing goal-progress correlation to the 95th percentile of circularly shifted permutations, individually for each neuron.

Note that these analyses are not comparing proportions of goal progress cells in day 1 versus day 2. Rather, they're comparing tuning when assessed within the same day to that assessed across days.

A-Within-day tuning is assessed by comparing the goal-progress tuning curve across 3 tasks in the same day. Then averaging the correlation values across days 1 and 2.

B-Across-day tuning is assessed by comparing the goal-progress tuning curve from task 1 in day1 against all tasks in day 2, and then repeating for task 2 in day 1 against all tasks in day 2...etc. Then averaging the correlation values.

Thus, our cross-day concatenation procedure conserves the basic tuning properties of the mFC neurons, indicating that we have successfully tracked the same neurons across days. These analyses have now been added in the Methods section under "**Electrophysiology, spike sorting and behavioural tracking**".

We have also now included additional evidence for spike sorting stability in **Extended data figure 2c** showing:

1-Drift traces, showing cross day stability of probe position .

2-Example waveforms from neurons spike sorted across days who's anchoring properties are displayed in **Figure 5** and **Extended data figure 6**

Extended data figure 2

- c) Data was spike sorted across concatenated sessions spanning two recording days for the GLM analyses below and later anchoring analysis in figures 5-7. Top: Here we show an example “Estimated drift trace” for a concatenated double day, showing a largely stable recording set up. The plot shows the estimated probe drift relative to the brain across the two recording days along the depth of the neuropixels probe. Bottom: Example mean spike waveforms from 3 different neurons across 3 different animals. The plots show the mean of the first 100 spikes on day1 (black) and the mean of the last 100 spikes on day2 (red), illustrating stability of spike detection across days. The spikes are from neuron 1 and neuron 2 in Extended data figures 6b and the neuron in Figure 5e respectively. Scale bars: Vertical: 200 μ V, Horizontal: 0.5 ms.

The main reason for using concatenated days is to be sufficiently powered for the generalisation analyses (in Figure 5). In essence the aim is to capture multiple instances where animals visit the same anchor points (e.g. same reward locations) but in different sequences. The more tasks we can get for this the more we can sample the same reward locations in different task sequences. With 3 tasks, a total of 12 reward locations are presented (4x3) meaning each of the 9 reward locations is seen in 1.33 different tasks on average. With 6 tasks, reward locations are experienced in 2.67 different tasks on average. This gives us the ability to assess the same anchor points in different task sequences in a cross-validated manner and hence assess whether mnemonic tuning to anchor is conserved across tasks. E.g. a neuron fires 2 states after reward in location 7 regardless of whether the animal is now in location 1 or 8. For non-rewarded locations, the situation is more complex and dependent on the animal’s behavioural trajectories between rewards, but the same qualitative principle applies: more tasks give more visits to a given behavioural step as part of different behavioural sequences. Ideally, we would be even more powered with more tasks and hence more concatenation, but practical limitations prevent reliably tracking the same neuron across extended periods of multiple days. This is now more clearly explained in the methods section under “**Mnemonic task space tuning**”.

Clarity of the manuscript

1. Data presentation: on more than several occasions, the style of data presentation appears to preclude important details or makes it very difficult for the reader to extract the information. For example:

- Fig. 2d: it is not obvious that the polar plots on the top row are organized into sets of polar plots from the same cell, while the polar plots on the bottom are from 6 different cells. Please consider using more informative labels in figures (e.g. “spatially tuned cell” rather than “spatially tuned”).

Thanks for pointing this out. We have amended this figure to clarify as suggested (see our answer to the point below).

- There are many details which are impossible to see due to their small font size and are often not explained in the legend. e.g. Fig. 2d, numbers showing firing rate in polar plots and the max firing rates in Fig. 5c heatmaps.

We have now adjusted these figures (and equivalent ones e.g. the new **Extended data figure 6b**) to at least double the size of the maximum firing rates. We have also added the detail in the legend where it was missing (e.g. **Figure 2c,d**). Note the change in figure panel numbers.

Figure 5e

e) Lagged spatial field analysis. Bottom: Each row represents a different task and each column a different lag in task space, starting from the animal’s current location (far right column) and then at successive task space lags in the past. Because of the circular nature of the task, past bins at lag X are equivalent to future bins at lag 360-X. Inset: zoomed in spatial maps at preferred lag. Top: the correlation of spatial maps across tasks at each lag. Colours are normalised per map to emphasise the spatial firing pattern, with maximum firing rates (in Hz) displayed at the top right of each map.

Extended data Figure 6b

b) Lagged spatial field analysis. Example plots showing spatial maps for 4 neurons. Each row represents a different task and each column a different lag in task space. Bottom: Activity of each neuron is plotted as a function of the animal's current location (far right column for each cell) and at successive task space lags in the past for the remaining columns. Because of the circular nature of the task, past bins at lag X are equivalent to future bins at lag 360-X. Top: the correlation of spatial maps across tasks at each lag. To avoid confounds due to goal-progress tuning, all firing rates are calculated only in each neuron's preferred goal-progress bin (i.e. one-third of the entire session). Colours are normalised per map to emphasise the spatial firing pattern, with maximum firing rates (in Hz) displayed at the top right of each map.

Figure 2

- c) Neurons are tuned to the animal's relative progress to goal ("goal-progress tuned"). Inset: a raster plot of firing activity in one state (C) of a cell that consistently fires shortly before a goal is reached. Top right: maximum firing rate (Hz).
- d) Some goal-progress-tuned cells are additionally modulated by state in a given task ("goal-progress + State tuned"). Inset: Polar plots and spatial maps for two "goal-progress + State tuned" neurons across two distinct task configurations.

Please note that the high quality version of these figures shows these details more clearly.

2. Terminology: The manuscript uses a variety of terminology (e.g. 'behavioral step', 'anchor', 'goal-progress/place conjunction') interchangeably and often in a confusing manner (e.g. Figure 5 includes the notion of 'behavioral steps' in the title but then doesn't refer to this again in the legend and instead use 'anchor' and 'goal-progress/place'. I'd recommend clarifying the manuscript by trying to use consistent terms and less jargon throughout.

We thank the reviewer for pointing this out. We have now gone through the manuscript, figure legends and methods sections and both simplified and clarified our usage of these terms.

Minor Comments

- I am still a bit confused about how animals achieve 'near perfect' performance. On one hand, authors show that more than half of the subjects exhibit suboptimal performance (i.e. taking longer routes) associated with persisting behavioral biases from before task exposure (Extended Data Fig. 1f). It is difficult to understand how this is compatible with the striking proportion of 'perfect trials' which require all transitions in a trial to take the shortest route (and presumably pooled from the same 13 subjects) (Extended Data Fig. 1i). Is this simply because the 'perfect trials' considers only the first trial's d→a transition?

We apologise for this misunderstanding. **Extended data figure 1f** (now **Extended data figure 1h**) isn't showing the overall performance of animals, rather its showing the per mouse correlation between:

1-relative path distance on a given trial in an ABCD task - i.e. ratio of taken path distance vs optimal (shortest) path distance

2-the mean **baseline** probability for all steps **actually taken by the animal** on that same trial (measured from exploration session before exposure to any ABCD task) - i.e. how likely was the animal to take this path on a given trial before task exposure?

This correlation is positive overall, indicating that **when animals take more suboptimal (longer) routes they do so through high probability steps**- i.e. ones that they were predisposed to take prior to any task exposure. This suggests that mistakes are associated with persisting behavioural biases. While this is true on average, the single dots show that some animals don't follow this trend. This has no bearing on the overall performance of each animal, just the relationship between suboptimal routes and prior biases. This has now been clarified in the Methods section under "**Behavioural Scoring**".

We also note that what we define as a perfect trial in **Extended data figure 1i** (now **Extended data figure 1k**) is a trial where the shortest route in the entire trial was taken (i.e. shortest route in all of the transitions (a→b, b→c, c→d and d→a) for a given trial) and not just the d→a transition. This is now clarified in the legend of **Extended data figure 1k**.

Extended data figure 1

- k) Mean proportion of "perfect trials" where all transitions (a→b, b→c, c→d and d→a) in a given trial were taken via the shortest route. Left: Scatter plot of mean proportion of perfect trials in the first 20 trials of early vs late tasks. Wilcoxon test (two-sided) N=13 animals, Statistic=11.0 P=0.028. Right: bar plot of the same data showing that, for both early and late tasks, the proportion of perfect trials is significantly

above chance: T-test (two-tailed) against chance (0.007): Early tasks statistic=2.55, P=0.025; Late tasks - statistic=4.06, P=0.002.

- Authors should provide more details on the partially connected maze (6 out of 9 ports available) used during pre-training. It appears that subjects were introduced to this partially configured maze twice before the full ABCD paradigm, so it's important to understand which parts of the ABCD task structure (if any) were present in this pre-training stage. The authors note that there was 'no explicit task structure' (pg. 44). How were the 6 out of 9 available ports connected? Did any of these configurations allow for a similar 'loop' structure that would be present in the ABCD paradigm? Furthermore, additional details of the task design/construction should be provided to facilitate reproducibility efforts. E.g. dimensions, materials of the maze walls; materials of the corridors, lighting conditions, etc.

Thank you for raising this important point. During pre-selection and pre-training, a partially connected version of the maze was used with only 5-7 accessible nodes out of 9. We note that the original text had a mistake where we stated 6 out of 9 ports were available, which actually denotes the average number of nodes available. We now clarify that the same animals could experience sessions with 5,6 and/or 7 nodes out of 9 in pre-selection/pre-training. The number of available nodes and their connectivity was fixed in the same 20 minute session but changed across sessions. Here, animals learned that poking in wells delivered water reward and that after gaining reward they must go to another node. Animals that obtained 40 or more rewards per 20 minute session were taken forward to training.

The available nodes were connected pseudorandomly (**Extended data figure 1a**), such that animals could access all available nodes but there were always "dead-ends". The identity, number and connectivity of the available nodes was changed for every new 20 minute session, to minimise any behavioural biases induced by the exact spatial structure. We now add this detail in the Methods section under "**Behavioural Training>Habituation/Pre-selection**" and "**Behavioural Training>Habituation/Pre-training**".

Extended data figure 1

- a) Three example connection configurations for the pre-selection and pre-training sessions done before exposure to the first ABCD task. Here a subset of 5-7 maze locations (nodes) were available to the mouse and each node was rewarded provided the animal did not just receive reward in the same node.

With regards to the possibility of pre-learning an ABCD loop, this is extremely unlikely as:

1-Animals were exposed to 20 minute sessions, each in a different configuration - and had no incentive to explore via any explicit structure as reward was always available provided they didn't return to the same node they just got reward from - this precludes any systematic learning of the ABCD structure (or any other consistent structure)

2-Given how long it takes animals to learn the explicitly presented ABCD structure (>6 weeks) it is extremely unlikely that animals would implicitly learn a structure in the short number of pre-training sessions provided (often as little as 2 x 20 minute sessions and a maximum of 8 x 20 minute sessions in total - all with different configurations) due to any accidental structure imposed by the animal's biased exploration.

3-Indeed we find no evidence of ABCD structural knowledge in the first 20 tasks (**Figure 1e (left)** and **Extended data figure 1q (left)**)

4-Our analyses deal with representations in well-trained animals, where there is clear evidence for ABCD-structure knowledge (**Figure 1e (right)** and **Extended data figure 1q (right)**) and pertain to the post-training representations rather than the learning process per se.

We now also provide more details of the Maze dimensions in the Methods section under a new section “**Behavioural Training>Apparatus**”. We also provide an open source link to the maze design files detailing the design and material of all maze components:

<https://github.com/pyControl/hardware/tree/master/GridMaze>

- The new histology-based mapping of probe channels, in addition to new analyses of subregional distribution of neuron properties, provides a better understanding of why the authors have chosen to use the blanket term “medial frontal cortex”. One detail which

remains unclear is if the authors actually included the 9.3% neurons that were histologically localized outside of mFC.

We thank the reviewer for prompting us to clarify this. We do indeed include all recorded neurons in all analyses throughout the manuscript but indicate where these pertain to different mFC regions in **Extended data figure 2i**, **Extended data figure 4 j,k** and **Extended data figure 7f**. This is stated in the methods section under “**Surgeries**”:

- The section “Tuning to basic task variables” could benefit from one or two citations to indicate if and how the methods of model fitting and performance assessment are standard practice.

We have now added a citation to Akam et al 2021 showing the use of the same model fitting and permutation-test based performance assessment under the section “**Tuning to basic task variables**”.

- Fig. 1a: the ‘d→a’ trajectory paths are hard to see. Extended Data Fig. 1b: these plots are difficult to see.

We assume the reviewer means **Figure 1b** for the ‘d→a’ trajectory paths (now moved to **Extended data figure 1c**). We have now changed the thickness/colour of these lines to make them clearer. For the old **Extended Data Figure 1b** (now **Extended Data Figure 1d**) we have re-scaled the y-axis scale to make the data points more visible.

Extended data figure 1

- c) Example paths from 3 different mice performing 3 different tasks. Each row is a set of 5 consecutive trials from the same mouse and task. Single trial paths are superimposed upon whole session coverage shown in grey. Mice rapidly converged on near-optimal routes and used only a subset of the available paths.
- d) Reward sequences showed minimal correlation across all tasks in all mice (left) and no correlations on all tasks for mice where neuronal data was recorded (7/13 mice middle) or on the late 3 task days from neural mice (right). Note that electrophysiological data in this manuscript is all collected from the late 3 task days. T-test (two-sided) against 0 correlation: All tasks: N=13 mice, $r=0.002$, statistic=2.91, $P=0.013$, $df=12$; All tasks from neural mice (mice where neuronal data was recorded): N=7 mice, $r=0.002$, statistic=1.81, $P=0.120$, $df=6$; tasks on Late 3-task days from neural mice (tasks where neural data was recorded): N= 7 mice, $r=4.5 \times 10^{-4}$, statistic=0.22, $P=0.634$, $df=6$.

- Extended Data Fig. 1a, right: it is unclear how the authors determined physical distance. How is the range of values on the y-axis so narrow? From the schematic on the left, the

optimal number of steps taken to each reward can range between 1 and 4 steps, yet, all values on the plot are close to 2.

In **Extended data figure 1a** (Now **Extended data figure 1b**), we quantify the physical distance as the minimum number of steps between rewards which indeed ranges from 1 to 4 as the reviewer states.

However, what is plotted in this figure is the **mean** physical distance between rewards that are 1, 2 or 3 states apart in the forward direction **averaged across all tasks experienced by a given mouse**. We have now corrected the figure legend to reflect this. The plot shows that the mean physical distance is ~ 2 across mice regardless of the number of states between rewards.

- b) Tasks were designed such that task space and physical space are orthogonal to each other. Left: schematic showing that optimal path lengths between rewarded goals differed both within and between tasks. Right: A bar plot showing that the task space “distances” between reward locations (how many task states are between the rewards) are not correlated with the physical distances in the maze (the optimal number of steps taken to reach reward). Data points represent individual mice, where physical distances are averaged across all tasks experienced by a given mouse. Pearson correlation: $r=6.04 \times 10^{-18}$ $P=1.0$; One-way ANOVA statistic=2.02, $P=0.147$, $df=12$

Referee #3 (Remarks to the Author):

I would like to thank the authors for their careful response to my (and other reviewer's) comments. They have clarified several aspects of their study, included new data and many new analyses. The central claim of the study will be of interest to the community. I have no further comment and the manuscript is suitable for publication in my opinion.

We are grateful to the reviewer for thoroughly and constructively assessing our manuscript. Their comments have helped make our manuscript stronger, clearer and more accessible.